# Historical greenhouse gas concentrations for climate modelling (CMIP6)

Malte Meinshausen[1,2,3] Elisabeth Vogel[1,2], Alexander Nauels[1,2], Katja Lorbacher[1,2], Nicolai Meinshausen[4], David M. Etheridge[5], Paul J. Fraser[5], Stephen A. Montzka[6], Peter J. Rayner[2], Cathy M. Trudinger[5], Paul B. Krummel[5], Urs Beyerle[7], Josep G. Canadell[8], John S. Daniel[9], Ian G. Enting[10], Rachel M. Law[5], Chris R. Lunder[11], Simon O'Doherty[12], Ron G. Prinn[13], Stefan Reimann[14], Mauro Rubino[5,15], Guus J.M. Velders[16], Martin K. Vollmer[14], Ray H.J. Wang[17], Ray Weiss[18]

[1]Australian-German Climate & Energy College, The University of Melbourne, Parkville, Victoria, Australia
[2]Department of Earth Sciences, The University of Melbourne, Parkville, Victoria, Australia
[3]Potsdam Institute for Climate Impact Research, Potsdam, Germany
[4]Seminar for Statistics, Swiss Federal Institute of Technology (ETH Zurich), Zurich, Switzerland.
[5]CSIRO, Oceans and Atmosphere, Aspendale, Victoria, Australia
[6]NOAA, Earth System Research Laboratory, Global Monitoring Division, Boulder, Colorado, USA
[7]Institute for Atmospheric and Climate Science, Swiss Federal Institute of Technology (ETH Zurich), Zurich, Switzerland
[8]Global Carbon Project, CSIRO Oceans and Atmosphere, Canberra, ACT, Australia
[9]NOAA, Earth System Research Laboratory, Chemical Sciences Division, Boulder, Colorado, USA
[10]The University of Melbourne, Victoria, Australia (retired)
[11]Norwegian Institute for Air Research, Kjeller, Norway
[12]University of Bristol, Bristol, United Kingdom
[13]MIT, Cambridge, MA, USA
[14]Empa, Swiss Federal Laboratories for Materials Science and Technology, Laboratory for Air Pollution and Environmental Technology, Switzerland
[15]Dipartimento di matematica e fisica, Seconda Università degli studi di Napoli, Caserta, Italy
[16]National Institute for Public Health and the Environment (RIVM), Bilthoven, Netherlands
[17] School of Earth and Atmospheric Sciences, Georgia Institute of Technology, Atlanta, Georgia, USA
[18]Scripps Institution of Oceanography, La Jolla, CA, USA

*Correspondence to:* M. Meinshausen (malte.meinshausen@unimelb.edu.au)

**Abstract.** Atmospheric greenhouse gas (GHG) concentrations are at unprecedented, record-high levels compared the last 800,000 years. Those elevated GHG concentrations warm the planet and - partially offset by net cooling effects by aerosols - are largely responsible for the observed warming over the past 150 years. An accurate representation of GHG concentrations is hence important to understand and model recent climate change. So far, community efforts to create composite datasets of GHG concentrations with seasonal and latitudinal information have focused on marine boundary layer conditions and recent trends since the 1980s. Here, we provide consolidated data sets of historical atmospheric concentrations (mole fractions) of 43 GHGs to be used in the Climate Model Intercomparison Project – Phase 6 (CMIP6) experiments. The presented datasets are based on AGAGE and NOAA networks, firn and ice core data, and archived air data, and a large set of published studies. In contrast to previous intercomparisons, the new datasets are latitudinally resolved and include seasonality. We focus on the period 1850 to 2014 for historical CMIP6 runs, but data are also provided for the last 2000 years. We provide consolidated datasets in various spatiotemporal resolutions for carbon dioxide ($CO_2$), methane ($CH_4$) and nitrous oxide ($N_2O$), as well as

other GHGs, namely 17 ozone depleting substances, 11 hydrofluorocarbons (HFCs), 9 perfluorocarbons (PFCs), sulfur hexafluoride ($SF_6$), nitrogen trifluoride ($NF_3$) and sulfuryl fluoride ($SO_2F_2$). In addition, we provide three equivalence-species that aggregate concentrations of GHGs other than $CO_2$, $CH_4$ and $N_2O$, weighted by their radiative forcing efficiencies. For the year 1850 that is used for pre-industrial control runs, we estimate annual global mean surface concentrations of $CO_2$ at 284.3 ppm, $CH_4$ at 808.2 ppb and $N_2O$ at 273.0 ppb. The data are available at https://pcmdi.llnl.gov/search/input4mips/ and www.climatecollege.unimelb.edu.au/cmip6. While the minimum CMIP6 recommendation is to use the global and annual mean time series, modelling groups can also choose our monthly and latitudinally resolved concentrations that imply a stronger radiative forcing in the northern hemisphere winter (due to the latitudinal gradient and seasonality).

## 1    Introduction

Emissions from the burning of fossil fuels, deforestation, agricultural activities and the production of synthetic GHGs are the primary reasons for the observed increases in GHG concentrations, defined as mole fractions in dry air. The elevated GHG concentrations induce a radiative forcing that in turn would cause more than the observed recent global warming if it were not for the cooling effect by aerosols (Fig. TS.10 in IPCC WG1 AR4 (IPCC)). An accurate quantification of anthropogenic and natural climate drivers is crucial for general circulation and Earth System models. Simulations by these models for the historical time periods, e.g. since 1850, can only be meaningfully compared to observations (e.g. surface temperature, ocean heat uptake) to the degree that input forcings are an accurate representation of the past. The difficulty with many anthropogenic climate drivers is that their global-mean magnitude, their latitudinal gradient and seasonal cycle are uncertain further back in time, even for the main GHGs carbon dioxide ($CO_2$), methane ($CH_4$) and nitrous oxide ($N_2O$). Systematic observational efforts started in 1957-1958, measuring $CO_2$ at the South Pole and Mauna Loa observatories (Keeling et al., 2001). Measurements of archived air, firn air and ice cores from both polar regions provide records for the pre-observational time. To date, reconstructions of millennial global-mean time series based on ice and firn data have been performed, e.g. for $CO_2$ over the last millennia (Ahn et al., 2012; MacFarling Meure et al., 2006; Rubino et al., 2013). For the more recent past, several studies investigated firn and ice data to constrain halocarbons (Buizert et al., 2012; Martinerie et al., 2009; Mühle et al., 2010; Sturrock et al., 2002; Trudinger et al., 2016a), some of them with hemispheric resolution. In terms of latitudinally-resolved monthly data, there have only been a few synthesis products, namely for $CO_2$, $CH_4$ and $N_2O$ over the instrumental record over the past 20 to 40 years (NOAA, 2013; NOAA ESRL GMD, 2014a, b, c). For this recent past, the World Data Centre for Greenhouse Gases (WDCGG) (ds.data.jma.go.jp/gmd/wdcgg/) also provides a synthesis with global and hemispheric means for $CO_2$, $CH_4$ and $N_2O$ (Tsutsumi et al., 2009). In light of the observational gaps further back in time, some studies, such as Keeling et al. (2011), used linear regressions between fossil fuel use and latitudinal $CO_2$ concentration trends to separate natural from anthropogenically-induced effects, which allows us to infer latitudinal gradients back in time.

In previous climate model inter-comparison projects (Meehl et al., 2005), global-mean concentrations have been prescribed (Meinshausen et al., 2011), with some models constraining internally generated fields of GHG concentrations to match those

global-mean values. Here, we update those global-mean and annual-mean GHG concentration time-series for the historical period over years 0 to 2014, with 'historical' simulations in the CMIP6 model intercomparison (Eyring et al., 2016) focussed on the most recent period 1850 to 2014. In addition, we provide hemispheric and latitudinal monthly-resolved fields for 43 GHGs in total. In the past, the large latitudinal and seasonal gradient of GHG radiative forcing has not been consistently applied

to model radiative forcing and climate change. The new datasets provide a more consistent starting point for climate model experiments. The monthly and latitudinal resolution of this new GHG dataset is designed to have a similar resolution as the monthly solar forcing (Matthes et al., 2016) and monthly and latitudinally resolved ozone and aerosol abundances. Many GHGs also have significant longitudinal (land/ocean) and diurnal variations but we do not attempt to resolve them. Neither do we provide vertical gradients of the GHGs concentrations and only discuss possible vertical extension methods (section 4.1

'The vertical dimension') in case models do not have their own methods to derive vertical gradients.

In this study, we compile one possible reconstruction of latitudinally and monthly resolved fields, as well as global annual means of surface GHG concentrations for 43 gases from year 0 to 2014, as input for the forthcoming model inter-comparison experiments that are part of the Phase-6 Coupled Model Inter-comparison project (CMIP6) (Eyring et al., 2016). Specifically, we provide the pre-industrial control runs at 1850 forcing levels (picontrol), the experiment with abruptly quadrupled $CO_2$

concentrations (abrupt4x), the standard experiment of a 1% annual $CO_2$ concentration increase (1pct2co2), and the historical runs that are driven with best-guess estimates of historical forcings since 1850. Species that are radiatively less important than $CO_2$, $CH_4$ and $N_2O$ ('importance' here being measured as radiative forcing exerted in year 2014 compared to 1750) are provided individually as well as aggregated as HFC-134a and CFC-12 equivalent concentrations. The description of the datasets geared towards CMIP6 modelling groups is provided in section 4, including a description of available data formats

and CMIP6 minimum recommendations.

The design principle for this long-term dataset is to provide a plausible reconstruction of past GHG concentrations to be used in climate models. Using various gap-filling procedures, reconstruction and extensions, this dataset aims to reflect observational evidence of both recent flask and *in situ* observations from the worldwide network of NOAA ESRL and AGAGE stations, as well as Antarctic and Greenland ice core and firn data over the last two thousand years, where available.

Furthermore, many detailed literature studies (Arnold et al., 2013; Arnold et al., 2014; Aydin et al., 2010; Butler et al., 1999; Ivy et al., 2012; Martinerie et al., 2009; Montzka et al., 2015; Mühle et al., 2010; Oram et al., 2012; Sturrock et al., 2002; Trudinger et al., 2004; Trudinger et al., 2016a; Velders and Daniel, 2014; Vollmer et al., 2016; Worton et al., 2006) for radiatively less important species are compared with our data product in the factsheet figures for the specific gases (Table 12 and Appendix A with Fig. 20 to Fig. 59) or synthesised where direct observational records from the above networks were not

available.

The predominant climate effect of GHG increases is captured by the global and annual mean concentrations throughout the atmosphere. The surface global and annual mean concentrations provided here, in combination with the models' approximations for the vertical concentration profile, are the minimum standard for CMIP6 models. Assimilating a latitudinally and seasonally resolved data product serves two purposes. Firstly, to derive the global and annual means from sparse

observations rests on knowledge or assumptions about spatial and seasonal distributions. Secondly, to open the opportunity for some modelling groups to go beyond the prescription of global and annual mean concentrations.

Undoubtedly, some of the assumptions stretch into unknown territory, such as the seasonality of the $CO_2$ concentrations in pre-observational times or the time-variability of latitudinal gradients, let alone the higher frequency fluctuations of global-mean concentrations during the time when only ice core data are available. Errors in the historical forcing do propagate and can hinder the comparison between observations and models. This study therefore had to find a workable compromise between providing a complete dataset that covers the whole time and space domain and being as close as possible to sometimes sparse observations. Hence, the remaining uncertainties in concentration gradients should be kept in mind, although they might not be of primary concern in regard to the inter-comparison aspect of the multi-model ensemble runs. Thus, while our CMIP6 community dataset will improve on the global- and annual-mean time-series prescribed for the last set of CMIP5 experiments on a number of key aspects, many research questions remain open.

The underlying reasons for meridional gradients of annual-mean concentrations are manifold (Keeling et al., 1989a; Keeling et al., 1989b; Tans et al., 1989). For one, the sources of anthropogenic GHGs from fossil fuel burning and cement production or industrial activities are not evenly distributed with latitude, but concentrated in the mid-northern land masses. In the case of $CO_2$, emissions from deforestation are not uniformly distributed with latitude either. The pattern of land-use related emissions is even less stationary, with $CO_2$ uptakes and sources predominantly focussed in the mid-northern latitudes up until earlier in the 20[th] century, shifting more towards lower latitudes in recent decades (Hurtt et al., 2011). This study uses an approach based on simple regressions that implicitly rest on the assumption of a fixed pattern approximation (such as Keeling et al., 2011). One complication to retrieve the latitudinal pre-industrial $CO_2$ concentration profile is that $CO_2$ fertilization and temperature effects on the carbon cycle, both over ocean and land, change both the magnitude and spatial patterns of natural $CO_2$ fluxes. Lastly, both the diurnal and seasonal cycle of photosynthesis and its covariance with vertical atmospheric mixing can have a pronounced effect on measured surface concentrations (the so-called 'rectifier' effect), increasing annual mean northern hemispheric $CO_2$ surface concentrations by up to 2.5 ppm (Denning et al., 1999).

To dissect and analyse the different causes for temporal and spatial heterogeneity in surface concentrations, a rich body of literature analyses observed latitudinal and seasonal gradients with various inversion techniques. Recent research provides a clearer picture in regard to the causes of the change in seasonality of $CO_2$ concentrations (Forkel et al., 2016), a topic researched already in 1989 (Kohlmaier et al., 1989) based on the $CO_2$ fertilization effect on northern hemispheric terrestrial biota. Generally, the research into meridional and seasonal variations employs various atmospheric inversion techniques (Enting and Mansbridge, 1991, 1989; Enting et al., 1995; Enting, 1998; Rayner et al., 1999) to match observed concentrations with source and sink pattern estimates (Baker et al., 2006; Enting et al., 1995; Gurney et al., 2002; Gurney et al., 2003; Gurney et al., 2004; Keeling et al., 1989a; Keeling et al., 1989b; Peylin et al., 2013; Rayner et al., 1999; Tans et al., 1989; Tans et al., 1990a). Similarly to $CO_2$, the spatial variation in $CH_4$ concentrations is used for model inversions to infer sources and sinks (Fung et al., 1991; Kirschke et al., 2013).

There is a substantial lack of observational evidence of both seasonality and latitudinal $CO_2$ gradients in pre-industrial times. Given that atmospheric $CO_2$ is not well preserved in the Greenland ice (Anklin et al., 1995; Barnola et al., 1995), the pre-observational north-south gradient cannot be inferred or derived from the Greenland and Antarctic ice core records. Alternatively, understanding biospheric sink and source dynamics could provide vital evidence to infer pre-industrial surface concentration patterns. In this study, we do not employ any such inversion models or results, and only note that our pre-industrial meridional and seasonal variations should be regarded as highly uncertain. However, some plausibility of the $CO_2$ gradients is gained by comparison with some model studies (section 5 'Discussion'). High-latitude records of $CH_4$ are available from both hemispheres (MacFarling Meure et al., 2006; Mitchell et al., 2013; Rhodes et al., 2013) allowing us to estimate pre-industrial large-scale $CH_4$ concentration gradients.

## 2    Methods

To achieve the goals of this study, several analytical steps were taken to assimilate the observational data. Global-mean and annual mean concentrations are of primary interest, but the discussion also covers latitudinal and seasonal variations. The assimilation procedure for sparse observational data requires accounting for this spatio-temporal heterogeneity to derive global

and annual means.

We consider a total of 43 GHGs: $CO_2$, $CH_4$, $N_2O$, a group of 17 ozone depleting substances made up of five CFCs (CFC-12, CFC-11, CFC-113, CFC-114, CFC-115), three HCFCs (HCFC-22, HCFC-141b, HCFC-142b), three halons (Halon-1211, Halon-1301, Halon-2402), methyl chloroform ($CH_3CCl_3$), carbon tetrachloride ($CCl_4$), methyl chloride ($CH_3Cl$), methylene chloride ($CH_2Cl_2$), chloroform ($CHCl_3$), and methyl bromide ($CH_3Br$), and 23 other fluorinated compounds made up of 11

HFCs (HFC-134a, HFC-23, HFC-32, HFC-125, HFC-143a, HFC-152a, HFC-227ea, HFC-236fa, HFC-245fa, HFC-365mfc, HFC-43-10mee), nine PFCs ($CF_4$, $C_2F_6$, $C_3F_8$, $C_4F_{10}$, $C_5F_{12}$, $C_6F_{14}$, $C_7F_{16}$, $C_8F_{18}$, and c-$C_4F_8$), $NF_3$, $SF_6$, and $SO_2F_2$.

All concentrations given here are dry air mole fractions and we use 'mole fractions' and 'concentrations' interchangeably and synonymously with 'molar mixing ratios'. For simplicity, we denote the dry air mole fractions 'μmol mol$^{-1}$', 'nmol mol$^{-1}$' and 'pmol mol$^{-1}$' as parts per million (ppm), parts per billion (ppb) and parts per trillion (ppt), respectively. Note that dry air mole

fractions are independent of temperature and pressure, while volume mixing ratios (e.g. ppmv) for mixtures of non-ideal real gases are not, and at standard temperature and pressure conditions can differ significantly from their corresponding mole ratios.

### 2.1    Summary of assimilation approach.

We perform three consecutive steps to synthesize the global mole fraction fields over the full-time horizon from year 0 to year 2014. First, we aggregate the available observational data over the recent instrumental period. Second, we estimate three

components of the global surface concentration fields from these data, namely global mean mole fractions, latitudinal gradients and seasonality. Third, we extend those components back in time with – *inter alia* – ice-core or firn data. The full historical GHG concentration field can then be generated by the time-varying components.

Under this basic assimilation model, the concentration $\hat{C}(l, t)$ at any point in time *t* and in a latitudinal band *l* can be written as:

$$\hat{C}(l,t) = \overline{C_{global}}(t) + \hat{S}_{l,m}(y) + \hat{L}_l(y) \hspace{4cm} (1)$$

Where $\overline{C_{global}}(t)$ is the global-mean dry air mole fraction at time t, and $\hat{S}_{l,m}$ is the seasonality in each latitude l and month m, and $\hat{L}_l(y)$ is the latitudinal annual-mean deviation in year *y* at latitude *l*. With this assimilation model, and the optimal low rank approximations of seasonality and latitudinal gradients, a regularisation of the data is performed by a principal components analysis, which creates a degree of robustness against data gaps or outliers. Other methods, like a harmonic

representation of station data, have, in principle, a similar smoothing and regularisation effect (Masarie and Tans, 1995), although quantitative differences exist (section 5.4 "Comparison").

A detailed data flow diagram of how the historical GHG mole fractions are derived in this study is provided in Fig. 1. The subsequent section will describe the method step-by-step as indicated by the green circles in Fig. 1 and also tabulated for the three main GHGs in Table 1.

### 2.1.1    Step 1: Aggregating raw station data.

Atmospheric measurements are taken in remote environments or locations that are closer to pollution sources, in continental or marine areas, at different times of the day or night, at different altitudes, and different seasons of the year, often using different calibration scales. This poses challenges for any synthesis of observational data.

The observational station data over the recent decades used in this study are predominantly sourced from the networks operated by NOAA (Earth System Research Laboratories: ESRL), and AGAGE. In general, we use monthly station data provided by

the respective networks as a starting point. In the case of the AGAGE network, monthly averages are provided with and without pollution events (http://agage.eas.gatech.edu/data_archive/agage/ and http://cdiac.ornl.gov/ftp/ale_gage_Agage/AGAGE/). We chose the monthly averages that include pollution events (file-endings '.mop', with the exception of $CH_2Cl_2$, in which case data issues warranted the use of monthly station averages without pollution events). The approach that we do not restrict our source data to background conditions is consistent with our approach elsewhere – and the NOAA network monthly station

averages - which do not screen out pollution events (although the dominant number of NOAA flask measurements will likely be biased towards background conditions rather than pollution events owing to their location and sampling protocols at most sites focussed on collecting background air). In total, $CO_2$ data from 81 stations from the NOAA flask network, and 3 stations from the NOAA *in situ* data stations are used (Table 2). For $CH_4$, 87 sampling stations from the NOAA flask network and 5 stations from the AGAGE *in situ* network are compiled (Table 3). For $N_2O$, data from flask and in situ measurements at 13

stations of the NOAA HATS global network are combined with data from 5 stations from the AGAGE network (Table 4). For other gases, the AGAGE and NOAA coverage and timeframes vary, with individual station's codes provided in panels f of the individual gases' factsheets (Appendix A with Fig. 20 to Fig. 59). We provide references to the used NOAA and AGAGE data in Table 12.

Calibration scales, i.e. the standardized gas mixtures that allow us to calibrate the instrumentation used for in-situ or flask

measurements, are different between the NOAA and AGAGE networks. Gas measurements on different measurement scales, and even when using the same scales by different laboratories, are subject to uncertainties (Hall et al., 2014). For halocarbons, the difference in calibration scales has been estimated as small, but not negligible, i.e. within 2.5%, often within 1% (Rhoderick et al., 2015).

While we use the station data that have already been converted to the latest scales of the respective networks, some older

comparison data products use previous scales (like the one published in the latest ozone assessment report (WMO, 2014)). Thus, where necessary, we convert those older data to the newer scales. For 7 gases, we use scale conversion factors to convert to the SIO14 scale, specifically 1.0826 for HFC-125 (from University of Bristol scale: UB98), 1.1226 for HFC-227ea (from Empa-2005), 1.1970 for HFC-236fa (from Empa-2009-p), and 1.1909 for HFC-245fa (from Empa-2005), 1.1079 for HFC-

365-mfc (from Empa-2003), 1.0485 for HFC-43-10-mee (from SIO-10-p), and 0.9903 for $CH_2Cl_2$ (from UB98), with all conversion factors taken from the Appendix in WMO (2012).

Apart from those scale conversions to the latest NOAA and SIO scales mentioned above, we only make sure that the three main gases each are on a unified scaled. In the case of $CO_2$, we source all our $CO_2$ station data from the NOAA network, which means no scale conversion is necessary. In the case of $CH_4$, we account for different calibration scales by converting AGAGE $CH_4$ data (Tohuko University scale) to the NOAA scale (NOAA04) (multiplication by 1.0003). In the case of $N_2O$, both the AGAGE (SIO1998) and NOAA network calibration scales (NOAA-2006) are compatible without the need for a conversion factor (WMO, 2012). The Law Dome data used here (Etheridge et al., 1998b; Etheridge et al., 1996; MacFarling Meure et al., 2006; Rubino et al., 2013) have been updated for minor dating changes and placed on current NOAA scales (http://www.esrl.noaa.gov/gmd/ccl/index.html).

Apart from those three main gases, we do not apply further scale conversions. Thus, given that our results are based on a mixture of the AGAGE and NOAA networks, they are de facto a weighted average between the respective two standard scales (SIO and NOAA) for each gas. The effective weight in this "weighted mean" depends on the station numbers and each network's station distribution given that our assimilation method implicitly gives less weight to stations that are geographically close, i.e. in the same latitude-longitude box. This mixture of scales is different from previous studies that either applied empirical scale conversions (so that global-mean or station averages are identical) or used both scales in parallel to estimate a measurement uncertainty (WMO, 2014), for example when estimating emissions with inverse techniques. Mathematically, our approach is similar to an approach where a station-by-station scale conversion would be applied towards an intermediate scale between NOAA and AGAGE. However, for some applications, this approach is clearly a limitation as it hides the uncertainty and would for example warrant a new data assimilation if one network updates its scales (section 6 'Limitations'). The reason this "weighted mean" approach is chosen in the context of this study is that we intend to reconstruct a single concentration history making use of the station data from both major measurement networks without giving preference to one or the other measurement scale. Given that different scales between the two major networks result in differences that are generally less than 2% (and are often for radiatively less important substances), this "middle of the road" approach seems justified given the other uncertainties in climate model forcings (vertical distributions, radiative forcing routines, other radiative forcings such as aerosols). Any conversion to a single scale would ease comparisons, but would not be able to address the inherent measurement uncertainty, and might even face a stronger bias (if the two scales SIO and NOAA are equally plausible representations of the "truth") (section 6 'Limitations').

However, in regard to the time of the day, month or year, we do not apply interpolation or adjustment techniques other than a simple monthly binning of all available data (see 2.1.2). The spatial and temporal coverages of the raw data used in this study are depicted in Fig. 2, Fig. 3, and Fig. 4 for $CO_2$, $CH_4$ and $N_2O$ data, respectively.

### 2.1.2   Step 2-4: Binning and spatial interpolation

We employ a simple monthly mean binning of all available data, separately averaged for each station. Stations with more than one measurement program, e.g. with flask and in-situ programs, are treated as distinct stations. Thus, the monthly average of

an in-situ data series with 1000 measurement points gets the same weight as the monthly average from a flask measurement program with few observations. In each latitudinal / longitudinal box, all available monthly mean station data are averaged, with the mean being assigned to the grid box centre before employing a 2-dimensional spatial interpolation to extend available data points to longitudinal and latitudinal grid points that do not have observed data for any particular month. Our method provides equal weight to each station within a longitude-latitude box, no matter whether the station reports a few flask measurement samples or sub-hourly *in situ* instrument readings in each month. The chosen assimilation grid has 72 boxes with 12 equal-latitude bands of 15 degrees and 6 longitudinal bands of 30 degree. Following the temporal monthly binning and subsequent spatial linear interpolation, we average all data across the longitudes to obtain 12 latitudinally resolved monthly time series of surface concentrations.

### 2.1.3    Step 5: Global mean mole fractions

The annual-global mean concentration $\overline{C_{global}}(y)$ is derived as the area-weighted arithmetic mean of the binned latitudinal data (grey small "5" in Fig. 1). In addition to the annual global mean, a time series of monthly values is derived as a smooth spline interpolation between the annual data points, with the constraint of being mean-preserving, i.e. that the average of the 12 monthly values is again the global annual average value initially-derived. Thus, the trend in the mole fraction data is reflected in the global-mean time series from month to month.

### 2.1.4    Step 6: Latitudinal Gradient

The annual-mean latitudinal gradients are derived as first and second empirical orthogonal function (EOFs) from the annual-average residuals per latitude after subtracting the global annual mean (step 6 in Fig. 1). Let $\mathbf{G}$ be the $n \times m$ matrix of $n$ years with observations, and $m$ latitudinal boxes, then $\mathbf{G}$ can be decomposed into its EOFs and scores by calculating the singular value decomposition of $\mathbf{G} = \mathbf{UDV}^T$, where $\mathbf{U}$ and $\mathbf{V}$ are orthogonal matrices in $\mathbb{R}_n$ and $\mathbb{R}_m$, respectively, and $\mathbf{D}$ is the $n \times m$ matrix with non-zero elements only on the diagonal. EOF$_i$ is the $i^{th}$ column of $\mathbf{V}$, and the score $S_i(y)$ of EOF$_i$ in year y is given as the (*y,i*) entry of the $\mathbf{UD}$ matrix. In other words, the EOFs are the eigenvectors of the Gram matrix *1/m×(*$\mathbf{G}$ *′*$\mathbf{G}$*)* and the scores are the projections of the observations $\mathbf{G}$ onto the EOFs.

Those EOF scores are regressed with suitable predictors or extended as constants. Thus, the term $\hat{L}(y)$ is the optimal low rank approximation of the latitudinal deviations from the global mean in year *y*. It is composed of the leading EOFs of latitudinal annual-mean variation multiplied with the observed or regressed scores *S* of that year *y*.

$$\hat{L}(y) = \sum_{i=1}^{imax} EOF_i \, S_i(y) \tag{2}$$

with *imax* being 1 or 2 if only the leading or the two leading EOFs are taken into account, respectively.

### 2.1.5    Step 7-10: Seasonality

The seasonality fulfils the condition that the sum of seasonal variations at each latitude is zero over the year, i.e.

$$\sum_{m=1}^{12} \hat{S}_{l,m} = 0 \tag{3}$$

This seasonality $\hat{S}_{l,m}(t)$ at time t is calculated for most gases as the relative seasonality $\frac{d\hat{S}_{l,m}}{dC_{\text{global}}}$, i.e. the monthly deviation in mole fraction divided by the global-mean mole fraction, multiplied by the global-mean mole fraction at time t (step 7 and 10 in Fig. 1).

An exception is the case of $CO_2$ (step 8 and 9 in Fig. 1). In this case, the seasonality pattern over the observational period is held fixed as absolute mole fractions, i.e. not relative to the global mean. However, the residuals between this fixed seasonality and the seasonality, which is derived from the observations by subtracting the latitudinal averages, are used for a singular value decomposition. Let $R_{l,m}(t)$ be the residuals at latitude $l$ and month $m$ at time $t$, the optimal lower rank representation of this seasonal change is then given by the first EOF of the gram matrix $1/n \times \mathbf{R'R}$ with $n$ being the number of observational data points. The derived score, i.e. the projection of the residuals onto the first EOF, is regressed against a time series $P$, a composite of global-mean $CO_2$ concentration and historical observed global-mean surface air temperatures. This simplified choice is taken because previous studies identified warmer temperatures and elevated $CO_2$ mole fractions as dominant reasons for increased seasonality (Forkel et al., 2016; Graven et al., 2013; Welp et al., 2016), although anthropogenically induced cropland productivity increases are also suggested to play some role (Gray et al., 2014). Specifically, $P$ is assumed as a composite of the product and the sum of normed global-mean surface air temperature and normed $CO_2$ mole fraction deviations from pre-industrial levels. The temperature and mole fraction deviations are normalized such that the 2000-2010 deviation from the 1850-1880 base period is set to one. Thus, the regressor $P$ can be described as:

$$P(t) = \frac{\Delta T(t) * \Delta C(t)}{2} + \frac{\Delta T(t) + \Delta C(t)}{2} \qquad (4)$$

With $\Delta T$ being the temperature deviation from the 1850-1880 period, specifically

$$\Delta T(t) = \left. \left( T(t) - \sum_{t=1850}^{1880} T(t) \right) \middle/ \sum_{t=2000}^{2010} \left( T(t) - \sum_{i=1850}^{1880} T(i) \right) \right. \qquad (5)$$

And $\Delta C$ being the normed mole fraction deviation. Note that this regressor $P$ is one of multiple options that were tested and could be regarded as a plausible regressor for seasonality changes. Specifically, we tested global-mean $CO_2$ concentrations, global-mean annual average surface air temperatures and lagged averages of surface air temperatures as regressors (see Fig. 5). The R-squared values of the regressions over the 1984-2014 period are relatively similar across all regressors, around 0.8. The marked difference is that the regression with only $CO_2$ concentrations would result in a stronger reduction of seasonality around 1940-1960 and before 1900. By 1850, the reduction of summertime $CO_2$ concentrations in the zonal band around 52.5°N would be around 8.6 ppm compared to 2014 (multiply the differences of the seasonality scaling difference between 1850 and 2014, about 21, with the 0.41 ppm maximum of the EOF pattern, shown in Fig. 9 a.2). In contrast, the other regression options would limit the maximal seasonality change to about 5.7ppm, closer to the maximal seasonality change detected within the period 1984-2014, of 4.5ppm (cf. Fig. 5e). Given the uncertainty in regard to pre-1960 seasonality, we opted for the more conservative extrapolation method that implies a less significant change outside the observational period and chose the regressor with the least variability, namely our composite regressor combining temperature and $CO_2$ concentrations.

Despite the differences in the regressors, it should be noted that early $CO_2$ observations are too sparse to come to a definite conclusion in regard to which regressor is best suited – given the induced differences around 1960s and 1970s are fairly small compared to the noise in the observations (see panel f and g of Fig. 5). Furthermore, seasonality changes in the case of $CO_2$ depend on a number of factors, inter alia: complex interaction of $CO_2$ fertilization of temperate, seasonal gross primary productivity, the influence of temperature, precipitation on biomass growth and respiration, as well as directly human-induced changes in land use areas and their productivity. Therefore, this extension of the observed seasonality changes beyond the observational period based on a regression with temperatures and $CO_2$ concentrations is just that: a plausible extrapolation that needs to be refined by further research to replace this study's *ad hoc* assumption.

The measured seasonality of $CH_4$ and $N_2O$ over the observational time period is found to be closely approximated by our default assumption of a seasonality that is proportional to global mean mole fractions. For several other substances, however, seasonality has been assumed to be zero – either because the diagnosed seasonality was very small or due to a lack of observational data.

### 2.1.6    Step 11-13: Extension of latitudinal gradients and global means with ice core and firn data

Historical GHG records from ice and firn provide high-latitude estimates of atmospheric GHG mole fractions before the instrumental record from air sampling stations. We rely mainly on the Law Dome data (Etheridge et al., 1998a; Etheridge et al., 1996; MacFarling Meure et al., 2006; Rubino et al., 2013), updated for minor dating changes and placed on current NOAA scales, and, for northern hemisphere $CH_4$, Greenland NEEM ice core data (Rhodes et al., 2013). Although we did not directly use their data, we acknowledge multiple other efforts, including, but not limited to Mitchell et al. (2013), Bauska et al. (2015), Schilt et al. (2010b), Flueckiger et al. (2002), and Sowers et al. (2003) (Fig. 6). Law Dome atmospheric composition records have the advantage of a very narrow air age spread that provides measurements with high temporal resolution and mean air ages up to the 1970s, where they overlap with the beginning of atmospheric observations for many gases.

Having obtained estimates of the latitudinal gradients over the observational period and having derived approximations back in time by regressing latitudinal gradients EOF scores with emissions (step 11 in Fig. 1, Table 4), we can estimate global mean mole fractions based on the Law Dome data for both $CO_2$ and $N_2O$ (step 12 in Fig. 1). In the case of $CH_4$, the advantage is that there are northern hemispheric data available from NEEM (Greenland) (Rhodes et al., 2013) over the past 2000 years. This NEEM record hence allows an optimisation of both the EOF scores and global means at past time points to match both the Law Dome and NEEM records (step 13 in Fig. 1). Some data gaps in the NEEM record are filled by linearly interpolating the optimised EOF scores of the latitudinal gradient. With an interpolated EOF score, the global-mean mole fraction can then be directly inferred from the Law Dome record. All optimisations are performed by minimising area-weighted squared residuals. The Law Dome ice core data are smoothed with a piecewise local $3^{rd}$ degree polynomial median regression, using ad hoc expert judgement assumptions of errors and smoothing window widths specific to each gas in order to approximately reflect their long-term median evolution. In the case of $CO_2$, a random error of 2 ppm was assumed, a percentage age error (reaching a maximum of 60 years at age 2000 years before present) with a bagging of 250 ensembles, a kernel width of 120 years, minimal number of data points of 7 and maximum of 25 (panel a in Fig. 8). Likewise, $CH_4$ Law Dome data ice core data are

smoothed with a 3rd degree polynomial median regression with a maximum kernel width of 100 years, 4 minimal data points (a constraint that overwrites the maximum kernel width, if necessary) and 10 maximal data points. As for $CO_2$, 250 ensembles were averaged, after adding noise of 3ppb, and an age uncertainty of 50 years per 2000 years. For $N_2O$, a kernel width of 300 years was chosen with a minimum number of 7 and maximum number of 15 data points to be included in the piecewise 3rd

degree polynomial regression. As for $CO_2$ and $CH_4$, 250 ensembles were used for bagging after injecting a random noise of 3 ppb and an age-dependent x-axis uncertainty of 90 years per 2000 years. The higher age uncertainty for $N_2O$ in comparison to $CO_2$ and $CH_4$ was chosen to account for the larger age gaps in the $N_2O$ Law Dome data that required a stronger horizontal smoothing for the median regression to converge. For $CO_2$, the slightly higher age uncertainty in comparison to $CH_4$ was chosen so that the smoothed record displays a comparable time evolution to the WAIS $CO_2$ record (Fig. 6).

The Greenland NEEM ice core $CH_4$ data (Rhodes et al., 2013) exhibits some outliers in the recent period (Fig. 6d) due to incursion of modern air into still-open pores of shallow ice. Spikes in deeper ice are likely due to impurities. Hence we use the 5-year smoothed data provided by Rhodes et al. (2013) as a proxy for Greenland atmospheric background mole fractions (open red circles in Fig. 6b and d). We used the NEEM $CH_4$ firn measurements from Buizert et al. (2012) (2008 campaign), with effective ages from Ghosh et al. (2015) based on the iterative dating method of Trudinger et al. (2002b), corrected for the effect

of gravity (as applied in other firn data) and put onto the NOAA 2006 primary calibration scale.

### 2.1.7    Step 14: Extension of latitudinal gradients and global means with literature data

For several gases, including ozone depleting substances, halons and PFCs, the available AGAGE and NOAA station data is sparse spatially. Before the start of systematic instrumental measurements, we use literature studies which make use of various data sources, such as air sample archives or firn records (step 14 in Fig. 1). Specifically, if a global mean is provided, we use

that global mean in conjunction with our derived and regressed latitudinal gradients. In the case of hemispheric data-points, we adapt the latitudinal gradient to match the literature studies, as in the case of $C_4F_{10}$, $C_5F_{12}$, $C_6F_{14}$, $C_7F_{16}$ or $C_8F_{18}$, where we based both the global mean and latitudinal gradients on the data of Ivy et al. (2012). Other key studies used were Velders and Daniel (2014), the data underlying the WMO Ozone Assessment Report (2014), Arnold et al. (2013; 2014), Trudinger et al. (2004), Mühle et al. (2010; 2009), Montzka et al. (2011), updated time series by Montzka et al. (1999) (updated at:

ftp://ftp.cmdl.noaa.gov/hats/Total_Cl_Br/), the recent study by Vollmer et al. (2016) in regard to Halons and by Trudinger et al (2016a) in regard to PFCs, and others (Arnold et al., 2013; Arnold et al., 2014; Butler et al., 1999; Ivy et al., 2012; Montzka et al., 2015; Mühle et al., 2010; Oram et al., 2012; Trudinger et al., 2016a; Velders and Daniel, 2014; Vollmer et al., 2016; Worton et al., 2007), as indicated in the gas-specific factsheet figures (Appendix A Fig. 20 to Fig. 59 with references provided in Table 12). In the case of $N_2O$ and $CH_2Cl_2$ we assumed a constant latitudinal gradient back in time before ongoing

measurement records are available (Fig. 12, and Appendix A Fig. 26, respectively).

### 2.1.8    Step 15: Extrapolation

For some limited data segments, an extrapolation has been used. Either a piecewise smoothing spline to converge concentrations back to zero or pre-industrial background concentrations, e.g. before the WMO (2014) or Velders and Daniel (2014) data started in 1978 or 1951, respectively. The three radiatively most important fluorinated species CFC-12, CFC-11

and HCFC-22 (Table 5) follow the global mean concentrations provided by Velders and Daniel (2014), in conjunction with separately derived latitudinal gradients and seasonality. Furthermore, a linear extrapolation was applied when there were not sufficient 2014 data available.

### 2.1.9   Step 16-19: Creating the composite surface concentration field

Following equation (1), the surface mole fraction fields over the full-time span are now synthesized from the lower rank representations of seasonality, latitudinal gradient and the smooth monthly representation of global mean mole fractions. As per the original station data aggregation, the latitudinal resolution is 15 degrees and the time resolution is monthly. In order to assist with application in climate models with finer grids, we also produced a finer grid interpolation to 0.5 degree latitudinal resolution using a mean-preserving smoothing. This finer grid interpolation should not be mistaken as a mole fraction field containing actual information at 0.5-degree level. The purpose is simply to offer a smooth interpolation that avoids errors that will arise from, e.g., a linear interpolation between the provided 15 degree latitude points, as the mean across those (linearly) interpolated values, would not match the original field. The mean-preserving smoothing code is available from the authors on request. Finally, the 15 degree fields are aggregated into global, northern and southern hemisphere monthly and annual means.

### 2.1.10   Step 20: Aggregating equivalent mole fractions

It is computationally inefficient to model the radiative effect of 43 individual GHGs in today's Earth system models or general circulation models. Climate models use different pathways to approximate the radiative effects of the full set of GHGs. As one strategy, only the radiatively-major GHGs are explicitly modelled, such as $CO_2$, $CH_4$, $N_2O$, CFC-12, CFC-11, which together cause 94.5% of GHG warming effect (measured in radiative forcing) in 2014 relative to 1750 and 98% of the total radiative effect compared to the full set of 43 GHGs (Table 5). Alternatively, radiatively-minor GHGs can be approximated by equivalent GHG concentrations of a marker gas. In this way, the radiative effect of the group of gases is expressed by a single gas mole fraction. One definitional issue is whether the radiative forcing since 1750, i.e. only the changes since pre-industrial levels, are expressed by the marker gas (here called 'marginal equivalence' $C_{eq,i}$). In this case, the marker gas' concentrations $C_{eq,i}$ are sought that would exert the same aggregate radiative forcing since 1750 as the group of summarized gases. Thus, let $Cj(t)$ be the concentration (mole fraction in dry air) of a GHG and $C_{0,j}$ the pre-industrial level, i.e. in year 1750 that is routinely used as base year for radiative forcing (IPCC, 2013). A marker equivalence mole fraction by gas $C_{eq,1}$ for group $Cj$ with $j = 1,...n$ is then given by:

$$C_{\mathrm{eq},i}(t) = R_i^{-1}\left( R_i(C_{0,i}) + \sum_{j=1}^{n}\left( R_j\left(C_j(t)\right) - R_j\left(C_{0,j}\right)\right)\right) \qquad (6)$$

With $R_j(C)$ being the radiative forcing function relating concentrations $C(t)$ at time $t$ to radiative forcing for gas $j$, in the linear case $R_j(C) = C * E_j$ with $E_j$ being the radiative efficiency. $R_i^{-1}(F)$ is the inverse of this radiative forcing function, so that the concentration $C$ that corresponds to a forcing $F$ is given by $C = R_i^{-1}(F)$.

In contrast, equivalent concentrations can express the radiative effects of the summarized GHGs including their natural background levels (here called 'full equivalence' $C'_{\mathrm{eq},i}$).

$$C'_{eq,i}(t) = R_i^{-1}\left(\sum_{j=1}^{n} R_j\left(C_j(t)\right)\right) \qquad (7)$$

While the former definition 'marginal equivalence' is often used to express the total GHG forcing in $CO_2$ equivalence concentrations, the latter 'full equivalence' is the more appropriate quantity to drive climate models, given that natural background concentrations of not-explicitly considered gases should nevertheless exert a radiative effect even in a pre-industrial control, even though that radiative effect does not count under a radiative forcing definition that looks at changes from 1750.

In the linear case, in which case radiative forcing is proportional to the gas' concentrations, equation (7) can be written as:

$$C'_{eq,i}(t) = \frac{\sum_{j=1}^{n} r_j^{eff} * C_j(t)}{r_i^{eff}} \qquad (8)$$

With $r_i^{eff}$ being the radiative efficiency of gas $i$ in W/m$^2$ per ppb.

Thus, climate models have the option to reduce the complexity of 43 GHGs and the associated computational burden by reducing the number of GHGs that are taken into account. With the top 5 GHGs, $CO_2$, $CH_4$, $N_2O$, CFC-11 and CFC-12, climate models would capture 98% of the total radiative effect in year 2014 and 94.5% of the radiative forcing since 1750, i.e. the change of the radiative effect between 1750 and 2014 (see Table 5). As an alternative, there is the option to use equivalent concentrations. For two such equivalence options, this study provides input data sets. Modelling groups should indicate the combination of files they employed:

a) **Option 1**: Climate models implement a subset of 43 GHGs.

b) **Option 2**: Climate models implement the four most important GHGs with their actual mole fractions explicitly, namely $CO_2$, $CH_4$, $N_2O$ and CFC-12 and summarize the effect of all other 39 gases in an equivalence concentration of CFC-11. For this purpose, we provide CFC-11-eq concentrations ('full equivalence').

c) **Option 3**: Like option 2, but with a different split up of gases other than $CO_2$, $CH_4$ and $N_2O$. Climate models implement the three most important GHGs with their actual mole fractions explicitly, namely $CO_2$, $CH_4$, and $N_2O$ and summarize the radiative effect of the ozone depleting substances in a CFC-12-eq concentration and the radiative effect of all other fluorinated gases as a HFC-134a-eq concentration. For this purpose, we provide CFC-12-eq and HFC-134a-eq concentrations ('full equivalence').

## 2.2 Data analysis for comparison with CMIP5 ESMs

We compare our results to various other datasets (see section 5), inter alia to $CO_2$ fields from CMIP5 Earth System Models (ESMs) (section 5.3). Here, we briefly describe the analytical steps that we performed for retrieving the ESM data. We analyse ten CMIP5 ESMs that have an interactive carbon cycle model and provided the mole fraction of carbon dioxide in the air as function of different pressure surfaces for the *esmhistorical* experiment. We diagnosed those *esmhistorical* experiments in terms of the simulated $CO_2$ mole fraction at surface pressure (1bar = 100000 Pa) for 10 CMIP5 ESMs, for which data were available: (1) BNU-ESM (BNU, China), (2) CanESM2 (CCCMA, Canada), (3) CESM1-BGC (NSF-DOE-NCAR, USA), (4) FIO-ESM (FIO, China), (5) GFDL-ESM2G (NOAA GFDL, USA), (6) GFDL-ESM2M (NOAA GFDL), (7) MIROC-ESM

(MIROC, Japan), (8) MPI-ESM-LR (MPI, Germany), (9) MRI-ESM1 (MRI, Japan), (10) NorESM1-ME (NCC, Norway). For the models CanESM2, MIROC-ESM and MPI-ESM-LR more than one realization is available. We calculated an ensemble mean based on the all available ensemble members. The climatological seasonal cycle (Fig. 62, Fig. 63) is calculated relative to the linear trend of the corresponding 30-year periods.

# 3    Results

Here, we describe the historical concentrations of the main GHGs and provide a fact sheet for all 43 individual gases.

## 3.1    Carbon Dioxide

The 800,000 years EPICA composite ice-core record (Ahn and Brook, 2014; Bereiter et al., 2015; Bereiter et al., 2012; Lüthi
et al., 2008; MacFarling Meure et al., 2006; Marcott et al., 2014; Monnin et al., 2004; Petit et al., 1999; Schneider et al., 2013; Siegenthaler et al., 2005) (available at ftp://ftp.ncdc.noaa.gov/pub/data/paleo/icecore/antarctica/antarctica2015co2.xls) indicates that $CO_2$ concentrations have fluctuated between 170 and 270ppm (Fig. 6a) in conjunction with glacial- and inter-glacial temperature variations. From the year 0 to 1000, our piecewise fit of the $3^{rd}$ degree polynomial of Law Dome ice core data allows a derivation of global mean concentrations of around 278.6 ppm (min-max range of 277.0 to 280.2 ppm).

Our smoothed Law Dome results do not reflect the higher frequency variations suggested by the individual data points (Etheridge et al., 1996; MacFarling Meure et al., 2006; Rubino et al., 2013) and are comparable to the frequency spectrum that would result from a smoothed median estimate of WAIS data by Bauska et al. (2015) and Ahn et al. (2012). The WAIS record is generally 3-6 ppm higher than the Law Dome record and is also higher than South Pole and EPICA DML ice cores (Ahn et al., 2012) and the Dronning Maud Land ice (Rubino et al., 2016). The cause for this difference is not yet known (Fig. 6b). The
differences between the WAIS and the Law Dome record persist in 1850 to 1890 with subsequent data points being more aligned with each other (Fig. 6c). CMIP6 modelling groups might want to test an alternative data set that captures those higher frequency characteristics of the Law Dome record (data can be generated by the authors on request). In that higher frequency data set, the minimum of global mean $CO_2$ concentrations is close to 270 ppm around the year 1610. The smoother version provided for CMIP6 has its minimum in year 1666 at 276.27 ppm (Fig. 6b). The reason for the 1610 dip in the Law Dome
record and why this does not show in the WAIS record is not yet fully understood. The current understanding of how the age kernel (to estimate the distribution of age of air at the time of bubble trapping) is different for the two sites cannot yet explain this difference in concentrations around 1610.

In regard to the latitudinal gradient, we explored various options. If we regress the scores of the first EOF of the latitudinal gradient (Fig. 9d) against global fossil $CO_2$ emissions, the pre-industrial latitudinal minimum of surface $CO_2$ concentrations
would be estimated in the mid-northern latitudes (approximately 1.8 ppm below the global-mean), where the maximum was observed in recent decades (e.g. 4.8 ppm above the global-mean in 2010). Previously a similar regression approach between concentrations and $CO_2$ emissions was used by Keeling et al. (2011) to separate the anthropogenic from the natural component in the concentration difference between Mauna Loa and the South Pole. This approach is not perfect due to the covariance of regional fossil fuel emissions with natural sinks over the same period, different patterns of anthropogenic land-use emissions,
and a latitudinal gradient component that merely results from seasonal $CO_2$ exchange (e.g. Denning et al., 1995). However, it can provide a first indication of the influence of anthropogenic emissions on the latitudinal gradient. Furthermore, this approach would result in an approximately 0.4 ppm higher pre-industrial Antarctic $CO_2$ concentration compared to the global mean, coinciding with the assumption taken by Rubino et al. (2013).

However, given the evidence by CMIP5 ESM models of a slight tropical local maximum (Fig. 9b) and large uncertainties regarding pre-industrial sinks and source distributions and hence the latitudinal gradients of $CO_2$, we assumed a zero pre-industrial latitudinal gradient. Thus we performed a zero-intercept regression of the scores of the latitudinal gradient EOF1 with global fossil $CO_2$ emissions and converging the score of the second EOF towards zero, resulting in a flat latitudinal gradient in pre-industrial times.

The second EOF of the latitudinal gradient of $CO_2$ does not exhibit the same linearity over time as the first EOF, and the reasons are currently unknown. Potential candidates for this pronounced spike (Fig. 9c) of mid-northern latitude concentrations in the case of $CO_2$ are a shift in station sampling locations with more 'polluted' land-station coming on line after 1995, the 'rectifier' effect due to an enhanced seasonal cycle (Denning et al., 1995), and the rise of Chinese emissions (the onset around year 2003 of the recent surge in Chinese $CO_2$ emissions is approximately coinciding with the respective EOF score becoming strongly positive (Fig. 9d) (Francey et al., 2013)). One suggested explanation for this 2010 change in north-south gradients are changes in interhemispheric transport (Francey and Frederiksen, 2016). Recently, i.e. after 2010, this spike in mid-latitude northern concentrations seemed to somewhat subside again according to our analysis (see scores for EOF1 and EOF2 in Fig. 9d). Future research could address the underlying reasons of this change in latitudinal patterns and a physical explanation will allow a more appropriate backward extension in time.

The diagnosed average seasonality of atmospheric $CO_2$ concentrations over the observational period reflects the standard carbon cycle pattern of strong $CO_2$ uptake in spring and release in autumn due to photosynthesis and heterotrophic respiration in the northern hemispheres ecosystems. Our EOF analysis of the residuals shows (Fig. 9a.2 and Fig. 9a.3) that the seasonality has increased over recent decades in line with previous studies, which explore the link to increased ecosystem productivity (Forkel et al., 2016; Graven et al., 2013; Welp et al., 2016) and increased cropland productivity (Gray et al., 2014). Specifically, our analysis shows a slight shift of the seasonality to earlier months in the year, i.e. the negative and positive deviations of the EOF pattern are shifted by a month compared to the average seasonality (cf. Fig. 9a.1 and Fig. 9a.2). The strongest change in $CO_2$ seasonality is derived for the latitudinal bins centred around 37.5 to 67.5 degree north bins with a maximum strengthening of negative deviations in the 52.5 degree north latitudinal band in July by around 4 ppm over 1984 to 2013 (4 ppm results from multiplying the EOF pattern value in July in the 52.5-degree bin with the EOF score difference of around 10, see Fig. 9a.2 and a.3). The maximum strengthening of the seasonal cycle happens in July in the 52.5-degree latitudinal band, however the maximum seasonal cycle deviation is still observed slightly later in August and extends also slightly more towards the northern latitudes (Fig. 9a.1).

In 1850, the start of the historical CMIP6 simulations, the estimated global-mean $CO_2$ concentration is 284.32ppm, rising to 295.67 ppm in 1900, 312.82 ppm in 1950, 369.12 ppm in year 2000 up to 397.55 ppm in 2014 (Table 6). Here and elsewhere (e.g. Table 6) we provide more significant figures than customary - not to claim a 5-digit precision of the data, but to avoid unnecessary (even if small) step changes in concentrations between the pre-industrial run and the historical and other runs. Our methodology does not include a formal uncertainty analysis. As a minimum uncertainty for the 1850's pre-industrial

values, we refer to the 1.2ppm variability stated by Etheridge et al. (1996), also used in Rubino et al. (2013) and Trudinger et al. (2002a) as minimum uncertainty for that period.

Global-mean surface $CO_2$ concentration growth slightly flattens off in the 1930s and a stronger flattening occurs during World War II until the 1950s (Bastos et al., 2016). The increase from 1970 onwards has a slightly positive curvature (accelerating trend) with small deviations around 1973, 1981 and the temporary flattening of $CO_2$ concentrations after the 1991 Pinatubo eruption (Jones and Cox, 2001; Peylin et al., 2005) (Fig. 9 and Fig. 10).

### 3.2 Methane

Over the 800,000 years before year 0, atmospheric $CH_4$ concentrations varied between 348.7 ppb and 728.4 ppb according to the EPICA ice core composite (Barbante et al., 2006; Capron et al., 2010; Loulergue et al., 2008) (Fig. 6c and Fig. 11). The Law Dome record (Etheridge et al., 1998a; MacFarling Meure et al., 2006) indicates an onset of increasing concentrations around the year 1720 (Fig. 6d, and Fig. 11). From year 1850 with slightly higher than 800 ppb concentrations, a slight rise is observed until the 1950s, when $CH_4$ concentrations markedly increase first in the latter half of the 1950s, then again from 1965 onwards. The Greenland firn and ice core data (Rhodes et al., 2013) are more difficult to interpret because part of the record is affected by high frequency ice core $CH_4$ signals, possibly of non-atmospheric origin. $CH_4$ spikes are accompanied by elevated concentrations of black carbon, ammonium and nitrate, suggesting that biological *in situ* production may be responsible – particularly in the later years of the record since 1940. Taking here the 5-yearly average measurement values with outliers removed (Rhodes et al., 2013) that approximate the lower bounds of the raw data points until 1942, we can then infer global gradients back in time and derive an estimate of global-mean concentrations. These global-mean concentrations are estimated to be around 30 ppb higher than the Law Dome record by 1850, with the difference growing to 45 ppb by 1940s, increasing further from there (Fig. 6d). This approximately matches the findings by Mitchell et al. (2013) of interpolar differences between about 35 and 45 ppb between 800 BC and 1700 AD.

Our analysis of $CH_4$ concentrations in the recent decades is based on a large number of stations (Table 3 and Fig. 11f). While the annual increase of global $CH_4$ concentrations slowed over the 1980s and markedly after 1992 towards stabilized concentrations between 1999 to 2005, $CH_4$ increased again after 2006 at about 5.4 ppb/yr (Fig. 11f) (Nisbet et al., 2016; Nisbet et al., 2014).

We retrieve a recent seasonal cycle of $CH_4$ that is similar in the spatial-temporal seasonality pattern as that of $CO_2$ (Fig. 11a). Each hemisphere exhibits its lowest $CH_4$ concentrations just after the summer solstice, up to 1.6% or 28 ppb lower than the global mean in the case of the high-latitude northern summer (Fig. 11a). Quantifying the underlying reasons is beyond the scope of this study, although the seasonally varying atmospheric sink by OH oxidization is likely the main contributor to that seasonal pattern – in combination with seasonally varying natural and anthropogenic sources.

The latitudinal annual-mean gradient of $CH_4$ concentrations is separated into its first two EOFs, with the first EOF being a continuous north-to-south gradient of about 90 ppb in the recent observational period (combination of EOF and its score, see Fig. 11c and d). The second EOF is a distinct mid-northern latitude local maximum with a high-latitude low, showing a slight but marked rise in 2008 within the 1985 to 2014 observational data window. Quantifying the reasons for this hump are again

beyond the scope of this study, with the possibility of a shift in locations of sampling stations or coal-seam gas-fracking related fugitive emissions being possible contributors. While we optimize the first EOF, the general north-south gradient to match the Greenland data and Antarctic Law Dome data in the past, we keep the second EOF of the latitudinal gradient constant at its 1985 value.

As a result of the constant extrapolation of the second EOF, and the optimization of the first EOF's score (Fig. 11d), we yield a total annual-mean meridional gradient for the last decades that features around 80 ppb higher surface $CH_4$ concentrations in mid-to-high northern latitudes compared to the global mean and around 60 ppb lower $CH_4$ concentrations at the high southern latitudes (Fig. 11b). In pre-industrial times, our approach of regressing the score of EOF1 with global emissions (Gütschow et al., 2016) suggests this gradient to be smaller, with only approximately 20-30 ppb higher northern and 20 ppb lower southern

latitude surface concentrations (Fig. 11b). These mean interpolar differences and their variations have earlier been quantified by Etheridge et al. (1998a) and Mitchell et al. (2013), yielding similar results (between 30 to 60 ppb) compared to our 40 to 50 ppb estimate.

### 3.3   Nitrous Oxide

$N_2O$ concentrations from ice cores dating back 800,000 years (Fluckiger et al., 2002; Schilt et al., 2010b) varied approximately

between 200 ppb and 300 ppb, with most recent glacial concentration minima of 180 ppb around 23 thousand years ago (Sowers et al., 2003) (Fig. 6a). The ice core record over the last 2000 years indicates marked difference between the Law Dome and GISPII record (Sowers et al., 2003), with the latter being up to 10 ppb lower. Here, as with $CH_4$, we use again a median quantile piecewise polynomial regression on the Law Dome record, assuming constant $N_2O$ concentrations between year 0 and the first Law Dome data point in year 154. In contrast to $CH_4$, there is not a monotonic increase of concentrations,

but rather an initial slight decrease until year 630 down to a minimum concentration of 265 ppb in our smoothed time series with a subsequent slow increase until the 9th century AD, then a slight decrease until 1650 in the smoothed global-mean mole fraction. A temporary local maximum indicated by individual Law Dome data in the 15th century is not resolved by our smoothing, and a similar spike in the 17th century is only just reflected (Fig. 6f). Several data points indicate a small decrease after a 1750 maximum with a minimum in 1850 of around 273.02 ppb. This maximum around 1750 and subsequent minimum

around 1800-1850 is also apparent in the H15 ice core record by Machida (1995) (we scale-corrected the Machida data downwards by 1 ppb as in Battle et al. (1996)) (Fig. 6b). After 1850, $N_2O$ concentrations increased markedly, reaching 1900, 1950, 2000 and 2014 values of 279.5, 289.7, 315.8 and 327.0 ppb, respectively (Table 6). Comparing the different firn and ice records, the 1920 – 1940 period seems particularly uncertain with some high measurements close to and beyond 290 ppb from both Law Dome and H15, while some of the Law Dome data is still at levels around 285 ppb or even 280 ppb in the case of

H15 (Fig. 6e). The South Pole firn data (Battle et al., 1996) suggest lower $N_2O$ concentrations in the 1920s and around 1960 – compared to both the smoothed Law Dome data (thin dashed line in Fig. 6e) and consequently our even higher global-mean estimate. Although the Ishijima estimate (Ishijima et al., 2007) (their Figure 6a) around 1952 is almost identical to our global-mean, their modelling study suggests slightly lower values around 1960 before being closely matching again from 1970

onwards. The Law Dome firn record (Park et al., 2012) suggests slightly higher $N_2O$ concentrations for the high southern latitudes compared to our global-mean Fig. 6e).

The variability of our derived $N_2O$ global-mean concentrations, in particular the steps in 1920s and 1940s, reflect the smoothing algorithm choices to noisy data (section 2.1.6), but should not be over-interpreted. Our algorithm does for example not include information on the lifetime of $N_2O$ that would guard against inferring too rapid declines of $N_2O$ mole fractions and mole fraction growth rates. The fit of the smoothing algorithm was chosen to balance the resolution of smaller scale features with the uncertainty present in the input data sources for the full-time horizon from year 0 to year 2014. Given overall uncertainties (Fig. 6e), a smoother representation between 1900 and 1980 seems equally justified.

Compared to $CH_4$ and $CO_2$, the seasonality and latitudinal gradient of $N_2O$ are relatively small. The $N_2O$ seasonality is only 0.1% of global mole fractions and is almost symmetric and seasonally time-synchronized between the northern and southern hemispheres with minima in the southern hemisphere late autumn and northern hemisphere summer/autumn (Fig. 12a). The seasonality is currently of the same size as the underlying trend, leading to global mean $N_2O$ mole fractions increasing in the latter months of any year with a subsequent flattening in the first half of any calendar year (e.g. Fig. 12h). Given a counter-intuitive slight decrease of the north-south gradient with flat or slightly increasing global $N_2O$ emissions (Gütschow et al., 2016) in recent years (Fig. 12d), we assumed constant scores for the latitudinal gradient EOFs for times before 1996 (Fig. 12d). Due to measurement fluctuations in the first years when systematic measurements started in 1978 that are larger compared to the recent period, we chose to interpolate $N_2O$ global-mean mole fractions over 1966 to 1987. For the period between 1978 and 1987, this interpolation is closely aligned with a smooth representation of the atmospheric measurements (Fig. 12f, cf. ALE/GAGE/AGAGE data as shown at http://agage.eas.gatech.edu/data_archive/data_figures/gcmd_month/n2o_monS5.pdf).

## 3.4 Ozone Depleting Substances and other chlorinated substances

Ozone depleting substances (ODSs), i.e. the substances destroying ozone and being controlled under the Montreal Protocol, also have a large warming effect (Velders et al., 2007; Velders et al., 2009). In particular CFC-12 and CFC-11 are important GHGs, as well as the replacement substance HCFC-22, which, unlike CFCs, continues to increase in the atmosphere, albeit at a declining rate. The radiative forcing of CFC-12 alone since 1750 is equivalent to that of $N_2O$, which is usually considered the third most important GHG after $CO_2$ and $CH_4$ (Table 5). The impact of ODSs on climate is somewhat complicated by their destruction of stratospheric ozone, which induces dynamical effects on circulation patterns, and has a net cooling effect on the global climate. The latest estimates suggest that this cooling might offset roughly two-thirds of the warming of the entire class of ODSs (Shindell et al., 2013). Note that we consider here also methylene chloride and methyl chloride, although these chlorinated substances are not controlled by the Montreal Protocol and hence often not termed ozone depleting substances (WMO, 2014).

The most abundant ozone depleting substances in the atmosphere (in 2014) were CFC-12 (520.6 ppt), CFC-11 (233.1 ppt) and HCFC-22 (229.5 ppt), with their mole fractions being about six orders of magnitude lower than currently measured for $CO_2$ (Table 7). In addition, methyl chloride $CH_3Cl$ has a high mole fraction (539.54 ppt), although is not considered an ODS here

as it is not controlled by the Montreal Protocol. Out of the 17 considered chlorinated and ozone depleting substances, only 6 have currently increasing concentrations. Those are the three HCFCs, of which the increase in HCFC-22 alone has offset the reducing radiative forcing of all other ODSs over the past decade (Fig. 8m). The other three substances that are still increasing are Halon-1301, methylene chloride ($CH_2Cl_2$) and chloroform ($CHCl_3$). Chloroform had been decreasing in the 1990s and

stabilized in the 2000s, but again recently showed an increase (Fig. 30).

Four of the considered chlorinated and ozone depleting substances are assumed to have natural emissions and hence above-zero pre-industrial concentrations. We estimate those pre-industrial natural background concentration by a simple budget equation under the assumption of a constant lifetime (IPCC, 2013) of 1 year for $CH_3Cl$ and 0.8 years for $CH_3Br$ – minimizing the error term when taking into account anthropogenic emission and atmospheric concentration estimates over 1950 to 1990

by Velders and Daniel (2014). Specifically, methyl chloride ($CH_3Cl$) is assumed to have pre-industrial global-mean concentrations of 457 ppt, and methyl bromide ($CH_3Br$) of 5.3 ppt. Chloroform ($CHCl_3$) is assumed to have a pre-industrial concentration of about 6 ppt, approximately in line with findings by Worton et al. (2006) and the estimation by Aucott et al. (1999) that in 1990 $CHCl_3$ was at about 8 ppt, with 80% of emissions assumed to be of natural origin. Lastly, in the absence of other information (a good understanding of the natural vs anthropogenic source fraction or historical industrial production

records) the available firn measurements (e.g., Trudinger et al., 2004) supplying information about methylene chloride ($CH_2Cl_2$) mole fractions in the early 20[th] century are used to suggest a 6.9 ppt pre-industrial mean concentration with a strong latitudinal gradient that results in northern (southern) hemisphere average concentrations of 12.8 (1.0) ppt. The transition of concentrations of some species between the observational station data and pre-industrial levels are also uncertain. For $CH_2Cl_2$, our derivation is in line with the smooth trajectory of Trudinger et al. (2004), indicating an almost monotonic transition between

1997 values and pre-industrial concentrations (Fig. 26f). Our assimilation approach (which is based on the Walker et al. data (2000)) causes our carbon tetrachloride ($CCl_4$) reconstruction to have a near-zero pre-industrial concentration of 0.025 ppt (0.025% of its peak value of 100ppt).We note that Walker et al. (2000) suggest zero pre-industrial concentrations before 1910, although the lowest empirical evidence from firn records suggest <5 ppt (Butler et al., 1999) or 3-4ppt as measured by S. Montzka for 1863 firn air and reported in Liang et al. (2016).

The seasonal cycle of ozone depleting substances and other synthetic GHGs can be influenced by seasonally varying stratospheric-tropospheric air exchanges, interhemispheric transport, tropopause heights, emissions and, for those substances with OH-related sinks, the seasonally varying OH concentrations. For 11 out of the 17 considered ozone depleting substances we find some indication of seasonal cycles based on the analyzed station data, namely for $CCl_4$, CFC-11, CFC-12, CFC-113, $CH_2Cl_2$, $CH_3Br$, $CH_3CCl_3$, $CH_3Cl$, $CHCl_3$, Halon-1211, and HCFC-22. Our analysis indicates that HCFC-141b also shows

some signs of a seasonal cycle, although we here assumed a zero seasonal cycle due to data sparsity (see Fig. 35a). We find the strongest seasonal cycles in case of the short-lived species $CH_3Cl$, $CHCl_3$, $CH_3Br$ and $CH_2Cl_2$ with absolute maximal seasonal deviations of -11%, -12%, ±9%, -32% compared to the annual mean, respectively. For the radiatively important and longer-lived species CFC-12, CFC-11 and HCFC-22, the seasonal cycle is much smaller, with ±0.2%, ±0.4%, ±0.8%, respectively.

Similar to the seasonality, the latitudinal gradient is found to be especially pronounced for the short-lived substances. Specifically, $CH_2Cl_2$ with a lifetime of 0.4 years, $CH_3Br$ with a lifetime of 0.8 years, $CH_3CCl_3$ with a lifetime of 5 years and $CH_3Cl$ with a lifetime of approximately 1 year and $CHCl_3$ with a lifetime of 0.4 years show substantial latitudinal gradients due to spatially heterogeneous sinks and sources (lifetimes following Table 8.A.1 in IPCC WG1 AR5 (2013)). While chemicals

with predominantly anthropogenic sources normally exhibit highest mole fractions at mid to high northern latitudes, the observations for several substances with substantial natural sources exhibit highest mole fractions in the tropics or lower northern latitudes in the recent observational period (e.g. $CH_3Cl$ in Fig. 29b and c).

### 3.5    Other fluorinated GHGs

The 23 other gases in this study are the hydrofluorocarbons (HFCs), which have recently been added to the substances

controlled under the Montreal Protocol (Kigali amendment in October 2016), and those substances whose production and consumption is not controlled under the Montreal Protocol, namely perfluorocarbons (PFCs) as well as sulfur hexafluoride ($SF_6$), nitrogen trifluoride ($NF_3$), and sulfuryl fluoride ($SO_2F_2$). Except for the latter, the emissions of all these species are controlled under the Kyoto Protocol and covered by most "nationally determined contributions" (NDCs) under the Paris Agreement. However, currently the aggregated greenhouse effect of this group of synthetic GHGs is still almost a factor of 10

smaller compared to the ODSs (cf. Fig. 8g and m). In contrast to the ODSs, nearly all of these other fluorinated gas concentrations are still rising; the exception is HFC-152a, which has stopped growing since 2012 and may now be in decline, (Fig. 52f). Thus, a primary concern with these gases is the potential for substantial climate forcing in the future if uncontrolled growth continues.

The most abundant of these gases is the refrigerant HFC-134a with 2014 concentrations estimated to be 80.5 ppt, followed by

HFC-23 (26.9 ppt), HFC-125 (15.4 ppt) and HFC-143a (15.2 ppt). At the other end of the concentration spectrum, we include results from Ivy et al. (2012) for some PFCs that exhibit low concentrations of 0.13 ppt ($C_5F_{12}$ and $C_7F_{16}$) or 0.09 ppt ($C_8F_{18}$) (Table 7). The only fluorinated gas considered to have substantial natural sources and hence a pre-industrial background concentration is $CF_4$ with an assumed pre-industrial concentration of 34.05 ppt (see Fig. 45), in line with findings by Trudinger et al. (2016a) and Mühle et al. (2010).

For a number of substances, especially the PFCs with lower abundances, there were not sufficient data available to estimate the seasonality of atmospheric concentrations. We consider seasonality only for 3 of the 23 species. HFC-134a has a somewhat atypical pattern of lowest mole fractions in the spring northern hemisphere (-2.6% compared to annual mean) as other gases normally show a summer or autumn low point of concentrations. This spring minimum results from a seasonality of sources of this refrigerant (Fig. 50a), although seasonality in loss also likely plays a role (Xiang et al., 2014). Secondly, the short-lived

HFC-152a (lifetime 1.5 years) shows seasonal variations of up to ±13% while the very long lived $SF_6$ (lifetime of 3200 years) exhibits a much smaller seasonality of up to ±0.5%.

For most of the considered substances, the latitudinal gradient is rather small. Exceptions are the shorter-lived species like HFC-32, whose concentration rose relatively quickly since 2000 due to rapidly increasing northern hemispheric sources (Fig. 47b), HFC-152a, and some other shorter lived HFCs. For the three heavier PFCs with very low abundances of well below 1

ppt in 2014, namely $C_6F_{14}$, $C_7F_{16}$ and $C_8F_{18}$, we incorporated hemispheric data from Ivy et al. (2012). Before about 1990, those three gases are suggested to have reversed latitudinal gradients with higher southern hemispheric concentrations. Due to the very low mole fractions near the limit of measurement, future studies may need to confirm whether those reverse gradients existed (and if so, why). Given the negligible radiative forcing from these gases to date, this uncertainty does not affect the overall results.

## 4    The CMIP6 recommendation and data format

We present the community CMIP6 data sets of historical GHG mole fractions. In conjunction with other data, these GHG surface mole fraction data sets are to be used in the historical concentration-driven runs for the climate model inter-comparison project phase 6 (CMIP6) (Eyring et al., 2016). Depending on the specific CMIP6 experiment, different protocols and recommendations can apply. Modellers should hence also check the experiment specific descriptions (see special issue available at http://www.geosci-model-dev.net/special_issue590.html), including protocols regarding the important other forcing input datasets like aerosols, their emissions and optical properties, landuse patterns, but also short-lived GHGs like tropospheric and stratospheric ozone for models without interactive ozone chemistry.

The historical GHG concentrations of this study are specifically designed to be useful for the historical run, and the idealized runs of abrupt4x, 1pctCO2 as well as the picontrol. Also, the PMIP4 related last millennium experiment will be based on the GHG concentrations of this study (Jungclaus et al., in preparation; Kageyama et al., 2016).

Regarding the historical runs of the DECK simulations, the CMIP6 recommendation as decided by the CMIP Panel is: "In the $CO_2$-concentration-driven historical simulations, time-varying global annual mean mole fractions for $CO_2$ and other long-lived GHGs are prescribed. If a modelling center decides to represent additional spatial and seasonal variations in prescribed GHG forcings, this needs to be adequately documented." (Eyring et al., 2016).

This study provides the data for both the global annual mean mole fractions as well as the mole fraction histories that take latitudinal and seasonal variations into account (see data description further below). CMIP6 modelling groups should indicate which time and space resolution of the data version they applied. All data are freely available via the PCMDI servers (https://pcmdi.llnl.gov/search/input4mips/) as netcdf files. The data is also available via ftp servers, and multiple data formats (netcdf, csv, xls and MATLAB mat) as described at climatecollege.unimelb.edu.au/cmip6.

In terms of the spatio-temporal resolution, four files for each of the 43 GHGs and the three equivalence species CFC-12-eq, HFC-134a-eq and CFC-11-eq (section 2.1.10) are provided as:

I.    latitudinal 15-degree bins with monthly resolution (filename-code: '_15degreelatXmonth'), with monthly means for each latitudinal band provided at the centre of the box, i.e. -82.5, -67.5, … 67.5, 82.6.

II.    interpolated latitudinal half degree bins with monthly resolution (filename-code: '_0p5degreelatXmonth'), with means for each latitudinal band provided at the center of the box, i.e. -89.75, -89.25, … 89.25, 89.75. The area-weighted mean over 15-degree latitudinal bands is the same as the files under (1).

III.    Global and hemispheric means with monthly resolution (filename-code '_GMNHSHmeanXmonth').

IV.    Global and hemispheric means with annual resolution (filename-code '_GMNHSHmeanXyear')

Given that climate effects will vary depending on whether global, annual-mean or seasonally varying latitudinally-resolved surface mole fractions are prescribed, modelling groups are asked to document which data set(s) they choose.

The CMIP6 recommendation for the **picontrol** experiment are to use the 1850 GHG mole fractions with annual means as provided in Table 8 ($CO_2$ annual-mean mole fractions of 284.32 ppm, $CH_4$ mole fractions of 808.25 ppb and $N_2O$ mole fractions of 273.02 ppb). Other gases are covered, depending on the choice of the modelling group by either following Option 1, Option 2, or Option 3 described in Table 5, or an equivalently suited method that aggregates the radiative effect of the remaining 40 GHGs or a large fraction thereof.

The **abrupt4x** experiment should keep all GHG mole fractions unchanged from the picontrol run except for the $CO_2$ mole fractions, which should be increased instantaneously in year 1 (=1850) of the experiment to four times the 1850 value, namely to 1137.27 ppm (Table 10).

The **1pctCO2** experiment should also keep all GHG mole fractions unchanged from the picontrol run except for $CO_2$ mole fractions. Starting in year 1 of the experiment, $CO_2$ mole fractions should increase by 1% per annum, reaching slightly over doubled $CO_2$ mole fractions in year 70 (or 1920, if the startyear is set to 1850) with 570.56 ppm and 1,264.76 ppm in year 150 (or year 2000) (Table 9).

As with the abrupt4x and 1pctCO2 scenarios, the **historical** experiment should diverge from the picontrol run. GHGs should then follow the historical observations as derived in this study, reaching e.g. $CO_2$ mole fractions of 397.55 ppm in 2014, and $CH_4$ and $N_2O$ mole fractions of 1831.47 ppb and 326.99 ppb, respectively. Modelling groups should document which spatial and temporal resolution (see above) of the provided data they use, as the climate effect will likely be different with different resolutions.

The future concentration pathways, the so-called 'SSP-RCP' scenarios, considered under ScenarioMIP (O'Neill et al., 2016) are planned to provide the same data formats and spatio-temporal resolutions. The methodological approach to derive and adapt both seasonality and latitudinal gradients in this study was designed such that a future extrapolation will be possible.

### 4.1    The vertical dimension

The purpose of our reconstructions is to provide radiative forcing for climate models. This radiative forcing depends on the vertical as well as horizontal distribution of a gases' mole fraction. Our reconstructions describe only surface concentrations and modellers need some method for calculating the three-dimensional distribution. If the model is capable of calculating tracer transport, includes any sinks and sources in the free atmosphere and has an appropriate treatment of the boundary layer, we recommend using this study's surface reconstruction as a mole fraction lower boundary condition for a mass balance inversion. If this is not possible, we propose a simple equation to reflect the relaxation of horizontal gradients with height and the upward propagation of mole fraction changes from the surface.

In case of $CO_2$, there are no sinks in the middle and upper troposphere or stratosphere and only slight sources due to the oxidization of $CH_4$ and carbon monoxide (CO). Evidence from Earth System Models (Fig. 13) indicates an almost well-mixed tropospheric column in the tropics and little or partly reversed vertical gradient in the southern troposphere, while the annual-mean gradient in the northern hemisphere is – depending on the season – variable. The annual average vertical gradient in the northern hemisphere is decreasing in all CMIP5 ESM models analysed here (Fig. 13).

In order to enable the implementation of surface mole fractions in models that do not have an inherent transport model to capture vertical gradients, we offer here simplified parameterizations as default options. While an assumption about a well-mixed atmospheric vertical column seem a justifiable simplification, these simple vertical extensions could increase the realism, vertical heating structure and overall climatic effect. Specifically, modelling teams could use the following approximation to extend surface concentration fields (at the 1000 hPa level) towards higher tropospheric and stratospheric levels. First, a bell-shaped concentration distribution is assumed at the 100 hPa level for the higher latitude tropopause and tropical upper troposphere:

$$C(l, 100\text{hPa}, t) = \bar{C}(global, 1000\text{hPa}, t) \ldots + \left(\bar{C}(global, 1000hPa, t - 5yrs) - \bar{C}(global, 1000hPa, t)\right) * \frac{\sin(l)^2}{2}$$

(9)

With $\bar{C}(global, 1000\text{hPa}, t)$ indicating global-average, annual-average concentrations at the surface 1000hPa level at time $t$. Ideally, a smoothed mean-preserving monthly time series of these annual-average global averages is used to prevent step changes from calendar month 12 to 1. Equivalently, $\bar{C}(global, 1000\text{hPa}, t - 5yrs)$ indicates the global-average, annual-average surface mole fraction 5 years earlier. The $\frac{\sin(l)^2}{2}$ factor depends on the latitude l and results in the bell-shaped concentration curve with concentrations at the tropical 100 hPa level to be identical to the global average surface concentrations, while the polar mole fractions are effectively of a medium age (2.5 years in the case of linearly increasing concentration history). Having defined this 100 hPa concentration level, the tropospheric mole fractions at latitude l and pressure level p (with p>100 hPa) can then be assumed as a simple linear interpolation between the surface mole fraction level at latitude l and the 100 hPa level, so that:

$$C(l, p, t) = C(l, 100\text{ hPa}, t) + \left(C(l, 1000\text{ hPa}, t) - C(l, 100\text{ hPa}, t)\right) * \frac{(\text{p} - 100\text{ hPa})}{(1000\text{ hPa} - 100\text{ hPa})} \qquad (10)$$

Above 100 hPa - i.e. in the tropical upper troposphere and stratosphere, the mole fraction is a simple linear interpolation between the 100 hPa level and the top-of-the atmosphere 1 hPa level that is assumed to have a median age of air of 5 years, so that for p<100 hPa:

$$C(l, p, t) = \bar{C}(global, 1000\text{hPa}, t - 5yrs) \ldots + \left(C(l, 100\text{hPa}, t) - \bar{C}(global, 1000\text{hPa}, t - 5yrs)\right) \ldots * \frac{(\text{p} - 1\text{hPa})}{(100\text{hPa} - 1\text{hPa})}$$

(11)

With $\bar{C}(global, 1000\text{hPa}, t - 5yrs)$ being again the global-mean surface concentration (1000hPa) five years ago and $C(l, 100\text{hPa}, t)$ the latitudinally-dependent concentration at the 100hPa level.

This equation captures the general form of the vertical $CO_2$ mole fraction gradient observed in CMIP5 ESM models – with the 100 hPa being an approximate division line of the vertical $CO_2$ gradient in all CMIP5 models (see bold red line in Fig. 13). The annual-average vertical gradient in the northern hemisphere will be somewhat reducing the effect of the strong surface latitudinal gradient. The idealized shaped of the above parameterization for a hypothetical flat surface mole fraction of 100 ppm is shown in Fig. 14b. Assuming linearly increasing surface mole fractions from a south pole minimum towards a 3 ppm higher north pole maximum will – under this simplified parameterization - result in an almost zero vertical tropospheric gradient in the southern hemisphere (Fig. 14a).

For non-$CO_2$ gases, we here suggest a scheme adapted from the CESM model current parameterization – in case that models do not have their own vertical extrapolation methods. These parameterisations assumed a simplified vertically well-mixed troposphere and define a tropopause height as:

$$p_{\text{tropopause}}(l) = 250\text{hPa} - 150\text{hPa} * \cos(l)^2 \tag{12}$$

With $p_{\text{tropopause}}(l)$ being the tropopause height in hPa, depending on the latitude l. Thus, below the tropopause, the zonal mean concentrations are assumed to be well-mixed vertically, so that:

$$C(l, p, t) = C(l, 1000\text{hPa}, t) \text{ for } p > p_{\text{tropopause}}$$

The stratospheric concentration can then be modelled for $p < p_{\text{tropopause}}$ as:

$$C(l, p, t) = \bar{C}(global, 1000\text{hPa}, \bar{t} - 1yrs) * \left( \frac{p}{p_{\text{tropopause}}(l)} \right)^s \tag{13}$$

with $\bar{C}(global, 1000\text{hPa}, t)$ being the global mean and annual-mean surface mole fraction of the previous year, $p/p_{\text{tropopause}}(l)$ being the ratio of the pressure at level p and the tropopause pressure at that latitude and *s* being a gas-dependent scaling factor (Table 11).

As mentioned above, this simple vertical extrapolation option of the provided surface data is only to be regarded as a simplified fall-back option in case that there are no model-intrinsic parameterisations available or active tracer transport part of the model. While this study provides the main step from global-mean and annual-mean concentration histories towards zonally and monthly resolved ones, future research will be needed to provide more robust 4-D fields of concentrations.

## 5 Discussion

We compare our results with a number of other data products. First, a comparison with the previous CMIP5 recommendation for historical GHG concentrations is provided (5.1). Second, we analyse and compare our CMIP6 recommendations to what the Earth System Models from the previous CMIP5 intercomparison produced in terms of $CO_2$ concentration fields in the emissions-driven runs (5.3). Third, we compare our data sets to the other global-mean, hemispheric and latitudinally-resolved data sets, namely the NOAA Marine Boundary Layer product and the WDCGG time series (5.4).

### 5.1 Comparison to CMIP5 input datasets.

For the CMIP5 inter-comparison, GHG concentrations were specified for historical times until 2005, followed by RCPs and their extensions until 2300. The recommendations for GHG concentrations were global and annual mean time series (Meinshausen et al., 2011), not including a seasonal cycle or latitudinal gradient. Those historical time series were composite products of existing ice core and instrumental data annual means (see references in Meinshausen et al. 2011). Global, annual-mean $CO_2$ concentrations over 1975 to 2005 were very close (<0.7 ppm different) to our current recommendations for CMIP6. The CMIP5 time series did not show the slight maximum in $CO_2$ concentrations around 1973 (difference 1.2 ppm), and was generally lower between 1940 and 1956 at about the time of the World War II, when $CO_2$ concentrations briefly plateaued (differences between 1.0 and 2.3 ppm) (Fig. 15). While the CMIP5 historical GHGs were an ad-hoc extension to the RCP pathways, our CMIP6 recommendation advanced the integration of historical data by accounting for latitudinal gradients (ice core data in CMIP5 has not been adjusted for the latitudinal gradients) and by taking into account a large array of additional data beyond a single network average for more recent times.

Recommended global-mean $CH_4$ concentrations for CMIP5 were generally lower than derived here, up to 50 ppb around 1910 and between 25-30 ppb more recently (2000-2005). The primary reason is that the CMIP5 data did not take into account the strong latitudinal gradient of $CH_4$ concentrations. For $N_2O$ concentrations, the CMIP5 historical timeseries did not capture some higher frequency variability, which caused the CMIP6 recommendation for the picontrol 1850 global-mean concentration being lower by around 2.5 ppb, and $N_2O$ concentrations in the 1910s being higher by up to 2.3 ppb (Fig. 15).

Overall, CMIP5 and CMIP6 recommendations are relatively similar. The 1850 picontrol values at the time of CMIP5 were slightly higher for $CO_2$ and $N_2O$ (0.14% or 0.4 ppm and 0.87% or 2.4 ppb, respectively), countered to some degree by slightly lower values for $CH_4$ (2.18% or 17.3 ppb). This is equivalent to a small net change in base year radiative forcing of 0.0065 $W/m^2$, when applying linear radiative efficiencies of IPCC AR5 (Appendix 8.A in IPCC WG1 AR5).

### 5.2 Comparison to $CO_2$ station data between 1958 and 1984.

As our data synthesis used monthly station data only from 1984 onwards (except for Mauna Loa annual averages back to 1958), a comparison to available station data from before 1984 is useful to qualitatively validate the extension method applied in this study. While latitudinal gradients (or rather: their first two EOFs) and seasonality changes are extended by regression (sections 2.1.4 to 2.1.7), the $CO_2$ fields' global-mean has been optimised to match both the annual average Mauna Loa record and the Law Dome ice record, specifically our smoothed version thereof (see Table 1 and Fig. 7k). Thus, it is informative to

compare our data product to available station data from the period before 1984 both in terms of seasonality and the absolute amplitude (which is derived from the global-mean and the regressed latitudinal gradient) (see Fig. 7). We here use the Scripps $CO_2$ data series, available at http://scrippsco2.ucsd.edu/data/atmospheric_co2/sampling_stations.

In general, the comparison suggests that this study's data product matches rather closely earlier station data, thereby validating our chosen extension approach to some degree. There are two noteworthy issues arising from this comparison though. For high southern latitudes, both Law Dome as well as SPO in-situ and flask station data are available. It seems that our CMIP6 high latitude data in the southern hemisphere could be ~1ppm too low over the period 1959 to 1972 (Fig. 7l). Earlier, in 1958, and subsequently from 1973 onwards, the match is rather close between SPO station data at -90° and our latitudinal average for the -90° to -75° zonal mean. Given our data product matches the MLO record quite closely (somewhat by design, given the optimisation to match the annual-average MLO record over that time), this points to a slightly exaggerated latitudinal gradient between 1959 and 1972.

The second issue relates to a bump in the concentration series centred around 1974. In our data assimilation, this bump is a propagation of an anomaly in the MLO record over that time and seems to a lesser degree to also show up in other northern hemisphere records. However, the southern hemispheric SPO record does not (or only minimally) show this slight upwards aberration from 1972 to 1974 and subsequent slowing and stagnating growth from 1974 to 1976 (while the lower precision Law Dome data would be consistent with that MLO pattern, see Fig. 7k). To what extent this bump has been present in the southern hemisphere is unknown, although earlier studies (Bacastow, 1976) relate the increased atmospheric $CO_2$ concentrations to decreased oceanic uptake during the El Nino back then. Such a process explanation would suggest the atmospheric signal also to be present throughout large parts of the southern hemisphere, while a predominantly extra-tropical land-related respiration increase during El Nino could imply the signal to be predominantly present in the Northern hemisphere. In summary, our assimilation's hemispheric upwards anomaly around 1974 of around ~2ppm could largely be an artefact of our methodology which propagates the MLO anomaly globally under the assumption of exogenously emission-regressed latitudinal gradients.

### 5.3    Comparison to CMIP5 ESM $CO_2$ concentration fields.

Several Earth System models during CMIP5 used prescribed $CO_2$ emissions instead of $CO_2$ concentrations and derived $CO_2$ concentration fields endogenously. For the year 1875, we see that models vary greatly, with some showing reverse latitudinal gradients with higher concentrations in the south (e.g. CanESM2), almost no gradient (CESM1-BCC), a local maximum in the tropics with lower poleward concentrations (MIROC-ESM) and very heterogeneous fields with high concentrations over the tropical rainforests (NorESM1-ME) (see Fig. 60). Similarly, for 1990 (Fig. 61), the fields are dissimilar, with some models exhibiting very strong north-south gradients (MPI-ESM-LR), while others show little gradients (CanESM2), although all models indicate an increase of northern hemispheric concentrations compared to the global mean between 1875 and 1990 (Fig. 64).

Though not as strong as NorESM1-ME, most models show a slight tropical maximum in the latitudinal gradient (exceptions are CanESM2, MIROC-ESM) both during 1875 and 1990 (Fig. 65 and Fig. 66). The high-latitude southern concentration

deviations from the global-mean in the 1875 time slices have different signs across the models, with some indicating clearly lower concentrations (BNU-ESM, MPI-ESM-LR, NorESM1-ME) and others suggesting slightly positive concentrations (CanESM2, MIROC-ESM in 1875). The average of three CMIP5 ESMs with full $CO_2$ data coverage at the surface 1000 hPa level and global mean $CO_2$ mole fraction values in line with observational records (CanESM2, MPI-ESM-LR, and NorESM1-ESM) shows a latitudinal gradient for 1990 comparable to the observed one derived in this study (Fig. 9b). Thus, given that the pre-industrial latitudinal gradient is almost flat for the models with the highest skill to replicate current observations, we assumed constant mole fractions with latitude for pre-industrial times.

In general, all ESMs show climatological seasonal cycles of $CO_2$ concentrations similar to the seasonality derived in this study (Fig. 9a). The climatological 1861-1890 average concentrations across the models clearly exhibits higher seasonality in the northern hemisphere, especially above 40°N. While the seasonality in some models is weaker, especially CESM1-BCC, others show variations of up to ±10 ppm (MPI-ESM-LR). In addition, the latter model exhibits a larger southern hemisphere seasonality than other models and what we observe. As expected from our analysis of observational data, this seasonality strengthens up to 1990 across all models (cf. Fig. 63 and Fig. 62). The latitudinal spread of the northern hemisphere minimum extends southwards towards the equator in August, September and October as we observe (Fig. 9a), with the exception of the BNU-ESM model (Fig. 63), which indicates a northward propagation of the minimum summer concentration values.

Overall, the basic features of the latitudinal gradient and seasonal cycle are represented in the ESMs as seen in the observational data. However, the variation across the models is substantial. This difference of several ppm in the latitudinal gradient or seasonal cycles will lead to follow-on differences in the climate response observed in those models.

As common input for the CMIP5 concentration-driven experiments, all models were provided with the same historical global and annual mean $CO_2$ concentrations. Some models had the capability to nudge internally-generated $CO_2$ concentration fields to match the prescribed annual and global mean $CO_2$ concentrations. Nevertheless, the differences in those internally-generated fields can be substantial, as our analysis from CMIP5 shows, and different from the observations.

For future model inter-comparisons, it seems preferable that any concentration-driven runs would use the same starting point. Of course, the longer-term aspiration has to be that emission-driven ESMs reliably reproduce observational concentration patterns. For CMIP6, modelling groups are requested to document their choice of concentration input data, specifically in relation to the chosen temporal and spatial resolutions.

## 5.4 Comparison of global-means to NOAA marine boundary layer products and WDCGG

The primary observational data product with coverage across all latitudes is the marine boundary layer (MBL) or GLOBALVIEW fields (NOAA, 2013; NOAA ESRL GMD, 2014c) produced by NOAA based on the Cooperative Global Air Sampling Network (Conway et al., 1994; Dlugokencky et al., 1994b; Trolier et al., 1996) for $CO_2$, $CH_4$ and $N_2O$ (available at http://www.esrl.noaa.gov/gmd/ccgg/mbl/mbl.html, with $N_2O$ data pers. comm. Pieter Tans). The aggregation method used to produce this data set is to first fit parametric functions to the weekly data of each station, thereby providing a gap-filling method. In a next step, the procedure fits smooth weekly latitudinal distributions to the various station data points (Tans et al., 1989). These latitudinal distributions are then combined into a 2-D field of latitude versus time, comparable to this study's

data product. The time period of these NOAA MBL data products is 1979 to 2014 for $CO_2$, 1983 to 2014 for $CH_4$ and 2001 to 2014 for $N_2O$.

The four main methodological differences between the NOAA MBL data product and ours are (1) the NOAA data product has a higher resolution in time (weekly instead of monthly) and latitudes, (2) the NOAA MBL data product includes only a subset of the NOAA network data (sites within the marine boundary layer), while this study mixes both NOAA and AGAGE network data in the case of $CH_4$ and $N_2O$, (3) this study characterizes the global fields by lower rank representations (EOFs) of annual mean latitudinal gradients and seasonality, while the NOAA product derives latitudinal gradients (and seasonality thereby only implicitly) directly from the observations at each time step. In other words, the main smoothing/regularization step in our study happens at a later level in the analysis, and (4) this study is extended by ice core and firn data, regressions and extra-/interpolation to span the full-time period between year 0 and 2014. Thus, this study seamlessly merges in situ observational, air archive, ice and firn data to generate a comprehensive data product.

For several applications, the NOAA data product has clear advantages. However, with the task to produce a continuous data product beyond the instrumental observations, this study had to choose a method that was readily extendable. Hence, this study chooses the characterization of global fields into global-means, latitudinal gradients and seasonality. This implies a high degree of regularizations by relying on EOFs and corresponding scores. By regression, these EOF scores for latitudinal gradients or seasonality changes can be easily extended to cover the full-time period of interest. Hence, our method allows an estimate of global-means even if there is only a single data point (such as a Law Dome ice core record for a specific year), under the assumption that latitudinal gradients and seasonality are captured by the derived EOFs and regressed EOF scores.

Global-average time series of monthly GHG mole fractions are also provided by the World Data Center for Greenhouse Gases (WDCGG) (Tsutsumi, 2009). The WDCGG product uses similar smoothing techniques as the NOAA product, but include, like this study, a broader set of measurement stations, both in terms of regional coverage (including continental stations) and different networks that use different calibration scales, sampling, gas handling etc.

We compare the results of this study and NOAA MBL and WDCGG products. Overall, our monthly hemispheric averages of $CO_2$ closely match the NOAA MBL product. The NOAA MBL product (which is not the same as NOAA network monthly averages) suggests a slightly faster increase of northern hemispheric concentrations in the latter months of each calendar year (cf. thick and thin orange lines in Fig. 16a). Specifically, this difference results from the mid-latitude northern hemispheric bands from about 1995 onwards (with monthly-average differences of up to 4 ppm) where our study is higher than the NOAA MBL product. This could be because this study does not screen out land stations closer to the pollution sources, as the NOAA MBL product does, hence named MBL for "marine-boundary-layer".

Likewise, the WDCGG includes a broader set of stations and matches very closely with our global-mean time series, with our study being very close to WDCGG or in between NOAA MBL and WDCGG (Fig. 16a). Given that the difference between the NOAA study and our study has a strong seasonality, the nature of those pollution sources and how they become mixed in the atmosphere, if these effects contribute to the differences, could be a combination of fossil fuel related and (more seasonally-varying) biospheric sources (Fig. 17c). The southern hemispheric means of our study and NOAA MBL are very closely

matched (cf. thick and thin blue lines in Fig. 16a). Consequently, the global-mean concentrations from NOAA MBL and our study are closely matched, although again our data suggests NH autumn concentrations rising slightly faster than the NOAA MBL product, reflecting the northern hemispheric difference (cf. thick and thin black lines in Fig. 16a).

For $CH_4$, the differences between this study and the NOAA MBL data are more systematic and stronger (~10 ppb), with
5 generally higher surface $CH_4$ concentrations implied by this study (Fig. 16b). Again, this study's global mean matches closely the WDCGG or sits in between the NOAA MBL and the WDCGG data products. There are some differences in the seasonality compared to the NOAA MBL product though. The seasonal variation is similarly shaped between our study and NOAA MBL for the southern hemisphere, although there seems to be a slight phase-shift of about a month with the NOAA MBL product in the southern hemisphere assuming a slightly earlier increase and decrease and slightly higher amplitude (Fig. 16b). This
phase-shift of the southern hemisphere together with sometimes lower peak northern hemispheric concentrations in the NOAA MBL product suggests global-mean NOAA MBL $CH_4$ concentration that show a double peak within any year, while our data assimilation and the WDCGG product suggests a smoother single-peak oscillation of global-mean $CH_4$ concentrations (Fig. 16b). This peak results from the mid northern latitudes, where in the summer months, our study suggests up to 40 or 50 ppb higher concentrations (Fig. 18c).

For $N_2O$, the WDCGG global-mean and our data match very closely, with our implicit smoothing due to our lower rank representation of seasonal cycles and latitudinal means resulting in a smoother global mean compared to WDCGG (Fig. 16c). Similarly, the draft data product of the NOAA MBL indicates almost identical mole fractions to our concentration fields over the available time period from 2001 to 2014, with maximal differences being 0.8 ppb (Fig. 19).

In summary, our dataset closely matches the global-means of WDCGG in many years, but provides a complete 2-D field of
20 mole fractions. In comparison to the NOAA MBL products, there is one more systematic difference. Our CMIP6 GHG concentration fields are meant to represent the mean monthly state of the latitudinally-averaged surface atmosphere, including land and polluted areas, i.e. not confined to areas with background concentrations (section 6 'Limitations'). This is a key difference to the NOAA Marine Boundary Layer product, which is a consistent background concentration product, resulting in slightly lower global-mean concentration estimates.

## 5.5    Comparison to mid-troposphere $CO_2$ concentrations by NASA Aqua satellite

Since its launch in 2002, the Aqua satellite and its infrared sounder provides an additional independent data product to estimate tropospheric $CO_2$ mole fractions. Rather than at ground level, this sensor provides an estimate of tropospheric concentrations with a maximum sensitivity around 7km height, i.e. in the mid-troposphere. In the tropics and the parts of the southern hemisphere that are covered by the Aqua satellite product, the agreement between our data and the AIRS level 3 data (available
at: ftp://acdisc.gsfc.nasa.gov/ftp/data/s4pa/Aqua_AIRS_Level3/AIRX3C2M.005/) is encouraging, although the overall gradient is lower in line with 3-D atmospheric transport model results (Olsen and Randerson, 2004). In the northern hemisphere, the difference in the phase and amplitude of the seasonal cycle is most apparent, with satellite data showing a later onset of the autumn concentration increase by about 4 months while the drawdown of concentrations seems closer in phase between mid-troposphere and surface concentrations (Fig. 16a). Overall the amplitude is less than half of the surface

hemispheric mean amplitude, leading to seasonally higher winter and lower summer concentrations of our surface data product in the northern hemisphere by up to 10 ppm (Fig. 17e).

This systematic difference between ground-level and mid-atmosphere concentrations, supported by 3-D transport modelling studies (Olsen and Randerson, 2004), has ramifications for the implementation of vertical concentration profiles in climate models. Without taking into account the dampened seasonal cycle and latitudinal gradient in the mid and higher troposphere, the models could overestimate the variations in the radiative effects, if our latitudinally and monthly resolved surface concentration fields are prescribed. On the other hand, if global- annual mean values are prescribed, the radiative forcing effect variations over latitudes and within a year will obviously be underestimated.

### 5.6   Comparison to other literature studies.

Our GHG derivations over the recent instrumental periods are based on the AGAGE and NOAA station-by-station data and we extended our 2-D concentration field results back in time by using e.g. global-mean estimates of previous studies' estimates (Methods). The AGAGE and NOAA networks themselves publish global-mean results, and WMO as well as other literature studies produce composite long-term global-mean and/or hemispheric concentration estimates. Thus, while often not entirely independent, as the studies use the same original data sources or we rely on some studies' previous derivations, we here provide a comparison to a selection of the literature. Specifically, in addition to the comparisons with NOAA marine boundary layer, WDCGG and NASA Aqua satellite data, we discuss some instances where our results show substantial differences compared to earlier studies that have derived hemispheric or global means from instrumental data (Montzka et al., 2015; Rigby et al., 2014), from firn data (Butler et al., 1999; Trudinger et al., 2016a) or are themselves composites of multiple data sources (Martinerie et al., 2009; Velders and Daniel, 2014; WMO, 2014). The comparisons are shown in the panels f, g, and h of the factsheets for each gas (Fig. 9, Fig. 11, Fig. 12, and Fig. 20 to Fig. 59) with the comparison data described in Table 12. High latitude northern hemisphere data for atmospheric mole fractions is reported in the supplement of Buizert et al. (2012), provided by Vas Petrenko and Patricia Martinerie (Table 12). For $CO_2$, the Petrenko data set has, as expected for the high northern latitudes, a very strong seasonal cycle, consistent with our less pronounced northern hemispheric-average cycle, as the data represents higher northern latitudes (Fig. 9f, g, and h). The long-term concentration trend over time in the Petrenko $CO_2$ record seems similar to the global CMIP5 data set which in turn was based on previous Law Dome data, indicating a slight local maximum in 1890 and lower 1940s plateau (cf. Fig. 9g and Fig. 15).

For $CH_4$, the Petrenko record shows a comparable, yet again stronger, seasonality. The annual means are very comparable to our derivation (compare the high latitude red circles, indicating annual-mean station averages of our analysis and Petrenko data as shown in Fig. 11f), although there are some steps in annual means in the Petrenko data set around 1956 and 1975, which are not present in our dataset (Fig. 11f). For earlier times, i.e. between 1860 and the 1920s, the Petrenko annual mean is closer to our global-mean, not the high-latitude estimates, as our study assumes a large latitudinal gradient based on the NEEM and Law Dome data differences (Methods) (Fig. 11g).

For $CCl_4$, the Martinerie data show a lower increase from 1955 to the late 1960s and strong increase around 1970. The firn data by Butler et al. (1999) suggests an earlier start of atmospheric concentration increases around 1890, and then slightly

lower levels over 1960 to 1990 compared to the WMO (2014) and Velders and Daniel (2014) timeseries which we use as optimisation target for our 2-D fields. The difference between the Butler and Velders datasets can probably be explained by the wider firn air age distribution in the study by Butler. The findings by Sturrock et al. (2002) suggest an onset of detectable atmospheric concentrations around 1920 (Figure 5f therein). The NOAA global mean that is available from 1992 onwards

(Montzka et al. (1999) updated at http://www.esrl.noaa.gov/gmd/hats/combined/CCl4.html) and indicates initially slightly higher global mean estimates than our derivation, which is for the instrumental period based on 6 AGAGE and 13 NOAA HATS stations (Fig. 20f, g, h).

For CFC-11 (Fig. 21g), the NOAA Montzka-ODS reconstruction of the global-mean is slightly higher (1 ppt) than ours, which is almost identical to WMO (2014) and data by Velders and Daniel (2014). Those differences presumably result from

differences in station coverage, different calibration scales and air sampling and analysis techniques between the NOAA and AGAGE networks. The seasonalities show comparable amplitudes, as they do for CFC-12 (Fig. 22h). With CFC-115, our study follows the historical shape of the WMO (2014) record, with Velders and Daniel (2014) being slightly lower (~0.5 ppt) (Fig. 25f).

For $CH_2Cl_2$, the in situ instrumental record we use only reaches back to 1994, although the Cape Grim air archive record goes

back to 1978. From 1994 to 2003, the northern latitude measurements imply a mole fraction reduction from 40 to 30 ppt, whereas the southern hemispheric measurements are almost flat during that time (also shown in Trudinger et al. (2004)) (Fig. 26f). We note that there are substantial uncertainties in the pre-1995 concentrations, as e.g. Koppmann (1993) reported 18 ppt and 36 ppt average concentrations for the southern hemispheric and northern hemispheric measurements from a 1989 Atlantic transect ship measurement campaign (not shown in the figure). This could imply a global average value of approximately 27

20  ppt in 1989, instead of the 20 ppt assumed in this study – although different calibration scales might contribute to this difference. Recent seasonality and increases of $CH_2Cl_2$ are closely matching other time series, such as the AGAGE and NOAA results from GCMS measurements (Fig. 26f), although there is a slight offset in the absolute level, possibly caused by our study not sorting out data points from so-called pollution events in the case of AGAGE data for $CH_2Cl_2$, whereas NOAA results are from flasks collected only in baseline-air conditions (Spivakovsky et al., 2000).

For $CH_3Br$, our CMIP6 recommendations match very closely the NOAA (Montzka et al. (2003) updated on ftp://ftp.cmdl.noaa.gov/hats/methylhalides/ch3br/flasks) and AGAGE global means (2014) after 1995. Before then, the Butler (1999) global-mean firn reconstruction coincides closely with our southern-hemispheric mean. The 2004 firn reconstruction by Trudinger (2004) is close to the southern hemispheric mean, but shows somewhat more variation than the smooth exponential increase assumed by this study, WMO (2014) and Velders and Daniel (2014).

For $CH_3CCl_3$, the overall agreement between the different (although not independent) studies considered here is excellent, for example the high northern latitude data from Martinerie (Buizert et al., 2012; Martinerie et al., 2009) in the South Pole firn data reconstruction (Montzka et al., 2010), approximately in line also with the findings by Sturrock et al. (2002).

The atmospheric concentrations of $CH_3Cl$ show a strong seasonal cycle, as is to be expected from the short lifetime due to the OH-related sink. As in the case of methyl bromide ($CH_3Br$), the pre-instrumental period before 1995 implies a number of

uncertainties in our CH₃Cl time series. Here, we follow again the WMO (2014) and (not independent) Velders and Daniel (2014) reconstruction that are based on Butler et al. (1999) firn reconstructions. However, we note that the more recent Trudinger et al. (2004) $CH_3Cl$ reconstruction indicates both a significantly lower concentration for southern latitudes in the 1970s and a smoother increase compared to the more sudden rise of concentrations around 1940 as implied in this study (Fig.

29g).

As briefly discussed in section 3.4, the $CHCl_3$ history in this study relies on the Worton et al. (2006) reconstruction, whose shape is similar to Trudinger et al. (2004), although the latter indicates lower global mean concentrations and not the diminishing latitudinal gradient suggested by Worton et al. (2006). As with other gases (e.g. $CH_2Cl_2$), the implied pre-industrial value of around 6 ppt should be investigated in the future (Fig. 30).

For Halon-1211, the recent study by Vollmer et al. (2016) and the earlier study by Sturrock et al. (2002) (not shown) suggest slightly higher initial concentrations (around 1975 to 1988) compared to the initially-lower and then larger exponential increase we assumed by following Velders and Daniel (2014). We follow the global-mean derivation in the CSIRO inversion from Vollmer et al. in case of Halon-1211. After 1990 the southern hemispheric reconstruction by the Bristol and CSIRO inversions (Vollmer et al., 2016) are slightly lower and hence the latitudinal gradient slightly larger than what we derived from the

AGAGE and NOAA station data, but the differences are small (Fig. 31f). The Cape Grim measurements analysed on the UEA volumetric scale (Newland et al., 2013) are also in good agreement with the small offset to our global mean consistent with the derived latitudinal gradient (Fig. 31f). Similarly to Halon-1211, the very early concentration increases of the Halon-1301 between 1970 and 1978 are higher in the Vollmer et al. (2016) study than in Velders and Daniel (2014), and again the more recent years from 2007 onwards (Fig. 32h) are higher in Vollmer. In those latter years, our aggregation of AGAGE and NOAA

station data however suggests slightly lower concentrations, although the absolute difference (0.05 ppt) is within the measurement uncertainty and the overall agreement is very good. The Newland et al. (Newland et al., 2013) study of southern hemispheric concentrations at Cape Grim would suggest slightly lower concentrations, although part of the slight offset could be related to differences in scales. However, our Halon-1301 record suffers from a potentially inadequate scaling of the latitudinal gradient. A low gradient around 2000 to 2002 (Fig. 32d and f) results from our scaling with global emissions that

are assumed to drop in that period (Velders and Daniel, 2014) although subsequent station data suggest again a slightly stronger gradient. Furthermore, a second issue with our Halon-1301 record is a slight drop of the monthly data in year 2014 (Fig. 33f), which is likely an artefact of our assimilation procedure to be corrected by assimilations that consider observational data beyond 2014.

Halon-2402 is likely the most obvious example where a shifting measurement spatial coverage density can lead to small jumps

in latitudinal gradients or global means (Fig. 33f and h). The overall mole fractions are very small and the early agreement between the WMO (2014) time series and the Vollmer et al. (2016) findings is very good. In 2009, when data coverage increased, the latitudinal gradient is suggested to suddenly decrease, which is likely an artefact of the assimilation procedure that is only able to cope with time-varying data coverage to a certain degree (Methods). However, overall, the implied shifts of 0.02 ppt are negligible in the larger picture, and certainly negligible for radiative forcing, as the shift in Southern hemispheric

radiative forcing is equivalent to only about 0.000003 W/m$^2$ (Fig. 33h). Halon-2402 is also an illustration of how big differences in some measurement scales can potentially be. The Cape Grim data analysed by Newland with a volumetric UEA scale indicates 10-15% lower concentrations (Fig. 33f) (Newland et al., 2013).

For HCFC-142b our derived global-mean is in the middle of the AGAGE and NOAA network averages, despite our study
including those data points that are subject to 'pollution' events in the case of HCFC-142b, with large positive outliers (Fig. 36f), similar as in the case of HFC-134a (Fig. 50f). Pollution events might however be contributing to the difference between our HFC-152a global-means and the two independently derived network global means for AGAGE and NOAA, which largely exclude pollution events by using statistical methods or conditional sampling (O'Doherty et al., 2001) (see Fig. 52f).Two more issues can be observed with HCFC-142b data. Firstly, our end of 2014 concentrations are somewhat uncertain and in this case
possibly wrongly decreasing, which results from the smooth annual mean representation and our assimilation procedure. The differences are again very small and negligible in radiative forcing terms, but a smooth connection will have to be designed for the adjacent datasets representing SSP-RCP scenarios. Secondly, since 2010, our estimates for the HCFCs, namely HCFC-22 (Fig. 34f), HCFC-141b (Fig. 35f) and HCFC-142b (Fig. 36f) indicate smaller increases than implied by the post-2010 non-observational scenario data represented by Velders and Daniel (2014). As in the early study by Sturrock et al. (2002), our study
represents the slow onset of HCFC-142b concentrations in between 1960 and 1990 as shown in WMO (2014) and Velders and Daniel (2014).

For the three main PFCs, i.e. $CF_4$ (Fig. 45), $C_2F_6$ (Fig. 37), and $C_3F_8$ (Fig. 38), we find a similar and good agreement of the main studies. The outliers are the previously recommended CMIP5 concentrations (Meinshausen et al., 2011) for these gases, which were at the time not yet based on either the Trudinger et al. (2016a) or Mühle et al. (2010) studies. As mentioned above,
the concentrations of the lesser important PFCs, $C_4F_{10}$ (Fig. 39), $C_5F_{12}$ (Fig. 40), $C_6F_{14}$ (Fig. 41), $C_7F_{16}$ (Fig. 42) and $C_8F_{18}$ (Fig. 43) are based on the Ivy et al. (2012) reconstructions, with reversing latitudinal gradients in the case of $C_6F_{14}$, $C_7F_{16}$, and $C_8F_{18}$, which are unexplained so far and require further confirmation. Our historical c-$C_4F_8$ concentrations are based on the study by Oram et al. (2012) with assumed conversions of the Cape Grim measurements to northern hemispheric and global-averages.

For HFC-43-10mee, we based our trajectory on the NH and SH estimates of Arnold et al. (2014) with relatively small latitudinal gradient and hemispheric means being informed by the recently available observations since 2010 from the AGAGE Medusa instruments (Fig. 48f). Note that for HFC-365mfc data (Fig. 56), the difference between the station data and those published in Montzka (2015) reflects a difference that is now much smaller after a calculation-related correction was applied to the NOAA calibration scale after the publication of Montzka et al. (2015). All studies are now in relatively close alignment
with the shown AGAGE network average, the Vollmer et al. (2011) study and our derivation (which is slightly lower, <0.1ppt). In addition, the air archive and AGAGE network analysis by Vollmer et al. (2011) investigated the HFCs HFCs-236fa, HFC-227ea, and HFC-245fa. Those results are closely aligned with the ones constructed here based on the WMO AGAGE network average estimates (Fig. 54, Fig. 53, Fig. 55).

Like our study, there are also studies that assimilate a wide range of gases with latitudinal and seasonal variation. For example, the AGAGE network assimilation with a 12-box model and optimization approach to reconcile emissions and concentrations (Rigby et al., 2011; Rigby et al., 2013) produces 4 semi-hemispheric concentration timeseries with 3 vertical levels (Rigby et al., 2014). Those studies based on AGAGE data are more comprehensive than this one, as both emissions and concentrations as well as lifetimes are optimized and reconciled. In our case, we only assimilate AGAGE and NOAA observations to derive atmospheric mole fractions in 15 degree latitudinal bands (methods).

## 6    Limitations

Even though the presented dataset of historical surface GHG concentrations is – to our knowledge - more comprehensive than other composite datasets before, there are several key limitations.

### 6.1    Specific use of dataset

First, the dataset was assimilated from several sources to provide a common starting point for global climate models as part of the CMIP6 experiments. Thus, for example, the data was not designed as a starting point for inversion studies, which estimate emissions, or studies of biogeochemical processes. Those studies tend to require pure observations, or at least products with appropriate uncertainty information (including auto-correlations) attached to it, rather than partly interpolated composite products. As mentioned earlier, our assimilation does not incorporate early atmospheric $CO_2$ measurements from the South Pole, which might result in a systematic bias for that latitude for some years of ~1ppm (Fig. 7, panel l). This warning in terms of our data use is especially important for the fine-grid interpolation we present. The 0.5-degree mean-preserving smooth interpolation should not be misinterpreted to portray measurement information at such a fine scale.

### 6.2    No vertical and longitudinal resolution

The purpose of forcing climate models correctly would best be accomplished by vertically resolved latitudinal and longitudinal fields, which (in the case of $CO_2$) even include a diurnal cycle. Our latitudinally and monthly resolved dataset offers climate models an option to capture some key variability compared to the global and annual mean CMIP5 concentration recommendation (Meinshausen et al., 2011). However, a correct implementation of this additional monthly and latitudinal variability is also dependent on an appropriate propagation of the surface signal throughout the troposphere and stratosphere. For example, some studies (Olsen and Randerson, 2004) find that column $CO_2$ is found to only exhibit roughly half of the latitudinal gradient and seasonal variation compared to the surface concentrations. In the CESM1 model (Hurrell et al., 2013) with prescribed surface GHG concentrations, the vertical propagation of the $CO_2$ concentration is assumed to be constant. In the case of the other GHGs ($CH_4$, $N_2O$ and CFCs) a constant concentration in the troposphere and a decrease of the concentration in the stratosphere is assumed in CESM1. In particular, the scale heights in the stratosphere of these trace gases depend on latitude, which produces a more realistic stratospheric distribution. We recommend vertical extensions to our surface concentration reconstructions only in the case that the model has no intrinsic transport model or extension parameterisation.

Furthermore, we do not include the longitudinal variation. Again, specifically for $CO_2$, this longitudinal variation might be systematic given the land/ocean contrast. For example, the MPI-ESM-LR model indicates systematically higher surface $CO_2$ concentration over land, which in turn would have a radiative effect (Fig. 60 and Fig. 61).

## 6.3    Limited filtering of station measurements

Our assimilation procedure is a rather simple one and does not attempt to offset potential biases due to day and night-time sampling biases for $CO_2$ in the case of some flask measurements, or whether including pollution events would bias the latitudinal averages towards higher than current average values. In a world with continuing point sources, screening out pollution effects might cause proposed averages to lag slightly behind the true average concentration. The question is whether the correlation between sampling locations and source locations will inherently bias the average concentrations towards higher-

than true average values in our assimilation for species, where we include pollution events. For most substances, we do not find any systematic difference between the network averages from AGAGE or NOAA, although there are some species (e.g. HFC-152a, see Fig. 52) for which our higher concentration reconstructions could in part be explained by this different method. The opposite might also be the case, i.e. that despite including some pollution events, there could still be an inherent underestimation of true zonal means. That is because the NOAA and AGAGE sampling stations, which we are sourcing our

raw data from, tend to be biased towards remote/clean-air/well mixed conditions and this will have implications for our latitudinal gradient and seasonal cycle. Where there are continental sites, they are often at altitude, and when flasks are sampled, they are generally for mid-afternoon when mixing is largest. Hence the fitted latitudinal gradient for $CO_2$ at least might be closer to the NOAA marine boundary layer product than to a true zonal mean. Also, the seasonal cycle will be more representative of marine conditions than continental ones (where a diurnal rectifier could potentially dampen or offset

seasonally low concentrations in summer in the case of $CO_2$). This bias towards remote measurements tends to increase the further back in time we go.

## 6.4    Calibration scales

Another limitation of our study is related to the different calibration scales of atmospheric gas measurements. In our data assimilation method with no scale conversion between the SIO and NOAA scales of the AGAGE and NOAA networks

(Methods), a time-varying difference between the scales or time-varying coverage from one network to another can lead to spurious trends in the derived concentrations. We argue that our "middle of the road" data assimilation method across the two networks is however one justifiable, yet not the only viable, assimilation method. The reasons for our chosen approach are a) uncertainties in absolute mole fractions estimates are small compared to other uncertainties that would affect the radiative forcing in climate models, b) alternative "pure" scale data assimilation could only deal with the trend uncertainty, not with the

uncertainty arising for absolute mole fraction values (assuming that both the SIO and NOAA scales are equally sound), c) we intend to be "network"-neutral and d) a single "in-between" concentration estimate is likely the most appropriate for the primary application purpose (historical simulations of climate models) of the provided data. However, future researchers are encouraged to work directly with the principal investigators of the two networks to devise data assimilation methods that would be better suited for alternative applications, such as uncertainty estimates of inverse emissions etc. A clear limitation of our

data product is hence our implicit "in between" scale, with time-varying influences from measurements under the one or other network. Thus, differences to "pure" SIO or NOAA scale will partly arise from this "scale" issue.

## 6.5 No uncertainty estimates

Another important limitation of our study is that we do not provide uncertainty estimates. This is primarily related to the fact that the purpose of this study was to provide a consolidated dataset for CMIP6 climate model experiments. Those models experiments can only be performed a limited number of times given today's computational resources. The experimental protocol hence does not foresee an ability to vary GHG mole fractions within its uncertainties, given that many aspects of climate models are affected by more substantial uncertainties, such as aerosols. The original AGAGE and NOAA (sometimes monthly averaged) sampling data points shown in the Factsheets (see panels f, g, and h) can however provide an indication of uncertainties and the spread in observations.

## 6.6 Uncertain scaling of seasonality changes and latitudinal gradients back in time

Our choice of predictor for the $CO_2$ seasonality change (namely the product of $CO_2$ concentration and global-mean temperature deviation since pre-industrial) is subjective and using only $CO_2$ concentration or temperature would have yielded a larger seasonality difference between current and pre-industrial times. Further research will be necessary to obtain an optimal proxy for presumed pre-observational $CO_2$ seasonality changes. Similarly, our common explanatory variable for regressions of latitudinal gradients, i.e. global emissions (Boden et al., 2013), is an approximation. Ideally, the time-changing latitudinal distribution of emissions would be considered in those backward extensions of the latitudinal gradient over time. More generally, further research into observational and modelling-derived constraints regarding pre-1950 latitudinal gradients of $CO_2$ could allow future studies to go beyond our simplified assumption of a zero pre-industrial gradient in the light of the uncertainty.

## 6.7 Broad, but not comprehensive data coverage

For the recent instrumental period, our study is predominantly based on the NOAA and the international AGAGE network data. Consistent quality control and consistent scales are advantages of that approach. Ideally however, our study should have started out from a yet more inclusive representation, e.g. including the multiple additional station datasets gathered and archived by the World Data Centre for Greenhouse Gases (WDCGG) that are neither part of AGAGE or NOAA networks. The WDCGG station raw data is available at: http://ds.data.jma.go.jp/gmd/wdcgg/cgi-bin/wdcgg/catalogue.cgi . While the methodology of our study could be maintained or built upon, we hence recommend for any future updates, that those additional datasets are considered – with the appropriate quality control and scale conversion efforts.

## 6.8 Known issues

There is one known issue in the historical dataseries before the year 2002 for CF4, C2F6 and C3F8. We use the Trudinger et al. (2016b) datasets and our algorithm categorised them as mid-year values, but the data were estimates for start-of-year values. Thus, while Trudinger et al. (2016b) is well aligned with the Mühle et al. (2010) over that time period (given that the same in-situ and archive data was used), our historical timeseries suggest half a year's growth rate, i.e. up to maximally 0.63 ppt, 0.065

and 0.015 ppt, too low mole fractions for $CF_4$, $C_2F_6$, $C_3F_8$, respectively for the pre-2002 timeframe. In terms of radiative forcing, this difference amounts to approximately 0.00022 $Wm^{-2}$, 0.000016 $Wm^{-2}$ and 0.0000043 $Wm^{-2}$ in the years with the maximal growth rates (1980, 1999 and 2002, respectively). Given that some CMIP6 models had started using the historical data by the time of discovering this error (which will have no significant effect on CMIP6 outputs), we opted for not revising this study's CMIP6 datasets.

# 7    Conclusion

Ice core measurements over the past 800'000 years reveal how atmospheric GHG concentrations of $CO_2$, $CH_4$ and $N_2O$ varied. These variations indicate various feedback mechanisms connected to the glacial and interglacial cycles driven by Milankovich cycles. With the arrival of homo sapiens, initially through activities such as deforestation and agriculture, and then through

fossil-fuel driven industrial activities from the start of the industrial revolution, the atmospheric composition changed. Unprecedented over the 800'000 years of the ice core record, $CO_2$, $CH_4$ and $N_2O$ concentrations suddenly rose to record levels, with global-mean $CO_2$ reaching a historical mark of 400 ppm in 2015 (Fig. 6). Recently, synthetic GHGs arising from refrigerants, solvents, foam-blowing agents and even gas-cushioned shoe soles added to the warming effect, the radiative forcing. As the IPCC AR5 found, the most likely warming contribution from these GHGs is now higher than the observed

warming (Figure TS.10 in IPCC AR5 (IPCC, 2013). That means that without the human activities that happen to cool the planet, namely the aerosols we emit, observed warming would have been even greater than what has already been experienced. In this study, we compile a set of GHG histories over the last 2000 years – based on numerous efforts by the scientific community to retrieve firn samples and ice cores in the most remote places on Earth, unlock their secrets by analysing the enclosed air and by investing in a large network of in-situ and flask measurement stations across the planet. Our understanding

of past climate change is vital to develop scenarios of the future and design humanity's response strategies in terms of mitigation and adaptation. The ongoing efforts to retrieve and monitor the composition of the planet's atmosphere efforts are sometimes threatened (Lewis, 2016). Without those efforts, the future ahead of us would remain shrouded in even greater uncertainty.

In this dataset, we attempted to provide a solid base for the next generation of climate and earth system models to further our

understanding of past and future climate changes. Providing seasonal and latitudinal differences of the radiative forcing that drives the climate change across the globe, we can hope for an even more appropriate comparison between models and past land-ocean, regional land and oceanic temperature observations. Ignoring these seasonal and latitudinal differences can lead to different calculated climate impacts of GHG emissions. Thus, accurately including this variability is a necessary condition to accurately compare model calculations and observations and to understand the reasons for the differences. Those agreements

and disagreements between what models and past observations tell us will then allow us to calibrate our understanding of the earth system, its non-linearities and its many feedback cycles, the human influences and natural variabilities – called 'detection and attribution'.

We have been engaging in a unique experiment with our climate. In order to stay below the warming limits, that were set forth in the Paris Agreement in 2015 (i.e. well below 2°C and 1.5°C relative to pre-industrial levels), the next generation of climate

models and the examination of their response to climate drivers will be vital as an information basis for decision makers. This study into the main past driver of human-induced climate change will hence contribute to our collective examination of the tremendous challenge in which we find ourselves.

## 8    Code Availability

The MATLAB and R code that was used to assimilate the raw data is available from the authors on request.

## 9    Data Availability

A supplementary data table is available with global and annual mean mole fractions. The complete dataset with latitudinally
and monthly resolved data in netcdf format is available via https://pcmdi.llnl.gov/search/input4mips/.Additional data formats, i.e. CSV, XLS, MATLAB .mat files of the same data are also available via www.climatecollege.unimelb.edu.au/cmip6. The respective raw data used in this study is available from the original referenced data providers on request or can be found at the web locations indicated in Table 12.

## 10    Acknowledgements

The authors would first and foremost like to thank the large community of scientists, research assistants and measurement technicians that were involved in collecting the firn, ice core and atmospheric in-situ and flask measurements across the world. The primary networks AGAGE and the Cooperative Air Sampling Network managed by NOAA deserve the most credit, including all its individual researchers and the networks' policy to make the raw data available to the broader scientific community. We thank in particular the following researchers for invaluable efforts to collect, screen and make available NOAA
network data: Ed Dlugokencky, Pieter Tans, David Nance, Bradley Hall, Geoff Dutton, James Elkins, Debra Mondeel, Carolina Siso, Ben Miller. We would also like to thank the Editor O. Morgenstern for helpful suggestions and Zebedee Nicholls for his comments.

Furthermore, we thank the ESM CMIP5 modelling groups who contributed to the 5th Climate Model Intercomparison project and whose data we analysed. Also, the reviewer comments from one anonymous reviewer and Piers Forster were most helpful
in improving the quality of the manuscript.

Attributions: MM designed the study. EV wrote most of the data analysis and read-in routines together with MM. KL analysed the CMIP5 ESM models and produced related figures. Other figures and the factsheets were produced by MM. AN provided an initial literature overview. All authors wrote, commented on and/or discussed the manuscript based on a first draft by MM. NM designed the mean-preserving interpolation routines. Multiple authors provided vital data.

MM thankfully acknowledges the support by the Australian Research Council Future Fellowship grant FT130100809. This work was undertaken in close collaboration with partners in the European Union's Horizon 2020 research and innovation programme CRESCENDO (grant No 641816), of which the University of Melbourne is an unfunded partner. CSIRO's contribution was supported in part by the Australian Climate Change Science Program (ACCSP).

# 11 Tables

**Table 1 - Derivation and construction of CMIP6 concentration fields for $CO_2$, $CH_4$ and $N_2O$, as shown in Fig. 1 and described in Methods.**

| Gas | Time period | Main data source | Global and annual-mean $\overline{C_{global}}$ | Seasonality $\hat{S}_{l,m}$ | Seasonality Change $\Delta S_{l,m}$ | Latitudinal gradient $\hat{L}$ |
|---|---|---|---|---|---|---|
| $CO_2$ | 1984-2013/2014 | NOAA ESRL Carbon Cycle Cooperative Global Air Sampling Network, 1968-2014. Version: 2015-08-03, monthly station averages (Dlugokencky, 2015b; NOAA ESRL GMD, 2014a, b, c) | Calculated based on observational data source (section 2.1.3). | Mean over 1984-2013 period. | Leading EOF of residuals from observation | Two leading EOFs and their scores derived from residuals from observations. (2014: optimized to match observational data). |
| | Before 1984 | See text. Updated Law Dome (Etheridge et al., 1998b; MacFarling Meure et al., 2006; Rubino et al., 2013) and annual-mean MLO station data (Keeling et al., 1976) | Optimized to match smoothed median approximation (section 2.1.6) of Law Dome record (0 to 1966) & Mauna Loa record (1959-1984) with interpolation between 1955-1958 | Kept constant as above | Regressed against product of $CO_2$ concentration and surface air temperature change since pre-industrial | The score for EOF1 is regressed against global annual fossil fuel & industry emissions (Boden et al., 2013). Score for EOF2 linearly returned to zero in 1850. See Fig. 9c. |
| $CH_4$ | 1985 to 2013/2014 | AGAGE monthly station means, incl. pollution events ('.mop') (Cunnold et al., 2002) & NOAA ESRL monthly station data (Dlugokencky, 2015a) | Calculated based on observational data source (section 2.1.3). | Mean over 1985-2013 period. Applied as relative seasonality | Assumed zero | Two leading EOFs and their scores derived from residuals from observations. (2014: optimized to match observational data). |
| | Before 1985 | Updated Law Dome (Etheridge et al., 1998a; MacFarling Meure et al., 2006) & NEEM (Rhodes et al., 2013) | Optimized to match smoothed Law Dome record & NEEM firn data | | | The score for EOF1 is regressed against global annual fossil fuel & industry emissions (Gütschow et al., 2016). Score for EOF2 kept constant before in situ instrumental period. |
| $N_2O$ | 1990 to 2013/2014 | AGAGE monthly station means, incl. pollution events (Prinn et al., 1990) & Combined Nitrous Oxide data (monthly station averages) from the NOAA/ESRL Global Monitoring Division. | Calculated based on observational data source (section 2.1.3). | Mean over 1990-2013 period. Applied as relative seasonality | Assumed zero | Two leadings EOF and their scores derived from residuals from observations. (2014: optimized to match observational data). |

| Gas | Time period | Main data source | Global and annual-mean $\overline{C_{global}}$ | Seasonality $\hat{S}_{l,m}$ | Seasonality Change $\Delta S_{l,m}$ | Latitudinal gradient $\hat{L}$ |
|---|---|---|---|---|---|---|
| | Before 1990 | Updated Law Dome (MacFarling Meure et al., 2006) until 1968 | Optimized to match smoothed Law Dome record until 1968. Interpolation until 1986 with optimization to sparse observational data until 1990. | | | Score for EOF1 and 2 kept constant before in situ instrumental period. |

**Table 2- Raw data used for CO₂ surface concentration field derivation**

| Dataset | Reference / URL | Stations / Location | Used for | Description / Filtering |
|---|---|---|---|---|
| NOAA ESRL GMD Surface Flask data. | (Conway et al., 1988; Conway et al., 1994; Komhyr et al., 1985; Komhyr et al., 1983; Tans et al., 1989; Tans et al., 1990a; Tans et al., 1990b; Thoning et al., 1995; Thoning, 1987; Thoning et al., 1989; Zhao and Tans, 2006) | 81 stations of the surface flask network[a]: ABP, ALT, AMS, AOC, ASC, ASK, AVI, AZR, BAL, BHD, BKT, BME, BMW, BRW, BSC, CBA, CGO, CHR, CIB, CMO, CPT, CRZ, DRP, DSI, EIC, GMI, GOZ, HBA, HPB, HSU, HUN, ICE, IZO, KCO, KEY, KUM, KZD, KZM, LEF, LLB, LLN, LMP, MBC, MEX, MHD, MID, MKN, MLO, NAT, NMB, NWR, OPW, OXK, PAL, PAO, POC, PSA, PTA, RPB, SCS, SDZ, SEY, SGI, SGP, SHM, SMO, SPO, STC, STM, SUM, SYO, TAP, THD, TIK, USH, UTA, UUM, WIS, WLG, WPC, ZEP | Observational period estimation of global mean, latitudinal gradient seasonality and seasonality change over 1984-2013. Optimization of global mean and latitudinal gradient in 2014 and before 1984. | This study used monthly average data that uses all sample points which have an 'accepted' flag, i.e. initial two dots ('..*') in the three digit flag. |
| Law Dome | (Etheridge et al., 1998b; Etheridge et al., 1996; Rubino et al., 2013) (MacFarling Meure et al., 2006) | Law Dome ice core | Used as input for piecewise 3rd-degree polynomial smoothing over remainder of years 0 to 1966. | |

[a] See station descriptions here: http://www.esrl.noaa.gov/gmd/dv/site/site_table.html

**Table 3 - Raw data used for CH₄ surface concentration field derivation**

| Dataset | Reference / URL | Stations / Location | Used for | Description / Filtering |
|---|---|---|---|---|
| NOAA ESRL GMD Surface Flask data. | (Dlugokencky et al., 2009; Dlugokencky et al., 1994a; Dlugokencky et al., 1998; Dlugokencky et al., 2005; Dlugokencky, 2015a; Dlugokencky et al., 1994c; Dlugokencky et al., 2001; Lang, 1990a, 1992, 1990b; Steele et al., 1992; Steele et al., 1987; Steele, 1991) | 87 stations of the surface flask network[a]: ABP, ALT, AMS, AMT, AOC, ASC, ASK, AVI, AZR, BAL, BHD, BKT, BME, BMW, BRW, BSC, CBA, CGO, CHR, CIB, CMO, CPT, CRZ, DRP, DSI, EIC, GMI, GOZ, HBA, HPB, HSU, HUN, ICE, ITN, IZO, KCO, KEY, KPA, KUM, KZD, KZM, LEF, LLB, LLN, LMP, MBC, MCM, MEX, MHD, MID, MKN, MLO, NAT, NMB, NWR, NZL, OPW, OXK, PAL, PAO, POC, PSA, PTA, RPB, SCS, SDZ, SEY, SGI, SGP, SHM, SIO, SMO, SPO, STM, SUM, SYO, TAP, THD, TIK, USH, UTA, UUM, WIS, WKT, WLG, WPC, ZEP | Observational period estimation of global mean, latitudinal gradient, seasonality and seasonality change over 1984-2013. Optimization of global mean and latitudinal gradient in 2014 and before 1984. | This study used monthly station averages that include all sample points which have an 'accepted' flag, i.e. initial two dots ('..*') in the three digit flag. |
| AGAGE GC-MD | (Prinn et al., 2000b) | AGAGE GC-MD network[b]: CGO, MHD, RPB, SMO, THD | | The monthly station averages that include pollution events ('.mop' file endings in case of AGAGE) were used. |
| Law Dome | (Etheridge et al., 1998a; MacFarling Meure et al., 2006) | Law Dome ice core at -66.73-degree south. | Long-term high-latitude southern hemisphere reference point with piecewise 3rd-degree polynomial smoothing over years 154 to 1974. | |
| EPICA Dronning Maud Land Ice Core | (Barbante et al., 2006; Capron et al., 2010) | Dronning Maud Land Ice Core | Used as input for piecewise 3rd-degree polynomial smoothing over remainder of years 0 to 153. | |
| NEEM Greenland | (Dahl-Jensen et al., 2013; Rhodes et al., 2013) | NEEM ice core Greenland data | Used for optimisation of global mean and latitudinal gradient score of EOF1 over timescale from year 0 to 1984, with linear interpolation of the score in between available 5-yearly NEEM datapoints. (section 2.1.4) | |

[a] NOAA station descriptions here: http://www.esrl.noaa.gov/gmd/dv/site/site_table.html

[b] AGAGE station descriptions here: https://agage.mit.edu/global-network

**Table 4 - Raw data used for N2O surface concentration field derivation**

| Dataset | Reference / URL | Stations / Location | Used for | Description / Filtering |
|---------|-----------------|--------------------|----------|------------------------|
| NOAA ESRL GMD Surface Flask data. | Combined $N_2O$ data from the NOAA/ESRL Global Monitoring Division, (ftp://ftp.cmdl.noaa.gov/hats/n2o/combined/HATS_global_N2O.txt, file date: Wed, Aug 19, 2015 2:40:55 PM) | 13 stations of the NOAA HATS global[a]: alt, brw, cgo, kum, mhd, mlo, nwr, psa, smo, spo, sum, tdf, thd | Observational period estimation of global mean, latitudinal gradient, seasonality and seasonality change over 1990-2013. Optimization of global mean and latitudinal gradient in 2014. | This study uses station averages, which include all sample points which have an 'accepted' flag, i.e. initial two dots ('..*') in the three digit flag. ('.mop' file endings in case of AGAGE) |
| AGAGE GC-MD | (Prinn et al., 1990; Prinn et al., 2000b) | AGAGE GC-MD network[b]: CGO, MHD, RPB, SMO, THD | | |
| Law Dome | (MacFarling Meure et al., 2006) | Law Dome ice core at -66.73 degree south. | Long-term high-latitude southern hemisphere reference point with piecewise 3rd-degree polynomial smoothing over years 155 to 1974. | |
| Gap | | | Sparse data availability in the period 1968 to 1986 suggestedagainst optimisations of global-means with annual datapoints, which is why an interpolation between 1968 (starting from smoothed Law Dome record) to 1986 (ending with optimized global mean to fit observational data) was assumed. | |

[a] NOAA station descriptions here: http://www.esrl.noaa.gov/gmd/dv/site/site_table.html

[b] AGAGE station descriptions here: https://agage.mit.edu/global-network

**Table 5- Options for reducing the number of GHGs to be taken into account in climate models to approximate full radiative forcing of all GHGs. The GHGs are ranked by their radiative forcing, with $CO_2$ having the highest radiative effect change between 1750 and 2014. The stated percentages in row X are cumulative, i.e. the radiative forcing of the GHG in row X plus the radiative forcing sum of all higher ranked GHGs. In Option 1, a climate model explicitly resolves actual GHG concentrations. With 8 and 15 species, 99.1% and 99.7% of the total radiative effect can be captured. In Option 2, only CFC-12 is modelled next to $CO_2$, $CH_4$, and $N_2O$; all other gases are summarized in a CFC-11-equivalence concentration. In Option 3, all ODS are summarized in a CFC-12-equivalence concentration, and all other fluorinated substances are summarized in HFC-134a-equivalence concentrations. Note that below shares are approximations, as linear radiative forcing efficiencies are assumed here for all gases, also for $CO_2$, $N_2O$ and $CH_4$.**

| Rank | The GHG contribution to climate change since 1750. Shares of change of total warming effect since 1750: Approx. Radiative forcing contribution between 1750 and 2014 relative to that of all GHGs | | Option 1 Using subset of actual concentrations, no equivalent gases | | Option 2 Summarizing all gases of lower importance than CFC-12 into CFC-11eq. | | Option 3 Summarizing all ODS into CFC-12-eq and all other fluorinated gases into HFC134a-eq | |
|---|---|---|---|---|---|---|---|---|
| | | | Shares of total warming effect: Approx. Radiative effect compared to effect of all GHGs (absolute in 2014, not relative to 1850) | | | | | |
| 1 | $CO_2$ | 64.0% | $CO_2$ | 72.9% | $CO_2$ | 72.9% | $CO_2$ | 72.9% |
| 2 | +$CH_4$ | 79.5% | +$N_2O$ | 86.1% | +$N_2O$ | 86.1% | +$N_2O$ | 86.1% |
| 3 | +CFC12 | 86.0% | +$CH_4$ | 95.0% | +$CH_4$ | 95.0% | +$CH_4$ | 95.0% |
| 4 | +$N_2O$ | 92.2% | +CFC12 | 97.2% | +CFC12 | 97.2% | +CFC12-eq | 99.5% |
| 5 | +CFC11 | 94.5% | +CFC11 | 98.0% | +CFC11-eq | 100.0% | +HFC134a-eq | 100% |
| 6 | +HCFC22 | 96.4% | +HCFC22 | 98.6% | | | | |
| 7 | +CFC113 | 97.2% | +CFC113 | 98.9% | | | | |
| 8 | +$CCl_4$ | 97.8% | +$CCl_4$ | 99.1% | | | | |
| 9 | +HFC134a | 98.3% | +HFC134a | 99.3% | | | | |
| 10 | +CFC114 | 98.5% | +$CF_4$ | 99.4% | | | | |
| 11 | +HFC23 | 98.7% | +$CH_3Cl$ | 99.5% | | | | |
| 12 | +$SF_6$ | 98.8% | +CFC114 | 99.5% | | | | |
| 13 | +$CF_4$ | 99.0% | +HFC23 | 99.6% | | | | |
| 14 | +HCFC142b | 99.2% | +$SF_6$ | 99.7% | | | | |
| 15 | +HCFC141b | 99.3% | +HCFC142b | 99.7% | | | | |
| ... | +28 additional GHGs | 100% | 28 additional GHGs | 100% | | | | |

**Table 6 – historical: Global- and annual-mean surface concentrations for the historical CMIP6 experiments. The year-to-year and monthly resolved global, hemispheric and latitudinally resolved concentrations for 43 GHGs and three aggregate equivalent concentrations are provided in the accompanying datasets over the time horizon year 0 (1 BC) to year 2014 AD. The complexity reduction options for capturing all GHGs with fewer species than 43 are indicated in the Table as Option 1, Option 2, and Option 3, with 'x' denoting relevant columns under each option (section 2.1.10).**

| Years | $CO_2$ | $CH_4$ | $N_2O$ | CFC-12-eq | HFC-134a-eq | CFC-11-eq | CFC-12 | Other |
|---|---|---|---|---|---|---|---|---|
| Option 1 | x | x | x | | | | x | x |
| Option 2 | x | x | x | | | x | x | |
| Option 3 | x | x | x | x | x | | | |
| Units: | ppm | ppb | ppb | ppt | ppt | ppt | ppt | |
| 1750 | 277.15 | 731.41 | 273.87 | 16.51 | 19.15 | 32.11 | 0.00 | All or a subset of other 39 individual gases, available online |
| 1850 | 284.32 | 808.25 | 273.02 | 16.51 | 19.15 | 32.11 | 0.00 | |
| 1851 | 284.45 | 808.41 | 273.09 | 16.51 | 19.15 | 32.11 | 0.00 | |
| 1852 | 284.60 | 809.16 | 273.17 | 16.51 | 19.15 | 32.11 | 0.00 | |
| 1853 | 284.73 | 810.40 | 273.26 | 16.51 | 19.15 | 32.11 | 0.00 | |
| 1854 | 284.85 | 811.73 | 273.36 | 16.51 | 19.15 | 32.11 | 0.00 | |
| 1855 | 284.94 | 813.33 | 273.47 | 16.51 | 19.15 | 32.11 | 0.00 | |
| 1856 | 285.05 | 814.80 | 273.58 | 16.51 | 19.15 | 32.11 | 0.00 | |
| 1857 | 285.20 | 816.45 | 273.68 | 16.51 | 19.15 | 32.11 | 0.00 | |
| 1858 | 285.37 | 818.36 | 273.76 | 16.51 | 19.15 | 32.11 | 0.00 | |
| 1859 | 285.54 | 820.40 | 273.90 | 16.51 | 19.15 | 32.11 | 0.00 | |
| 1860 | 285.74 | 822.31 | 274.06 | 16.51 | 19.15 | 32.11 | 0.00 | |
| 1861 | 285.93 | 824.40 | 274.24 | 16.51 | 19.15 | 32.11 | 0.00 | |
| 1862 | 286.10 | 827.03 | 274.42 | 16.51 | 19.15 | 32.11 | 0.00 | |
| 1863 | 286.27 | 830.17 | 274.57 | 16.51 | 19.15 | 32.11 | 0.00 | |
| 1864 | 286.44 | 833.60 | 274.72 | 16.51 | 19.15 | 32.11 | 0.00 | |
| 1865 | 286.61 | 836.89 | 274.88 | 16.51 | 19.15 | 32.11 | 0.00 | |
| 1866 | 286.78 | 840.36 | 275.05 | 16.51 | 19.15 | 32.11 | 0.00 | |
| 1867 | 286.95 | 844.00 | 275.21 | 16.51 | 19.15 | 32.11 | 0.00 | |
| 1868 | 287.10 | 847.25 | 275.39 | 16.51 | 19.15 | 32.11 | 0.00 | |
| 1869 | 287.22 | 850.13 | 275.56 | 16.51 | 19.15 | 32.11 | 0.00 | |
| 1870 | 287.35 | 852.44 | 275.72 | 16.51 | 19.15 | 32.11 | 0.00 | |
| 1871 | 287.49 | 853.99 | 275.90 | 16.51 | 19.15 | 32.11 | 0.00 | |
| 1872 | 287.66 | 855.23 | 276.08 | 16.51 | 19.15 | 32.11 | 0.00 | |
| 1873 | 287.86 | 856.17 | 276.25 | 16.51 | 19.15 | 32.11 | 0.00 | |
| 1874 | 288.06 | 857.82 | 276.42 | 16.51 | 19.15 | 32.11 | 0.00 | |
| 1875 | 288.29 | 859.47 | 276.59 | 16.51 | 19.15 | 32.11 | 0.00 | |
| 1876 | 288.52 | 860.86 | 276.74 | 16.51 | 19.15 | 32.11 | 0.00 | |
| 1877 | 288.75 | 862.38 | 276.86 | 16.51 | 19.15 | 32.11 | 0.00 | |
| 1878 | 288.99 | 864.14 | 277.00 | 16.51 | 19.15 | 32.11 | 0.00 | |
| 1879 | 289.22 | 866.28 | 277.13 | 16.51 | 19.15 | 32.11 | 0.00 | |
| 1880 | 289.47 | 868.70 | 277.27 | 16.51 | 19.15 | 32.11 | 0.00 | |
| 1881 | 289.74 | 870.98 | 277.37 | 16.51 | 19.15 | 32.11 | 0.00 | |
| 1882 | 290.02 | 873.25 | 277.49 | 16.51 | 19.15 | 32.11 | 0.00 | |
| 1883 | 290.26 | 875.60 | 277.59 | 16.51 | 19.15 | 32.11 | 0.00 | |
| 1884 | 290.51 | 878.15 | 277.70 | 16.51 | 19.15 | 32.11 | 0.00 | |
| 1885 | 290.80 | 881.03 | 277.80 | 16.51 | 19.15 | 32.11 | 0.00 | |
| 1886 | 291.10 | 883.84 | 277.89 | 16.51 | 19.15 | 32.11 | 0.00 | |
| 1887 | 291.41 | 886.93 | 278.00 | 16.51 | 19.15 | 32.11 | 0.00 | |
| 1888 | 291.76 | 889.93 | 278.08 | 16.51 | 19.15 | 32.11 | 0.00 | |
| 1889 | 292.11 | 893.16 | 278.19 | 16.51 | 19.15 | 32.11 | 0.00 | |

| Years | CO$_2$ | CH$_4$ | N$_2$O | CFC-12-eq | HFC-134a-eq | CFC-11-eq | CFC-12 | Other |
|---|---|---|---|---|---|---|---|---|
| 1890 | 292.46 | 896.38 | 278.27 | 16.51 | 19.16 | 32.11 | 0.00 | |
| 1891 | 292.82 | 899.67 | 278.35 | 16.51 | 19.16 | 32.11 | 0.00 | |
| 1892 | 293.17 | 903.53 | 278.44 | 16.51 | 19.16 | 32.11 | 0.00 | |
| 1893 | 293.48 | 907.27 | 278.55 | 16.51 | 19.16 | 32.11 | 0.00 | |
| 1894 | 293.79 | 910.48 | 278.69 | 16.51 | 19.16 | 32.11 | 0.00 | |
| 1895 | 294.08 | 913.23 | 278.83 | 16.51 | 19.16 | 32.11 | 0.00 | |
| 1896 | 294.36 | 914.77 | 278.94 | 16.51 | 19.16 | 32.11 | 0.00 | |
| 1897 | 294.65 | 916.27 | 279.05 | 16.51 | 19.16 | 32.11 | 0.00 | |
| 1898 | 294.95 | 919.02 | 279.16 | 16.51 | 19.16 | 32.11 | 0.00 | |
| 1899 | 295.30 | 922.28 | 279.31 | 16.51 | 19.16 | 32.11 | 0.00 | |
| 1900 | 295.67 | 925.55 | 279.45 | 16.51 | 19.16 | 32.11 | 0.00 | |
| 1901 | 296.01 | 928.80 | 279.61 | 16.51 | 19.16 | 32.11 | 0.00 | |
| 1902 | 296.32 | 932.73 | 279.86 | 16.51 | 19.16 | 32.11 | 0.00 | |
| 1903 | 296.65 | 936.78 | 280.16 | 16.51 | 19.16 | 32.11 | 0.00 | |
| 1904 | 296.95 | 942.11 | 280.43 | 16.51 | 19.16 | 32.12 | 0.00 | |
| 1905 | 297.29 | 947.44 | 280.71 | 16.51 | 19.16 | 32.12 | 0.00 | |
| 1906 | 297.66 | 953.09 | 280.98 | 16.51 | 19.17 | 32.12 | 0.00 | |
| 1907 | 298.10 | 959.16 | 281.28 | 16.51 | 19.17 | 32.12 | 0.00 | |
| 1908 | 298.52 | 964.09 | 281.61 | 16.51 | 19.18 | 32.13 | 0.00 | |
| 1909 | 298.94 | 969.40 | 281.95 | 16.51 | 19.18 | 32.13 | 0.00 | |
| 1910 | 299.38 | 974.79 | 282.31 | 16.51 | 19.19 | 32.13 | 0.00 | |
| 1911 | 299.83 | 979.47 | 282.72 | 16.54 | 19.20 | 32.18 | 0.00 | |
| 1912 | 300.35 | 983.61 | 283.02 | 16.55 | 19.21 | 32.20 | 0.00 | |
| 1913 | 300.91 | 986.24 | 283.36 | 16.56 | 19.23 | 32.22 | 0.00 | |
| 1914 | 301.42 | 988.61 | 283.72 | 16.60 | 19.24 | 32.28 | 0.00 | |
| 1915 | 301.94 | 991.46 | 284.05 | 16.67 | 19.26 | 32.37 | 0.00 | |
| 1916 | 302.48 | 998.45 | 284.31 | 16.78 | 19.28 | 32.51 | 0.00 | |
| 1917 | 303.01 | 1,003.57 | 284.62 | 16.90 | 19.31 | 32.68 | 0.00 | |
| 1918 | 303.45 | 1,010.13 | 284.81 | 16.99 | 19.34 | 32.81 | 0.00 | |
| 1919 | 303.81 | 1,017.63 | 284.85 | 17.08 | 19.37 | 32.94 | 0.00 | |
| 1920 | 304.25 | 1,025.07 | 284.93 | 17.12 | 19.40 | 33.01 | 0.00 | |
| 1921 | 304.60 | 1,032.20 | 285.04 | 17.16 | 19.43 | 33.08 | 0.00 | |
| 1922 | 304.94 | 1,039.10 | 285.17 | 17.24 | 19.44 | 33.18 | 0.00 | |
| 1923 | 305.27 | 1,045.13 | 285.47 | 17.37 | 19.46 | 33.36 | 0.00 | |
| 1924 | 305.63 | 1,049.45 | 285.61 | 17.50 | 19.49 | 33.53 | 0.00 | |
| 1925 | 305.81 | 1,052.16 | 285.65 | 17.65 | 19.54 | 33.74 | 0.00 | |
| 1926 | 305.95 | 1,053.60 | 285.69 | 17.84 | 19.58 | 34.00 | 0.00 | |
| 1927 | 306.18 | 1,055.77 | 285.74 | 17.97 | 19.62 | 34.19 | 0.00 | |
| 1928 | 306.33 | 1,060.64 | 285.83 | 18.15 | 19.67 | 34.45 | 0.00 | |
| 1929 | 306.49 | 1,066.66 | 285.89 | 18.42 | 19.73 | 34.82 | 0.00 | |
| 1930 | 306.62 | 1,072.64 | 285.94 | 18.72 | 19.80 | 35.22 | 0.00 | |
| 1931 | 306.82 | 1,077.49 | 286.12 | 19.08 | 19.85 | 35.71 | 0.00 | |
| 1932 | 307.09 | 1,081.96 | 286.22 | 19.46 | 19.89 | 36.19 | 0.00 | |
| 1933 | 307.40 | 1,086.54 | 286.37 | 19.85 | 19.92 | 36.69 | 0.00 | |
| 1934 | 307.78 | 1,091.77 | 286.47 | 20.30 | 19.95 | 37.26 | 0.00 | |
| 1935 | 308.23 | 1,097.08 | 286.59 | 20.86 | 19.98 | 37.97 | 0.00 | |
| 1936 | 309.01 | 1,101.83 | 286.75 | 21.57 | 20.04 | 38.88 | 0.00 | |
| 1937 | 309.76 | 1,106.32 | 286.95 | 22.34 | 20.11 | 39.87 | 0.00 | |
| 1938 | 310.29 | 1,110.63 | 287.19 | 23.09 | 20.21 | 40.86 | 0.00 | |
| 1939 | 310.85 | 1,116.91 | 287.39 | 23.89 | 20.32 | 41.90 | 0.00 | |
| 1940 | 311.36 | 1,120.12 | 287.62 | 24.80 | 20.45 | 43.11 | 0.00 | |
| 1941 | 311.81 | 1,123.24 | 287.86 | 25.89 | 20.59 | 44.53 | 0.00 | |
| 1942 | 312.17 | 1,128.19 | 288.14 | 27.25 | 20.77 | 46.32 | 0.00 | |
| 1943 | 312.39 | 1,132.66 | 288.78 | 28.89 | 21.00 | 48.48 | 0.00 | |
| 1944 | 312.41 | 1,136.27 | 289.00 | 30.85 | 21.31 | 51.06 | 0.02 | |

| Years | CO$_2$ | CH$_4$ | N$_2$O | CFC-12-eq | HFC-134a-eq | CFC-11-eq | CFC-12 | Other |
|-------|--------|--------|--------|-----------|-------------|-----------|--------|-------|
| 1945 | 312.38 | 1,139.32 | 289.23 | 32.67 | 21.53 | 52.94 | 0.42 | |
| 1946 | 312.39 | 1,143.66 | 289.43 | 35.15 | 21.59 | 54.53 | 1.64 | |
| 1947 | 312.49 | 1,149.64 | 289.51 | 37.73 | 21.67 | 56.29 | 2.84 | |
| 1948 | 312.52 | 1,155.63 | 289.56 | 40.53 | 21.79 | 58.34 | 4.03 | |
| 1949 | 312.63 | 1,160.35 | 289.60 | 43.44 | 21.92 | 60.53 | 5.22 | |
| 1950 | 312.82 | 1,163.82 | 289.74 | 46.41 | 22.04 | 62.83 | 6.38 | |
| 1951 | 313.01 | 1,168.81 | 289.86 | 49.53 | 22.18 | 65.04 | 7.78 | |
| 1952 | 313.34 | 1,174.31 | 290.03 | 52.53 | 22.37 | 66.80 | 9.44 | |
| 1953 | 313.73 | 1,183.36 | 290.33 | 55.93 | 22.58 | 68.92 | 11.21 | |
| 1954 | 314.09 | 1,194.43 | 290.55 | 59.82 | 22.80 | 71.41 | 13.20 | |
| 1955 | 314.41 | 1,206.65 | 290.84 | 64.26 | 23.04 | 74.27 | 15.44 | |
| 1956 | 314.70 | 1,221.10 | 291.19 | 69.32 | 23.29 | 77.48 | 18.01 | |
| 1957 | 314.99 | 1,235.80 | 291.51 | 75.05 | 23.54 | 81.04 | 20.98 | |
| 1958 | 315.34 | 1,247.42 | 291.77 | 81.16 | 23.78 | 84.76 | 24.18 | |
| 1959 | 315.81 | 1,257.32 | 291.99 | 87.55 | 24.03 | 88.56 | 27.61 | |
| 1960 | 316.62 | 1,264.12 | 292.28 | 94.78 | 24.30 | 92.70 | 31.61 | |
| 1961 | 317.30 | 1,269.46 | 292.60 | 103.17 | 24.60 | 97.52 | 36.24 | |
| 1962 | 318.04 | 1,282.57 | 292.95 | 112.78 | 24.94 | 103.11 | 41.48 | |
| 1963 | 318.65 | 1,300.79 | 293.33 | 123.96 | 25.33 | 109.56 | 47.60 | |
| 1964 | 319.33 | 1,317.37 | 293.69 | 136.86 | 25.73 | 116.84 | 54.80 | |
| 1965 | 319.82 | 1,331.06 | 294.05 | 151.46 | 26.15 | 124.93 | 63.03 | |
| 1966 | 320.88 | 1,342.24 | 294.45 | 167.71 | 26.60 | 133.86 | 72.25 | |
| 1967 | 321.48 | 1,354.27 | 294.86 | 185.88 | 27.09 | 143.77 | 82.61 | |
| 1968 | 322.39 | 1,371.65 | 295.27 | 206.27 | 27.67 | 154.88 | 94.26 | |
| 1969 | 323.25 | 1,389.34 | 295.68 | 229.03 | 28.28 | 167.24 | 107.29 | |
| 1970 | 324.78 | 1,411.10 | 296.10 | 254.09 | 28.94 | 180.81 | 121.65 | |
| 1971 | 325.40 | 1,431.12 | 296.52 | 281.15 | 29.69 | 195.51 | 137.14 | |
| 1972 | 327.35 | 1,449.29 | 296.96 | 310.64 | 30.51 | 211.74 | 153.86 | |
| 1973 | 329.91 | 1,462.86 | 297.40 | 343.56 | 31.41 | 230.16 | 172.26 | |
| 1974 | 330.76 | 1,476.14 | 297.86 | 379.95 | 32.40 | 250.57 | 192.56 | |
| 1975 | 330.83 | 1,491.74 | 298.33 | 416.91 | 33.51 | 271.30 | 213.24 | |
| 1976 | 331.54 | 1,509.11 | 298.81 | 453.19 | 34.60 | 292.30 | 233.00 | |
| 1977 | 333.35 | 1,527.68 | 299.32 | 489.38 | 35.78 | 314.19 | 251.99 | |
| 1978 | 335.01 | 1,546.89 | 299.85 | 524.85 | 37.12 | 336.51 | 270.00 | |
| 1979 | 336.60 | 1,566.16 | 300.39 | 557.73 | 38.90 | 357.76 | 286.49 | |
| 1980 | 338.70 | 1,584.94 | 300.97 | 588.51 | 40.76 | 377.49 | 302.18 | |
| 1981 | 340.06 | 1,602.65 | 301.56 | 621.21 | 42.65 | 397.68 | 319.42 | |
| 1982 | 340.64 | 1,618.73 | 302.19 | 652.90 | 44.48 | 418.45 | 335.14 | |
| 1983 | 342.27 | 1,632.62 | 302.84 | 685.20 | 46.14 | 437.87 | 352.51 | |
| 1984 | 344.01 | 1,643.50 | 303.53 | 715.67 | 47.82 | 458.80 | 366.80 | |
| 1985 | 345.46 | 1,655.91 | 304.25 | 753.45 | 49.69 | 486.19 | 383.27 | |
| 1986 | 346.90 | 1,668.79 | 305.00 | 789.53 | 51.62 | 508.22 | 402.41 | |
| 1987 | 348.77 | 1,683.75 | 305.79 | 831.33 | 53.55 | 535.08 | 423.35 | |
| 1988 | 351.28 | 1,693.94 | 306.62 | 879.94 | 55.70 | 564.26 | 449.32 | |
| 1989 | 352.89 | 1,705.63 | 307.83 | 921.47 | 57.93 | 593.68 | 468.07 | |
| 1990 | 354.07 | 1,717.40 | 308.68 | 953.43 | 60.21 | 616.35 | 482.76 | |
| 1991 | 355.35 | 1,729.33 | 309.23 | 979.87 | 62.66 | 636.82 | 493.78 | |
| 1992 | 356.23 | 1,740.14 | 309.73 | 1,001.60 | 65.13 | 650.21 | 505.87 | |
| 1993 | 356.92 | 1,743.10 | 310.10 | 1,012.33 | 67.79 | 657.53 | 511.99 | |
| 1994 | 358.25 | 1,748.62 | 310.81 | 1,021.09 | 70.74 | 662.45 | 518.21 | |
| 1995 | 360.24 | 1,755.23 | 311.28 | 1,029.02 | 74.60 | 666.66 | 524.66 | |
| 1996 | 362.00 | 1,757.19 | 312.30 | 1,038.98 | 79.14 | 673.40 | 531.41 | |
| 1997 | 363.25 | 1,761.50 | 313.18 | 1,041.17 | 84.42 | 674.97 | 534.96 | |
| 1998 | 365.93 | 1,770.29 | 313.91 | 1,046.23 | 90.45 | 681.59 | 537.67 | |
| 1999 | 367.84 | 1,778.20 | 314.71 | 1,048.71 | 96.94 | 685.59 | 540.14 | |

| Years | CO$_2$ | CH$_4$ | N$_2$O | CFC-12-eq | HFC-134a-eq | CFC-11-eq | CFC-12 | Other |
|---|---|---|---|---|---|---|---|---|
| 2000 | 369.12 | 1,778.01 | 315.76 | 1,051.12 | 104.52 | 690.46 | 542.38 | |
| 2001 | 370.67 | 1,776.53 | 316.49 | 1,052.91 | 113.35 | 697.10 | 543.20 | |
| 2002 | 372.83 | 1,778.96 | 317.10 | 1,053.74 | 121.44 | 702.52 | 543.66 | |
| 2003 | 375.41 | 1,783.59 | 317.73 | 1,053.52 | 129.89 | 707.84 | 543.35 | |
| 2004 | 376.99 | 1,784.23 | 318.36 | 1,053.30 | 139.31 | 713.98 | 542.85 | |
| 2005 | 378.91 | 1,783.36 | 319.13 | 1,053.46 | 150.43 | 721.88 | 542.15 | |
| 2006 | 381.01 | 1,783.42 | 319.93 | 1,053.71 | 160.64 | 730.31 | 540.65 | |
| 2007 | 382.60 | 1,788.95 | 320.65 | 1,053.94 | 171.15 | 739.81 | 538.43 | |
| 2008 | 384.74 | 1,798.42 | 321.57 | 1,054.80 | 181.99 | 750.11 | 536.33 | |
| 2009 | 386.28 | 1,802.10 | 322.28 | 1,054.17 | 191.13 | 758.10 | 533.78 | |
| 2010 | 388.72 | 1,807.85 | 323.14 | 1,054.37 | 203.07 | 768.76 | 531.28 | |
| 2011 | 390.94 | 1,813.07 | 324.16 | 1,053.45 | 216.23 | 779.12 | 528.53 | |
| 2012 | 393.02 | 1,815.26 | 325.00 | 1,051.97 | 227.84 | 787.77 | 525.83 | |
| 2013 | 395.72 | 1,822.58 | 325.92 | 1,051.74 | 244.88 | 801.30 | 523.11 | |
| 2014 | 397.55 | 1,831.47 | 326.99 | 1,049.51 | 257.06 | 809.19 | 520.58 | |

**Table 7 – Global- and annual mean GHG surface concentrations for year 2011 and 2014, including a comparison to 2011 NOAA, AGAGE and UCI estimates – as provided in IPCC AR5 WG1. Unit is ppt, unless otherwise stated.**

| Rank of Abundance | Species | 2014 CMIP6 (This Study) | 2011 CMIP6 (This Study) | 2011 UCI | 2011 SIO b/AGAGE | 2011 NOAA |
|---|---|---|---|---|---|---|
| 1 | $CO_2$ (ppm) | 397.55 | 390.94 | | 390.48 ± 0.28 | 390.44 ± 0.16 |
| 2 | $CH_4$ (ppb) | 1831.47 | 1813.07 | 1798.1 ± 0.6 | 1803.1 ± 4.8 | 1803.2 ± 1.2 |
| 3 | $N_2O$ (ppb) | 326.99 | 324.16 | | 324.0 ± 0.1 | 324.3 ± 0.1 |
| 4 | $CH_3Cl$ | 539.54 | 534.17 | | | |
| 5 | CFC-12 | 520.58 | 528.53 | 525.3 ± 0.8 | 529.5 ± 0.2 | 527.4 ± 0.4 |
| 6 | CFC-11 | 233.08 | 238.25 | 237.9 ± 0.8 | 236.9 ± 0.1 | 238.5 ± 0.2 |
| 7 | HCFC-22 | 229.54 | 214.56 | 209.0 ± 1.2 | 213.4 ± 0.8 | 213.2 ± 1.2 |
| 8 | $CCl_4$ | 83.07 | 86.06 | 87.8 ± 0.6 | 85.0 ± 0.1 | 86.5 ± 0.3 |
| 9 | $CF_4$ | 81.09 | 79.04 | | 79.0 ± 0.1 | |
| 10 | HFC-134a | 80.52 | 62.85 | 63.4 ± 0.9 | 62.4 ± 0.3 | 63.0 ± 0.6 |
| 11 | CFC-113 | 72.71 | 74.64 | 74.9 ± 0.6 | 74.29 ± 0.06 | 74.40 ± 0.04 |
| 12 | $CH_2Cl_2$ | 36.35 | 29.49 | | | |
| 13 | HFC-23 | 26.89 | 24.13 | | 24.0 ± 0.3 | |
| 14 | HCFC-141b | 23.81 | 21.56 | 20.8 ± 0.5 | 21.38 ± 0.09 | 21.4 ± 0.2 |
| 15 | HCFC-142b | 22.08 | 21.35 | 21.0 ± 0.5 | 21.35 ± 0.06 | 21.0 ± 0.1 |
| 16 | CFC-114 | 16.31 | 16.36 | | | |
| 17 | HFC-125 | 15.36 | 10.46 | | 9.58 ± 0.04 | |
| 18 | HFC-143a | 15.25 | 11.92 | | 12.04 ± 0.07 | |
| 19 | $CHCl_3$ | 9.90 | 8.95 | | | |
| 20 | CFC-115 | 8.43 | 8.39 | | | |
| 21 | HFC-32 | 8.34 | 5.17 | | | |
| 22 | $SF_6$ | 8.22 | 7.31 | | 7.26 ± 0.02 | 7.31 ± 0.02 |
| 23 | HFC-152a | 7.73 | 7.89 | | 6.4 ± 0.1 | |
| 24 | $CH_3Br$ | 6.69 | 7.11 | | | |
| 25 | $C_2F_6$ | 4.40 | 4.17 | | 4.16 ± 0.02 | |
| 26 | Halon-1211 | 3.75 | 4.05 | | | |
| 27 | $CH_3CCl_3$ | 3.68 | 6.31 | 6.8 ± 0.6 | 6.3 ± 0.1 | 6.35 ± 0.07 |
| 28 | Halon-1301 | 3.30 | 3.23 | | | |
| 29 | HFC-245fa | 2.05 | 1.56 | | | |
| 30 | $SO_2F_2$ | 2.04 | 1.74 | | | |
| 31 | c-$C_4F_8$ | 1.34 | 1.23 | | | |
| 32 | $NF_3$ | 1.24 | 0.83 | | | |
| 33 | HFC-227ea | 1.01 | 0.74 | | | |
| 34 | HFC-365mfc | 0.77 | 0.56 | | | |
| 35 | $C_3F_8$ | 0.60 | 0.56 | | | |
| 36 | Halon-2402 | 0.43 | 0.45 | | | |
| 37 | $C_6F_{14}$ | 0.28 | 0.27 | | | |
| 38 | HFC-43-10mee | 0.25 | 0.22 | | | |
| 39 | $C_4F_{10}$ | 0.18 | 0.17 | | | |
| 40 | HFC-236fa | 0.13 | 0.10 | | | |
| 41 | $C_5F_{12}$ | 0.13 | 0.12 | | | |
| 42 | $C_7F_{16}$ | 0.13 | 0.12 | | | |
| 43 | $C_8F_{18}$ | 0.09 | 0.09 | | | |

**Table 8 – picontrol: Global- and annual-mean surface concentrations for the picontrol CMIP6 experiment. The hemispheric and latitudinally resolved concentrations for 43 GHGs and three aggregate equivalent concentrations are provided in the accompanying historical run dataset for the year 1850. The complexity reduction options for capturing all GHGs with fewer species than 43 are indicated in the Table as Option 1, Option 2, and Option 3, with 'x' denoting relevant columns under each option.**

| Years | $CO_2$ | $CH_4$ | $N_2O$ | CFC-12-eq | HFC-134a-eq | CFC-11-eq | CFC-12 | Other |
|---|---|---|---|---|---|---|---|---|
| Option 1 | x | x | x | | | | x | x |
| Option 2 | x | x | x | | | x | x | |
| Option 3 | x | x | x | x | x | | | |
| Units: | ppm | ppb | ppb | ppt | ppt | ppt | ppt | |
| 1850 | 284.317 | 808.25 | 273.02 | 16.51 | 19.15 | 32.11 | 0.00 | All or a subset of other 39 individual gases, available in Supplementary |

**Table 9 – 1pctCO2: Global-mean annual-mean surface CO$_2$ concentrations for idealized CMIP6 experiments 1pctCO2. All other gases, as in picontrol run (see Table 8). The value 284.317 ppm with 3-digit precision in year 1850 is increased by 1% per year.**

| YEAR | CO$_2$ (PPM) | YEAR | CO$_2$ (PPM) | YEAR | CO$_2$ (PPM) | YEAR | CO$_2$ (PPM) | YEAR | CO$_2$ (PPM) |
|---|---|---|---|---|---|---|---|---|---|
| 1850 | 284.32 | 1900 | 467.60 | 1950 | 769.02 | 2000 | 1264.76 | 2050 | 2080.07 |
| 1851 | 287.16 | 1901 | 472.27 | 1951 | 776.71 | 2001 | 1277.41 | 2051 | 2100.87 |
| 1852 | 290.03 | 1902 | 477.00 | 1952 | 784.48 | 2002 | 1290.18 | 2052 | 2121.88 |
| 1853 | 292.93 | 1903 | 481.77 | 1953 | 792.33 | 2003 | 1303.09 | 2053 | 2143.10 |
| 1854 | 295.86 | 1904 | 486.58 | 1954 | 800.25 | 2004 | 1316.12 | 2054 | 2164.53 |
| 1855 | 298.82 | 1905 | 491.45 | 1955 | 808.25 | 2005 | 1329.28 | 2055 | 2186.17 |
| 1856 | 301.81 | 1906 | 496.36 | 1956 | 816.34 | 2006 | 1342.57 | 2056 | 2208.03 |
| 1857 | 304.83 | 1907 | 501.33 | 1957 | 824.50 | 2007 | 1356.00 | 2057 | 2230.11 |
| 1858 | 307.87 | 1908 | 506.34 | 1958 | 832.74 | 2008 | 1369.56 | 2058 | 2252.42 |
| 1859 | 310.95 | 1909 | 511.40 | 1959 | 841.07 | 2009 | 1383.25 | 2059 | 2274.94 |
| 1860 | 314.06 | 1910 | 516.52 | 1960 | 849.48 | 2010 | 1397.08 | 2060 | 2297.69 |
| 1861 | 317.20 | 1911 | 521.68 | 1961 | 857.98 | 2011 | 1411.06 | 2061 | 2320.67 |
| 1862 | 320.38 | 1912 | 526.90 | 1962 | 866.56 | 2012 | 1425.17 | 2062 | 2343.87 |
| 1863 | 323.58 | 1913 | 532.17 | 1963 | 875.22 | 2013 | 1439.42 | 2063 | 2367.31 |
| 1864 | 326.82 | 1914 | 537.49 | 1964 | 883.97 | 2014 | 1453.81 | 2064 | 2390.98 |
| 1865 | 330.08 | 1915 | 542.87 | 1965 | 892.81 | 2015 | 1468.35 | 2065 | 2414.89 |
| 1866 | 333.38 | 1916 | 548.29 | 1966 | 901.74 | 2016 | 1483.03 | 2066 | 2439.04 |
| 1867 | 336.72 | 1917 | 553.78 | 1967 | 910.76 | 2017 | 1497.86 | 2067 | 2463.43 |
| 1868 | 340.09 | 1918 | 559.31 | 1968 | 919.87 | 2018 | 1512.84 | 2068 | 2488.07 |
| 1869 | 343.49 | 1919 | 564.91 | 1969 | 929.07 | 2019 | 1527.97 | 2069 | 2512.95 |
| 1870 | 346.92 | 1920 | 570.56 | 1970 | 938.36 | 2020 | 1543.25 | 2070 | 2538.08 |
| 1871 | 350.39 | 1921 | 576.26 | 1971 | 947.74 | 2021 | 1558.68 | 2071 | 2563.46 |
| 1872 | 353.89 | 1922 | 582.03 | 1972 | 957.22 | 2022 | 1574.27 | 2072 | 2589.09 |
| 1873 | 357.43 | 1923 | 587.85 | 1973 | 966.79 | 2023 | 1590.01 | 2073 | 2614.98 |
| 1874 | 361.01 | 1924 | 593.72 | 1974 | 976.46 | 2024 | 1605.91 | 2074 | 2641.13 |
| 1875 | 364.62 | 1925 | 599.66 | 1975 | 986.22 | 2025 | 1621.97 | 2075 | 2667.55 |
| 1876 | 368.26 | 1926 | 605.66 | 1976 | 996.08 | 2026 | 1638.19 | 2076 | 2694.22 |
| 1877 | 371.95 | 1927 | 611.71 | 1977 | 1006.04 | 2027 | 1654.57 | 2077 | 2721.16 |
| 1878 | 375.67 | 1928 | 617.83 | 1978 | 1016.11 | 2028 | 1671.12 | 2078 | 2748.38 |
| 1879 | 379.42 | 1929 | 624.01 | 1979 | 1026.27 | 2029 | 1687.83 | 2079 | 2775.86 |
| 1880 | 383.22 | 1930 | 630.25 | 1980 | 1036.53 | 2030 | 1704.71 | 2080 | 2803.62 |
| 1881 | 387.05 | 1931 | 636.55 | 1981 | 1046.89 | 2031 | 1721.76 | 2081 | 2831.65 |
| 1882 | 390.92 | 1932 | 642.92 | 1982 | 1057.36 | 2032 | 1738.97 | 2082 | 2859.97 |
| 1883 | 394.83 | 1933 | 649.35 | 1983 | 1067.94 | 2033 | 1756.36 | 2083 | 2888.57 |
| 1884 | 398.78 | 1934 | 655.84 | 1984 | 1078.62 | 2034 | 1773.93 | 2084 | 2917.46 |
| 1885 | 402.76 | 1935 | 662.40 | 1985 | 1089.40 | 2035 | 1791.67 | 2085 | 2946.63 |
| 1886 | 406.79 | 1936 | 669.02 | 1986 | 1100.30 | 2036 | 1809.58 | 2086 | 2976.10 |
| 1887 | 410.86 | 1937 | 675.71 | 1987 | 1111.30 | 2037 | 1827.68 | 2087 | 3005.86 |
| 1888 | 414.97 | 1938 | 682.47 | 1988 | 1122.41 | 2038 | 1845.95 | 2088 | 3035.92 |
| 1889 | 419.12 | 1939 | 689.29 | 1989 | 1133.64 | 2039 | 1864.41 | 2089 | 3066.28 |
| 1890 | 423.31 | 1940 | 696.19 | 1990 | 1144.97 | 2040 | 1883.06 | 2090 | 3096.94 |
| 1891 | 427.54 | 1941 | 703.15 | 1991 | 1156.42 | 2041 | 1901.89 | 2091 | 3127.91 |
| 1892 | 431.82 | 1942 | 710.18 | 1992 | 1167.99 | 2042 | 1920.91 | 2092 | 3159.19 |
| 1893 | 436.14 | 1943 | 717.28 | 1993 | 1179.67 | 2043 | 1940.12 | 2093 | 3190.78 |
| 1894 | 440.50 | 1944 | 724.46 | 1994 | 1191.46 | 2044 | 1959.52 | 2094 | 3222.69 |
| 1895 | 444.90 | 1945 | 731.70 | 1995 | 1203.38 | 2045 | 1979.11 | 2095 | 3254.91 |
| 1896 | 449.35 | 1946 | 739.02 | 1996 | 1215.41 | 2046 | 1998.90 | 2096 | 3287.46 |
| 1897 | 453.84 | 1947 | 746.41 | 1997 | 1227.57 | 2047 | 2018.89 | 2097 | 3320.34 |
| 1898 | 458.38 | 1948 | 753.87 | 1998 | 1239.84 | 2048 | 2039.08 | 2098 | 3353.54 |
| 1899 | 462.97 | 1949 | 761.41 | 1999 | 1252.24 | 2049 | 2059.47 | 2099 | 3387.08 |
| | | | | | | | | 2100 | 3420.95 |

**Table 10 – abrupt4x: Global- and annual-mean surface concentrations for the idealized abrupt4x CMIP6 experiment. The hemispheric and latitudinally resolved concentrations for 43 GHGs and three aggregate equivalent concentrations are provided in the accompanying historical run dataset, with the 1850 $CO_2$ concentration of 284.317 being multiplied by four. The complexity reduction options for capturing all GHGs with fewer species than 43 are indicated in the Table as Option 1, Option 2, and Option 3, with 'x' denoting relevant columns under each option.**

| Years | $CO_2$ | $CH_4$ | $N_2O$ | CFC-12-eq | HFC-134a-eq | CFC-11-eq | CFC-12 | Other |
|---|---|---|---|---|---|---|---|---|
| Option 1 | x | x | x | | | | x | x |
| Option 2 | x | x | x | | | x | x | |
| Option 3 | x | x | x | x | x | | | |
| Units: | ppm | ppb | ppb | ppt | ppt | ppt | ppt | |
| 0 -150 | 1137.268 | 808.25 | 273.02 | 16.51 | 19.15 | 32.11 | 0.00 | All or a subset of other 39 individual gases, available in Supplementary |

**Table 11 – Exponents 's' to estimate vertical gradient of concentrations for gases with stratospheric sinks in the stratospheric column – depending on the latitude 'lat'. See text. For HFC-134a and other species with stratospheric lifetimes shorter than 30 years, the $CH_4$ exponent parameterization can be used as approximation. This exponent scale parameterization is taken from the CESM model, implemented by J. Kiehl.**

| | TROPICS AND MID-LATITUDES ABS(LAT)<45° | MID TO HIGH LATITUDES, ABS(LAT)≥45° |
|---|---|---|
| $CH_4$ | 0.2353 | 0.2353 + 0.0225489 × (abs(lat) - 45); |
| $N_2O$ | 0.3478 + 0.00116 × abs(lat) | 0.40 + 0.013333 × (abs(lat) - 45) |
| CFC-11 | 0.7273 + 0.00606 × abs(lat) | 1.00 + 0.013333 × (abs(lat) - 45); |
| CFC-12 | 0.4000 + 0.00222 × abs(lat) | 0.50 + 0.024444 × (abs(lat) - 45) |

**Table 12 – Description of data labels shown in Factsheets, namely Fig. 9, Fig. 11, Fig. 12, and Appendix A with Fig. 20 to Fig. 59.**

| Label | Gases | Description / Source |
|---|---|---|
| **NOAA_surface_flask** | $CO_2$ | Atmospheric Carbon Dioxide Dry Air Mole Fractions from the NOAA ESRL Carbon Cycle Cooperative Global Air Sampling Network, 1968-2014, Version: 2015-08-03Surface flask, available at data ftp://aftp.cmdl.noaa.gov/data/trace_gases/co2/flask/surface/ (Dlugokencky, 2015b) |
| **NOAA_surface_insitu** | $CO_2$ | Atmospheric Carbon Dioxide Dry Air Mole Fractions from quasi-continuous measurements at Barrow, Alaska; Mauna Loa, Hawaii; American Samoa; and South Pole, 1973-2013; National Oceanic and Atmospheric Administration (NOAA); Earth System Research Laboratory (ESRL), Global Monitoring Division (GMD), Carbon Cycle Greenhouse Gases (CCGG); Version: 2014-11-10, available at: ftp://aftp.cmdl.noaa.gov/data/trace_gases/co2/in-situ/surface/ (NOAA ESRL GMD, 2014a, b, c, d) |
| **NOAA_surface_flask** | $CH_4$ | Atmospheric Methane Dry Air Mole Fractions from the NOAA ESRL GMD Carbon Cycle Cooperative Global Air, Sampling Network, 1983-2014, Fileversions: 2015-08-03, available at: ftp://aftp.cmdl.noaa.gov/data/trace_gases/ch4/flask/ (Dlugokencky, 2015a) |
| **HATS_global_combined** | $N_2O$, $CCl_4$, CFC-11, CFC-113, CFC-12, $SF_6$ | Combined data from the NOAA/ESRL Global Monitoring Division and two or more measurement programs: Wed, Aug 19, 2015 2:40:55 PM, available at: ftp://ftp.cmdl.noaa.gov/hats/n2o/combined/HATS_global_N2O.txt, ftp://ftp.cmdl.noaa.gov/hats/cfcs/cfc113/combined/HATS_global_F113.txt ftp://ftp.cmdl.noaa.gov/hats/cfcs/cfc11/combined/HATS_global_F11.txt ftp://ftp.cmdl.noaa.gov/hats/cfcs/cfc12/combined/HATS_global_F12.txt ftp://ftp.cmdl.noaa.gov/hats/sf6/combined/HATS_global_SF6.txt ftp://ftp.cmdl.noaa.gov/hats/solvents/CCl4/combined/HATS_global_CCl4.txt |
| **Montzka_NOAA_GMD** | $CCl_4$, CFC-11, CFC-113, $CH_3CCl_3$, $CH_3Br$, $CH_3Cl$, $CH_2Cl_2$, HCFC-22, HCFC-141b, HCFC-142b, HFC-134a, HFC-152a, HFC-32, HFC-125, HFC-143a, HFC-365mfc, HFC-227ea, Halon-1211, Halon-1301, Halon-2402 | Flask data provided from the Global Monitoring Division of the National Oceanic and Atmospheric Administration's Earth System Research Laboratory (NOAA/ESRL/GMD) as a result of analysis on gas chromatography with mass spectrometry instrumentation. Principal investigators S. Montzka and James W. Elkins. Version 13 Nov. 2015. Data available at: ftp://ftp.cmdl.noaa.gov/hats/cfcs/cfc113/flasks/GCMS/CFC113_GCMS_flask.txt |

| Label | Gases | Description / Source |
|---|---|---|
| | | ftp://ftp.cmdl.noaa.gov/hats/solvents/CH3CCl3/flasks/GCMS/CH3CCL3_GCMS_flask.txt |
| | | ftp://ftp.cmdl.noaa.gov/hats/methylhalides/ch3br/flasks/CH3BR_GCMS_flask.txt |
| | | ftp://ftp.cmdl.noaa.gov/hats/methylhalides/ch3cl/flasks/CH3Cl_GCMS_flask.txt |
| | | ftp://ftp.cmdl.noaa.gov/hats/solvents/CH2Cl2/flasks/ch2cl2_GCMS_flask.txt |
| | | ftp://ftp.cmdl.noaa.gov/hats/hcfcs/hcfc22/flasks/HCFC22_GCMS_flask.txt |
| | | ftp://ftp.cmdl.noaa.gov/hats/hcfcs/hcfc141b/HCFC141B_GCMS_flask.txt |
| | | ftp://ftp.cmdl.noaa.gov/hats/hcfcs/hcfc142b/flasks/HCFC142B_GCMS_flask.txt |
| | | ftp://ftp.cmdl.noaa.gov/hats/hfcs/hfc134a_GCMS_flask.txt |
| | | ftp://ftp.cmdl.noaa.gov/hats/hfcs/hf152a_GCMS_flask.txt |
| | | ftp://ftp.cmdl.noaa.gov/hats/hfcs/HFC-32_M2_MS_flask.txt |
| | | ftp://ftp.cmdl.noaa.gov/hats/hfcs/HFC-125_M2_MS_flask.txt |
| | | ftp://ftp.cmdl.noaa.gov/hats/hfcs/HFC-143a_M2_MS_flask.txt |
| | | ftp://ftp.cmdl.noaa.gov/hats/hfcs/HFC-365mfc_GCMS_flask.txt |
| | | ftp://ftp.cmdl.noaa.gov/hats/hfcs/HFC-227ea_GCMS_flask.txt |
| | | ftp://ftp.cmdl.noaa.gov/hats/halons/flasks/HAL1211_GCMS_flask.txt |
| | | ftp://ftp.cmdl.noaa.gov/hats/halons/flasks/H-1301_M2_MS_flask.txt |
| | | ftp://ftp.cmdl.noaa.gov/hats/halons/flasks/HAL2402_GCMS_flask.txt |
| AGAGE_gc-md_monthly | CFC-11, CFC-12, $CH_3CCl_3$, $CCl_4$, $N_2O$, CFC-113, $CH_4$, $CHCl_3$, | Chemical species measured by AGAGE GC-ECD/FID/MRD system. Version 20 June 2015; Data available at: http://agage.eas.gatech.edu/data_archive/agage/gc-md/monthly/ (Cunnold et al., 2002; Cunnold et al., 1997; Fraser et al., 1996; O'Doherty et al., 2001; Prinn et al., 1990; Prinn et al., 2005; Prinn et al., 2001; Reimann et al., 2005; Simmonds et al., 1998) |
| AGAGE_gc-ms_monthly | HFC-134a, HCFC-22, HCFC-141b, HCFC-142b, $CH_3Cl$, $CH_3Br$, Halon-1211, Halon-1301, HFC-152a, $CH_2Cl2$, $CHClCCl_3$, $CCl_2CCl_2$ | Chemical compounds measured by AGAGE GC-MS (ADS) system. Version 20 June 2015; Data available at: http://agage.eas.gatech.edu/data_archive/agage/gc-ms/monthly/ (Cox et al., 2003; Miller et al., 1998; O'Doherty et al., 2004; Simmonds et al., 2004) |
| AGAGE_gc-ms-medusa_monthly | CFC-11, CFC-12, CFC-113, CFC-114, CFC-115, HCFC-22, HCFC-141b, HCFC-142b, HFC-125, HFC-134a, HFC-152a, HFC-365mfc, HFC-23, HFC-4310mee, Halon-1211, Halon-1301, Halon-2402, $CH_3Cl$, $CH_2Cl_2$, $CHCl_3$, $CH_3Br$, $CH_3CCl_3$, $CCl_4$, $SF_6$, $SO_2F_2$, $NF_3$, PFC-14, PFC-116, PFC-218, HFC-32, HFC-143a, HFC-227ea HFC-236fa HFC-245fa | Chemical compounds measured by Medusa GCMS system. Version 20 June 2015; Data available at: http://agage.eas.gatech.edu/data_archive/agage/gc-ms-medusa/monthly/ (Prinn et al., 2000a) |

| Label | Gases | Description / Source |
|---|---|---|
| **Montzka - NOAA ODS update 7/2015** | HCFC-22, CFC-113, CFC-11, HCFC-141b, CCl4, CFC-12, HCFC-142b, $CH_3CCl_3$, H-1211, H-1301, H2402, $CH_3Br$, HFC-134a, HFC-152a, HFC-143a, HFC-125, HFC-32, HFC-365mfc, HFC-227ea | Data from 7/2014 update of NOAA compilation of monthly global mean concentrations, made available on web as '2015 update total Cl Br & F July update.xls' by S. Montzka at: ftp://ftp.cmdl.noaa.gov/hats/Total_Cl_Br/. The substances HCFC-22, CFC-113, CFC-11, HCFC-141b, CCl4, CFC-12, HCFC-142b, $CH_3CCl_3$, Halon-1211, Halon-1301, are Halon-2402, are updated from data displayed in Figure 1 in Montzka et al. (1999), with $CH_3Br$ data published in Montzka et al. (2003) and with HFCs data published in Montzka et al. (2015). |
| **Martinerie-2010** | $SF_6$, CFC-11, CFC-12, CFC-113, $CCl_4$, $CH_3CCl_3$, HFC-134a | Monthly high-latitude NH data by Patricia Martinerie, made available as supplementary by Buizert et al. (2012) in files SCENARIO_NEEM08_XX.txt |
| **Petrenko-2010** | $CO_2$, $CH_4$ | Monthly high-latitude NH data by Vas Patrenko, made available as supplementary by Buizert et al. (2012) in files SCENARIO_NEEM08_CO2.txt and SCENARIO_NEEM08_CH4.txt |
| **WDCGG (2015)** | $CO_2$, $CH_4$, $N_2O$ | Data synthesis as available from the World Data Centre of Greenhouse Gas Emissions (Tsutsumi, 2009), available at: http://ds.data.jma.go.jp/gmd/wdcgg/. Version: co2_monthly_20151109.csv, ch4_monthly_20151109.csv and n2o_monthly_20151109.csv |
| **NOAA MBL** | $CO_2$, $CH_4$ | NOAA Greenhouse Gas Marine Boundary Layer Reference, derived from atmospheric carbon dioxide, methane and nitrous oxide concentrations, from the NOAA ESRL Carbon Cycle Cooperative Global Air, Sampling Network, available at http://www.esrl.noaa.gov/gmd/ccgg/mbl/ for CO2 and CH4. Zonal means for SH and NH, as well as global means. File creation dates: 2016-02-11 |
| **CMIP5 hist.** | Many | The global-mean annual average concentrations that were used as default recommendation for concentration-driven runs in the CMIP5 experiment (Meinshausen et al., 2011) |
| **CMIP5 ctrl.** | Many | The global-mean annual average concentrations in 1850 that were recommended as picontrol concentrations in the CMIP5 experiment (Meinshausen et al., 2011). |
| **Firn – Montzka-(2009)** | CFC-12. HFC-134a, HCFC-22, and $CH_3CCl_3$ | "Southern Hemisphere atmospheric trace-gas histories used in the analysis of firn air" data compiled by Montzka in 2009 (available at ftp://ftp.cmdl.noaa.gov/hats/firnair/ in file "SH Atmosphere Trace Gas Histories.xls"), based on several earlier studies (Butler et al., 1999; Elkins et al., 1993; Montzka et al., 1993; Montzka et al., 1996; Montzka et al., 2000; Prinn et al., 2005), and e.g. reported in Aydin et al. (2010) for CFC-12 and underlying (Montzka et al., 2010). |
| | | |
| **WMO (2014)** | CFC-11, CFC-12, CFC-113, CFC-114, CFC-115, CCl4, $CH_3CCl_3$, HCFC-22, HCFC-141b, HCFC-142b, Halon-1211, Halon-1202, Halon-1301, Halon-2402, $CH_3Br$, $CH_3Cl$ | Data from Table 5A2 in the 2014 Ozone Assessment (WMO, 2014), starting with 5-year intervals from 1955 to 1980 then annually. We interpolated the data to annual values using a local polynomial regression between 1955-1980. |

| Label | Gases | Description / Source |
|---|---|---|
| WMO2014/AGAGE 'late'/'early' | HFC125, HFC134a, HFC152a, HFC143a, HFC32, HFC245fa, HFC365mfc, HFC227ea, HFC236fa, $CF_4$, HFC23, $C_2F_6$, $C_3F_8$, $SF_6$, $SO_2F_2$, $NF_3$ | The network average global-mean mole fractions from the AGAGE network as shown in the WMO (2014) Ozone Assessment Report. (WMO, 2014) |
| WMO2014/NOAA | HFC-134a, HFC-152a, $SF_6$ | NOAA global-mean annual average time series as shown in WMO Ozone Assessment Report (WMO, 2014). |
| WMO2014/PFC | $C_4F_{10}$, $C_5F_{12}$, $C_6F_{14}$, $C_7F_{16}$, $C_8F_{18}$ | PFC data compiled and shown in WMO Ozone Assessment Report (WMO, 2014) |
| AGAGE – Global Monthly Average | HFC-23, HFC-125, HFC-134a, HFC-152a, HFC-227ea, HFC-236fa, HFC-245fa, HFC-365mfc, HCFC-22, HCFC-141b, HCFC-142b, H-1211, H-1301, $CH_3Br$, $CH_3Cl$, $CH_2Cl_2$, $CCl_2CCl_2$, $CHClCCl_2$, $SF_6$, $SO_2F_2$, PFC-14, PFC-116, PFC-218, CFC-113, CFC-114, CFC-115, HFC-4310mee | Monthly global means of baseline data derived from AGAGE measurements based on AGAGE GC-MS/Medusa measurements (from 2004 to current) from file global_mean_ms.txt available at: http://agage.eas.gatech.edu/data_archive/global_mean/ |
| Binned annual observations | All | These are the monthly averages for each 15-degree zonal mean derived from the analysed station data points (with three-digit station names provided in the top left corner of panel f of each factsheet). An "n/a" indication behind the latitude indicator means that not enough raw station data points were available to create zonal means for that latitude. The estimate of the latitudinal gradient is then based on the remainder available latitudinal bands. |
| OTHER LABELS, NAMELY: MONTZKA ET AL. (2015) VELDERS ET AL. (2014) MUEHLE ET AL. (2010) TRUDINGER ET AL. (2016) IVY ET AL. (2012) WORTON (2007) BUTLER ET AL. (1999) ARNOLD ET AL. (2013) ARNOLD ET AL. (2014) | Various | See respective literature studies (Arnold et al., 2013; Arnold et al., 2014; Butler et al., 1999; Ivy et al., 2012; Montzka et al., 2015; Mühle et al., 2010; Newland et al., 2013; Oram et al., 2012; Trudinger et al., 2016a; Velders and Daniel, 2014; Vollmer et al., 2016; Walker et al., 2000; Worton et al., 2007). Note, the $CCl_4$ data by Walker et al. is used as 1910 to 1950 amendment to the Velders and Daniel (2014) timeseries. |

| Label | Gases | Description / Source |
|---|---|---|
| **VOLLMER ET AL (2016)** | | |
| **ORAM ET AL. (2012)** | | |
| **WALKER ET AL. (2000)** | | |
| **Newland et al. (2013)** | | |

## 12 Figure Captions

**Fig. 1 -** Data flow diagram of how historical GHG concentrations are derived in this study. See text.

**Fig. 2 -** Availability of instrumental carbon dioxide data from 1968 to 2015 from the NOAA ESRL network, shown as data samples per month, per latitudinal band (panels a to l) and per longitudinal bin within each latitudinal band.

**Fig. 3 -** Availability of instrumental $CH_4$ data from 1983 to 2015 from the AGAGE and NOAA ESRL networks, shown as data samples per month, per latitudinal band (panels a to l) and per longitudinal bin within each latitudinal band.

**Fig. 4 -** Availability of instrumental $N_2O$ data from 1983 to 2015 from the AGAGE and NOAA ESRL networks, shown as data samples per month, per latitudinal band (panels a to l) and per longitudinal bin within each latitudinal band.

**Fig. 5 –** Comparison of various scaling options for the change of seasonality of $CO_2$ concentrations over time. The first EOF of the residual fields of observations minus the mean 1984-2014 $CO_2$ seasonality (Fig. 9 a.2) is scaled with an EOF score. Before 1984, this EOF score is regressed against a composite of global-mean $CO_2$ concentrations and global-mean surface air temperatures (see text and panel b). Alternative regressors include global-mean $CO_2$ concentrations (panel a), lagged averages of monthly global-mean surface air temperatures (panel c) and raw global-mean annual average surface air temperatures (HadCRUT4v) (Morice et al., 2012) (panel d). The regressed EOF score back in time is shown in panel e. A comparison to the first $CO_2$ measurements of higher northern latitudes at so-called Station P (STP) and Point Barrow in Alaska (PTB), where the seasonality change is most pronounced, is provided in panels f and g, respectively (see text for discussion).

**Fig. 6 -** Atmospheric $CO_2$, $CH_4$ and $N_2O$ concentrations over different time-scales, from 800 thousand years ago until today (panel a), over the last 2000 years (panel b) and over 1850 to 2014 (panel c, d, e). The shown data is for $CO_2$: Mauna Loa data by Keeling et al. (Keeling et al., 1976); the Law Dome ice record (Etheridge et al., 1998b; MacFarling Meure et al., 2006; Rubino et al., 2013), updated for minor dating changes and placed on current NOAA scales; NOAA ESRL station data (NOAA, 2013; NOAA ESRL GMD, 2014a, b, c); the EPICA composite data (Ahn and Brook, 2014; Bereiter et al., 2015; Bereiter et al., 2012; Lüthi et al., 2008; MacFarling Meure et al., 2006; Marcott et al., 2014; Monnin et al., 2004; Petit et al., 1999; Rubino et al., 2013; Schneider et al., 2013; Siegenthaler et al., 2005) and the WAIS data (Bauska et al., 2015). For $CH_4$, the shown data is the Law Dome data (Etheridge et al., 1998a; MacFarling Meure et al., 2006), the instrumental data from the NOAA and AGAGE networks (see Table 3), NEEM ice core measurements (Rhodes et al., 2013) the EPICA Dronning Maud Land ice core record {Barbante, 2006 #4721;Schilt, 2010 #4563;Capron, 2010 #4519} the long record by Loulergue et al. (2008) as well as the GISP2D, WDC05A and WDC06A records by Mitchell et al. (2013). In case of $N_2O$, the shown data is the Law Dome record (MacFarling Meure et al., 2006), the Talos Dome record (Schilt et al., 2010b), the GISPII record (Sowers et al., 2003) and the EPICA Dome C record {Schilt, 2010 #4644;Spahni, 2005 #4645;Fluckiger, 2002 #3541;Stauffer, 2002 #4750} in addition to the H15 ice core record from Antarctica (Machida et al., 1995), the South Pole firn record (Battle et al., 1996), the Law Dome firn record "Park" (Park et al., 2012) and a modelling synthesis by Ishijima (2007). For data sources behind "this study's" composite product, see Table 2, Table 3 and Table 4.

**Fig. 7 –** Comparison of 1950 to 1990 $CO_2$ concentrations with early Scripps station data (Keeling et al., 2001) for each 15°-degree latitudinal band. Also, the Law Dome ice record data is shown (panel k) with our 3rd degree polynomial smoothing. This study's monthly $CO_2$ zonal means were derived from station data from 1984 onwards. Before that, this study used Mauna Loa MLO annual average and smoothed Law Dome data (see Table 1 and section 2 "Methods"). The shown comparison with monthly Scripps station data before 1984 is a qualitative validation of the applied methodology to regress latitudinal gradient and seasonality changes to times before 1984. See text.

**Fig. 8 –** Historical GHG concentrations from 1750 to 2014 as global-mean (right panels), northern hemispheric (middle panels) and southern hemispheric averages (right panels). The top row comprises all GHGs, the middle row comprises HFCs, PFCs, $SF_6$, $NF_3$ and $SO_2F_2$. The lower row comprises all ozone depleting substances, expressed as equivalent CFC-12eq concentrations. In the narrow boxes, the last data year from 15 Jan 2014 to 15 Dec 2015 is shown, indicating the intra-annual trend (top row), increasing gradient (middle row) or relatively flat concentration levels (lower row).

**Fig. 9 -** Overview of historical $CO_2$ concentrations. Panel a.1, the average seasonality of $CO_2$ over the observational period, a.2, the change of seasonality over time. a.3, the observationally derived and extended EOF score of the seasonality change. The first EOF1's score is almost linearly increasing over the time of instrumental data from 1984 to 2014. b, the latitudinal variation of mole fractions (dashed lines), shown for example years from 1500 to 2014, including (for comparison) the average of three CMIP5 ESM models

(solid lines). c, the first and second EOF of latitudinal variation. The second EOF exhibits a strong signal around middle northern latitudes d, the EOF scores derived from the observational data (dots) and regression (dashed line) as well as the ultimately used EOF score (solid line). The second EOF's score indicates that the mid-latitude northern spike was only a recent phenomenon and the score is here assumed to linearly converge to zero. The first EOF's score is more linearly increasing, and regressed against global fossil emissions. e, the resulting latitudinal-monthly concentration field, here shown between 1950 and 2014. f, global and hemispheric means of the derived concentration field over the same time period 1950 to 2014 in comparison to monthly station data (grey dots), latitudinal average station data (coloured circles), and various literature studies (see legend). g, same as panel f, except for time period 1750 to 2014. h, same as panel f but for time period 2005 to 2010.

Fig. 10 - Annual Growth Rate of $CO_2$ concentrations for global-mean, northern hemispheric average and southern hemispheric average concentrations. Before 1960, the smooth growth rate results from interpolated global mean values. After 1960, the growth rates are diagnosed from the surface station data, as shown in Fig. 9f. Noticeable are fluctuations of the annual growth rate around 1973, 1981, and 1992.

Fig. 11 - Overview of historical $CH_4$ concentrations. Panel a.1, the relative seasonality of $CH_4$ over the observational period. b, the latitudinal variation of concentrations (dashed lines), shown for example years. c, the first and second EOF of latitudinal variation. d, the EOF scores derived from the observational data (dots) and regression against global emissions (dashed line) as well as the ultimately used EOF score (solid line). e, the resulting latitudinal-monthly concentration field, here shown between 1950 and 2014. f, global and hemispheric means of the derived concentration field over the same time period 1950 to 2014 in comparison to monthly station data (grey dots), latitudinal average station data (coloured circles), and various literature studies (see legend). g, same as panel f, except for time period 1750 to 2014. h, same as panel f but for time period 2005 to 2010.

Fig. 12 - Overview over historical $N_2O$ concentrations. As Fig. 11, but for $N_2O$.

Fig. 13 - CMIP5 ESMs vertical mole fraction averages at the provided pressure levels - averaged over the 30-year period 1976 to 2005. The black line indicates surface mole fractions at the 1000hPa pressure level. The red bold line indicates mole fractions at the 100hPa level (cf. Fig. 14a, and b).

Fig. 14 – Idealized vertical gradients recommended for implementation of surface concentration fields. For parametric formulas, see text. Note that tropospheric columns of non-$CO_2$ gases are – for simplicity – assumed to be well-mixed. The assumed age of air at the 1haPa level for $CO_2$ is 5 years.

Fig. 15 - Comparison between the recommended annual global mean surface concentrations of $CO_2$, $CH_4$ and $N_2O$ for CMIP5 and CMIP6 historical experiments. Fig. 16 - Comparison of global-mean, and hemispheric monthly average concentrations of $CO_2$ (panel a), $CH_4$ (panel b) and $N_2O$ (panel c) between the CMIP6 surface mole fractions (this study), the NOAA Marine Layer Boundary products, the World Data Centre of Greenhouse gases (WDCGG) products and the NASA AQUA satellite data of tropospheric $CO_2$ concentrations. For comparison, individual (monthly average) NOAA and AGAGE station data across all latitudes is shown in the background (grey dots).

Fig. 17 - Comparison of the CMIP6 historical $CO_2$ emissions (panel a) with the NOAA Marine Boundary Layer MBL product from 1979 to 2014 (panel b). Differences indicate that a seasonal higher $CO_2$ concentration is implied by the CMIP6 data of up to 5 ppm in mid-latitude northern bands, whereas some monthly tropical $CO_2$ mole fractions tend to be slightly lower in the CMIP6 product compared to NOAA MBL (panel c).

Fig. 18 - Comparison of the surface $CH_4$ monthly mean concentrations between CMIP6 (panel a), the NOAA Marine Boundary Layer product (panel b) and the difference (panel c). Since around 1992, there are seasonal differences in the mid northern latitudes with the CMIP6 data being up to 50ppb higher than the NOAA MBL product. Similarly, higher concentrations are apparent in the areas of tropical southern and lower latitude southern areas, presumably due to differences of data over land areas.

Fig. 19 - The comparison between latitudinal and monthly $N_2O$ concentrations to the NOAA Marine Boundary Layer product (panel b). The differences (panel c) show that the CMIP6 historical GHG concentrations are slightly higher in the southern hemisphere (0.5ppb) and slightly lower in the tropics (0.5ppb), as the stronger latitudinal gradient from tropics to southern latitudes is not reproduced in CMIP6 data. Note: Data submitted by Pieter Tans, pers. Communication.

# 13    Appendix A: Factsheets of GHGs other than CO$_2$, CH$_4$ and N$_2$O

Fig. 20 - CCl$_4$ Factsheet

Fig. 21 - CFC-11 Factsheet

Fig. 22 – CFC-12 Factsheet

Fig. 23 - CFC-113 Factsheet

Fig. 24 - CFC-114 Factsheet

Fig. 25 - CFC-115 Factsheet

Fig. 26 - CH$_2$Cl$_2$ Factsheet

Fig. 27 - CH$_3$Br Factsheet

Fig. 28 - CH$_3$CCl$_3$ Factsheet

Fig. 29 - CH$_3$Cl Factsheet

Fig. 30 - CHCl$_3$ Factsheet

Fig. 31 - Halon-1211 Factsheet

Fig. 32 - Halon-1301 Factsheet

Fig. 33 - Halon-2402 Factsheet

Fig. 34 - HCFC-22 Factsheet

Fig. 35 - HCFC-141b Factsheet

Fig. 36 - HCFC-142b Factsheet

Fig. 37 - C$_2$F$_6$ Factsheet

Fig. 38 - C$_3$F$_8$ Factsheet

Fig. 39 - C$_4$F$_{10}$ Factsheet

Fig. 40 - C$_5$F$_{12}$ Factsheet

Fig. 41 - C$_6$F$_{14}$ Factsheet

Fig. 42 - C$_7$F$_{16}$ Factsheet

Fig. 43 - C$_8$F$_{18}$ Factsheet

Fig. 44 - c-C$_4$F$_8$ Factsheet

Fig. 45 - CF$_4$ Factsheet

Fig. 46 - HFC-23 Factsheet

## 14    Appendix B: CMIP5 Analysis of CO₂ concentration fields

**Fig. 60 - Annual mean $CO_2$ concentrations in 8 CMIP5 ESM models in the year 1875. The CMIP5 recommended value was 288.7 ppm for 1875. Two more models with higher average $CO_2$ concentrations, namely BNU-ESM and FIO-ESM, are shown in and Fig. 66.**

**Fig. 61 - Annual mean $CO_2$ concentrations in 8 CMIP5 ESM models in the year 1990. The CMIP5 recommended value was 353.885 ppm for 1990 in the historical experiment.**

**Fig. 62 – Climatological seasonal cycle of $CO_2$ concentrations in 9 CMIP5 ESM models for the historical experiment's 30-year period 1861-1890.**

**Fig. 63 – Climatological seasonal cycle of $CO_2$ concentrations in 9 CMIP5 ESM models for the historical experiment's 30-year period**
**1976-2005.**

**Fig. 64 - Latitudinal gradient of surface atmospheric $CO_2$ concentrations exhibited in 9 considered CMIP5 ESM models for both the preindustrial period (grey lines) and recent period 1976-2005 (red lines). The bold dotted lines indicate the annual means. The 12 finer lines represent the individual twelve monthly averages over the respective 30 year periods (shaded areas show the min-max of those monthly averages). The lowest panel shows an ensemble mean for three CMIP5 ESMs, namely CanESM2, MPI-ESM-LR and**
**NorESM1-ME.**

**Fig. 65 - Annual average $CO_2$ concentration fields diagnosed from CMIP5 ESM models for the years 1875 (left column), 1960 (middle column), and 1990 (right column). All models are on the same colour scale, with colouring steps at 5 ppm. 1990 annual average $CO_2$ concentrations are estimated in this study to be 354.07 ppm and had been specified for CMIP5 with 353.855 ppm.**

**Fig. 66 – As Fig. 65, but for a different set of five CMIP5 ESM models.**

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

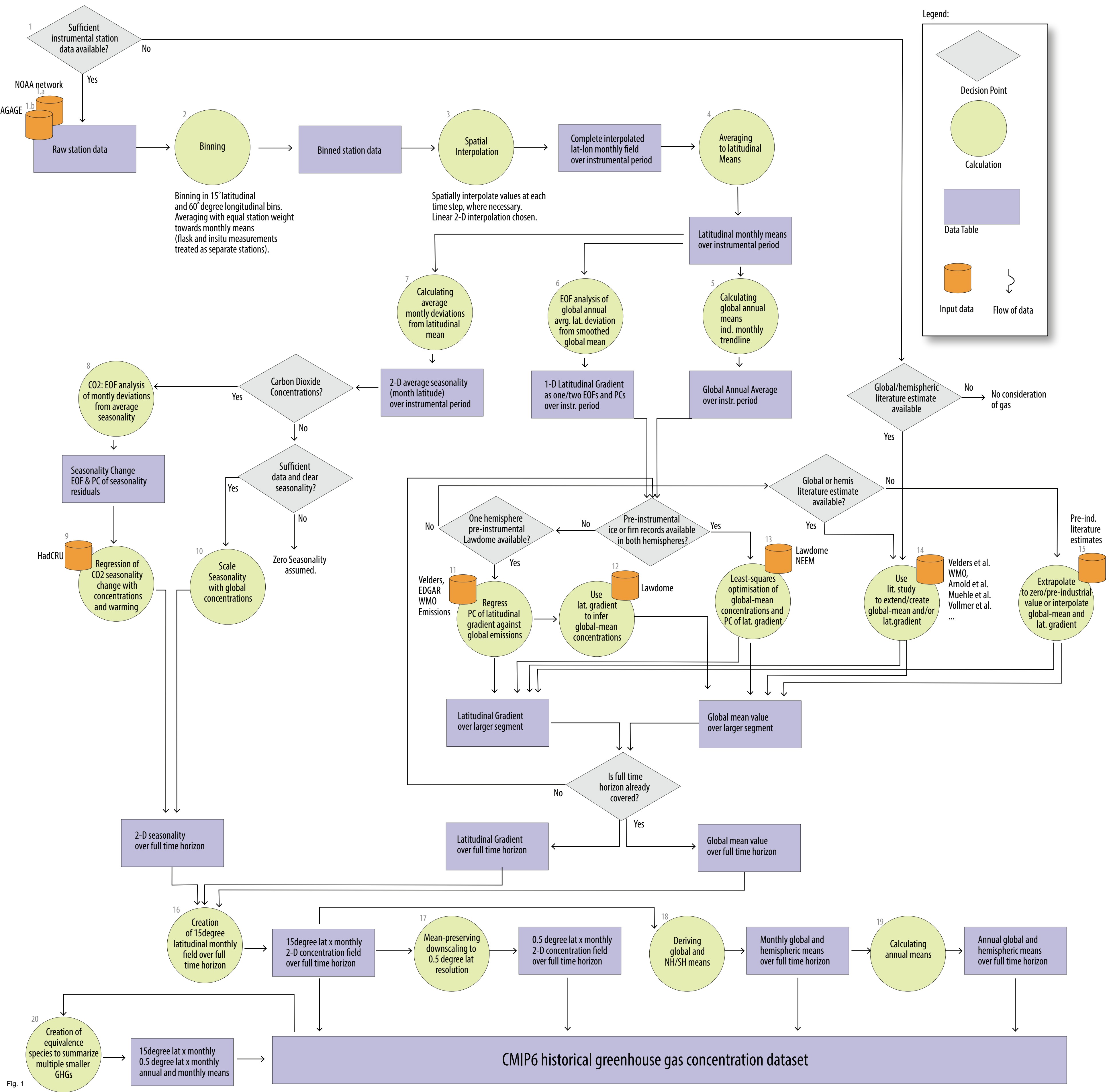

Fig. 1

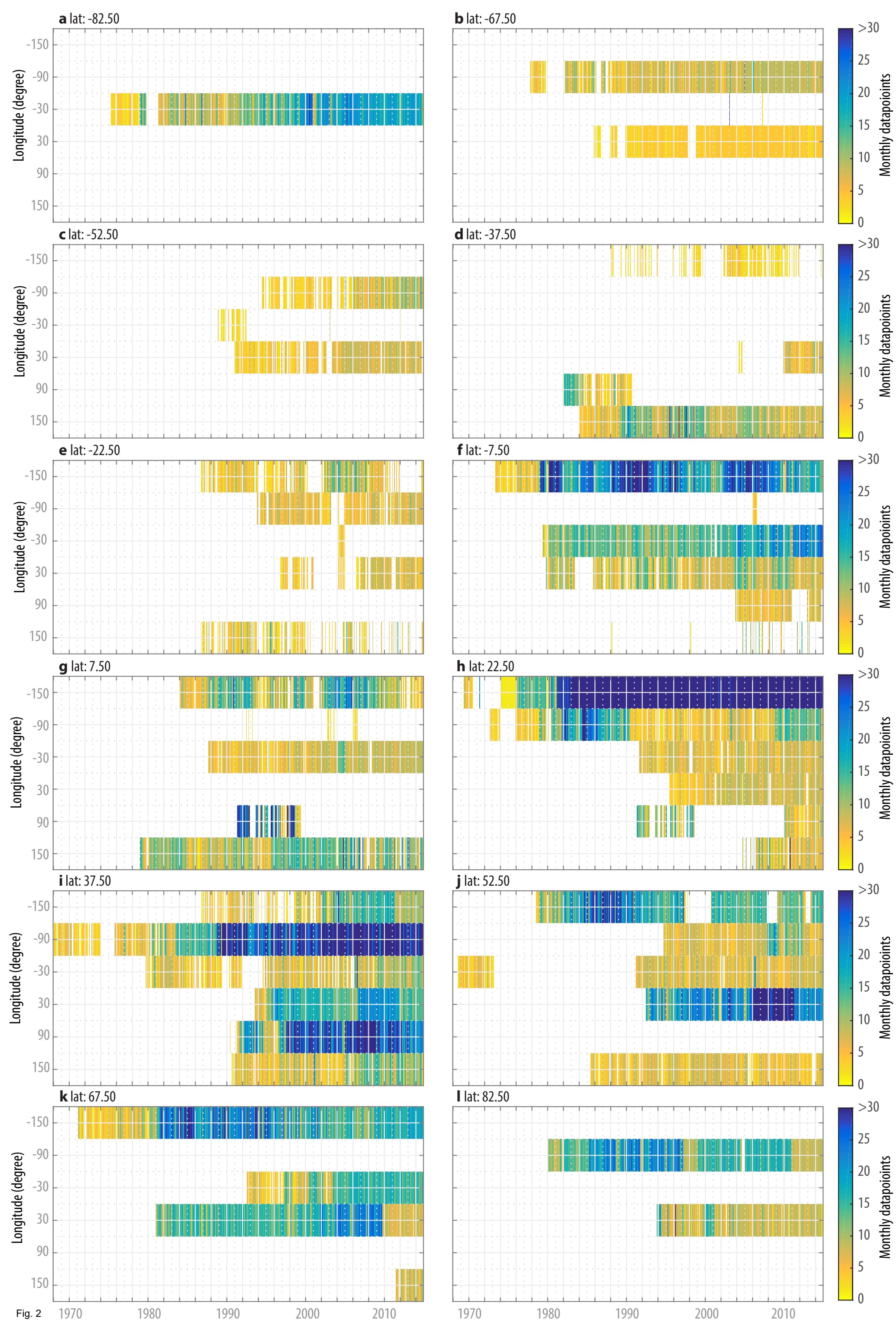

Fig. 2

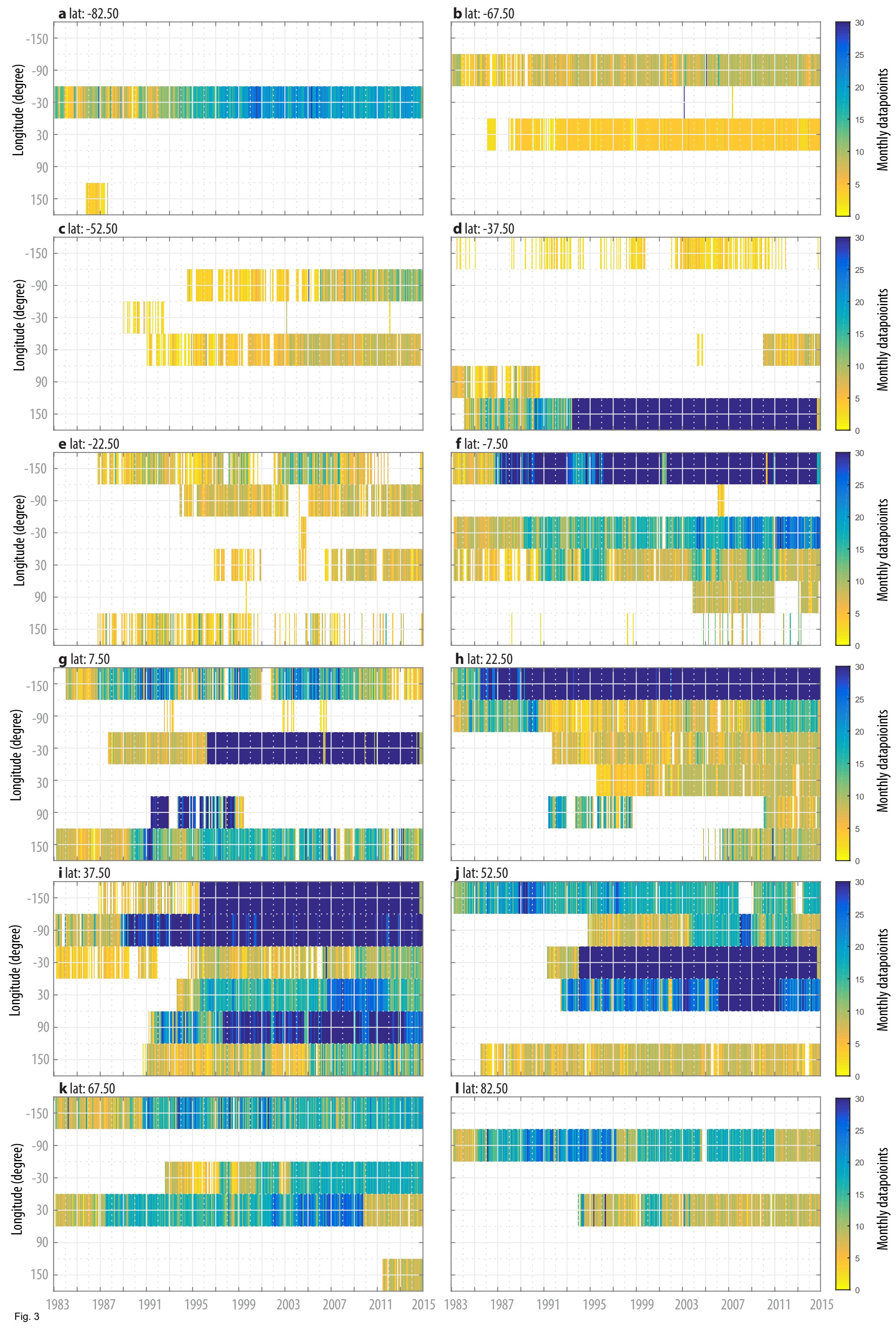

Fig. 3

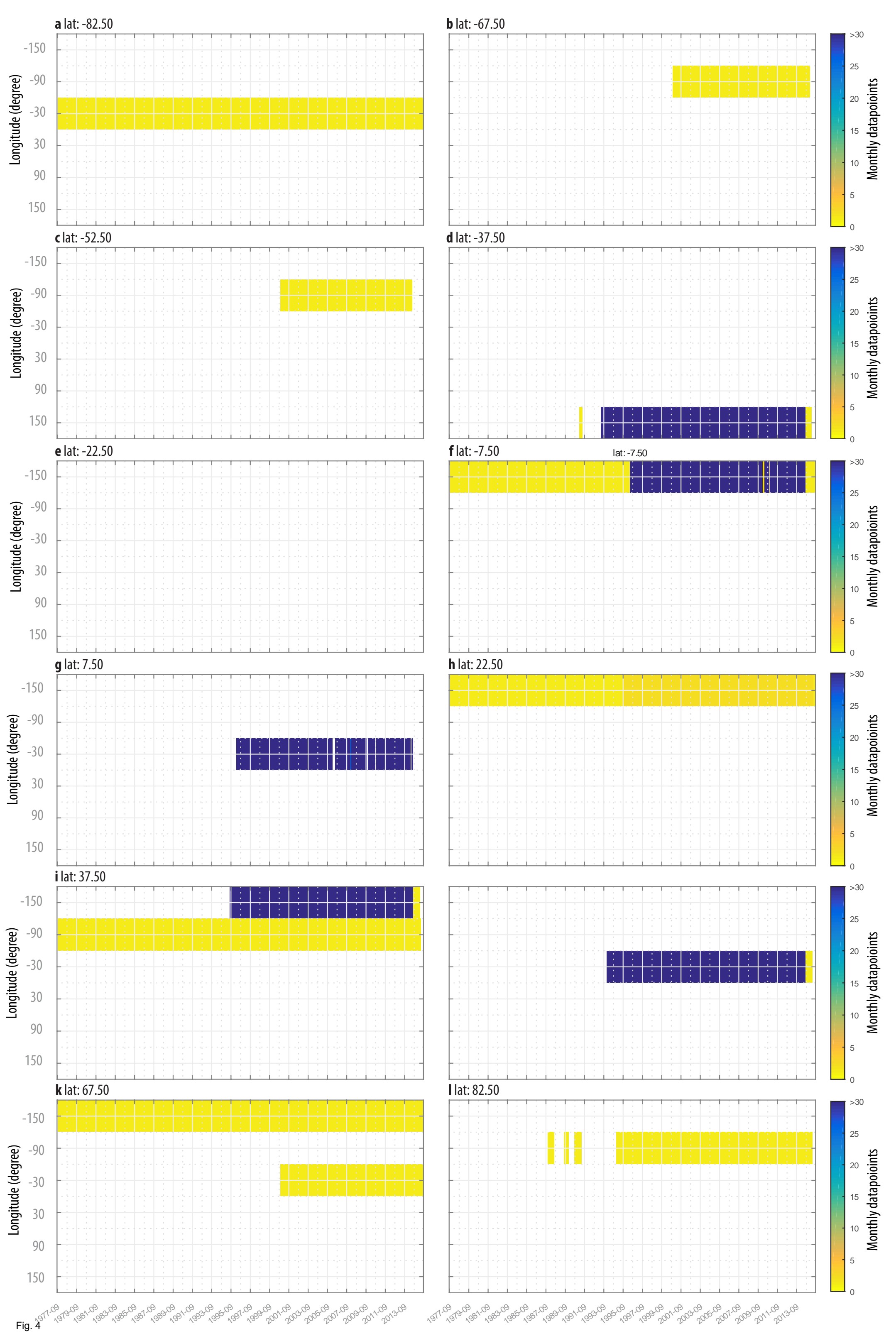

Fig. 4

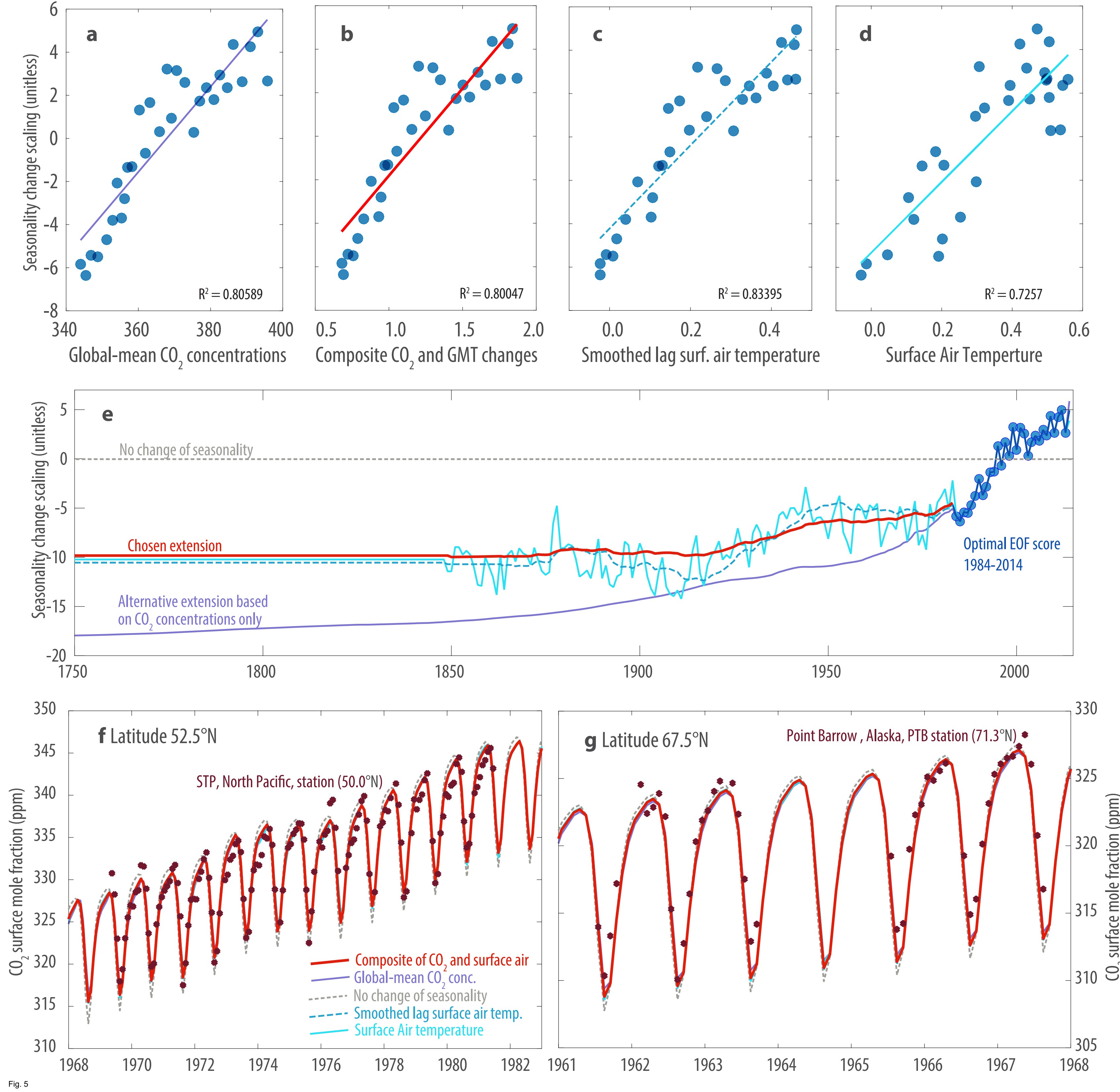

Fig. 5

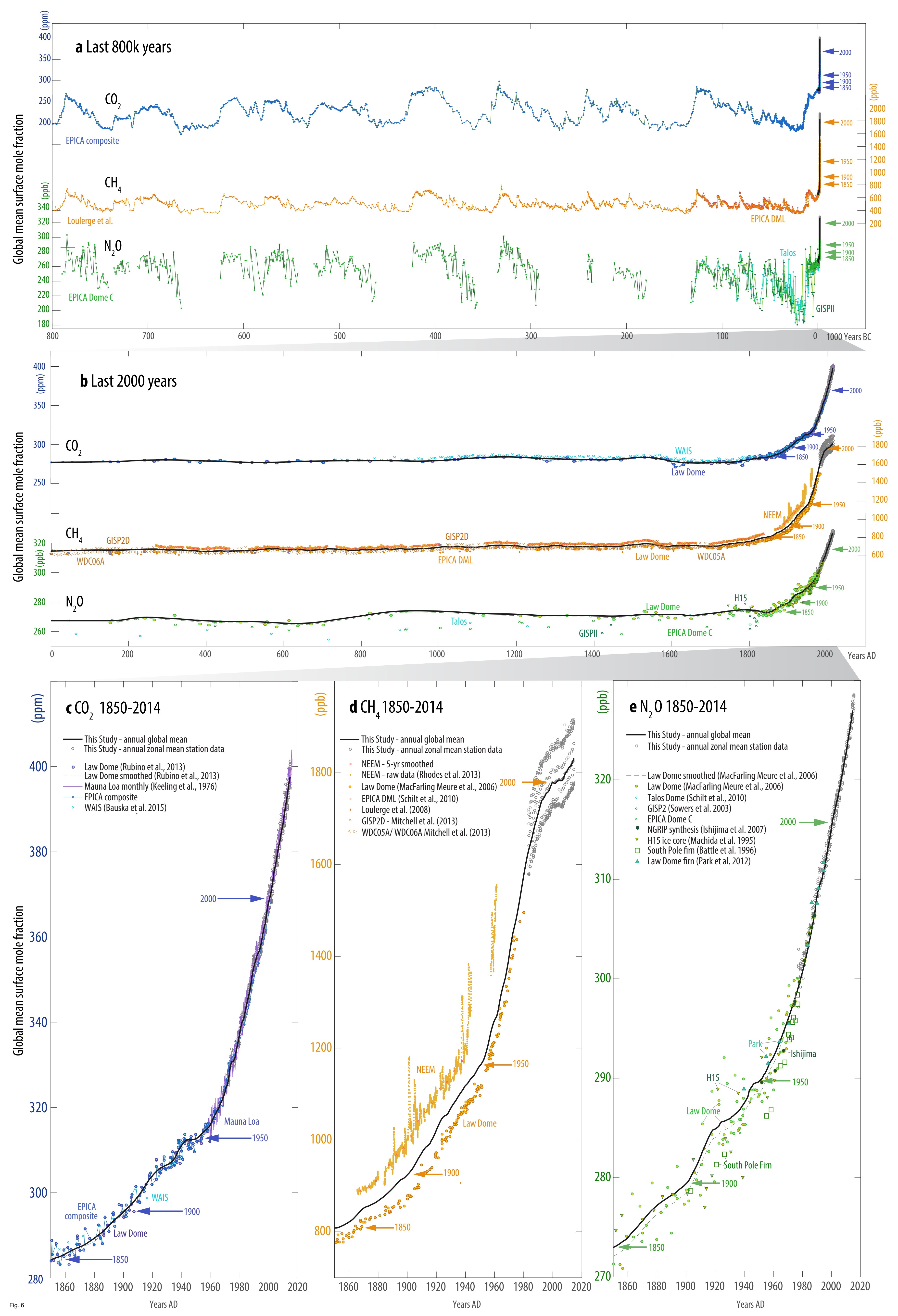

Fig. 6

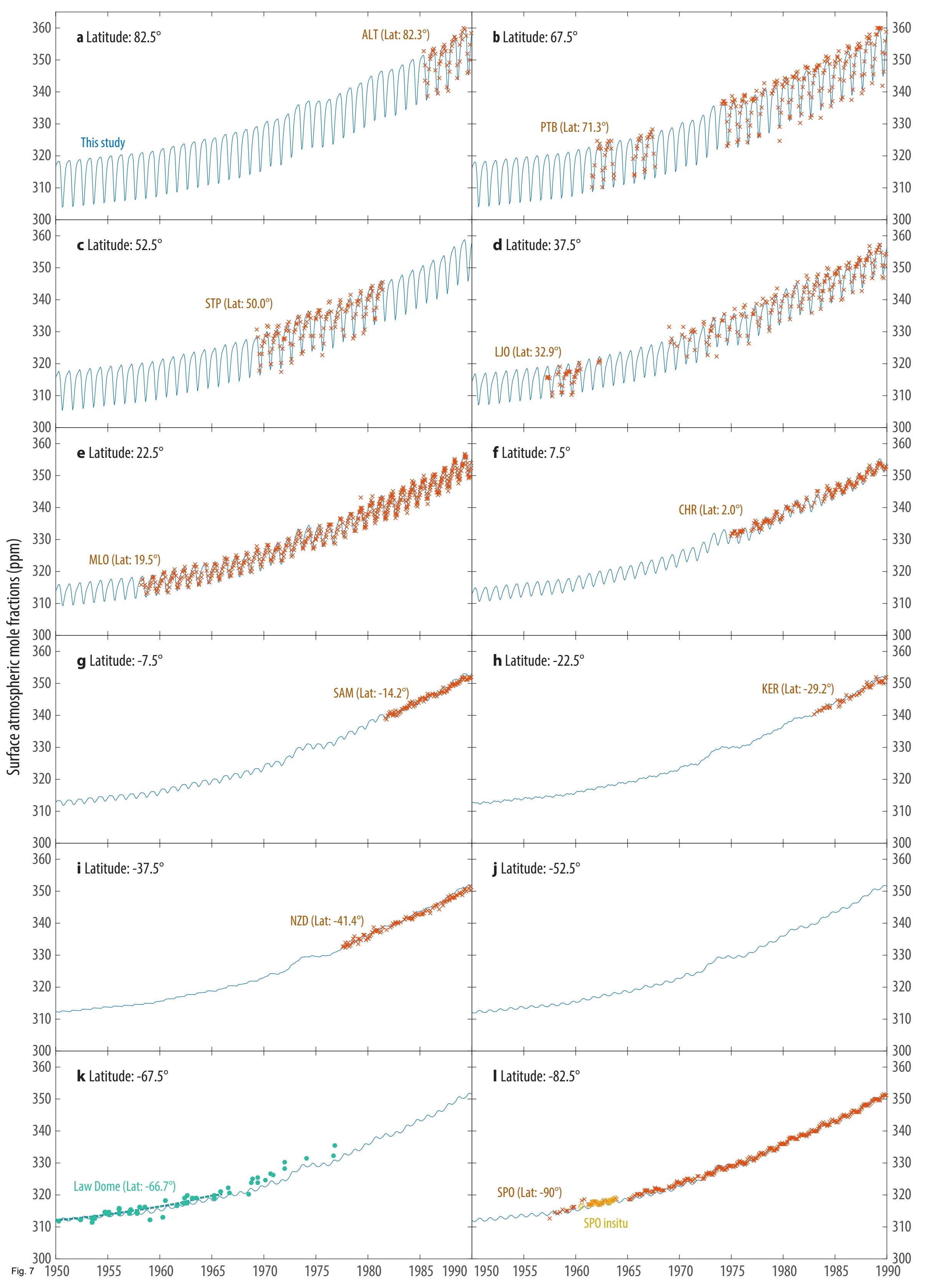

Fig. 7

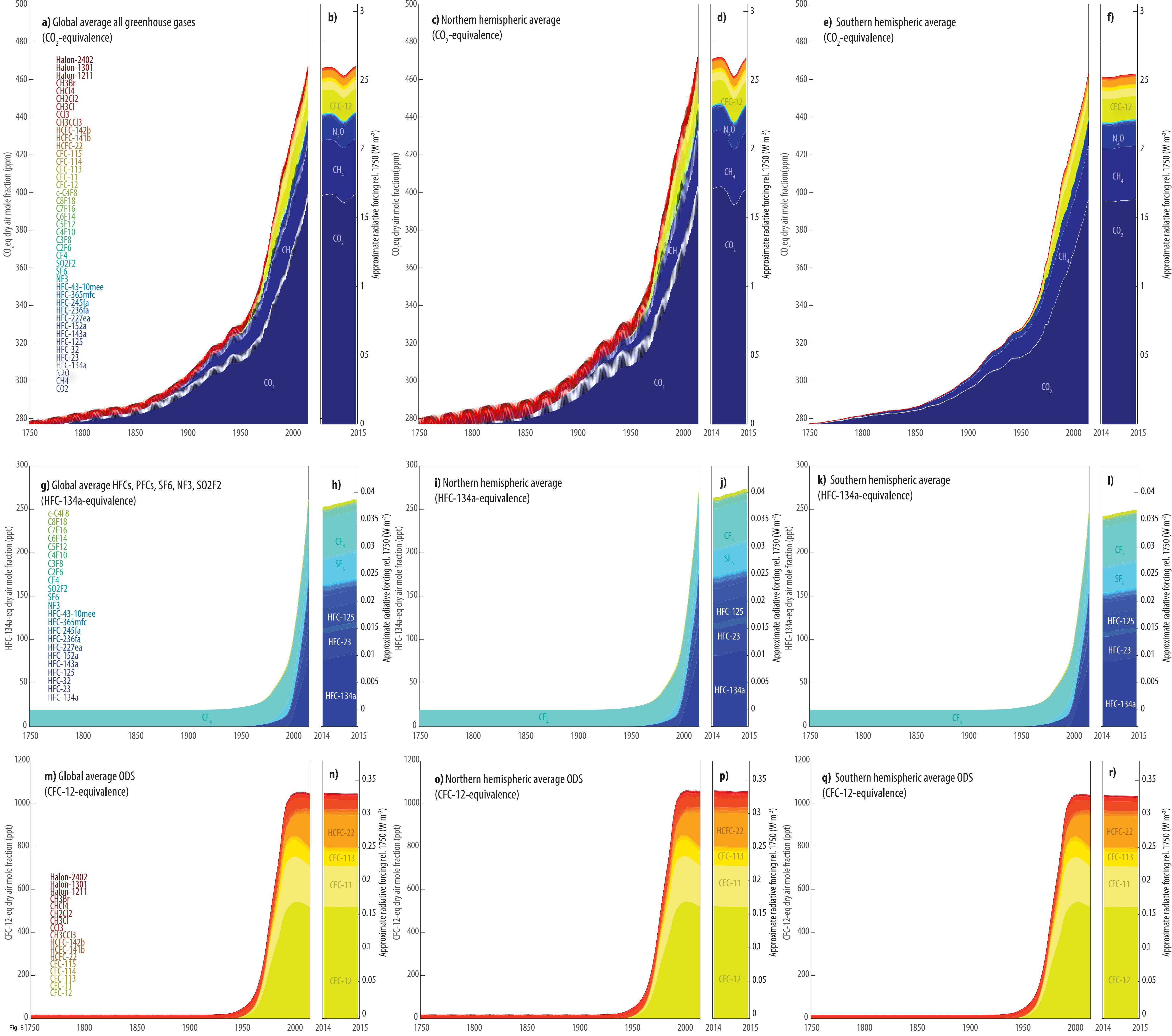

Fig. 8

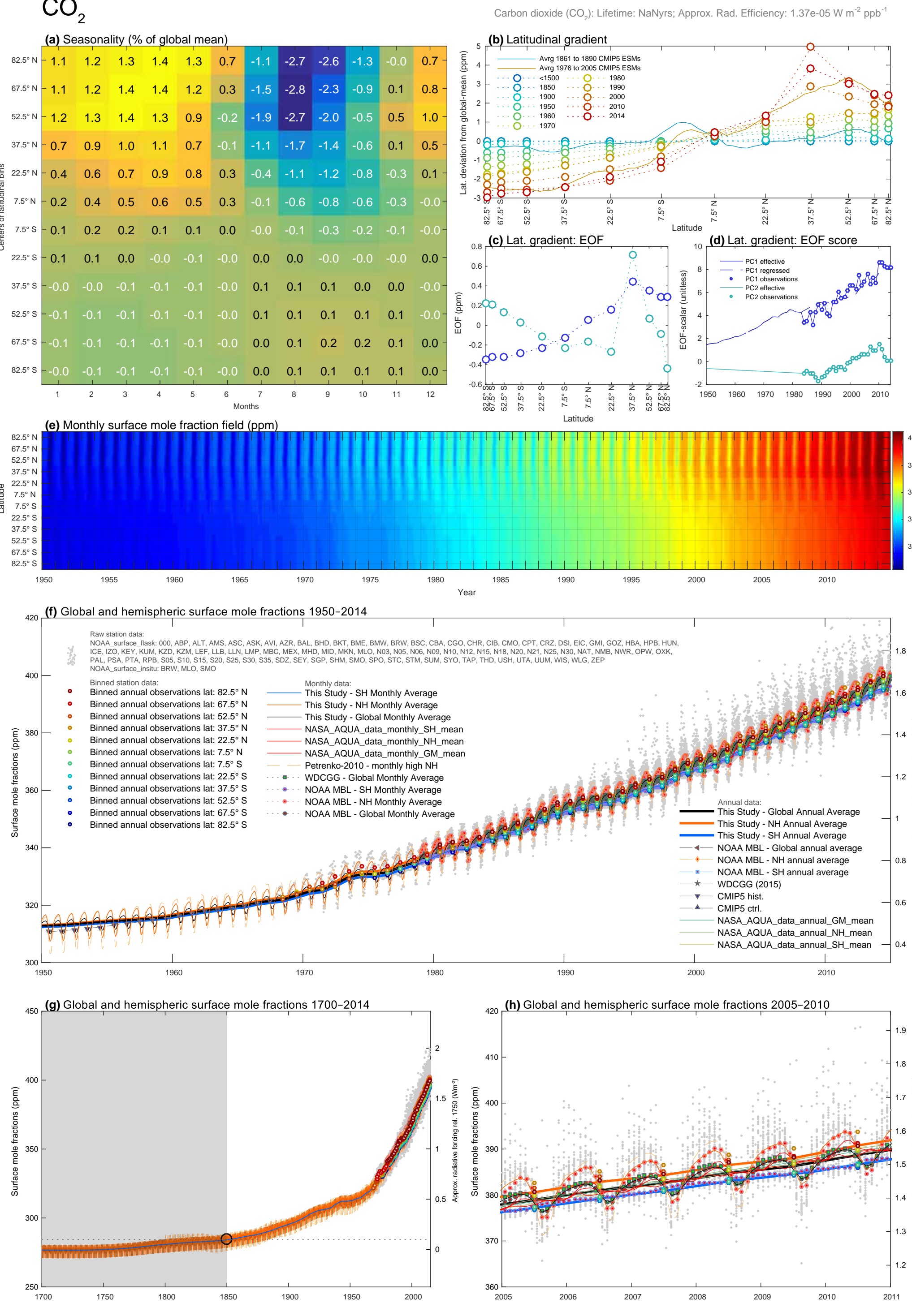

Fig. 9

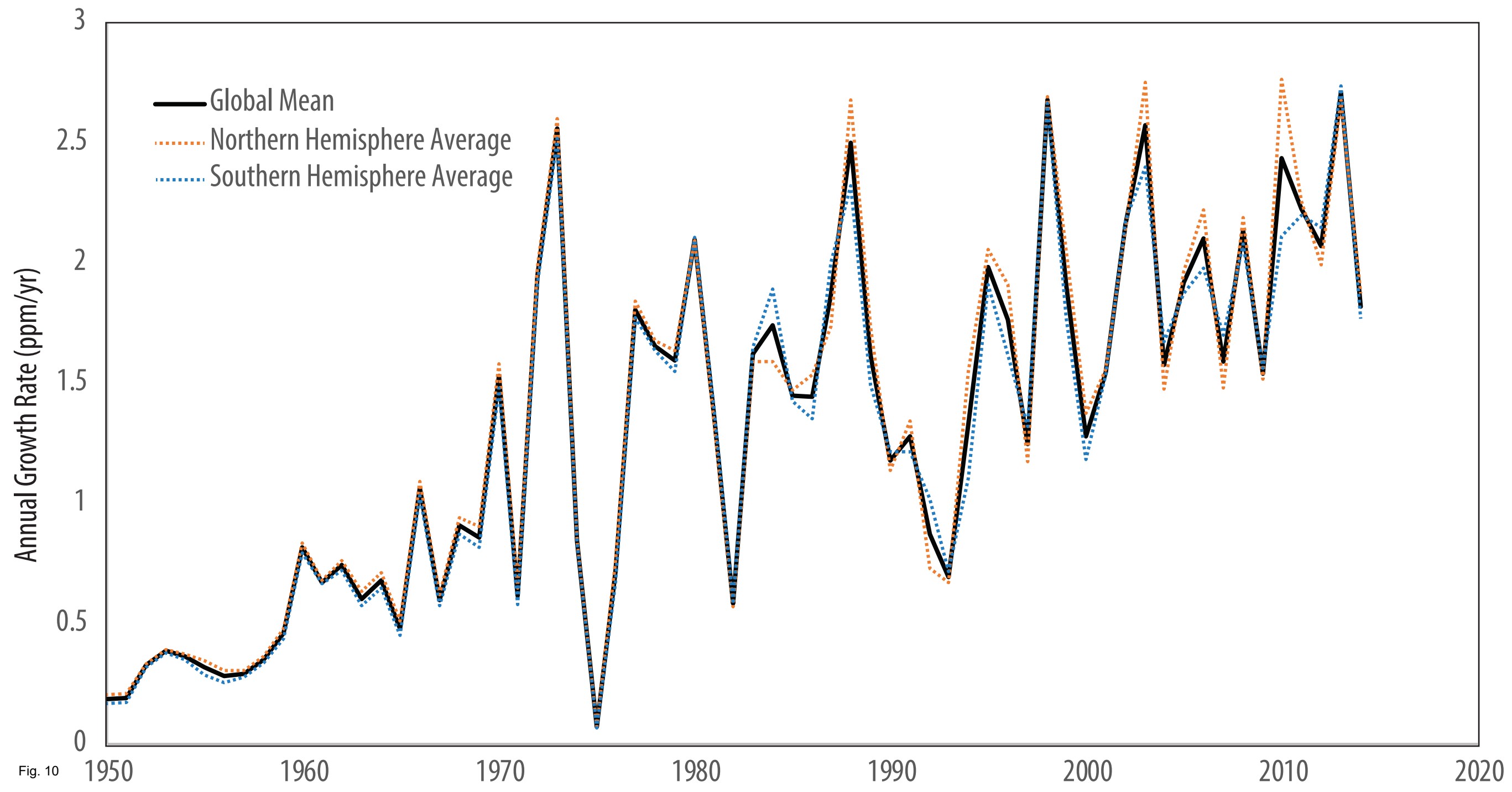

Fig. 10

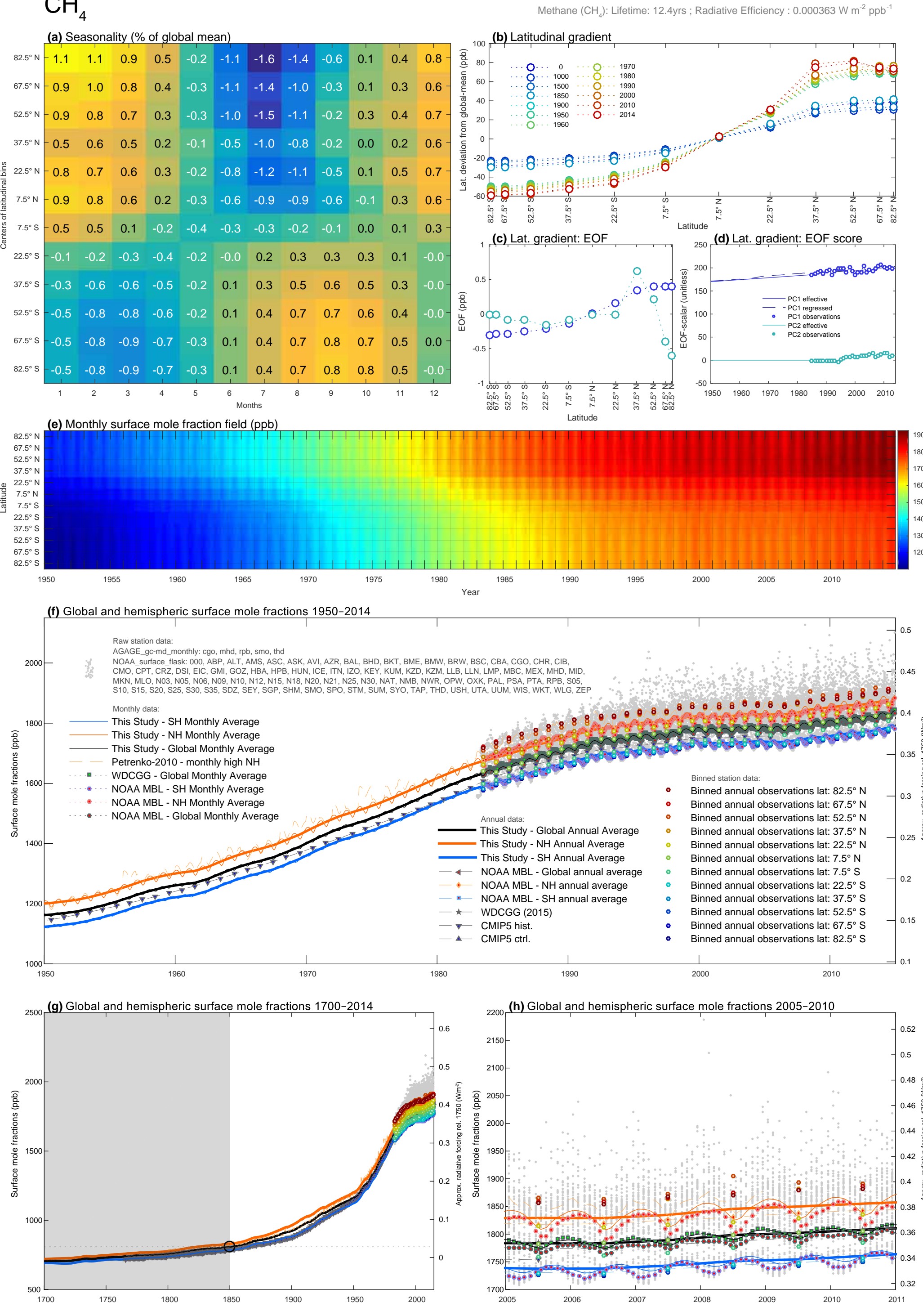

**Fig. 11**

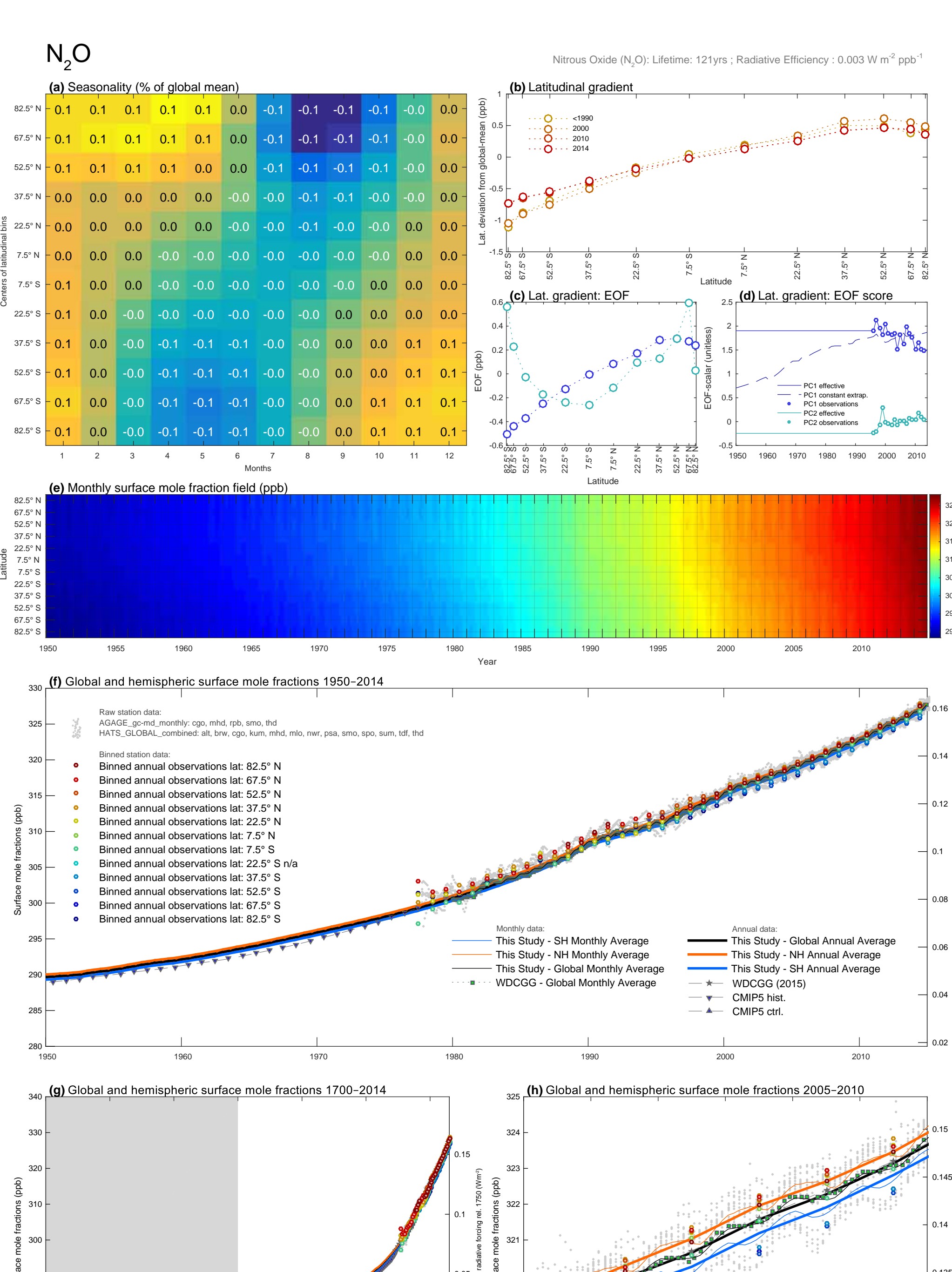

**Fig. 12**

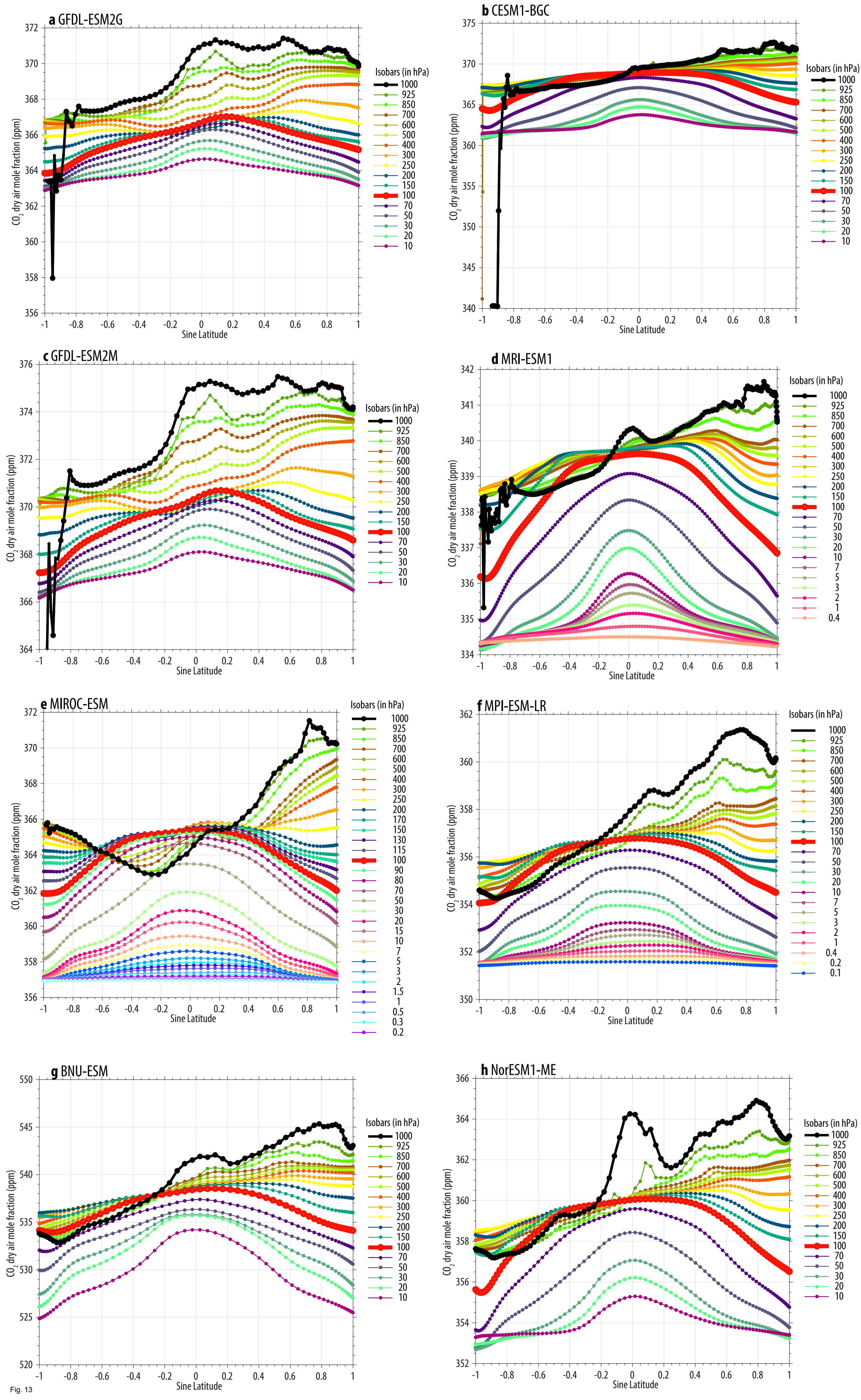

Fig. 13

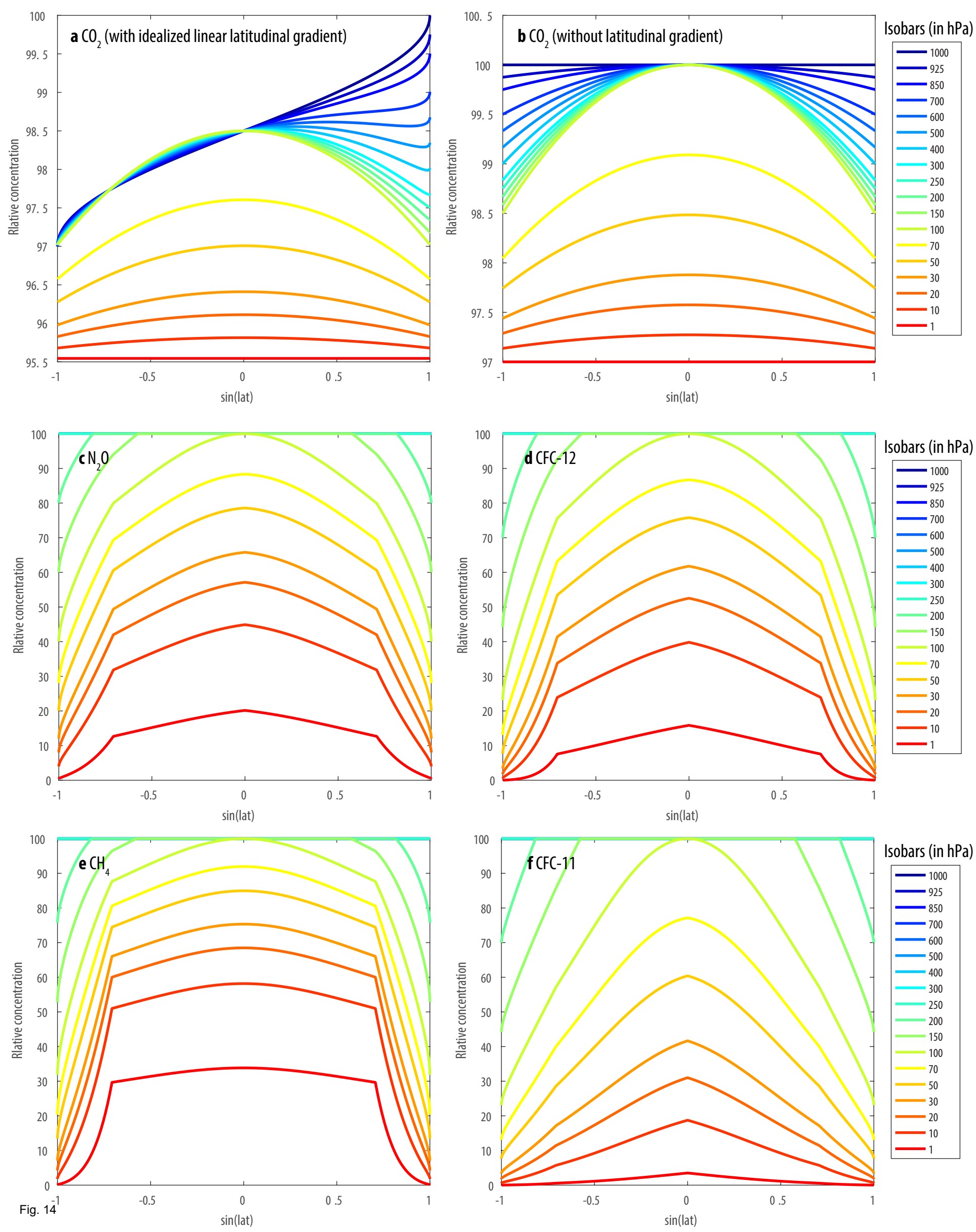

Fig. 14

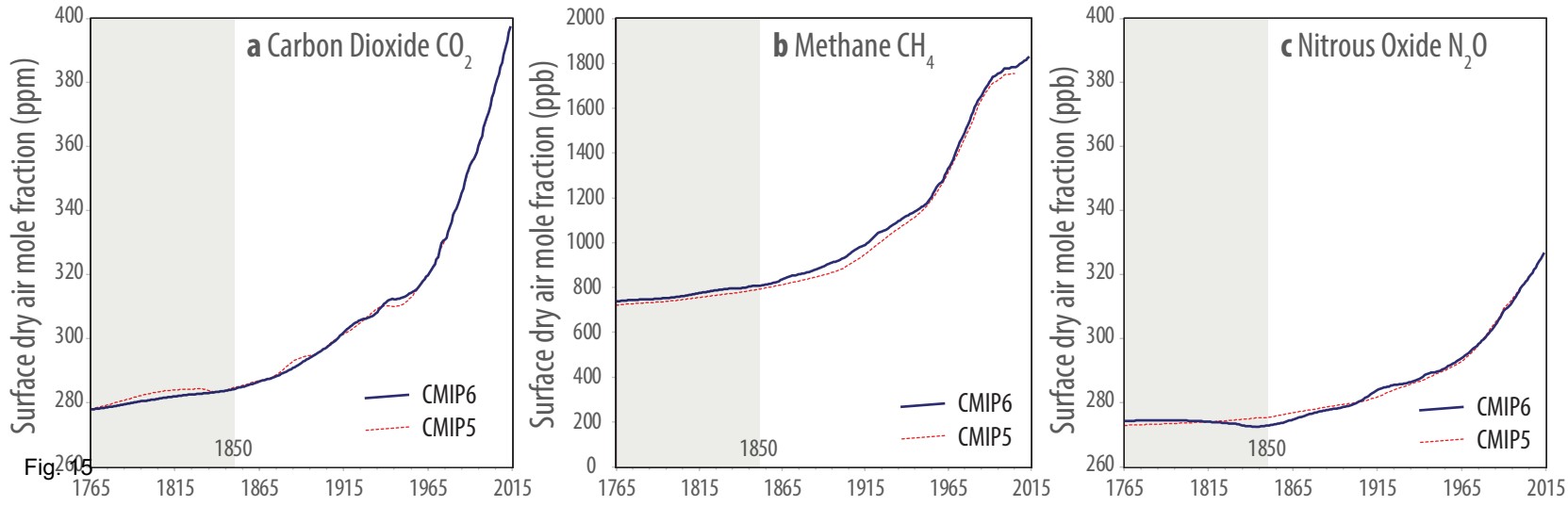

Fig. 15

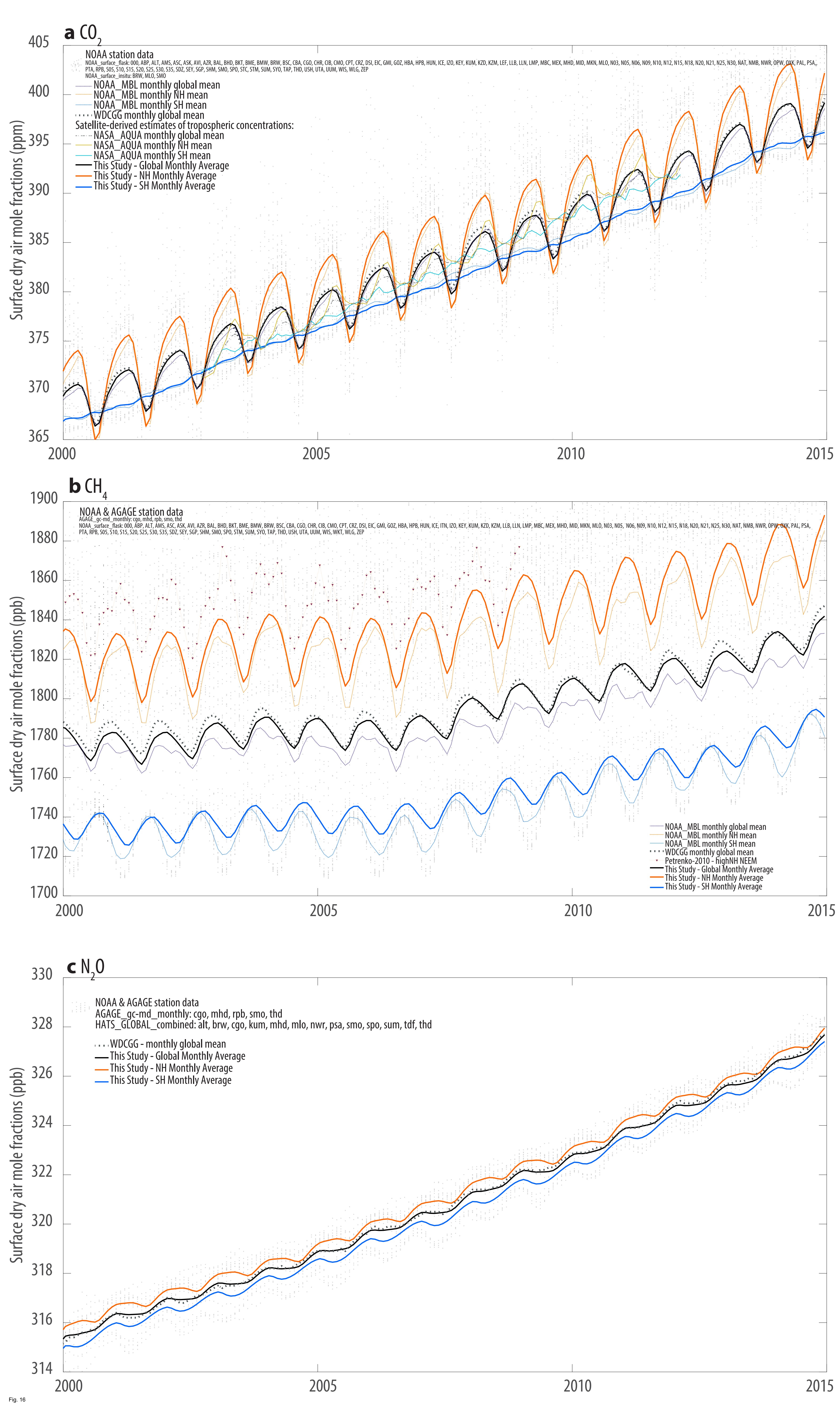

Fig. 16

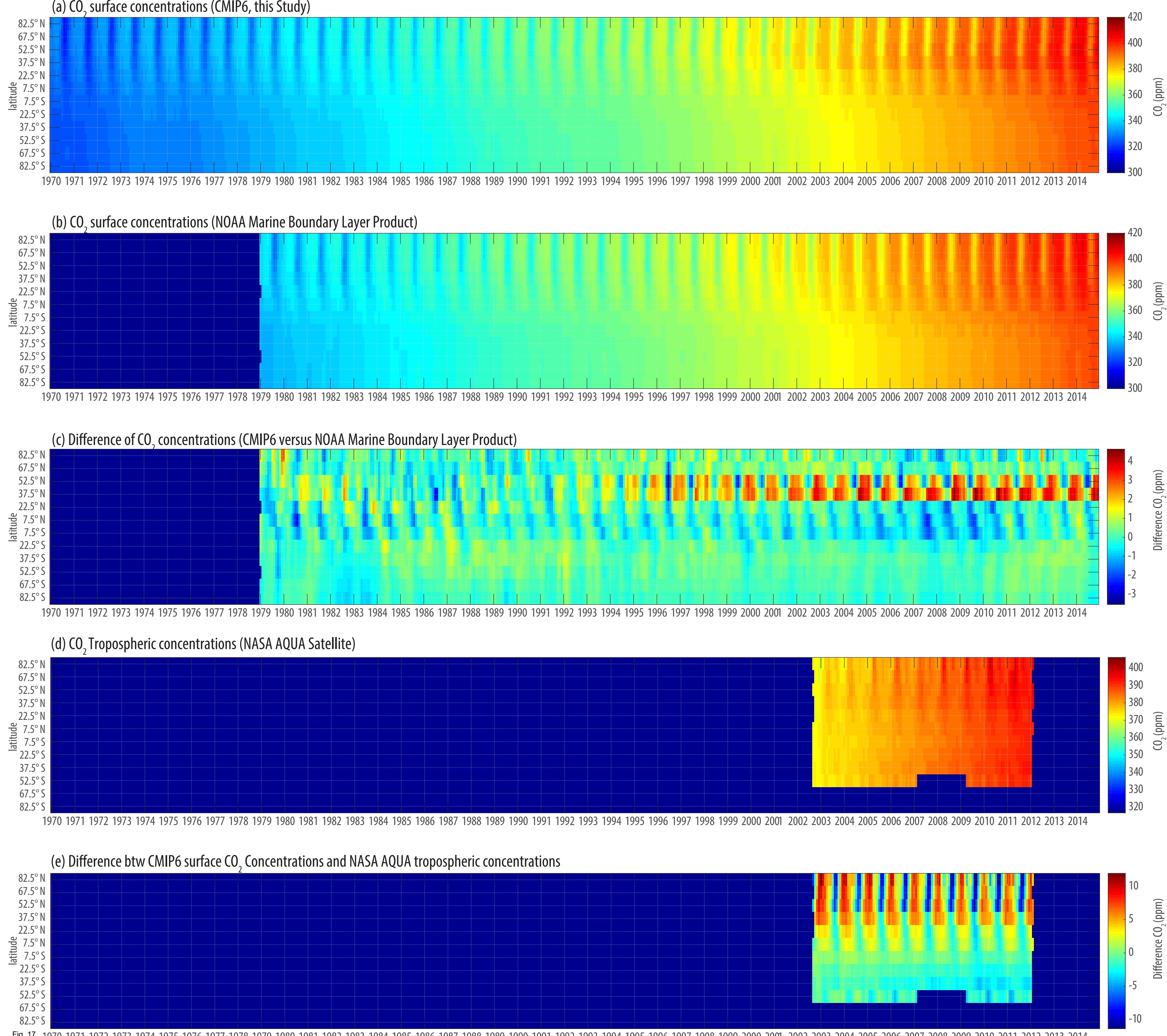

Fig. 17

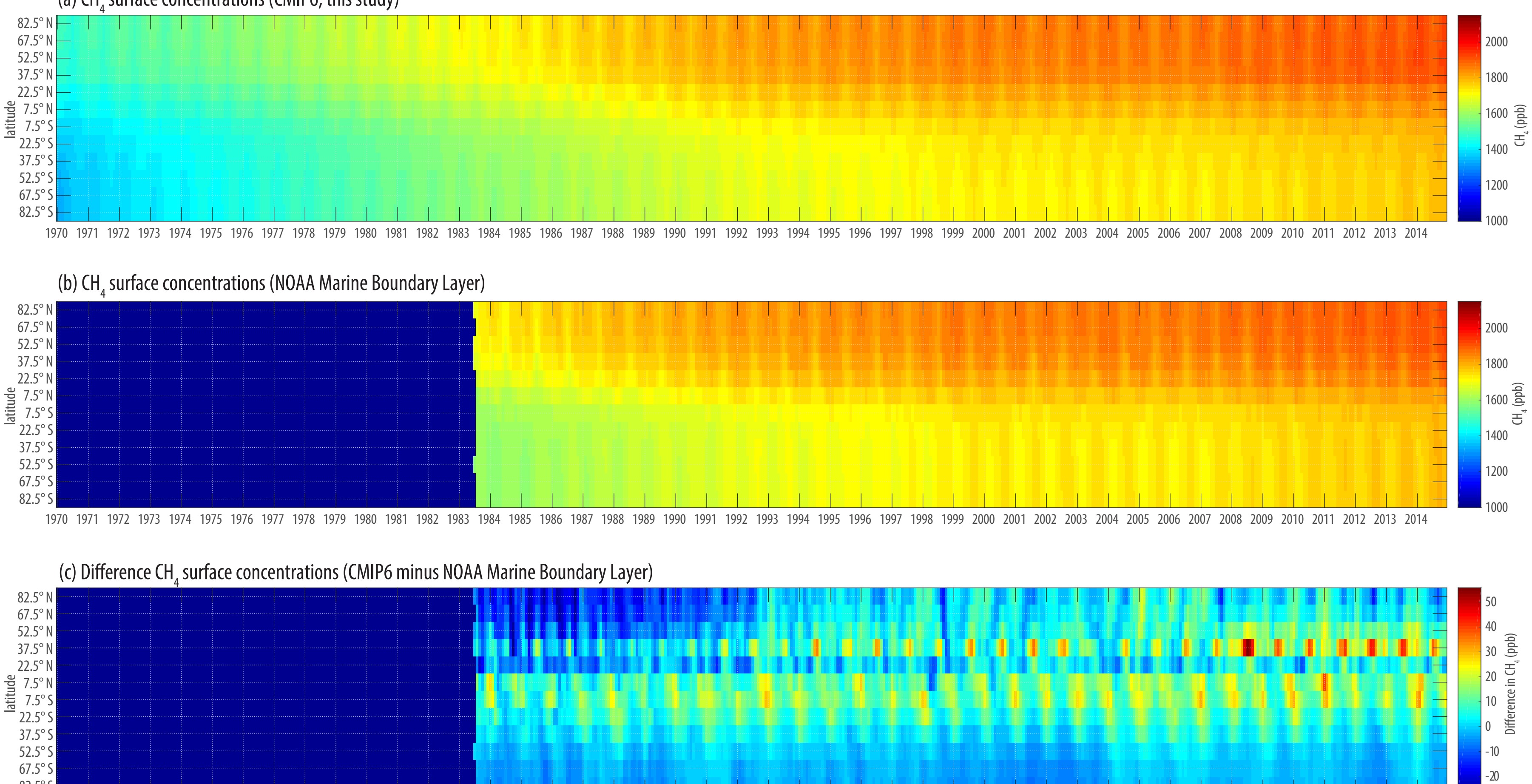

(a) CH$_4$ surface concentrations (CMIP6, this study)

(b) CH$_4$ surface concentrations (NOAA Marine Boundary Layer)

(c) Difference CH$_4$ surface concentrations (CMIP6 minus NOAA Marine Boundary Layer)

Fig. 18

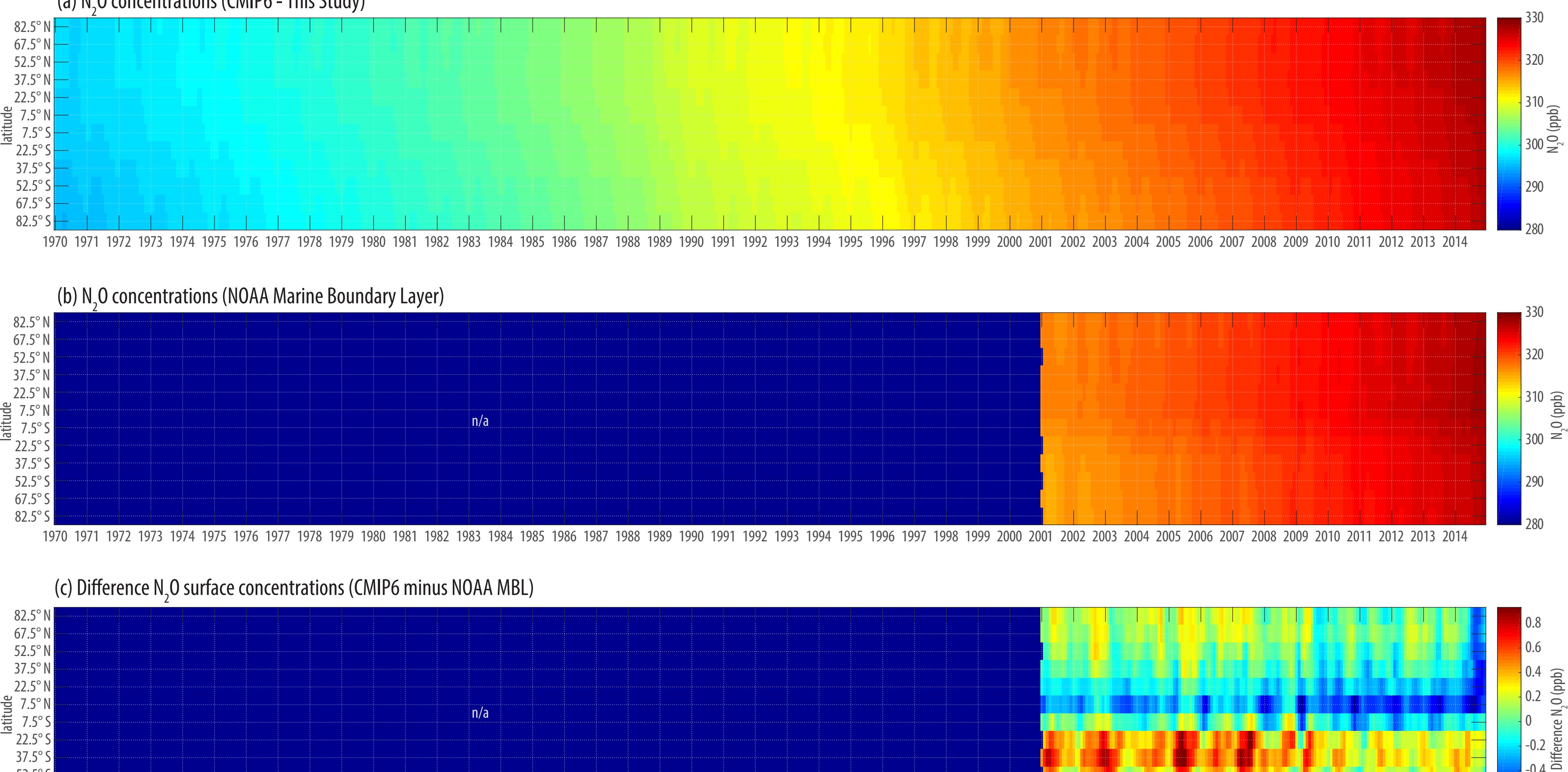

(a) N$_2$O concentrations (CMIP6 - This Study)

(b) N$_2$O concentrations (NOAA Marine Boundary Layer)

(c) Difference N$_2$O surface concentrations (CMIP6 minus NOAA MBL)

Fig. 19

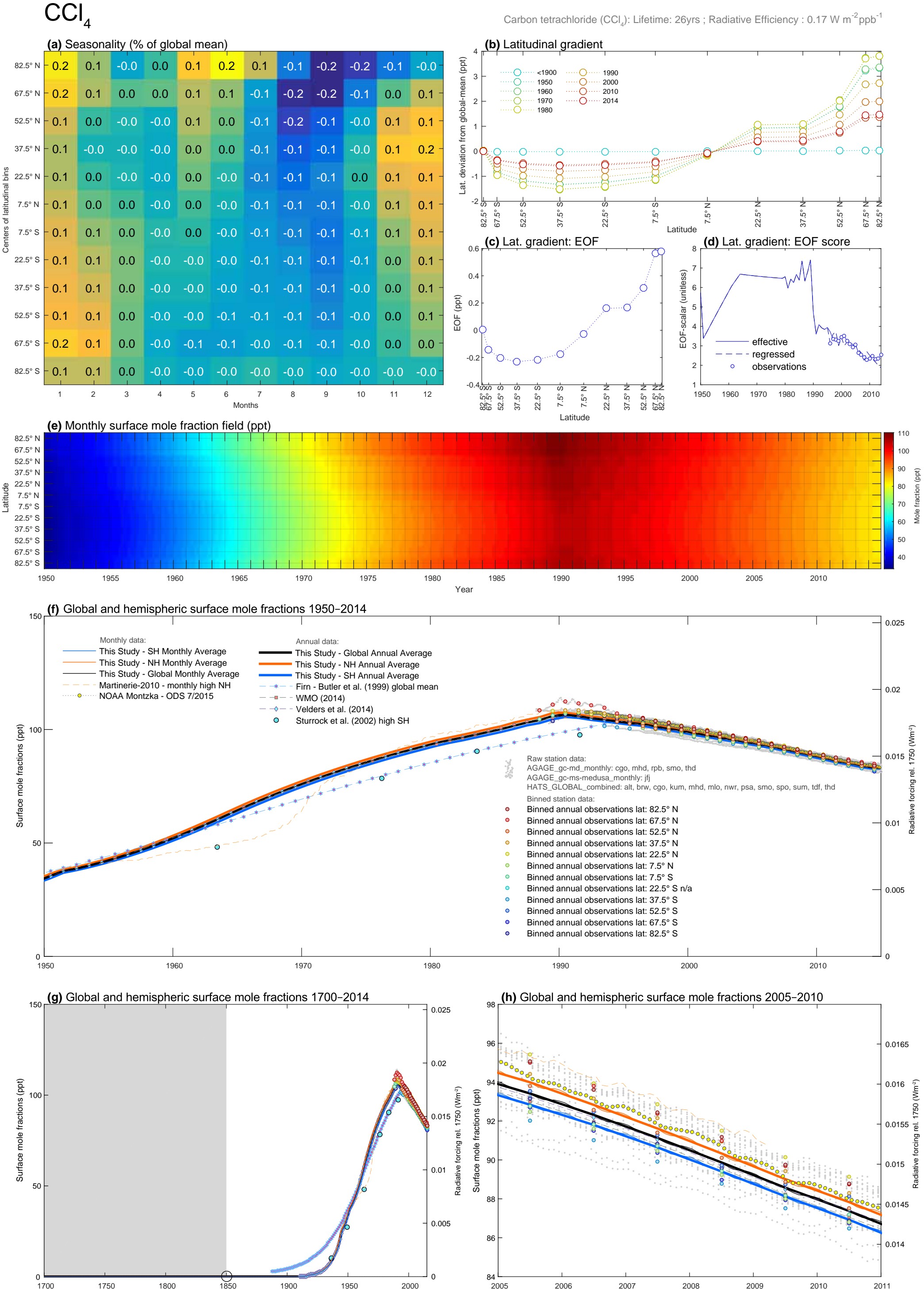

Fig. 20

# CFC-11

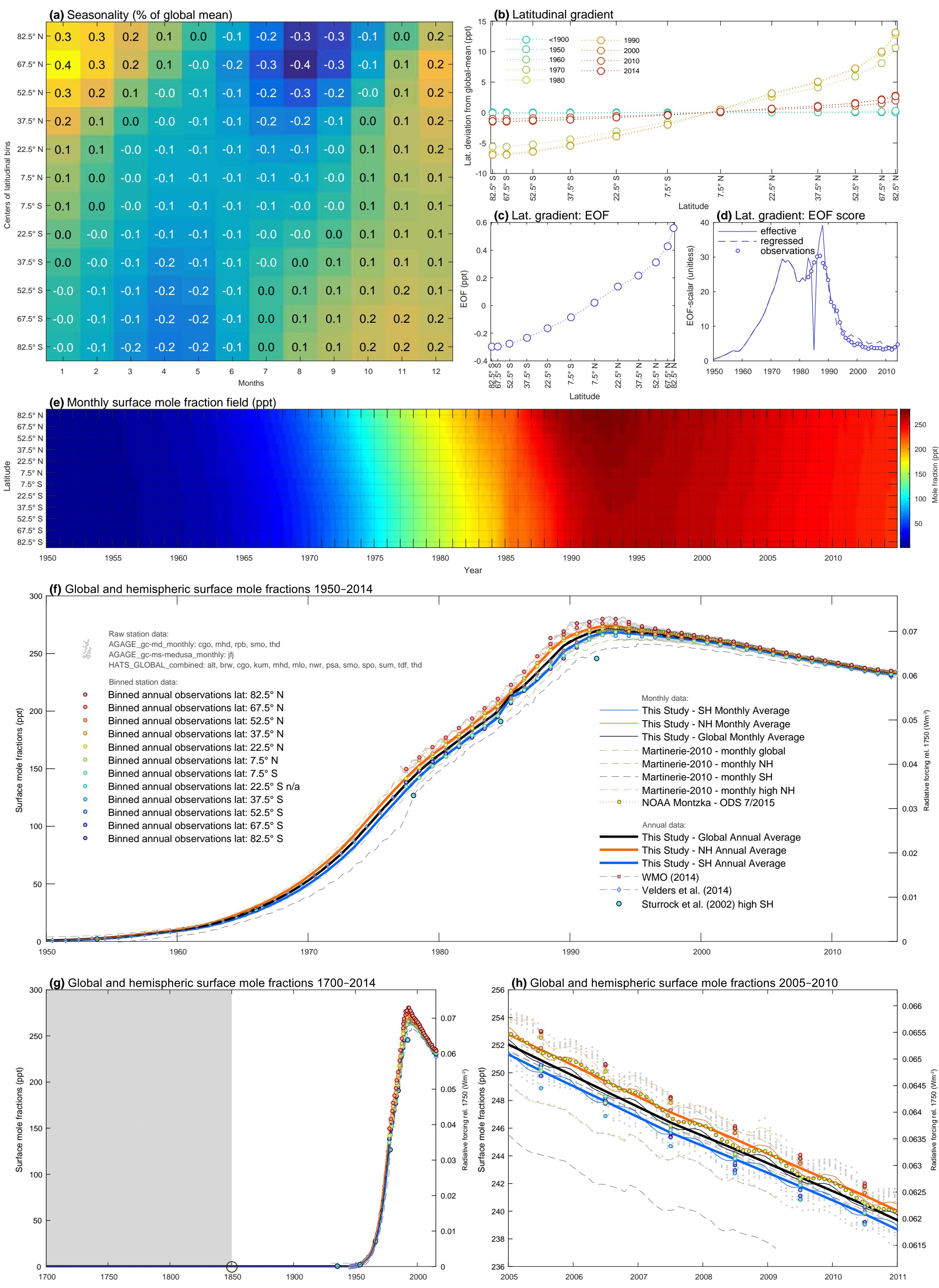

# CFC-12

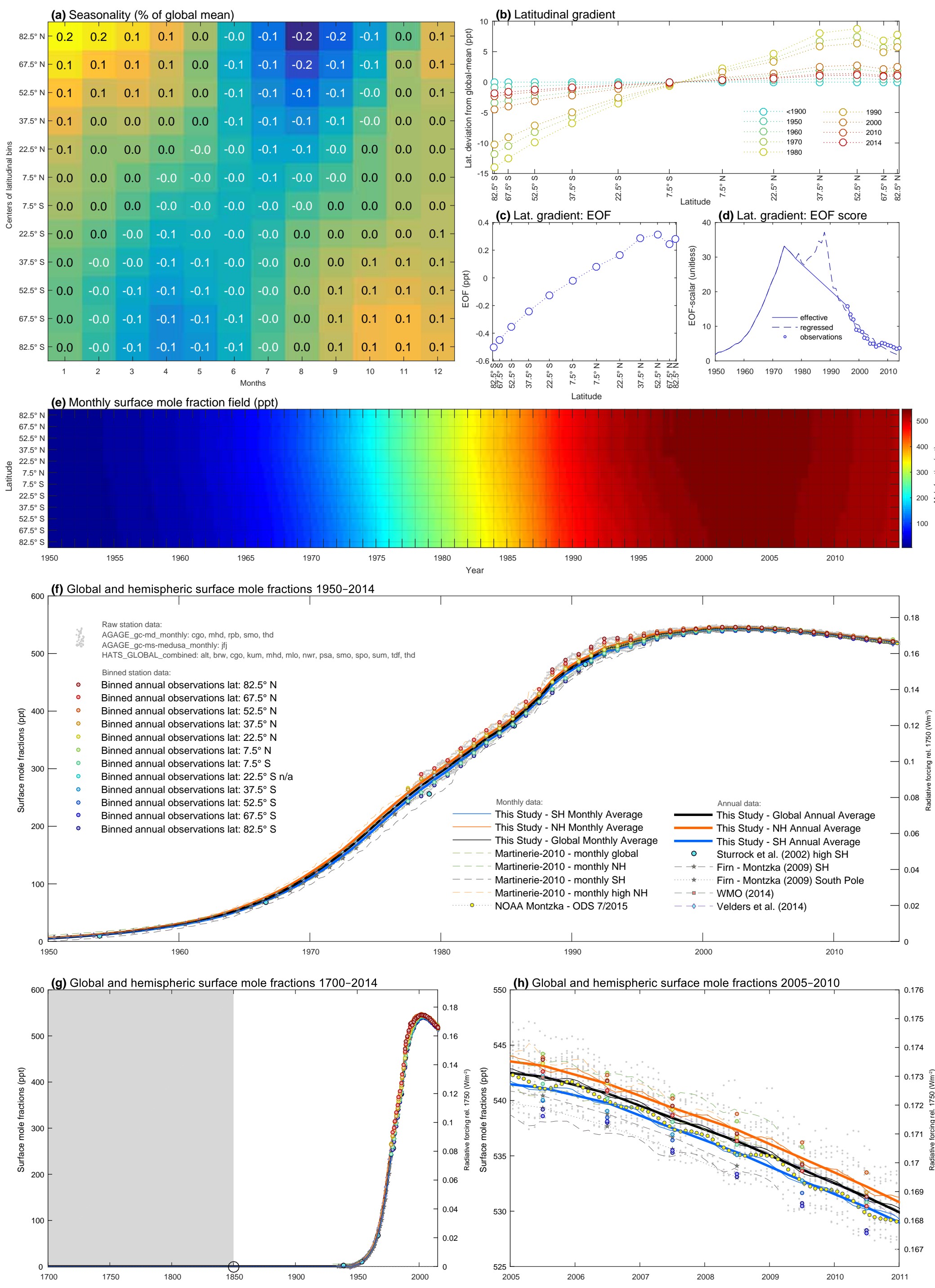

CFC-12 (CCl$_2$F$_2$): Lifetime: 100yrs ; Radiative Efficiency : 0.32 W m$^{-2}$ ppb$^{-1}$

**(a)** Seasonality (% of global mean)

**(b)** Latitudinal gradient

**(c)** Lat. gradient: EOF

**(d)** Lat. gradient: EOF score

**(e)** Monthly surface mole fraction field (ppt)

**(f)** Global and hemispheric surface mole fractions 1950–2014

**(g)** Global and hemispheric surface mole fractions 1700–2014

**(h)** Global and hemispheric surface mole fractions 2005–2010

Fig. 22

# CFC-113

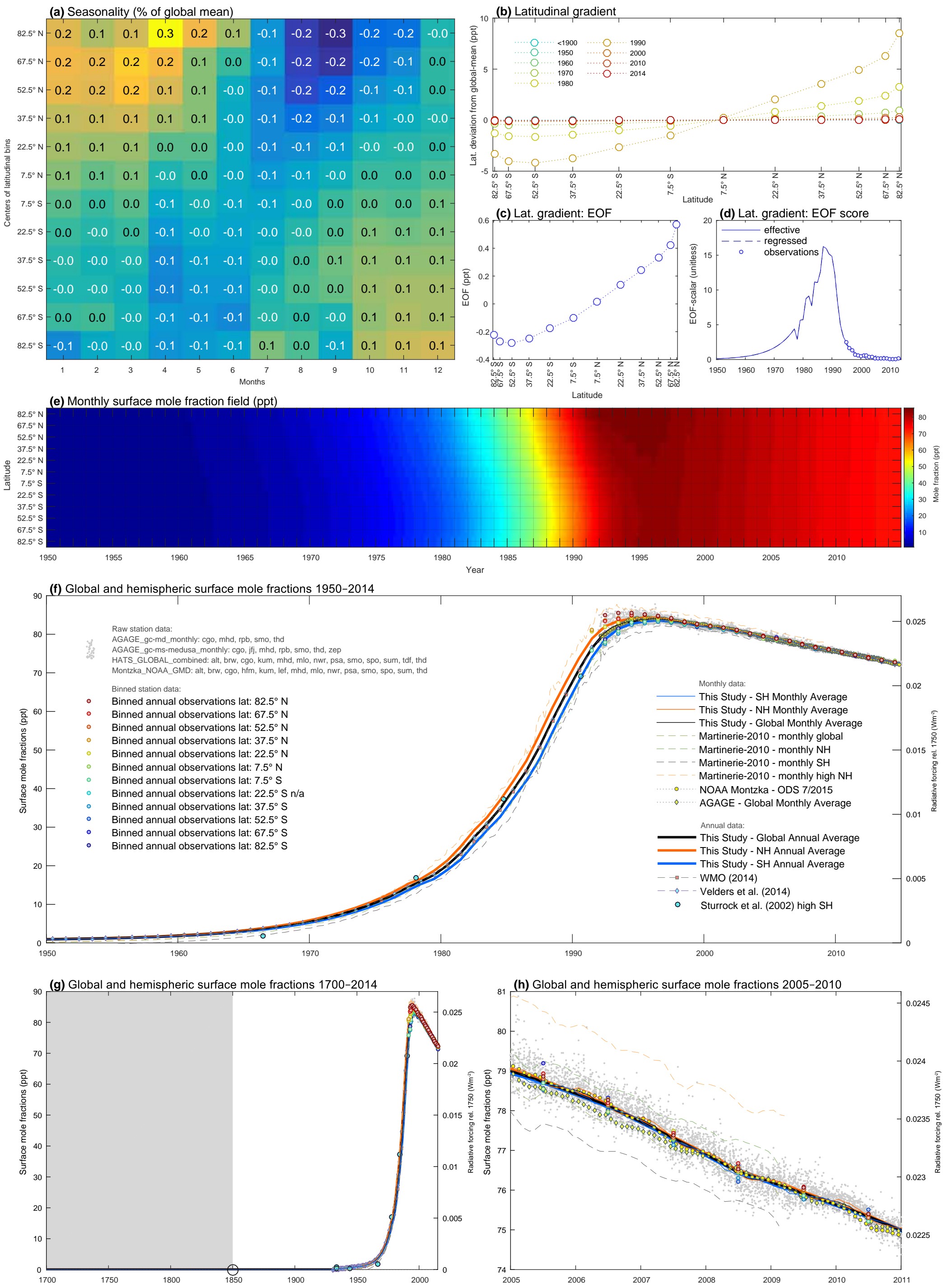

CFC-113 (CCl$_2$FCClF$_2$): Lifetime: 85yrs ; Radiative Efficiency : 0.3 W m$^{-2}$ ppb$^{-1}$

(a) Seasonality (% of global mean)

(b) Latitudinal gradient

(c) Lat. gradient: EOF

(d) Lat. gradient: EOF score

(e) Monthly surface mole fraction field (ppt)

(f) Global and hemispheric surface mole fractions 1950–2014

(g) Global and hemispheric surface mole fractions 1700–2014

(h) Global and hemispheric surface mole fractions 2005–2010

Fig. 23

# CFC-114

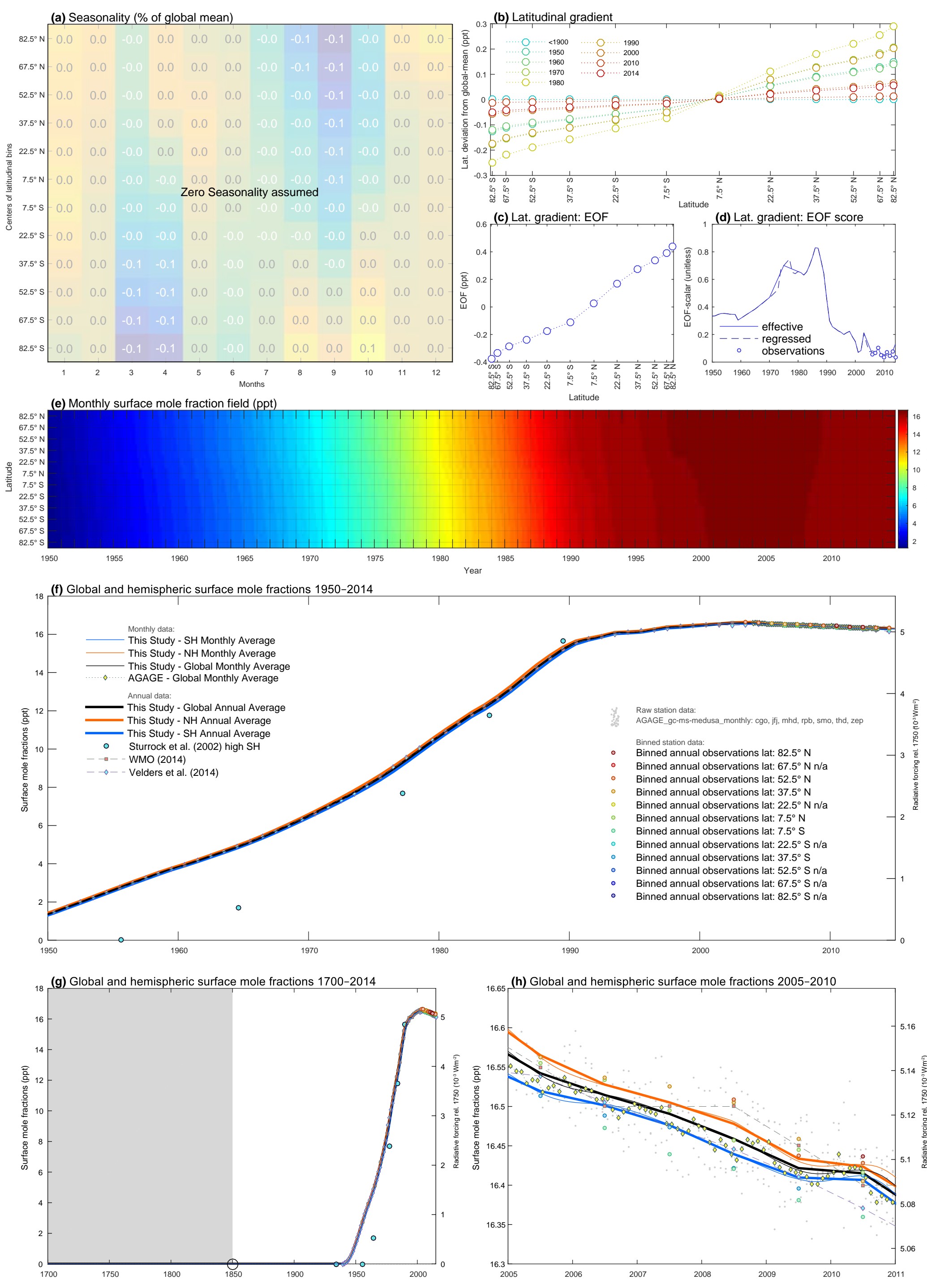

CFC-114 (CCIF₂CCIF₂): Lifetime: 190yrs ; Radiative Efficiency : 0.31 W m⁻² ppb⁻¹

Fig. 24

# CFC-115

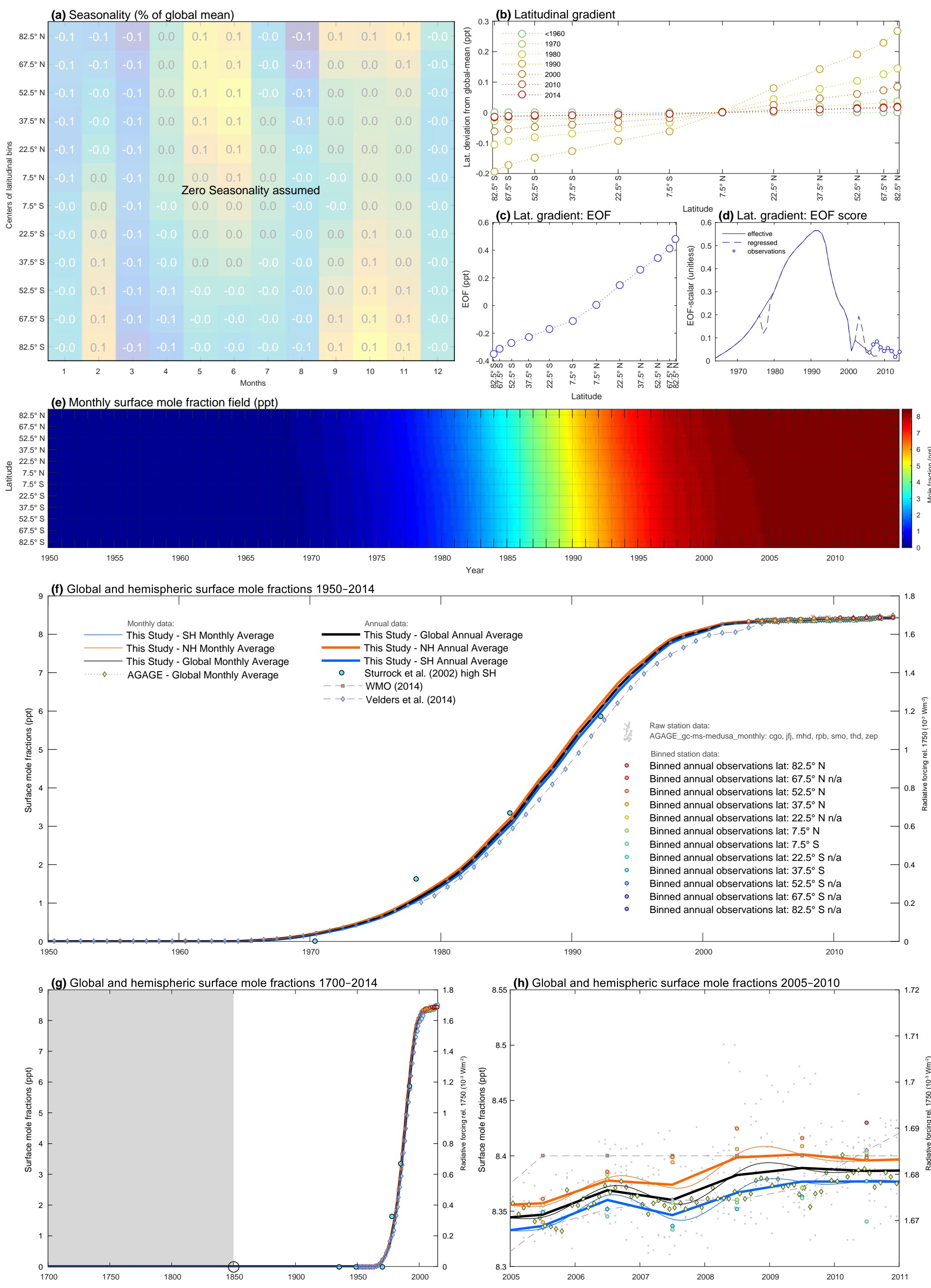

**(a)** Seasonality (% of global mean)

**(b)** Latitudinal gradient

**(c)** Lat. gradient: EOF

**(d)** Lat. gradient: EOF score

**(e)** Monthly surface mole fraction field (ppt)

**(f)** Global and hemispheric surface mole fractions 1950–2014

**(g)** Global and hemispheric surface mole fractions 1700–2014

**(h)** Global and hemispheric surface mole fractions 2005–2010

CFC-115 (CClF$_2$CF$_3$): Lifetime: 1020yrs ; Radiative Efficiency : 0.2 W m$^{-2}$ ppb$^{-1}$

Fig. 25

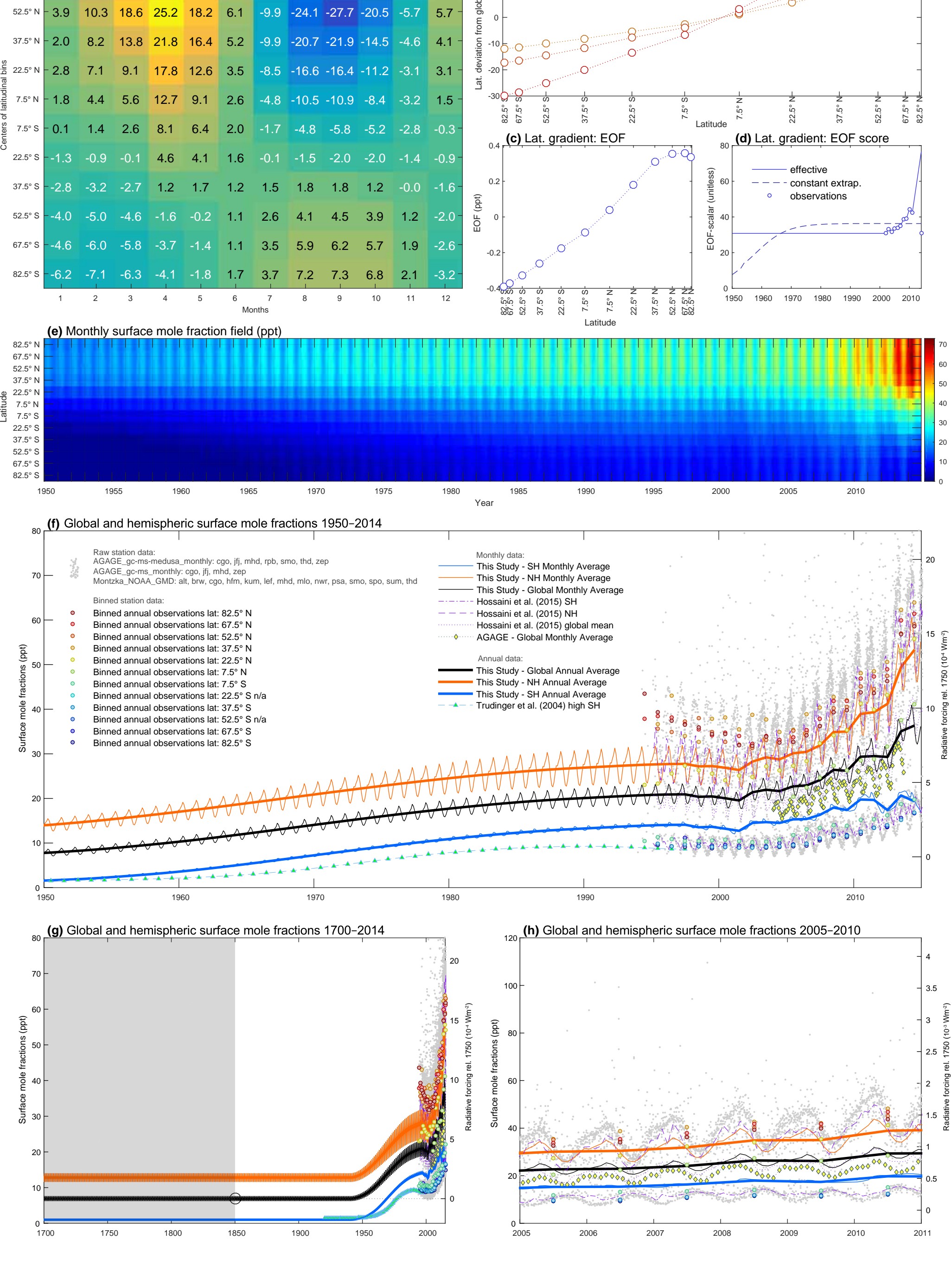

Fig. 26

# CH₃Br

Methyl bromide (CH₃Br): Lifetime: 0.8yrs ; Radiative Efficiency : 0.004 W m⁻² ppb⁻¹

**(a) Seasonality (% of global mean)**

**(b) Latitudinal gradient**

**(c) Lat. gradient: EOF**

**(d) Lat. gradient: EOF score**

**(e) Monthly surface mole fraction field (ppt)**

**(f) Global and hemispheric surface mole fractions 1950–2014**

**(g) Global and hemispheric surface mole fractions 1700–2014**

**(h) Global and hemispheric surface mole fractions 2005–2010**

Fig. 27

# CH₃CCl₃

Methyl chloroform (CH₃CCl₃): Lifetime: 5yrs ; Radiative Efficiency : 0.07 W m⁻² ppb⁻¹

**(a)** Seasonality (% of global mean)

**(b)** Latitudinal gradient

**(c)** Lat. gradient: EOF

**(d)** Lat. gradient: EOF score

**(e)** Monthly surface mole fraction field (ppt)

**(f)** Global and hemispheric surface mole fractions 1950–2014

**(g)** Global and hemispheric surface mole fractions 1700–2014

**(h)** Global and hemispheric surface mole fractions 2005–2010

Fig. 28

# CH₃Cl

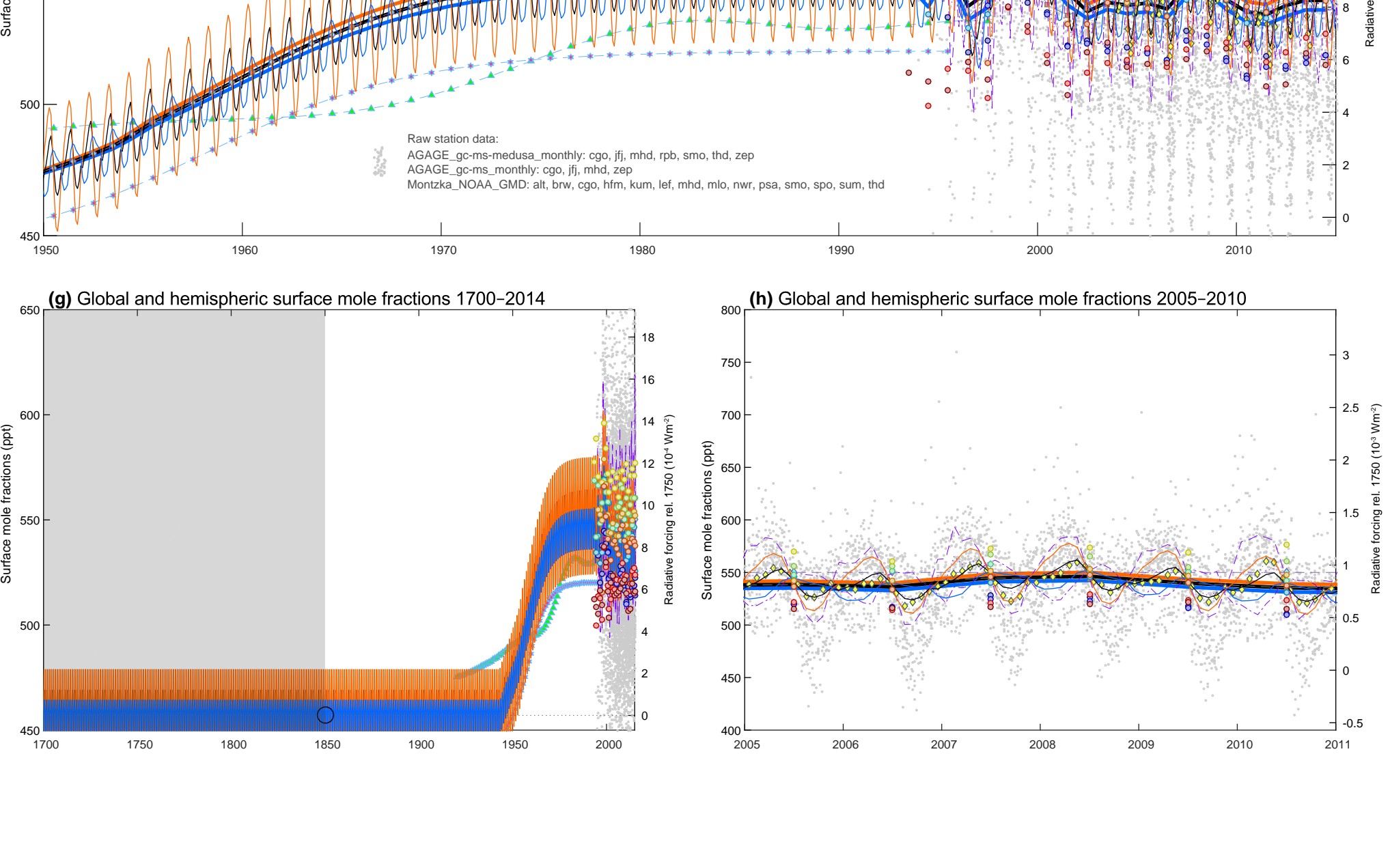

Fig. 29

# CHCl₃

**(a)** Seasonality (% of global mean)

| Centers of latitudinal bins | 1 | 2 | 3 | 4 | 5 | 6 | 7 | 8 | 9 | 10 | 11 | 12 |
|---|---|---|---|---|---|---|---|---|---|---|---|---|
| 82.5° N | 5.6 | -1.7 | -0.6 | -1.3 | -1.5 | -12.3 | -7.3 | -4.3 | 4.6 | 3.9 | 6.8 | 8.0 |
| 67.5° N | 4.2 | -1.6 | -0.2 | -0.6 | -1.2 | -10.4 | -6.7 | -4.6 | 4.4 | 4.5 | 5.9 | 6.3 |
| 52.5° N | 2.9 | -1.5 | 0.2 | 0.0 | -0.8 | -8.6 | -6.2 | -4.8 | 4.1 | 5.1 | 5.1 | 4.6 |
| 37.5° N | 2.7 | -0.4 | 0.5 | 0.4 | -0.6 | -7.4 | -6.2 | -5.2 | 3.0 | 4.8 | 4.9 | 3.5 |
| 22.5° N | 3.3 | 0.9 | 1.0 | 0.8 | 0.3 | -5.5 | -5.0 | -4.8 | 1.0 | 2.4 | 3.6 | 2.1 |
| 7.5° N | 4.4 | 2.0 | 1.4 | 1.0 | 1.9 | -2.9 | -2.6 | -3.9 | -1.6 | -1.4 | 1.4 | 0.4 |
| 7.5° S | 4.0 | 1.7 | 0.9 | 1.1 | 3.6 | -0.3 | -0.3 | -2.8 | -2.6 | -3.3 | -0.1 | -1.9 |
| 22.5° S | 2.9 | 1.2 | 0.1 | 1.4 | 4.7 | 1.1 | 0.3 | -2.5 | -2.9 | -3.1 | 0.3 | -3.7 |
| 37.5° S | 1.8 | 0.6 | -0.6 | 1.8 | 5.8 | 2.6 | 1.0 | -2.2 | -3.1 | -2.9 | 0.7 | -5.5 |
| 52.5° S | 0.7 | -0.0 | -1.4 | 2.1 | 6.7 | 3.7 | 1.5 | -1.9 | -3.1 | -2.5 | 1.2 | -7.0 |
| 67.5° S | -0.3 | -0.7 | -2.2 | 2.3 | 7.6 | 4.9 | 2.0 | -1.6 | -3.1 | -2.1 | 1.6 | -8.5 |
| 82.5° S | -1.3 | -1.4 | -3.0 | 2.5 | 8.5 | 6.0 | 2.5 | -1.2 | -3.0 | -1.7 | 2.1 | -9.9 |

Months

Chloroform (CHCl₃): Lifetime: 0.4yrs ; Radiative Efficiency : 0.08 W m⁻² ppb⁻¹

**(b)** Latitudinal gradient

**(c)** Lat. gradient: EOF

**(d)** Lat. gradient: EOF score

**(e)** Monthly surface mole fraction field (ppt)

**(f)** Global and hemispheric surface mole fractions 1950–2014

**(g)** Global and hemispheric surface mole fractions 1700–2014

**(h)** Global and hemispheric surface mole fractions 2005–2010

Fig. 30

# Halon-1211

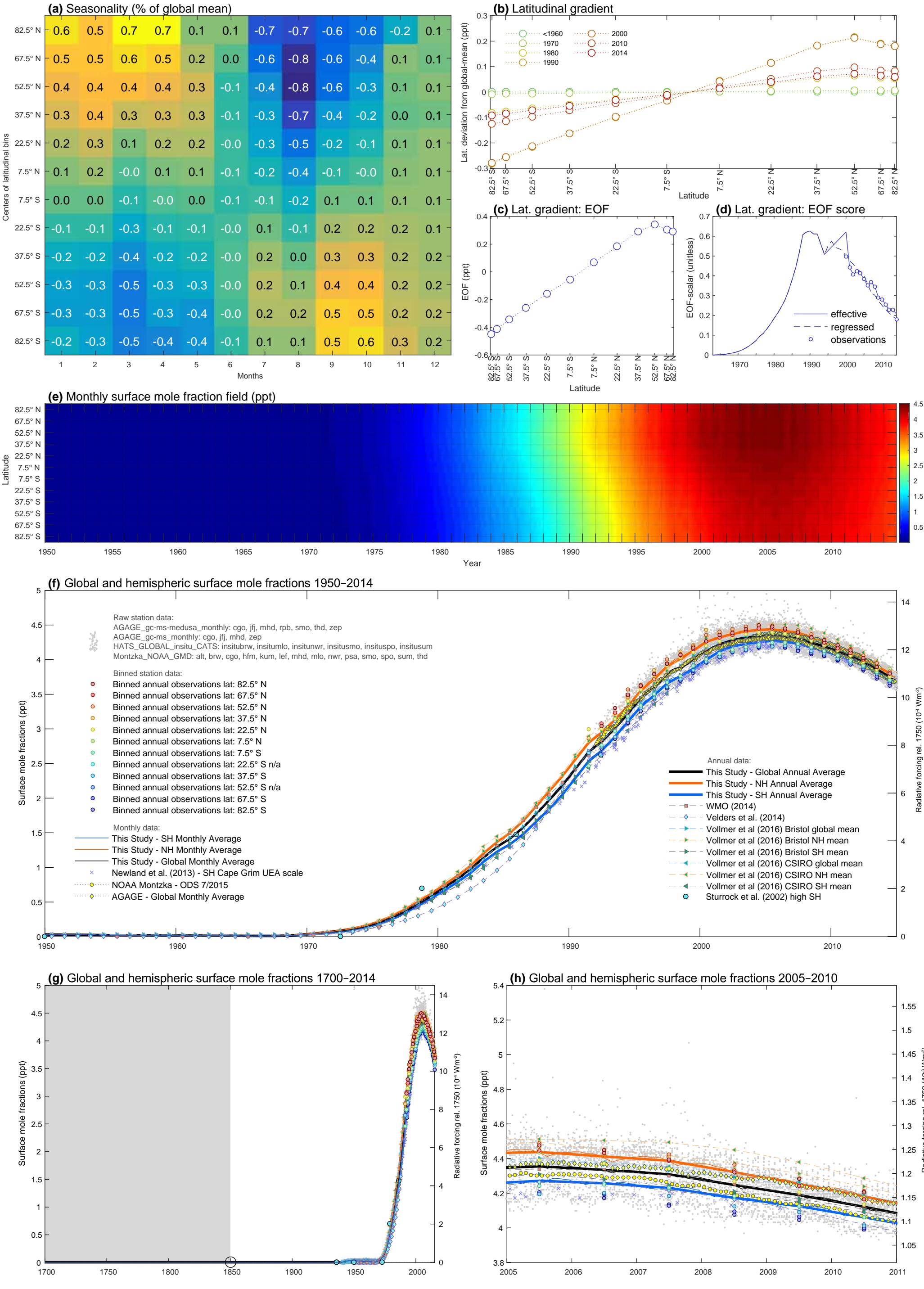

Halon-1211 (CBrClF$_2$): Lifetime: 16yrs ; Radiative Efficiency : 0.29 W m$^{-2}$ ppb$^{-1}$

**(a)** Seasonality (% of global mean)

**(b)** Latitudinal gradient

**(c)** Lat. gradient: EOF

**(d)** Lat. gradient: EOF score

**(e)** Monthly surface mole fraction field (ppt)

**(f)** Global and hemispheric surface mole fractions 1950–2014

**(g)** Global and hemispheric surface mole fractions 1700–2014

**(h)** Global and hemispheric surface mole fractions 2005–2010

Fig. 31

# Halon-1301

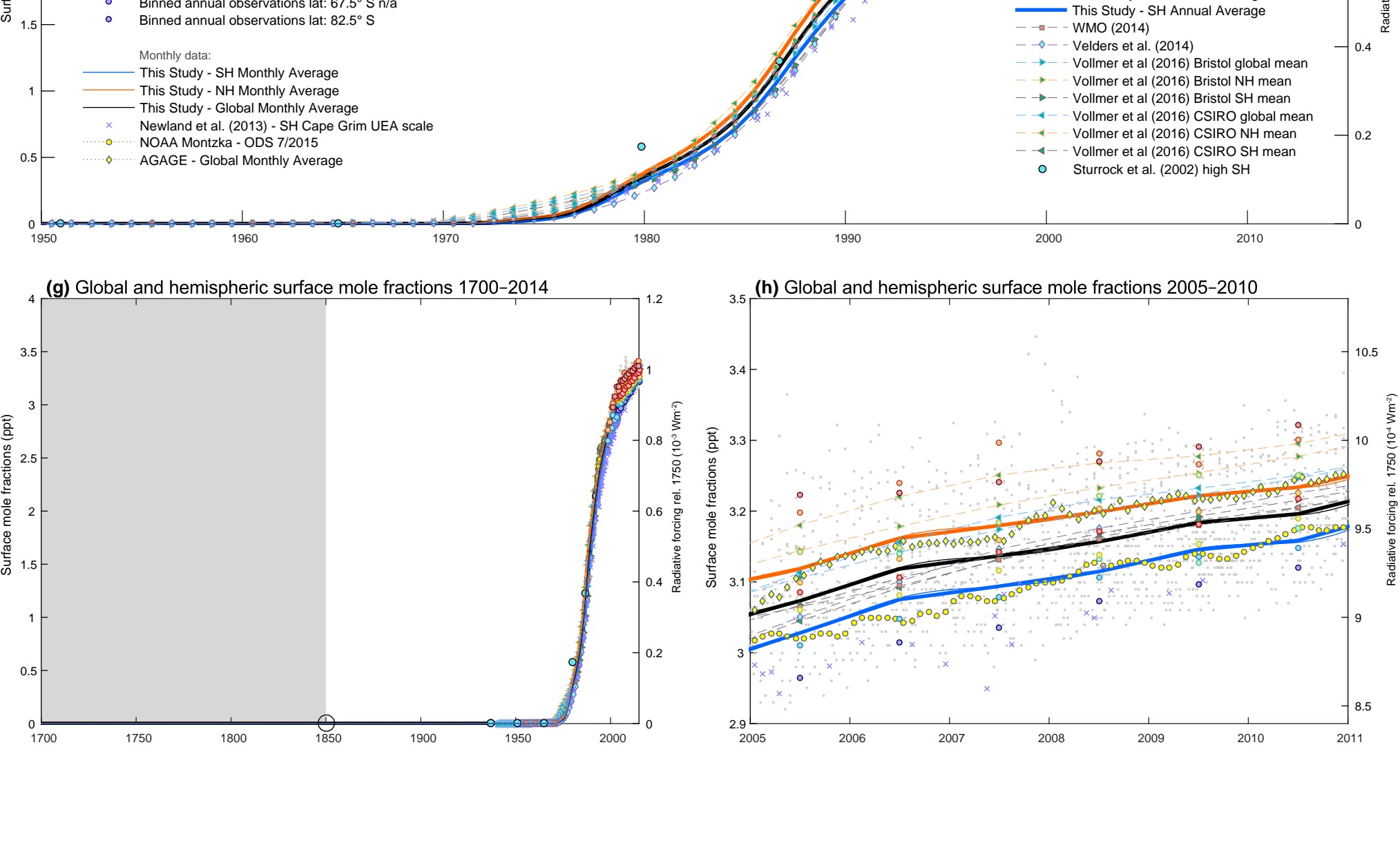

**(a)** Seasonality (% of global mean)

**(b)** Latitudinal gradient

Halon-1301 (CBrF$_3$): Lifetime: 65yrs ; Radiative Efficiency : 0.3 W m$^{-2}$ ppb$^{-1}$

**(c)** Lat. gradient: EOF

**(d)** Lat. gradient: EOF score

**(e)** Monthly surface mole fraction field (ppt)

**(f)** Global and hemispheric surface mole fractions 1950–2014

**(g)** Global and hemispheric surface mole fractions 1700–2014

**(h)** Global and hemispheric surface mole fractions 2005–2010

# Halon-2402

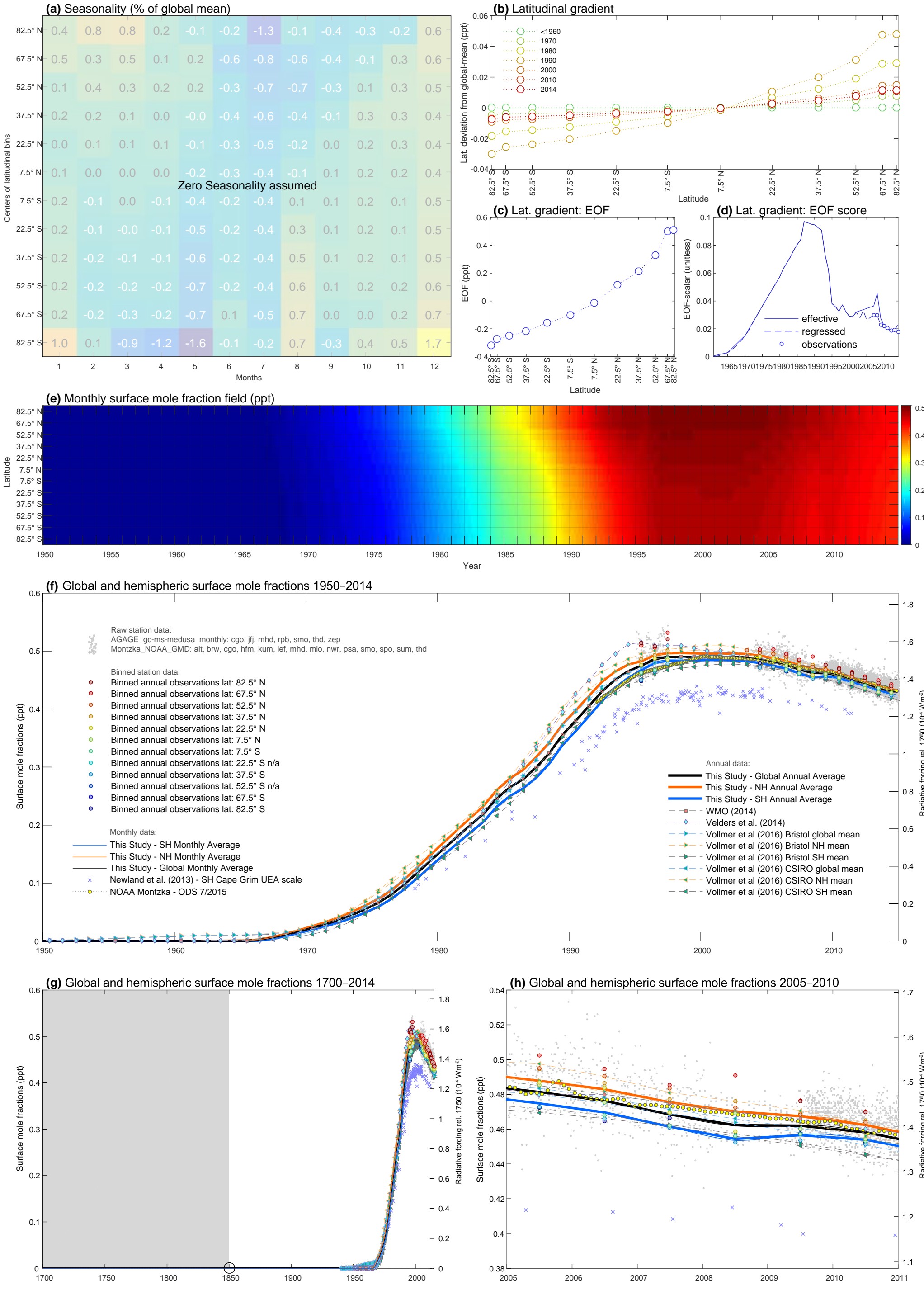

Fig. 33

# HCFC-22

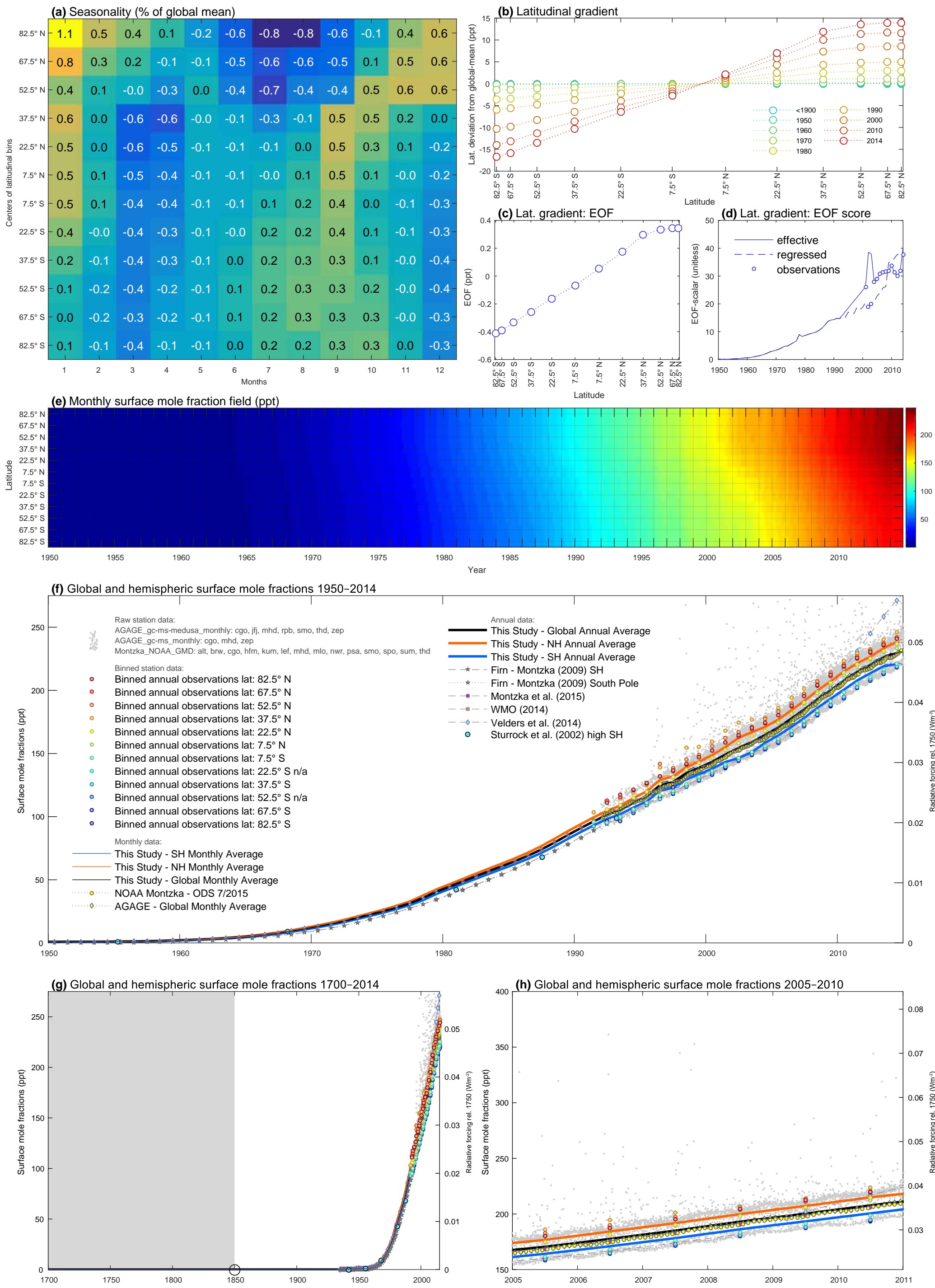

Fig. 34

# HCFC-141b

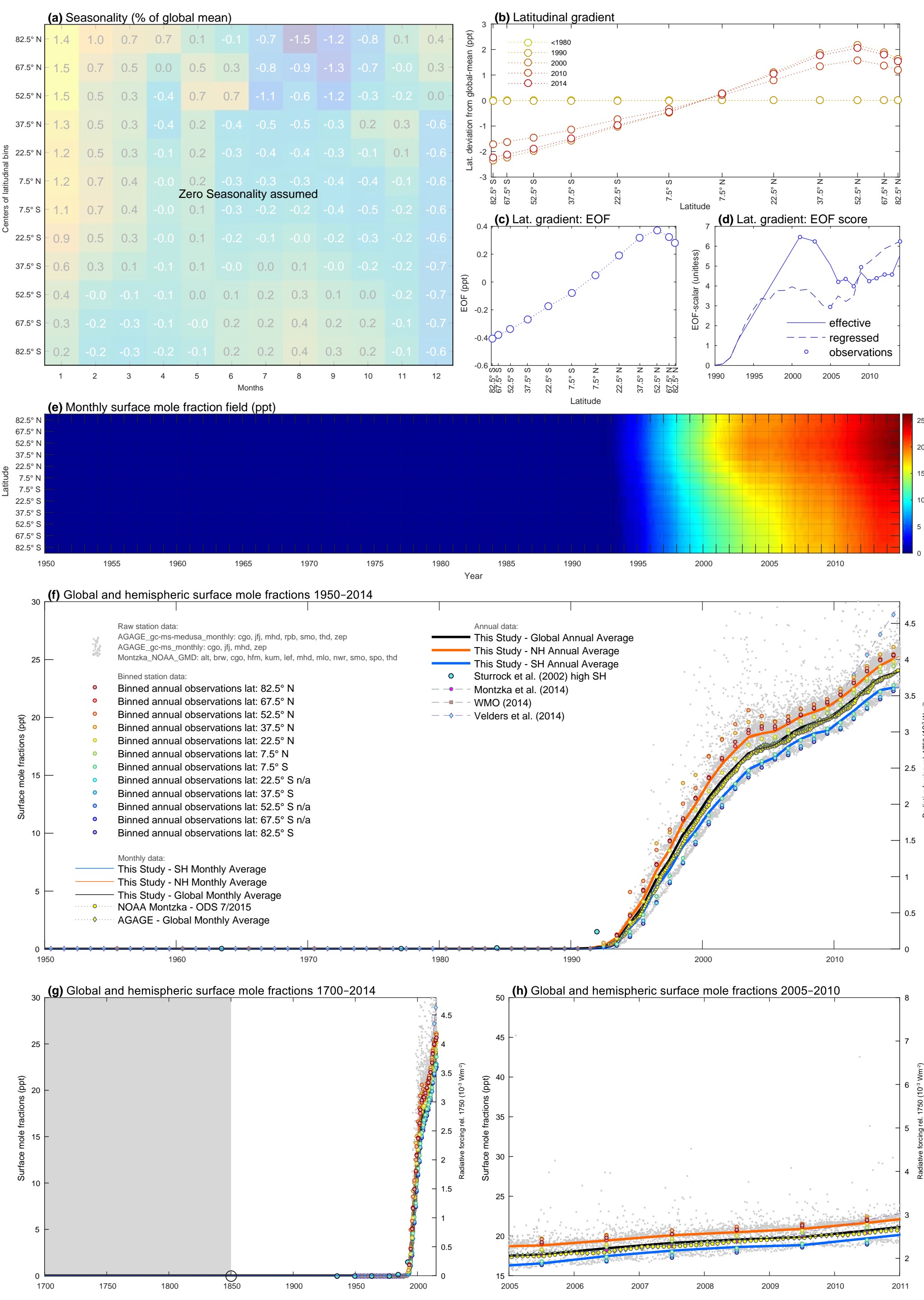

# HCFC-142b

HCFC-142b (CH₃CClF₂): Lifetime: 17.2yrs ; Radiative Efficiency : 0.19 W m⁻² ppb⁻¹

**(a)** Seasonality (% of global mean)

**(b)** Latitudinal gradient

**(c)** Lat. gradient: EOF

**(d)** Lat. gradient: EOF score

**(e)** Monthly surface mole fraction field (ppt)

**(f)** Global and hemispheric surface mole fractions 1950–2014

**(g)** Global and hemispheric surface mole fractions 1700–2014

**(h)** Global and hemispheric surface mole fractions 2005–2010

Fig. 36

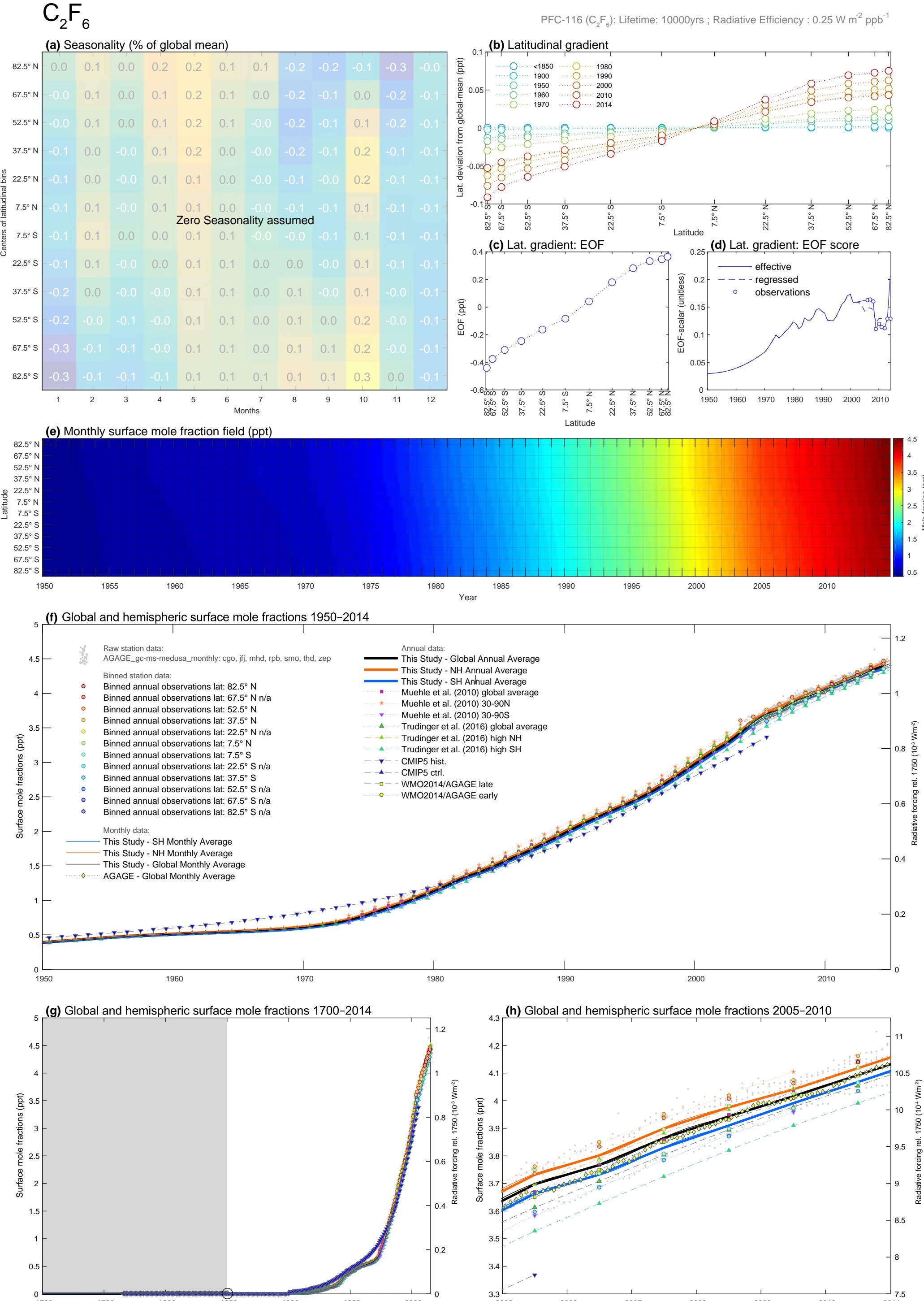

Fig. 37

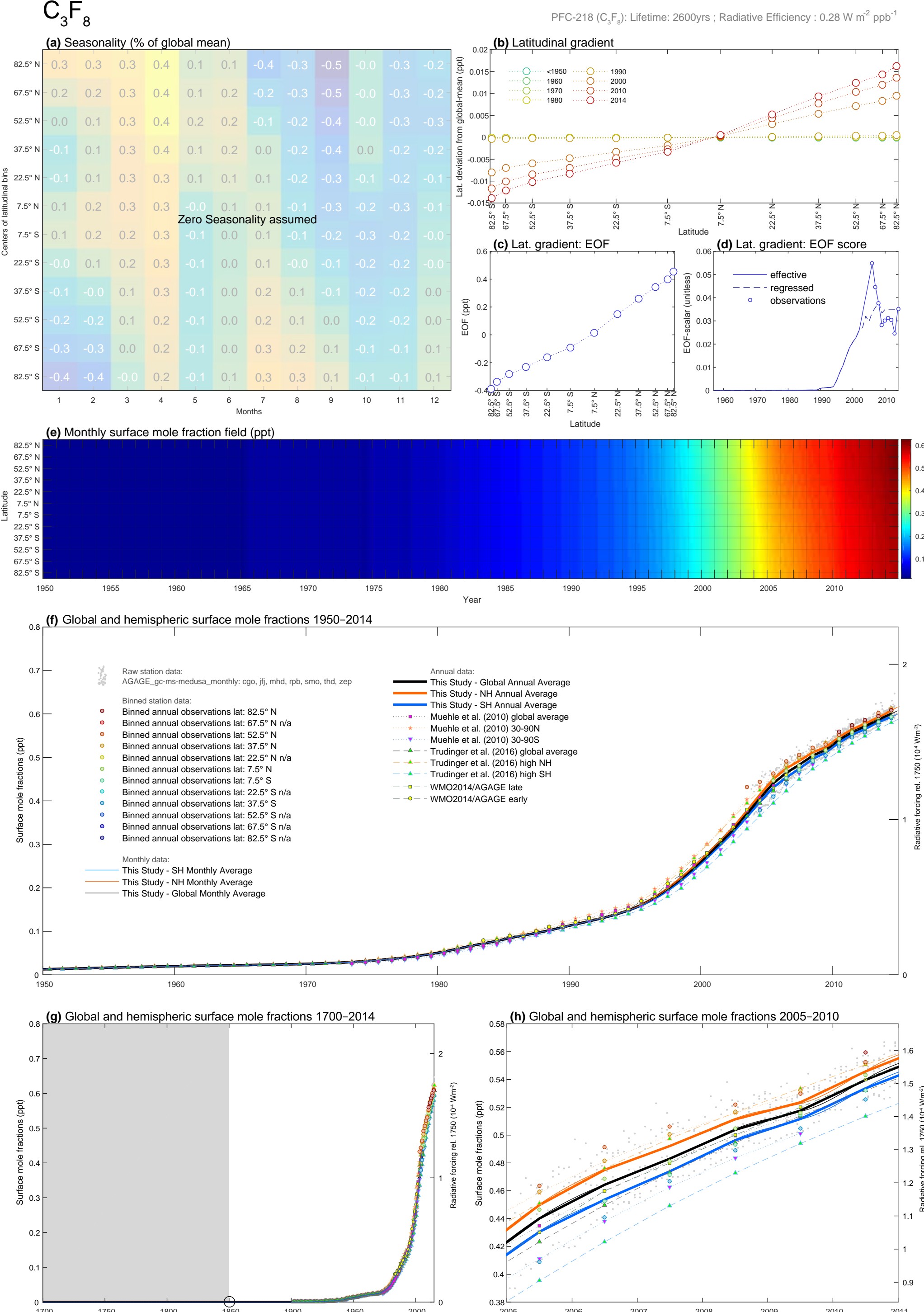

Fig. 38

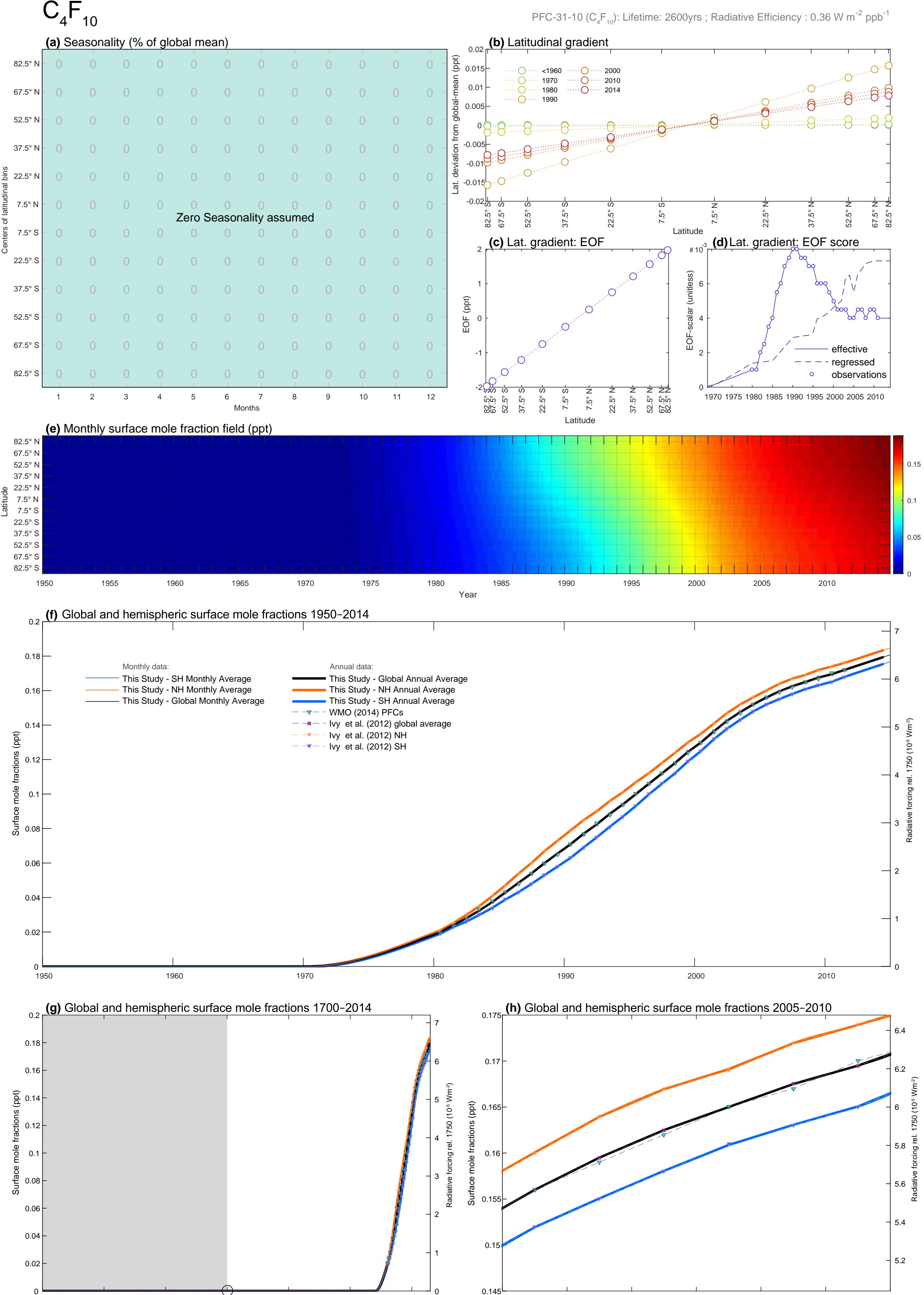

$C_4F_{10}$

PFC-31-10 ($C_4F_{10}$): Lifetime: 2600yrs ; Radiative Efficiency : 0.36 W m$^{-2}$ ppb$^{-1}$

**(a)** Seasonality (% of global mean)

Zero Seasonality assumed

**(b)** Latitudinal gradient

**(c)** Lat. gradient: EOF

**(d)** Lat. gradient: EOF score

**(e)** Monthly surface mole fraction field (ppt)

**(f)** Global and hemispheric surface mole fractions 1950–2014

**(g)** Global and hemispheric surface mole fractions 1700–2014

**(h)** Global and hemispheric surface mole fractions 2005–2010

Fig. 39

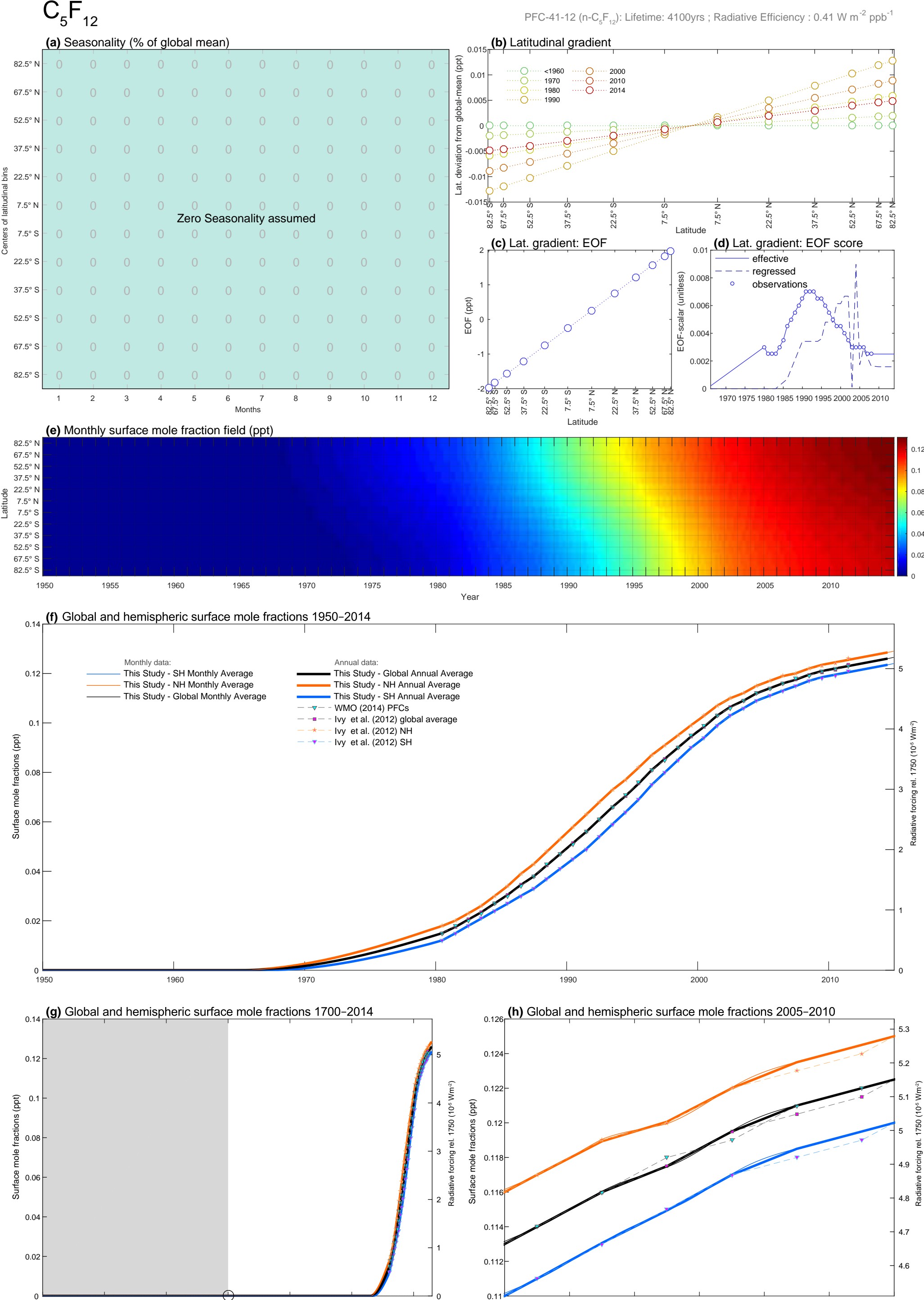

$C_5F_{12}$

PFC-41-12 (n-$C_5F_{12}$): Lifetime: 4100yrs ; Radiative Efficiency : 0.41 W m$^{-2}$ ppb$^{-1}$

**(a)** Seasonality (% of global mean)

Zero Seasonality assumed

**(b)** Latitudinal gradient

**(c)** Lat. gradient: EOF

**(d)** Lat. gradient: EOF score

**(e)** Monthly surface mole fraction field (ppt)

**(f)** Global and hemispheric surface mole fractions 1950–2014

Monthly data:
— This Study - SH Monthly Average
— This Study - NH Monthly Average
— This Study - Global Monthly Average

Annual data:
— This Study - Global Annual Average
— This Study - NH Annual Average
— This Study - SH Annual Average
WMO (2014) PFCs
Ivy et al. (2012) global average
Ivy et al. (2012) NH
Ivy et al. (2012) SH

**(g)** Global and hemispheric surface mole fractions 1700–2014

**(h)** Global and hemispheric surface mole fractions 2005–2010

Fig. 40

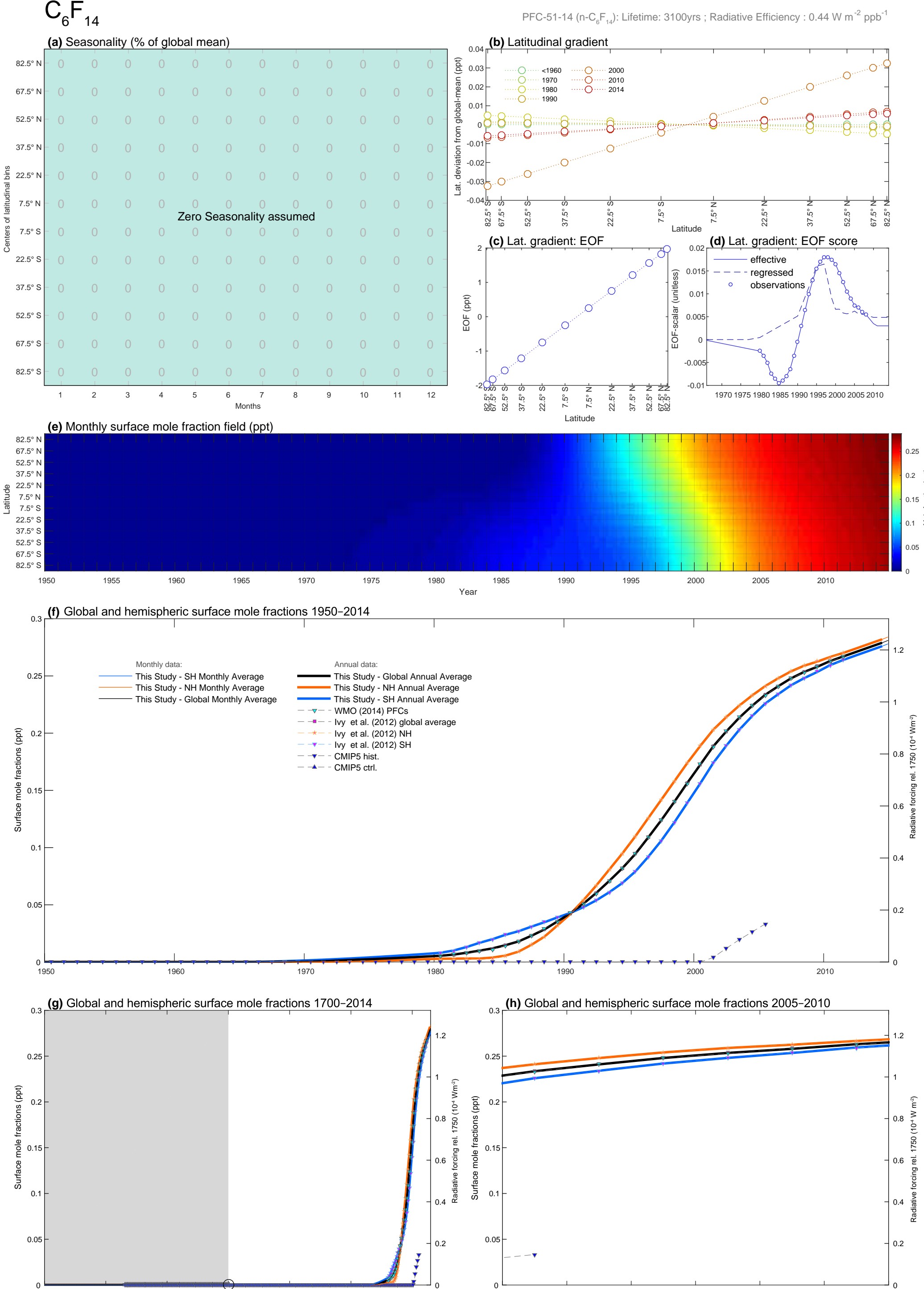

Fig. 41

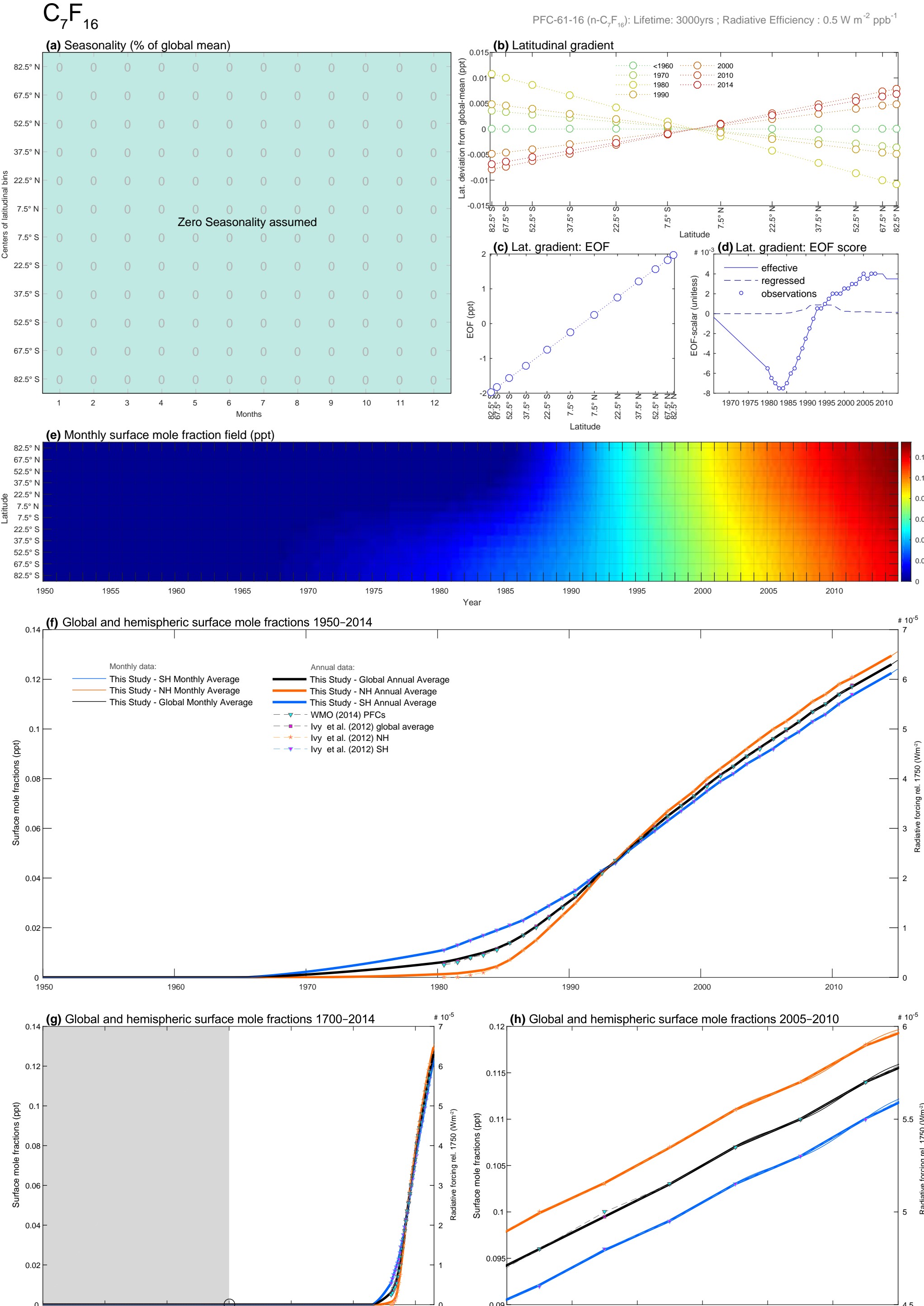

C₇F₁₆

PFC-61-16 (n-C₇F₁₆): Lifetime: 3000yrs ; Radiative Efficiency : 0.5 W m⁻² ppb⁻¹

(a) Seasonality (% of global mean)

(b) Latitudinal gradient

(c) Lat. gradient: EOF

(d) Lat. gradient: EOF score

(e) Monthly surface mole fraction field (ppt)

(f) Global and hemispheric surface mole fractions 1950–2014

(g) Global and hemispheric surface mole fractions 1700–2014

(h) Global and hemispheric surface mole fractions 2005–2010

Fig. 42

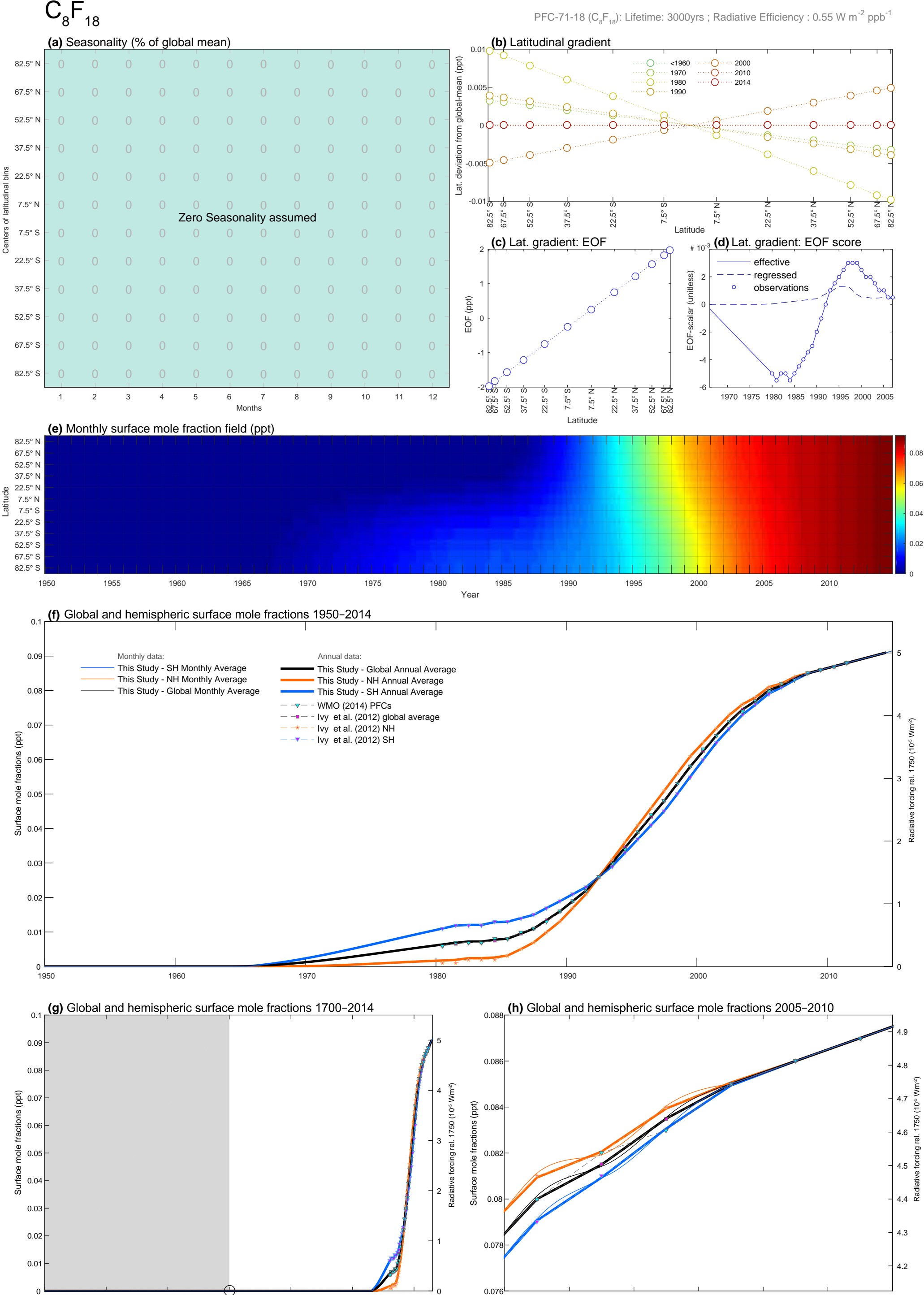

$C_8F_{18}$

(a) Seasonality (% of global mean)

Zero Seasonality assumed

(b) Latitudinal gradient

<1960  2000
1970  2010
1980  2014
1990

(c) Lat. gradient: EOF

(d) Lat. gradient: EOF score

effective
regressed
observations

(e) Monthly surface mole fraction field (ppt)

(f) Global and hemispheric surface mole fractions 1950–2014

Monthly data:
This Study - SH Monthly Average
This Study - NH Monthly Average
This Study - Global Monthly Average

Annual data:
This Study - Global Annual Average
This Study - NH Annual Average
This Study - SH Annual Average
WMO (2014) PFCs
Ivy et al. (2012) global average
Ivy et al. (2012) NH
Ivy et al. (2012) SH

(g) Global and hemispheric surface mole fractions 1700–2014

(h) Global and hemispheric surface mole fractions 2005–2010

Fig. 43

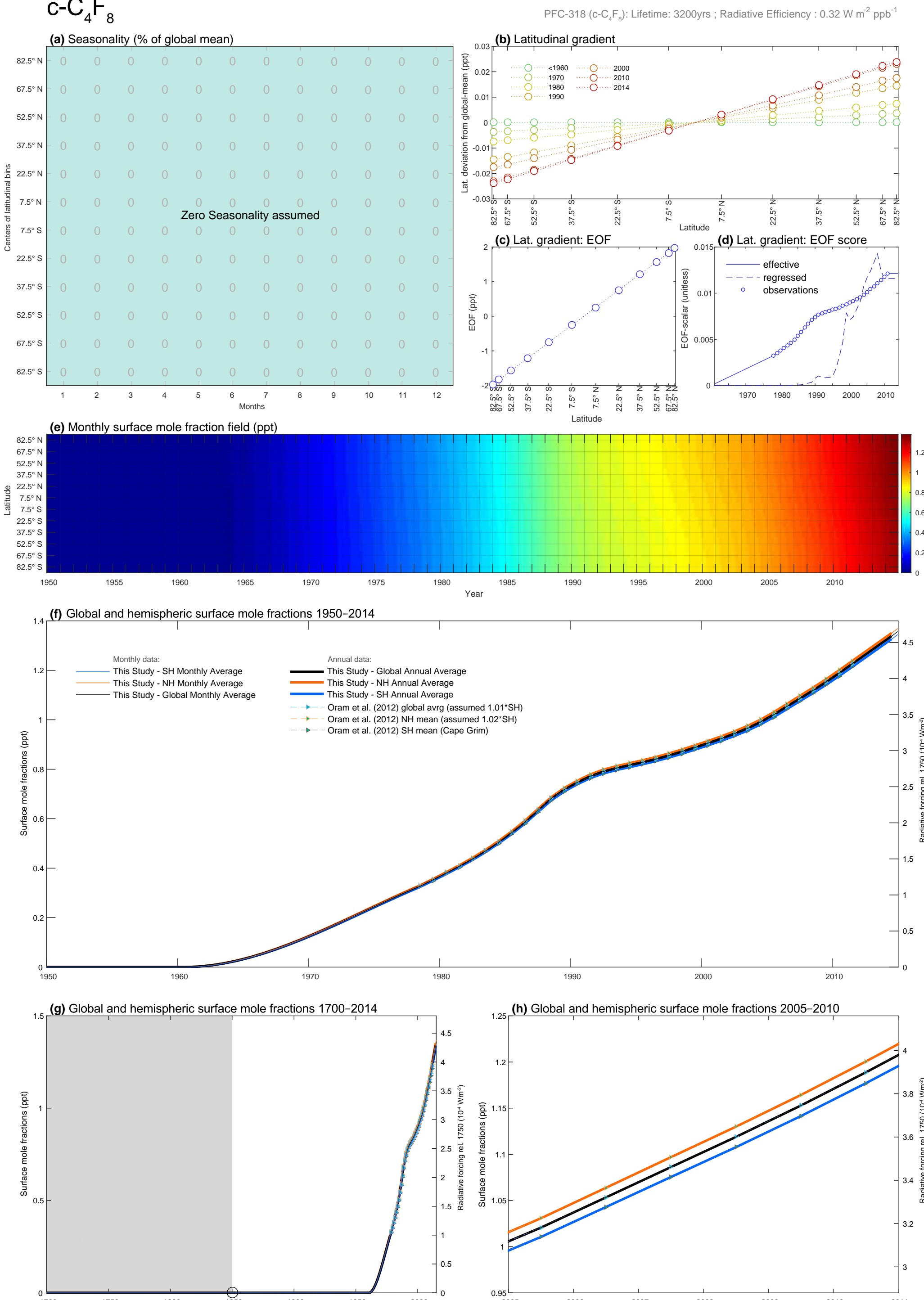

Fig. 44

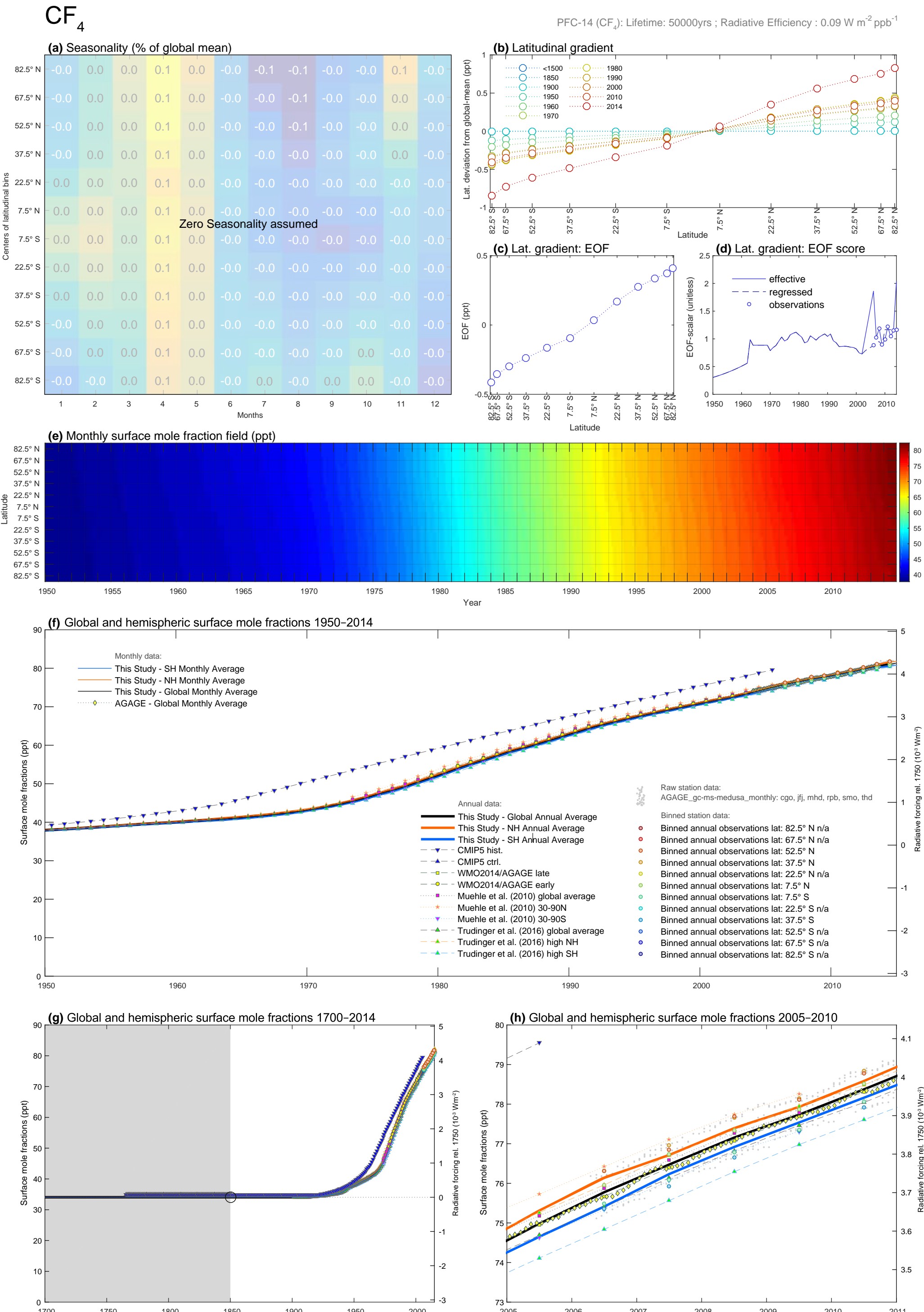

CF$_4$

PFC-14 (CF$_4$): Lifetime: 50000yrs ; Radiative Efficiency : 0.09 W m$^{-2}$ ppb$^{-1}$

**(a)** Seasonality (% of global mean)

**(b)** Latitudinal gradient

**(c)** Lat. gradient: EOF

**(d)** Lat. gradient: EOF score

**(e)** Monthly surface mole fraction field (ppt)

**(f)** Global and hemispheric surface mole fractions 1950–2014

**(g)** Global and hemispheric surface mole fractions 1700–2014

**(h)** Global and hemispheric surface mole fractions 2005–2010

Fig. 45

# HFC-23

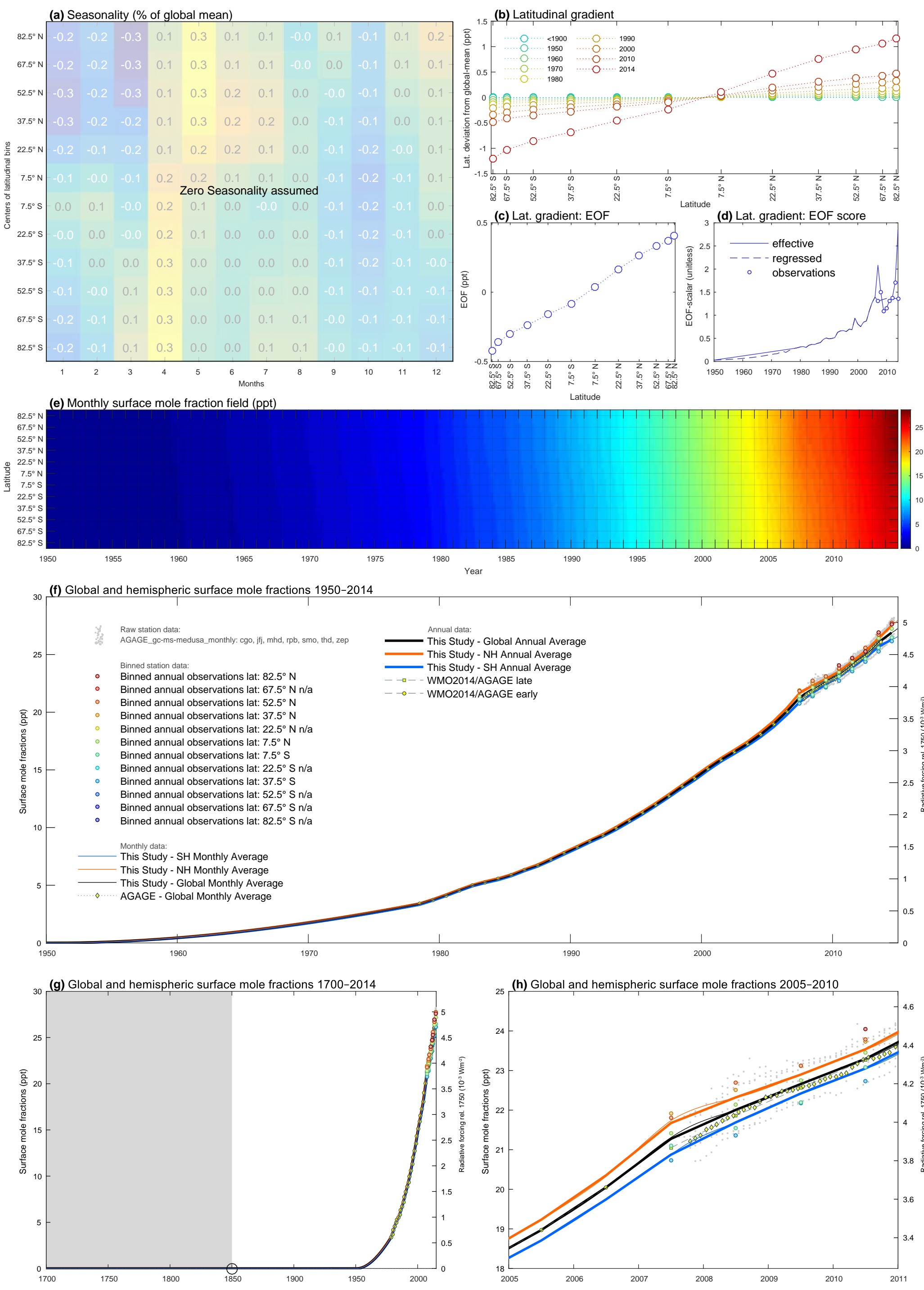

HFC-23 (CHF₃): Lifetime: 222yrs ; Radiative Efficiency : 0.18 W m⁻² ppb⁻¹

(a) Seasonality (% of global mean)

(b) Latitudinal gradient

(c) Lat. gradient: EOF

(d) Lat. gradient: EOF score

(e) Monthly surface mole fraction field (ppt)

(f) Global and hemispheric surface mole fractions 1950–2014

(g) Global and hemispheric surface mole fractions 1700–2014

(h) Global and hemispheric surface mole fractions 2005–2010

# HFC-32

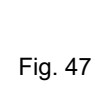

Fig. 47

# HFC-43-10mee

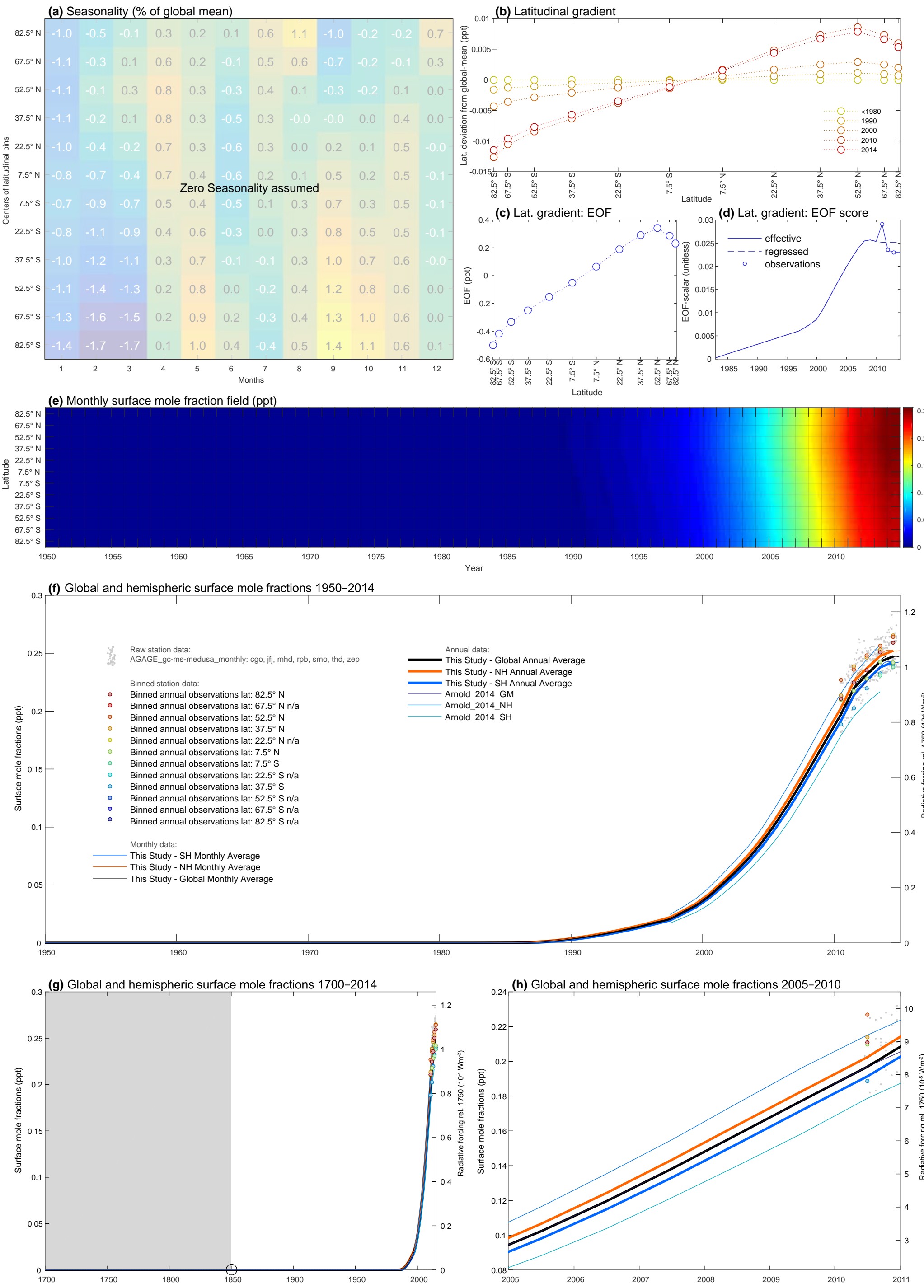

# HFC-125

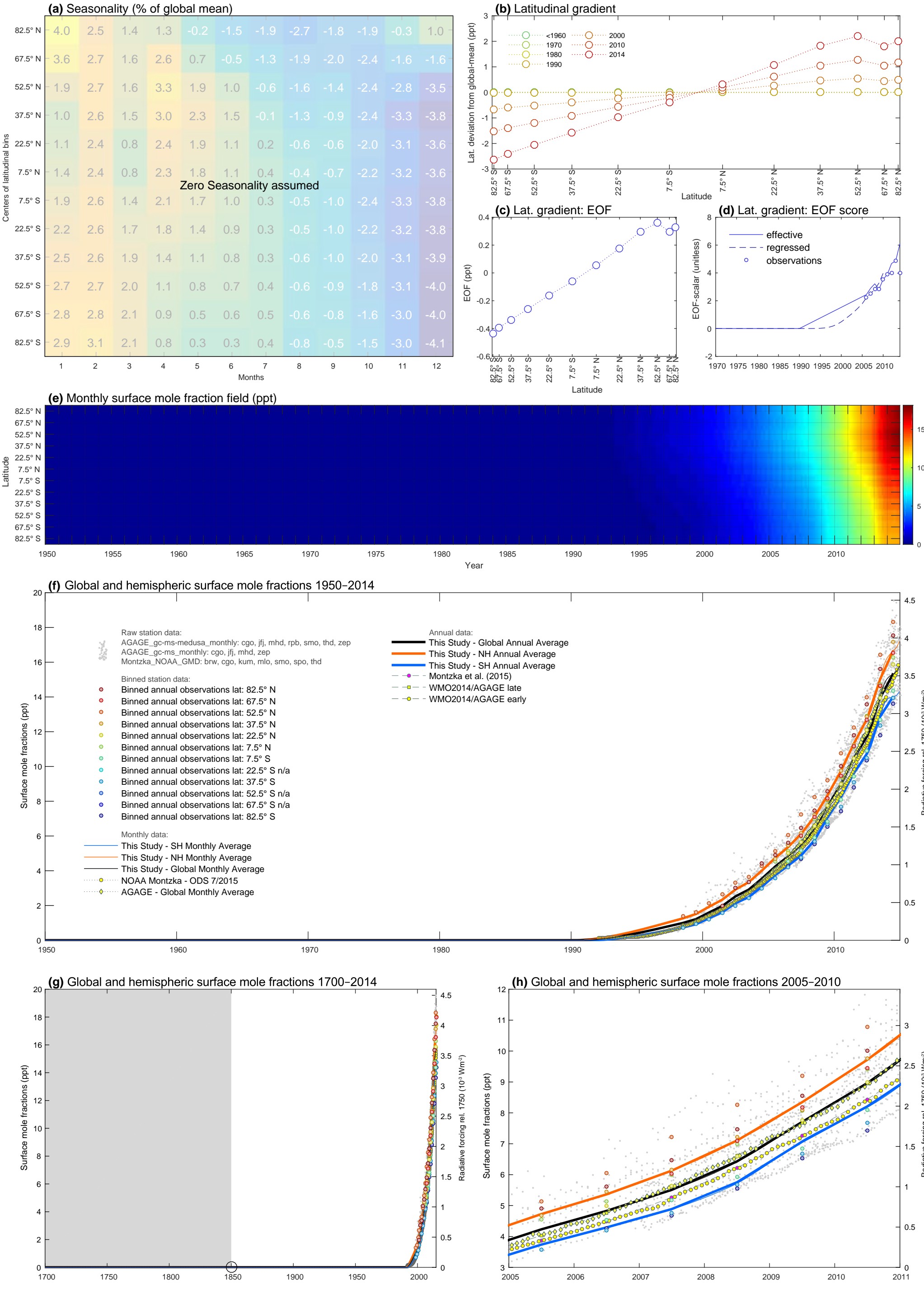

# HFC-134a

HFC-134a (CH$_2$FCF$_3$): Lifetime: 13.4yrs ; Radiative Efficiency : 0.16 W m$^{-2}$ ppb$^{-1}$

**(a)** Seasonality (% of global mean)

| | 1 | 2 | 3 | 4 | 5 | 6 | 7 | 8 | 9 | 10 | 11 | 12 |
|---|---|---|---|---|---|---|---|---|---|---|---|---|
| 82.5° N | 1.2 | 0.4 | -0.4 | -0.9 | -1.3 | -1.3 | -1.2 | -0.9 | -0.1 | 0.7 | 1.8 | 2.1 |
| 67.5° N | -0.0 | -0.2 | -1.1 | -0.9 | -0.4 | -0.6 | -0.7 | -0.3 | 0.4 | 0.8 | 1.6 | 1.4 |
| 52.5° N | -1.6 | -1.3 | -2.0 | -1.2 | 0.4 | -0.1 | 0.0 | 0.2 | 1.2 | 1.4 | 2.5 | 0.4 |
| 37.5° N | -1.4 | -1.5 | -2.9 | -1.5 | -0.0 | 0.5 | 1.3 | 1.0 | 2.5 | 1.4 | 1.5 | -0.9 |
| 22.5° N | -1.3 | -1.6 | -2.7 | -1.2 | 0.0 | 0.5 | 1.4 | 1.1 | 1.8 | 0.9 | 2.1 | -1.0 |
| 7.5° N | -0.9 | -1.3 | -2.2 | -1.0 | 0.0 | 0.4 | 1.3 | 1.1 | 1.4 | 0.5 | 1.7 | -1.0 |
| 7.5° S | -0.6 | -1.0 | -1.7 | -0.8 | -0.0 | 0.2 | 1.0 | 0.9 | 1.0 | 0.3 | 1.7 | -0.9 |
| 22.5° S | -0.5 | -0.9 | -1.4 | -0.7 | -0.0 | 0.2 | 0.8 | 0.7 | 0.8 | 0.3 | 1.3 | -0.8 |
| 37.5° S | -0.4 | -0.8 | -1.1 | -0.5 | -0.0 | 0.2 | 0.6 | 0.6 | 0.6 | 0.3 | 0.9 | -0.6 |
| 52.5° S | -0.3 | -0.7 | -0.8 | -0.4 | -0.0 | 0.2 | 0.4 | 0.5 | 0.5 | 0.4 | 0.6 | -0.4 |
| 67.5° S | -0.1 | -0.5 | -0.6 | -0.3 | -0.0 | 0.2 | 0.4 | 0.4 | 0.3 | 0.4 | 0.3 | -0.2 |
| 82.5° S | -0.0 | -0.4 | -0.6 | -0.3 | -0.1 | 0.2 | 0.1 | 0.3 | 0.4 | 0.5 | 0.1 | -0.1 |

Centers of latitudinal bins / Months

**(b)** Latitudinal gradient

Legend: <1980, 1990, 2000, 2010, 2014

**(c)** Lat. gradient: EOF

**(d)** Lat. gradient: EOF score

effective / regressed / observations

**(e)** Monthly surface mole fraction field (ppt)

**(f)** Global and hemispheric surface mole fractions 1950–2014

Raw station data:
AGAGE_gc-ms-medusa_monthly: cgo, jfj, mhd, rpb, smo, thd, zep
AGAGE_gc-ms_monthly: cgo, jfj, mhd, zep
Montzka_NOAA_GMD: alt, brw, cgo, hfm, kum, lef, mhd, mlo, nwr, psa, smo, spo, sum, thd

Binned station data:
○ Binned annual observations lat: 82.5° N
○ Binned annual observations lat: 67.5° N
○ Binned annual observations lat: 52.5° N
○ Binned annual observations lat: 37.5° N
○ Binned annual observations lat: 22.5° N
○ Binned annual observations lat: 7.5° N
○ Binned annual observations lat: 7.5° S
○ Binned annual observations lat: 22.5° S n/a
○ Binned annual observations lat: 37.5° S n/a
○ Binned annual observations lat: 52.5° S n/a
○ Binned annual observations lat: 67.5° S
○ Binned annual observations lat: 82.5° S

Monthly data:
— This Study - SH Monthly Average
— This Study - NH Monthly Average
— This Study - Global Monthly Average
— Martinerie-2010 - monthly global
— Martinerie-2010 - monthly NH
— Martinerie-2010 - monthly SH
— Martinerie-2010 - monthly high NH
— NOAA Montzka - ODS 7/2015
— AGAGE - Global Monthly Average

Annual data:
— This Study - Global Annual Average
— This Study - NH Annual Average
— This Study - SH Annual Average
Firn - Montzka (2009) South Pole
Montzka et al. (2014)
WMO2014/AGAGE late

**(g)** Global and hemispheric surface mole fractions 1700–2014

**(h)** Global and hemispheric surface mole fractions 2005–2010

Fig. 50

# HFC-143a

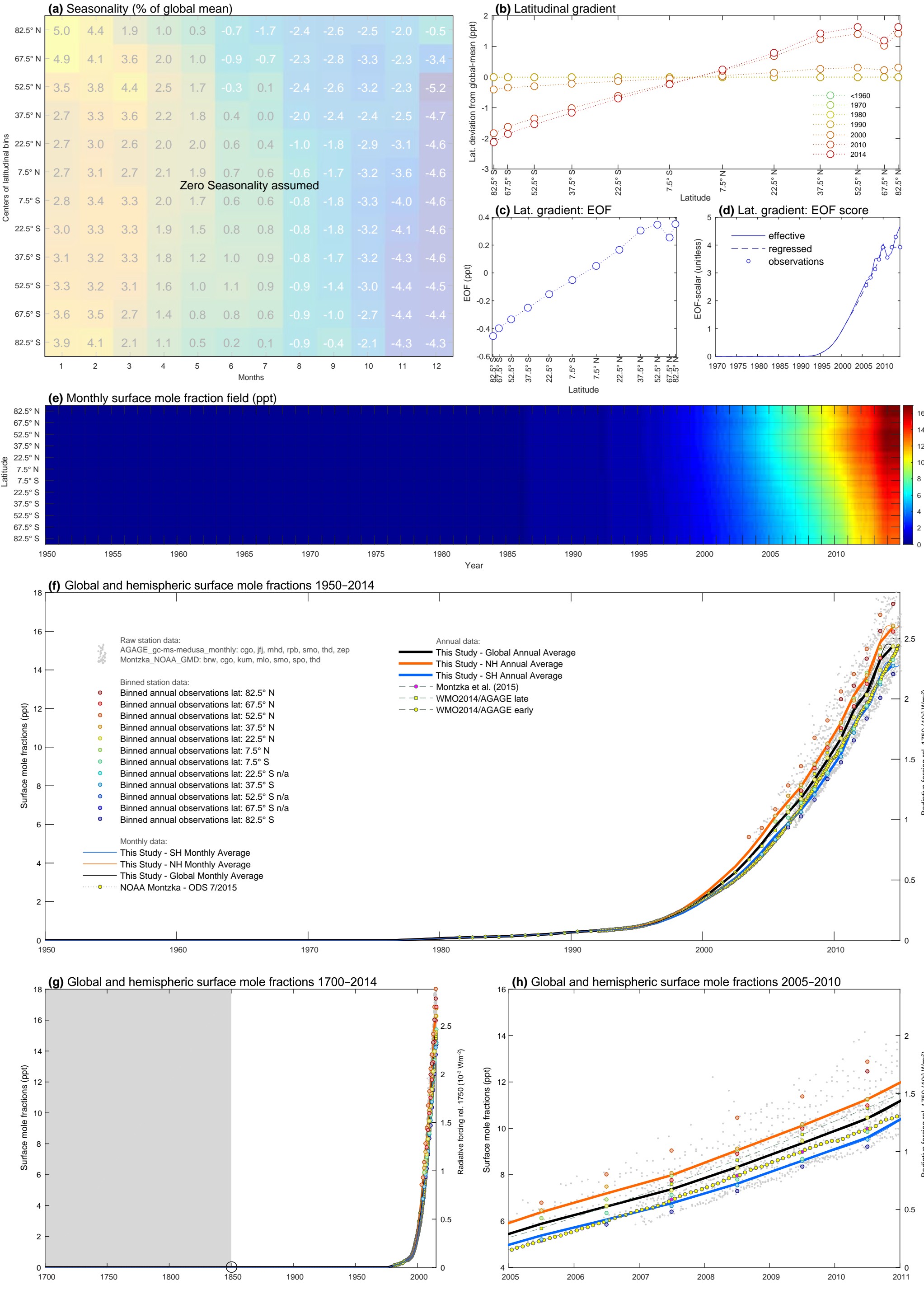

HFC-143a (CH₃CF₃): Lifetime: 47.1yrs ; Radiative Efficiency : 0.16 W m⁻² ppb⁻¹

**(a)** Seasonality (% of global mean)

**(b)** Latitudinal gradient

**(c)** Lat. gradient: EOF

**(d)** Lat. gradient: EOF score

**(e)** Monthly surface mole fraction field (ppt)

**(f)** Global and hemispheric surface mole fractions 1950–2014

**(g)** Global and hemispheric surface mole fractions 1700–2014

**(h)** Global and hemispheric surface mole fractions 2005–2010

Fig. 51

# HFC-152a

HFC-152a (CH$_3$CHF$_2$): Lifetime: 1.5yrs ; Radiative Efficiency : 0.1 W m$^{-2}$ ppb$^{-1}$

**(a)** Seasonality (% of global mean)

**(b)** Latitudinal gradient

**(c)** Lat. gradient: EOF

**(d)** Lat. gradient: EOF score

**(e)** Monthly surface mole fraction field (ppt)

**(f)** Global and hemispheric surface mole fractions 1950–2014

**(g)** Global and hemispheric surface mole fractions 1700–2014

**(h)** Global and hemispheric surface mole fractions 2005–2010

Fig. 52

# HFC-227ea

HFC-227ea (CF$_3$CHFCF$_3$): Lifetime: 38.9yrs ; Radiative Efficiency : 0.26 W m$^{-2}$ ppb$^{-1}$

**(a)** Seasonality (% of global mean)

**(b)** Latitudinal gradient

**(c)** Lat. gradient: EOF

**(d)** Lat. gradient: EOF score

**(e)** Monthly surface mole fraction field (ppt)

**(f)** Global and hemispheric surface mole fractions 1950–2014

**(g)** Global and hemispheric surface mole fractions 1700–2014

**(h)** Global and hemispheric surface mole fractions 2005–2010

Fig. 53

# HFC-236fa

HFC-236fa (CF$_3$CH$_2$CF$_3$): Lifetime: 242yrs ; Radiative Efficiency : 0.24 W m$^{-2}$ ppb$^{-1}$

**(a)** Seasonality (% of global mean)

**(b)** Latitudinal gradient

**(c)** Lat. gradient: EOF

**(d)** Lat. gradient: EOF score

**(e)** Monthly surface mole fraction field (ppt)

**(f)** Global and hemispheric surface mole fractions 1950–2014

Raw station data:
AGAGE_gc-ms-medusa_monthly: cgo, jfj, mhd, rpb, smo, thd, zep

Binned station data:
Binned annual observations lat: 82.5° N
Binned annual observations lat: 67.5° N n/a
Binned annual observations lat: 52.5° N
Binned annual observations lat: 37.5° N
Binned annual observations lat: 22.5° N n/a
Binned annual observations lat: 7.5° N
Binned annual observations lat: 7.5° S
Binned annual observations lat: 22.5° S n/a
Binned annual observations lat: 37.5° S
Binned annual observations lat: 52.5° S n/a
Binned annual observations lat: 67.5° S n/a
Binned annual observations lat: 82.5° S n/a

Monthly data:
This Study - SH Monthly Average
This Study - NH Monthly Average
This Study - Global Monthly Average
AGAGE - Global Monthly Average

Annual data:
This Study - Global Annual Average
This Study - NH Annual Average
This Study - SH Annual Average
WMO2014/AGAGE late
WMO2014/AGAGE early

**(g)** Global and hemispheric surface mole fractions 1700–2014

**(h)** Global and hemispheric surface mole fractions 2005–2010

Fig. 54

# HFC-245fa

HFC-245fa (CHF$_2$CH$_2$CF$_3$): Lifetime: 7.7yrs ; Radiative Efficiency : 0.24 W m$^{-2}$ ppb$^{-1}$

**(a)** Seasonality (% of global mean)

**(b)** Latitudinal gradient

**(c)** Lat. gradient: EOF

**(d)** Lat. gradient: EOF score

**(e)** Monthly surface mole fraction field (ppt)

**(f)** Global and hemispheric surface mole fractions 1950–2014

**(g)** Global and hemispheric surface mole fractions 1700–2014

**(h)** Global and hemispheric surface mole fractions 2005–2010

Fig. 55

# HFC-365mfc

HFC-365mfc (CH₃CF₂CH₂CF₃): Lifetime: 8.7yrs ; Radiative Efficiency : 0.22 W m⁻² ppb⁻¹

**(a)** Seasonality (% of global mean)

**(b)** Latitudinal gradient

**(c)** Lat. gradient: EOF

**(d)** Lat. gradient: EOF score

**(e)** Monthly surface mole fraction field (ppt)

**(f)** Global and hemispheric surface mole fractions 1950–2014

**(g)** Global and hemispheric surface mole fractions 1700–2014

**(h)** Global and hemispheric surface mole fractions 2005–2010

Fig. 56

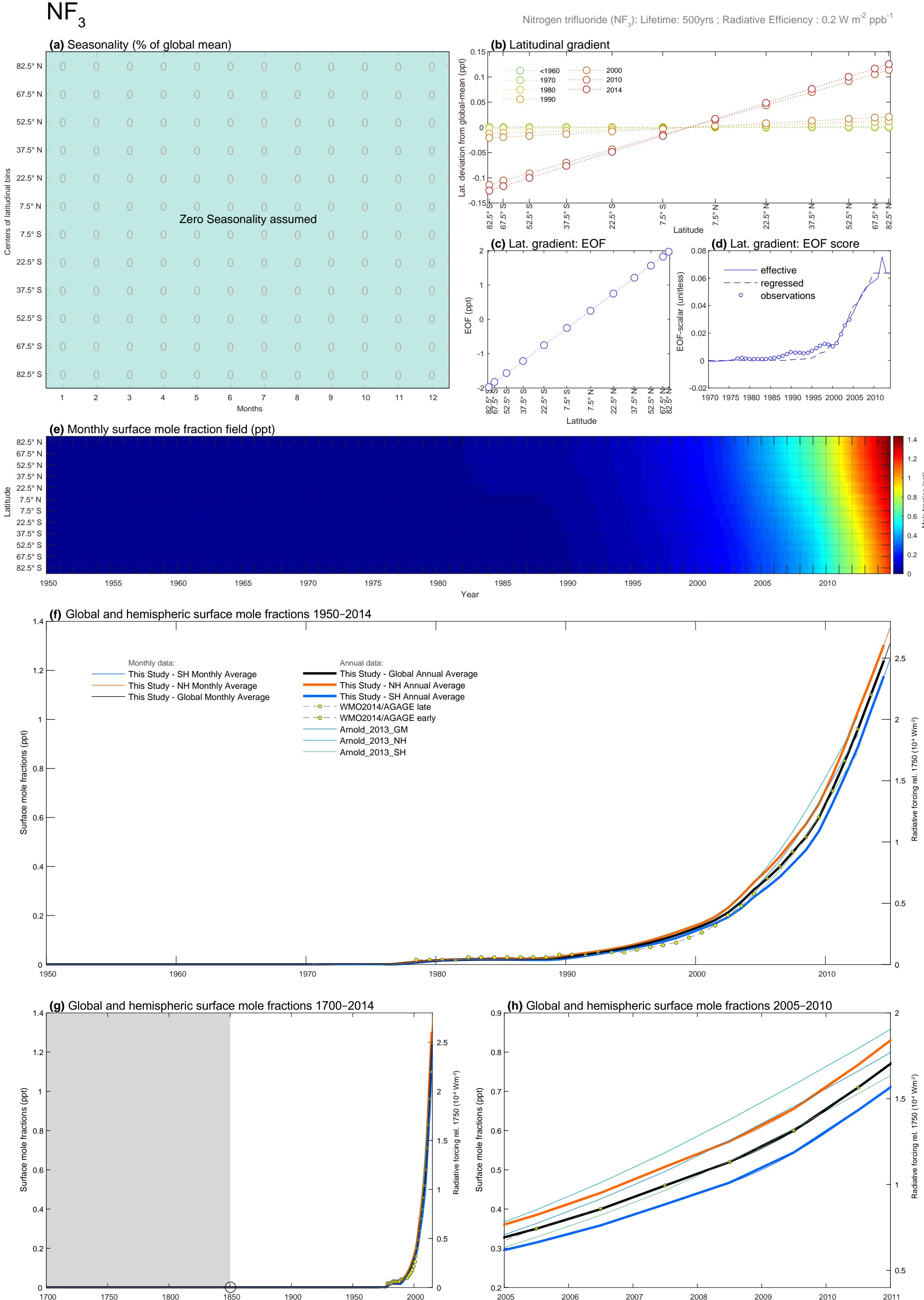

Fig. 57

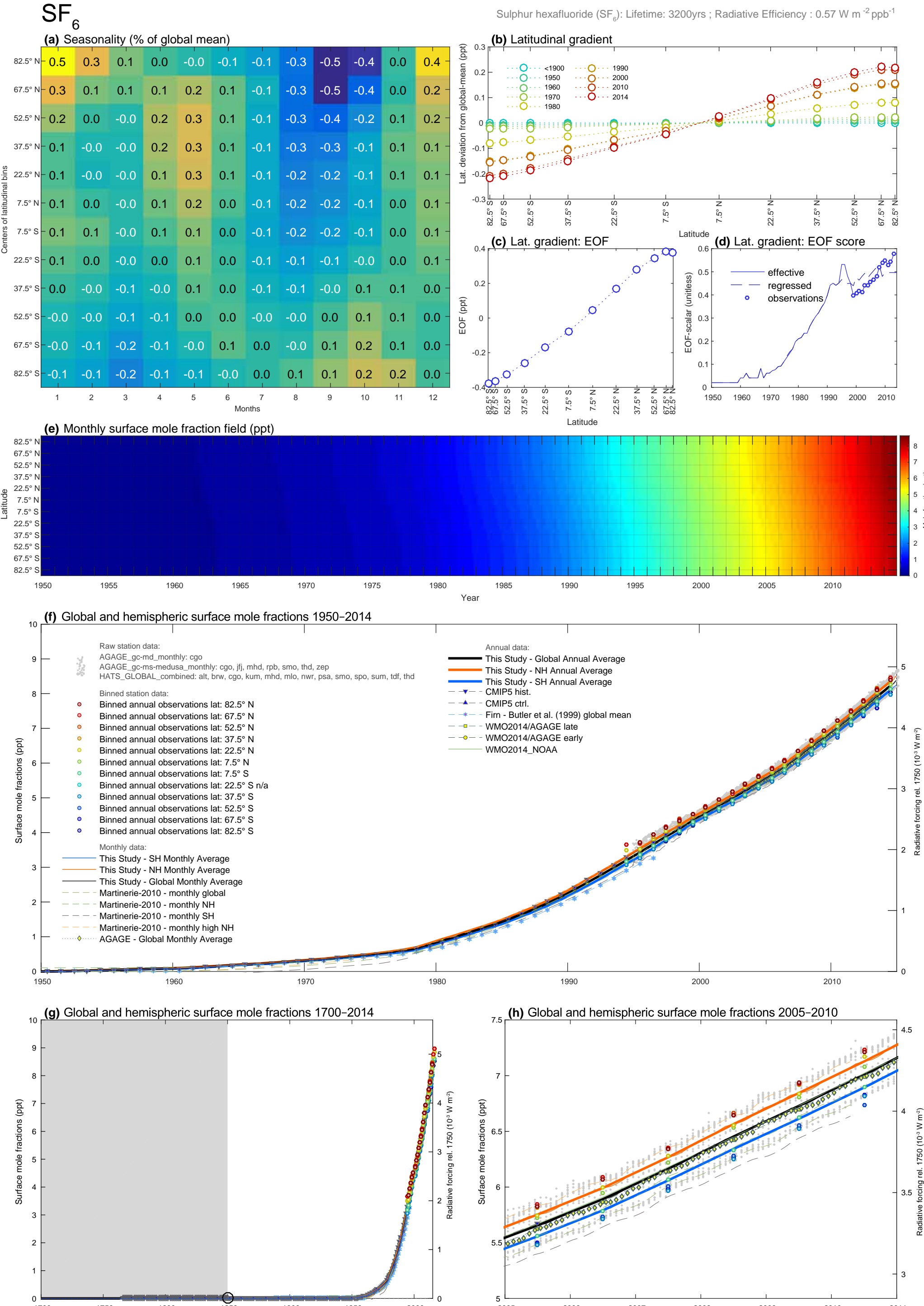

Fig. 58

# SO$_2$F$_2$

Sulphuryl fluoride (SO$_2$F$_2$): Lifetime: 36yrs ; Radiative Efficiency : 0.2 W m$^{-2}$ ppb$^{-1}$

**(a)** Seasonality (% of global mean)

**(b)** Latitudinal gradient

**(c)** Lat. gradient: EOF

**(d)** Lat. gradient: EOF score

**(e)** Monthly surface mole fraction field (ppt)

**(f)** Global and hemispheric surface mole fractions 1950–2014

**(g)** Global and hemispheric surface mole fractions 1700–2014

**(h)** Global and hemispheric surface mole fractions 2005–2010

Fig. 59

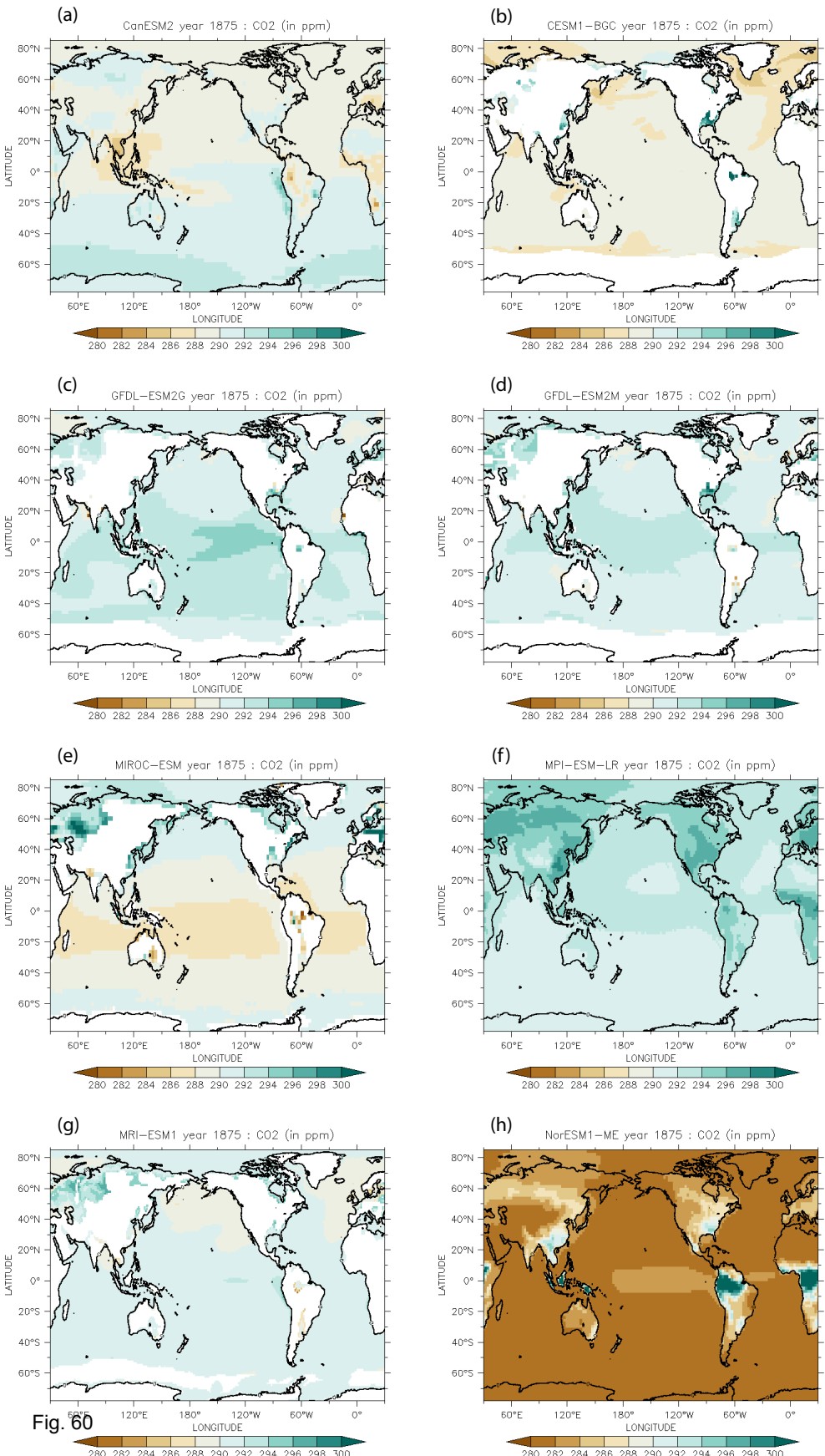

Fig. 60

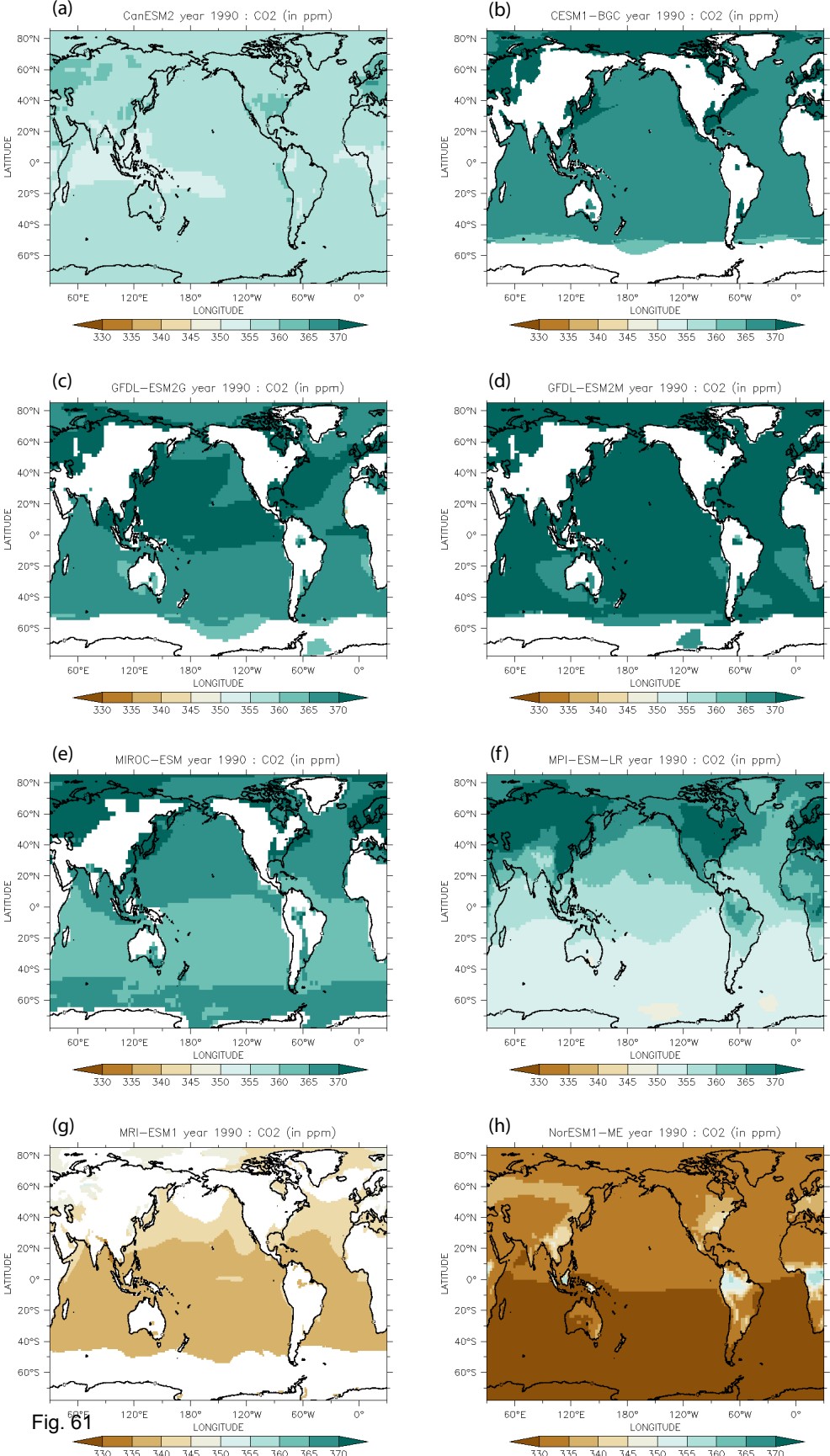

Fig. 61

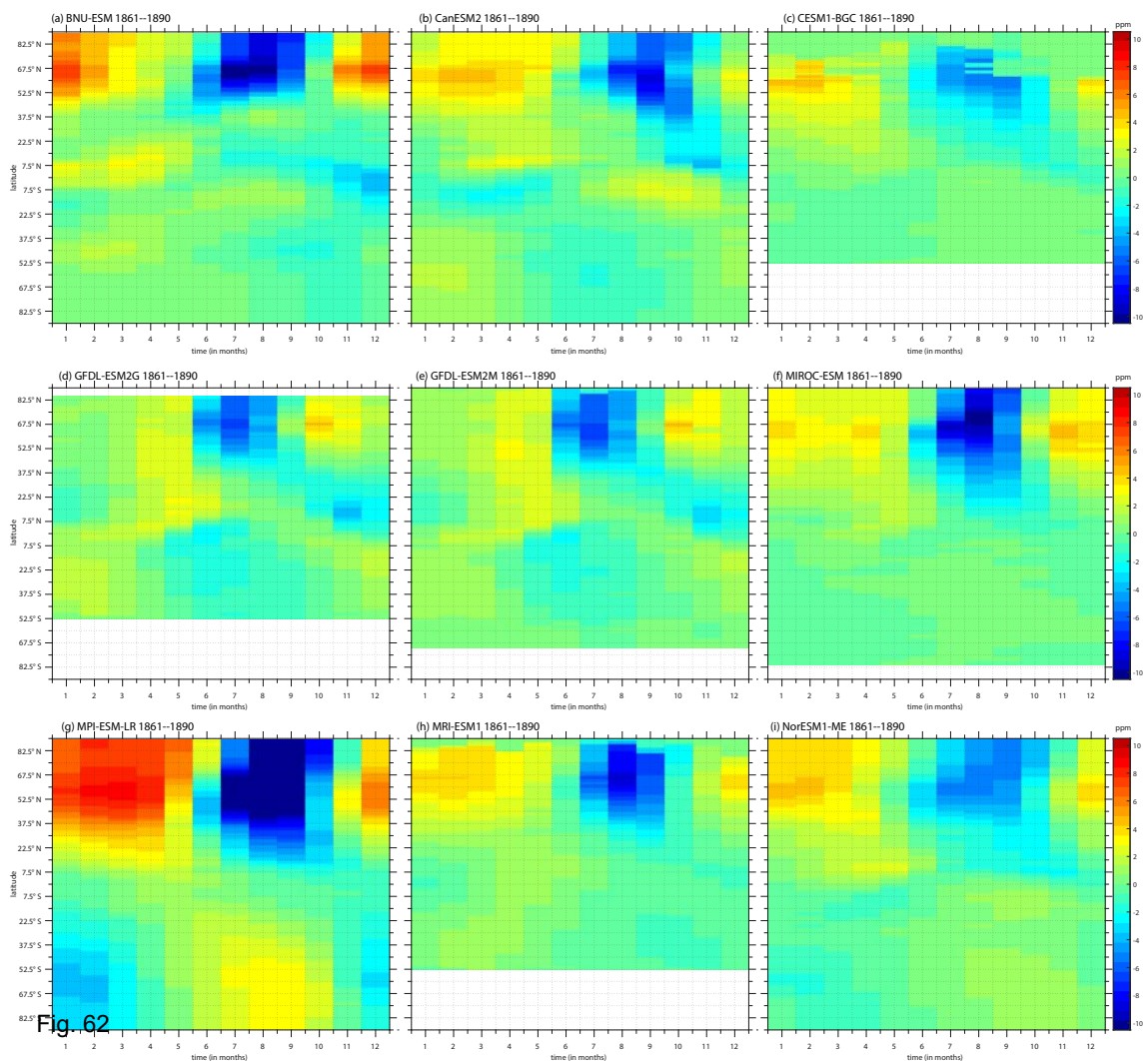

Fig. 62

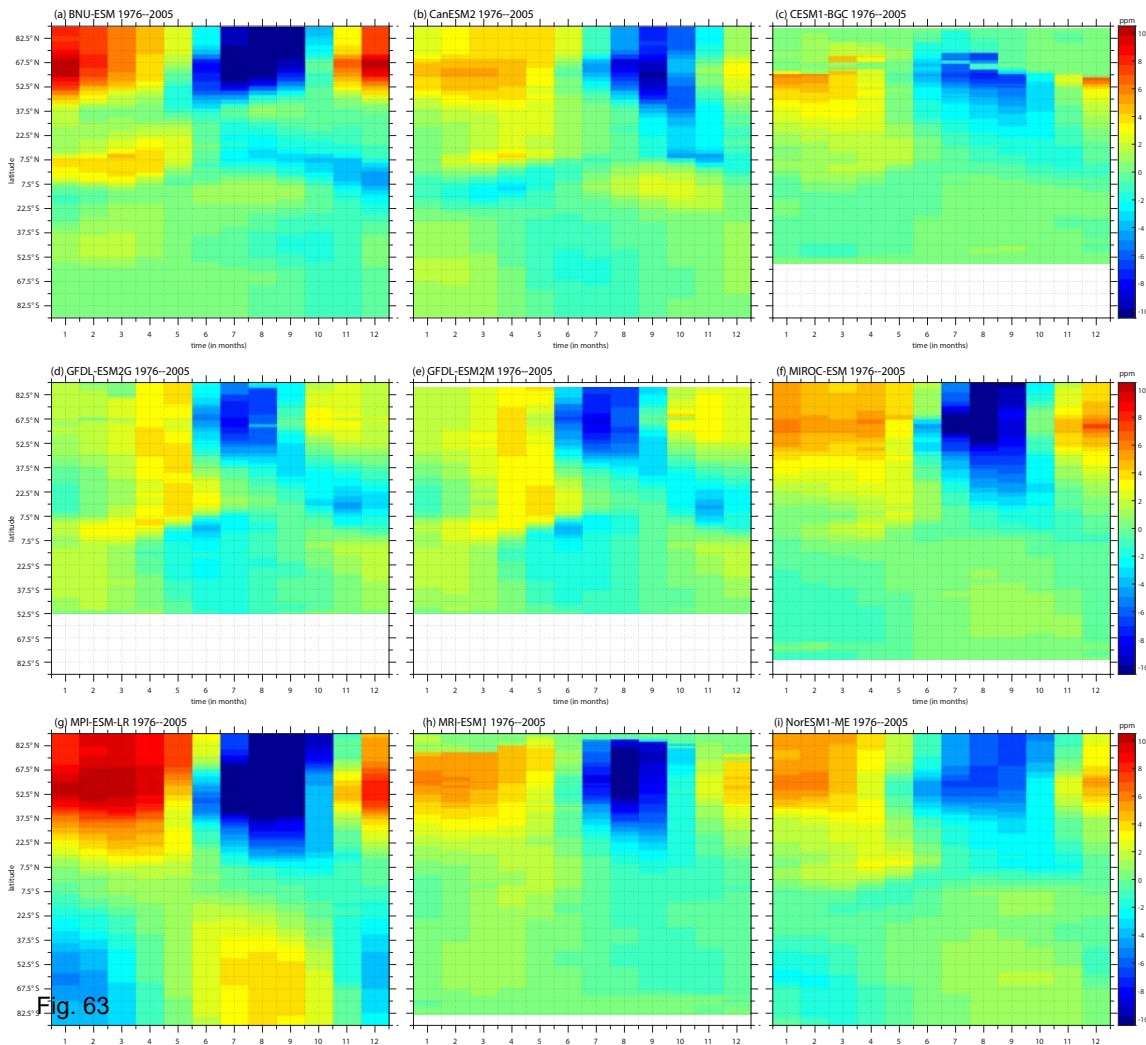

Fig. 63

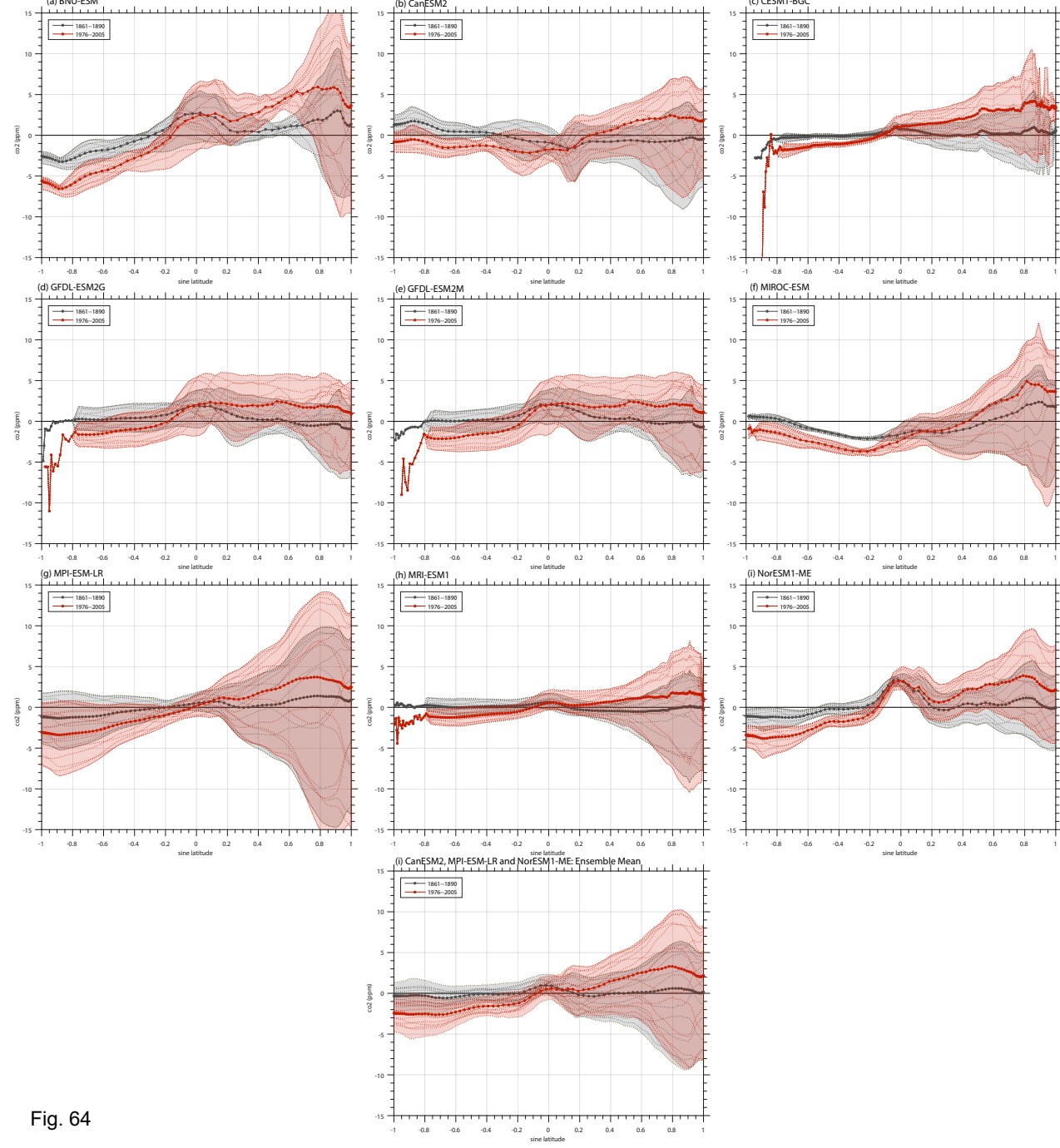

Fig. 64

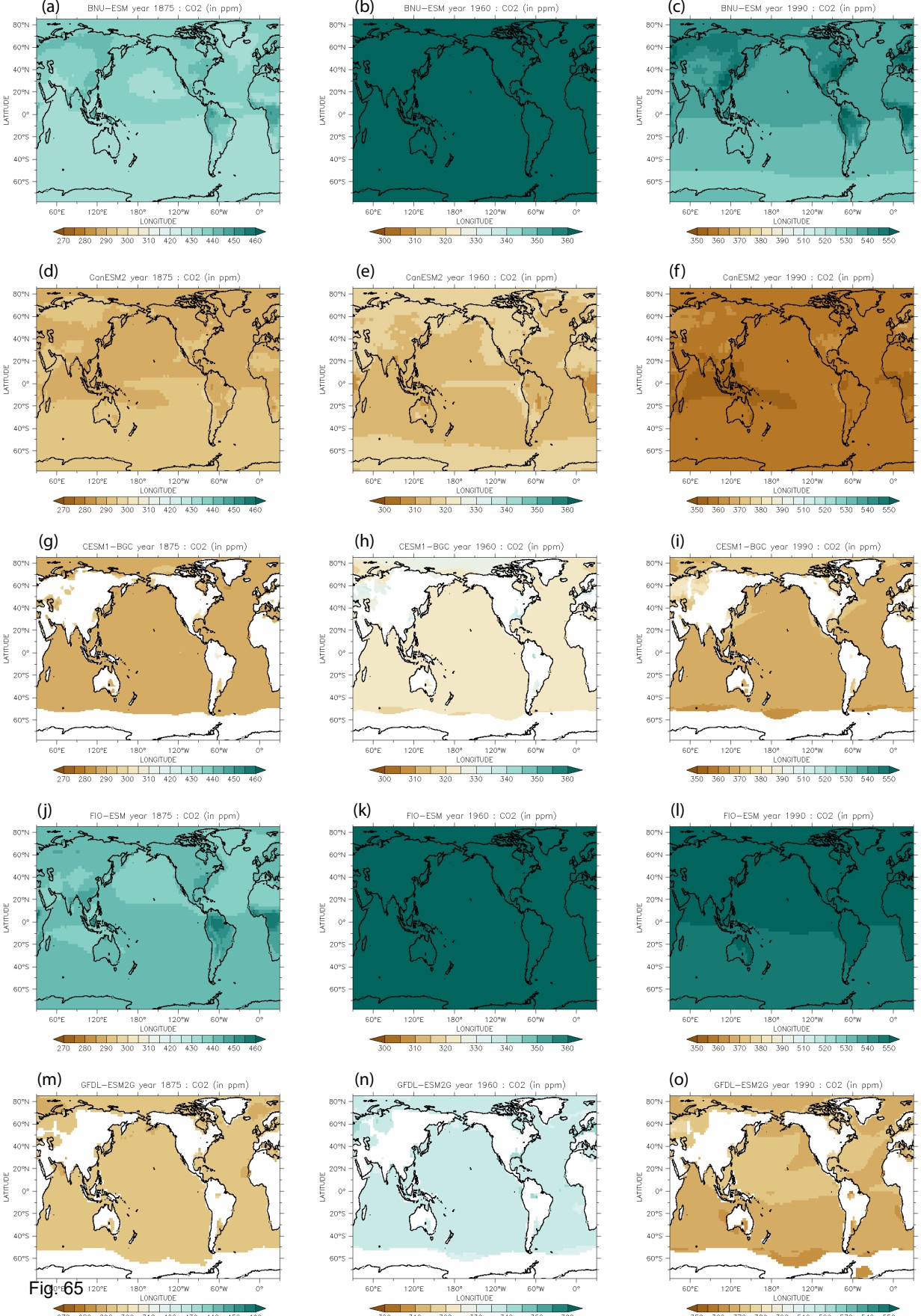

Fig. 65

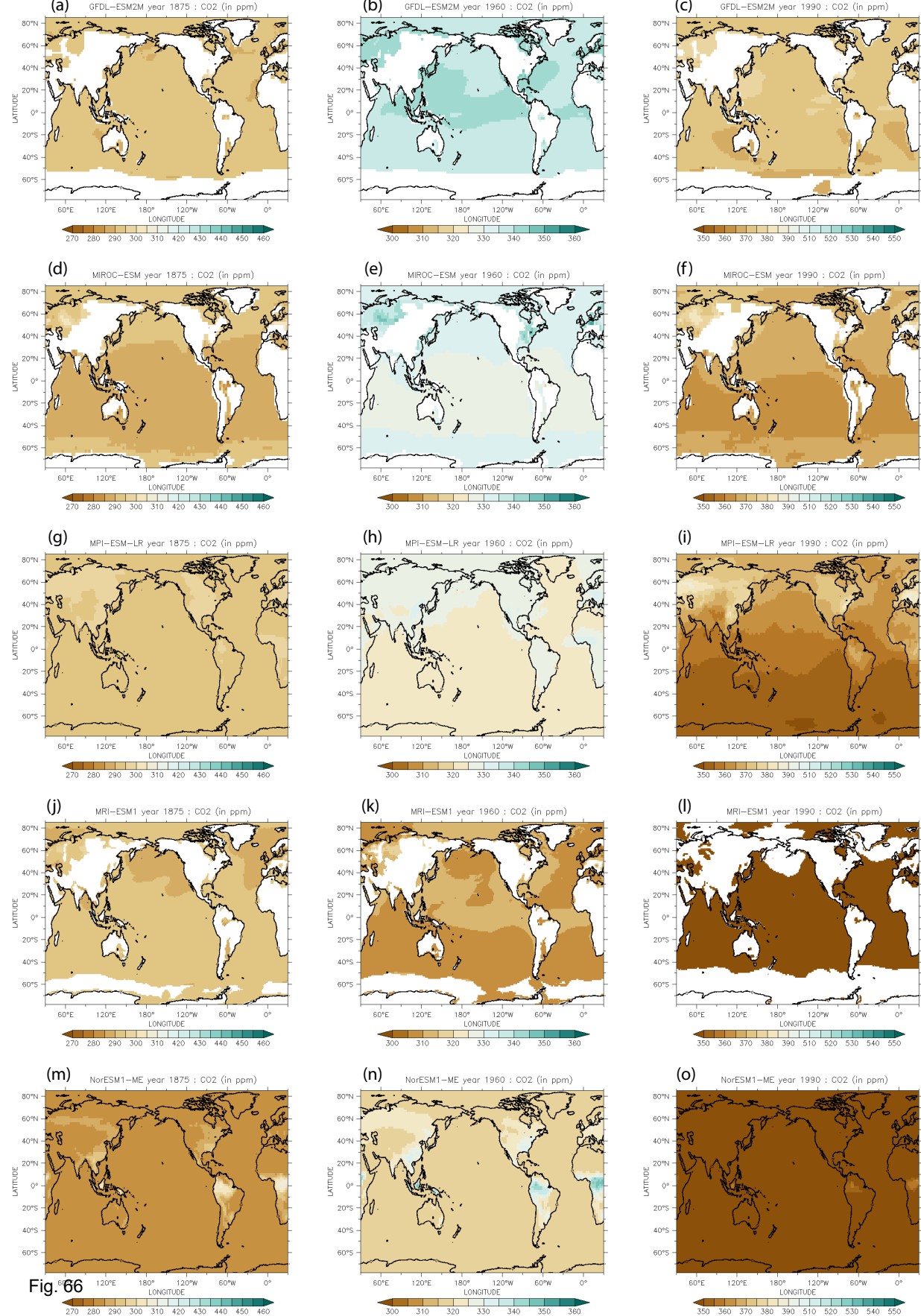

Fig. 66