# Peer review of "Historical greenhouse gas concentrations for climate modelling (CMIP6)"

_Geoscientific Model Development, 2016_

## Short Comment (SC1) · 8 Aug 2016

Dear authors,

In agreement with the CMIP6 panel members, the Executive editors of GMD would like to establish a common naming convention for the titles of the CMIP6 experiment description papers.

The title of CMIP6 papers should include both the acronym of the MIP, and CMIP6, so that it is clear this is a CMIP6-Endorsed MIP.

Good formats for the title include:

'XYZMIP contribution to CMIP6: Name of project'

or

'Name of Project (XYZMIP) contribution to CMIP6'

If you want to include a more descriptive title, the format could be along the lines of,

'XYZMIP contribution to CMIP6: Name of project - descriptive title'

or

'Name of Project (XYZMIP) contribution to CMIP6: descriptive title.'

In your case a Project name XYZMIP might not exist, but the title should contain the information that the Historical greenhouse gas concentrations have been provided for CMIP6. When you revise your manuscript, please correct the title of your manuscript accordingly.

Yours,

Astrid Kerkweg

---

## Short Comment (SC2) · 12 Aug 2016

In the initial editing step while posting the manuscript to the discussion forum, all units were changed from ppm, ppb, ppt to ppmv, ppbv, and pptv. This step was unfortunately incorrect. Reported "concentrations" in the manuscript are dry air mole fractions of real gases. The ppmv, ppbv and pptv units would be correct for ideal gases, but not for real gases. We will correct the units back to ppm, ppb, ppt and insert a more detailed explanation in the final manuscript. My apologies.

---

## Editor Comment (EC1) · OM Morgenstern (Editor) · 29 Aug 2016

Thanks for the clarification. I still suggest to retain the notation "ppmv", "ppbv", and "pptv". This is to avoid ambiguity with mass mixing ratios. You can add in the main text that the volume mixing ratios discussed in the paper are dry air mole fractions. In the modelling world, this is what they typically are.

---

## Referee Comment (RC1) · P. Forster (Referee) · 12 Sep 2016

The paper by Meinshausen and colleagues presents historical climate model scenarios of GHGs. There are numerous extensions over their CMIP5 efforts that together make considerable progress on a number of fronts. Particularly impressive is is knitting together of observations and models. This will also help the two communities understand the others needs. The paper represents a huge amount of work by the author team and it provides huge community good. I was very impressed.

The paper is long, has complex figures and contains a lot of technical detail. I would argue that this is appropriate and necessary for the GMD approach to allow clarity of methods and their reproducibility. The paper goes beyond simply documenting the method and showing the data. It details comparison to other data, and has a very interesting discussion and limitations sections that has insights into ESM uncertainty and possible effects on climate.

[Figure]

My suggestions of corrections are of only a technical nature, outlined below

Ln 26-27. I would argue that lots of things change climate not just GHGs and aerosol. I would maybe phrase as GHGs largely responsible for the warming and associated climate change ?

line 28 and 40. The future climate change comments seemed strange in abstract as to me the future is another but related problem - the paper really helps sort the past, but your call

Abstract. Time period is not mentioned - only 1850 date. Also not at all clear you are talking about historical scenarios - or changes through time. 0-2014 ins mentioned but analysts concentrates on 1850-2014?

Will historical runs end at the end of 2014? I thought it was 2015, but I may well be wrong!

Maybe I'm stupid but it did not seem clear where the data could be accessed?

The paper would benefit from a careful proof read. I am afraid that it is beyond my community spirit to do this! But examples are 1. Sections are not referred to consistently, sometimes by names, sometimes by numbers sometimes both cf ln 70,128,210 (e.g.) 2. There are typos in places e.g.1026-1027 (to prove i read to end!) 3. The odd statement is repeated 4. Equations are not presented as uniformly as they might be e.g. 230-236 e.g. nxm 5. Do you want asterisks or for multiply or something else?

line 210 Figures 20 and 21 might also be useful here -these don't seem to be referred to in text?

Figures don't seem to appear in the correct order - I'm not sure what your logic is here?

line 167 - that are these AGAGE? files - maybe giving a web address early on where files could be found would help? Or adding more explanation?

189-193 - it is not clear which scalings are being referred to for what gases?

I found Table 1 really useful in helping me understand your methods - could this be referred to earlier?

The figures are generally good considering their complexity - but details are hard to see even when zooming in online, such as the small "5" on fig 22 referred to in the text. I also found it hard to see the CMIP5 lines on the CMIP6/CMIP5 comparison figures.

On the science I had a few questions

1. It might be useful to quote 1750 PI concentrations and 2011 concentrations to compare to IPCC. A comparison might also be fun with IPCC historical forcings? 2. I guess your forcing estimates were all made with global radiative efficiency formula - are you going to run your fields through a radiation call to estimate actual forcings. Give me a shout if you would like someone to do this!

---

## Referee Comment (RC2) · Anonymous Referee #2 · 16 Sep 2016

The manuscript presents a data synthesis and assimilation study which aims at producing historical greenhouse gas concentration fields to be used in CMIP6 historical simulations. While I am overall confident that the results are of sufficient quality, I found the manuscript difficult to read and confusing in several aspects. It is not enough focussed on the most relevant time period (1850-2014) and species (in terms of radiative forcing). The task of precisely documenting and illustrating the data for all considered periods and 43 species may have been too ambitious, and results in incomplete documentation of the data even for the most important period and species: incomplete and hard to read figure captions, match of the data and scenarios hard to appreciate on figures in 1850-1950, missing references etc. (see below), however it should not be too difficult to improve the results presentation and discussion for the most important species and CMIP6 simulations relevant period. The scientific aspects of the study would be better highlighted if figures and tables that are little or not commented in the text were placed in a supplement more focussed on the technical

documentation of the data. I noted several potential circular arguments that should be clarified. In a worldwide IPCC context study, I was sad to read that the data used are nearly exclusively American network data for the atmospheric measurements (whereas for example WDCGG conveniently provides a large dataset in consistent format) and Australian data firn and ice core data (whereas considering all existing firn/ice datasets for the CMIP6 historical simulations period and most important species should not require a tremendous bibliographic effort). In a CMIP7 perspective, I think that more efforts could be made to relate the building of model inputs to an IPCC worldwide data synthesis, and include uncertainty estimates.

Detailed comments:

CMIP6 could be mentioned in the abstract.

The lists and number of species at lines 33-36 and 138-142 are not consistent.

The introduction or Section 4 could mention how other important greenhouse gases (e.g. $O_3$), greenhouse gas producers (e.g. CO, organics), aerosol source species (e.g. organics and sulfur compounds) and/or aerosols should be handled in historical simulations.

lines 52-56: this sentence is misleading, other contributions could be emphasized such as Buizert et al. (2012), WDCGG etc.

lines 76-78 and Tables 3 and 10: it would be useful to provide a radiative forcing ranking of the 43 species considered.

lines 83-85: only a few references are provided here, as well as in Sections 3.4 and 3.5 and the Supplement.

lines 101-112: the role of the ocean could be mentioned in the discussion of past latitudinal $CO_2$ gradients.

line 121: more recent references could be provided for $CH_4$.

lines 176-180: references are provided for a subset of AGAGE data (not $CH_2Cl_2$ discussed at lines 167-168) but not NOAA data. More generally, it is not clear to me if the datasets for all species are published and/or publicly available in AGAGE or NOAA databases.

lines 181-196 (calibration scales): for the major halocarbons in terms of radiative forcing, calibration scale intercomparison studies (e.g. Hall et al., 2014; Rhoderick et al., 2015) could be used at least to evaluate uncertainties. Scale names for the seven species mentioned at lines 186-188 suggest that measurements were not made by AGAGE or NOAA. Is it the case? If yes, could the data source / reference be provided?

lines 252-262: I'm not at ease with the principle of scaling $CO_2$ variations with temperature variations while producing inputs for models aiming at evaluating the impact of $CO_2$ on temperature. On Figure 2 a.3 the seasonality change is provided only after about 1950. I think that the earlier $CO_2$ seasonality change should be illustrated and discussed.

lines 278-281, 292-294 and Figures 1, 2, 4, 5 and 9. The ad hoc smoothing of Law Dome data and very high sampling resolution of non Law Dome recent ice core data (e.g. Bauska et al., 2015, Mitchell et al., 2013, Rhodes et al., 2013) makes the choice of using Law Dome only data less obvious than some years ago. The choices of time scales in the Figures make it very hard to appreciate the match of the scenarios with available firn and ice data especially for the beginning of the CMIP6 historical runs (1850-1950). One specific issue is that in view of the $N_2O$ data dispersion, I'm not convinced that the dip in the $N_2O$ scenario around 1850 is really reliable.

lines 305-308: in this very technical description, I did not understand the main message. Why does the Gosh et al. (2015) data need to be updated? Why excluding North GRIP? Does that induce significant changes?

lines 312-315: the methodology is unclear to me here. Mixing ratio data are always local. Global or hemispheric means should already be the result of an assimilation procedure. Is there some circularity in constraining an assimilation procedure with assimilated data?

lines 314-321: the list of key studies should be focused on key species in terms of radiative forcing. It would be useful to provide the references of all data used in the supplementary tables.

lines 323-325: what are the data used to constrain the major halocarbon trends (e.g. CFC-11, CFC-12, HCFC-22) before about 1978?

lines 373-377: I don't understand the motivation for grouping the species as ozone depleting versus non ozone depleting. Splitting the species between those destroyed in the troposphere or not seems more obvious to me as they have very different vertical structures. Could the ozone depleting choice be commented?

lines 380-390: are the inputs used for CMIP5 simulations ($CO_2$ fluxes?) and the CMIP6 input scenarios discussed here fully independent?

lines 399-409 and Figure 1: the consistency of the different datasets for the CMIP6 simulation period (after 1850) should be made more visible on the figure and should be commented in the manuscript.

lines 448-449: Section 3.1 starts with discussing discrepancies of several ppb between ice core datasets and large uncertainties on meridional gradients. Providing estimates of the uncertainties on the global mean $CO_2$ at the dates mentioned would be useful.

lines 497-498 and Figure 1f: in view of the large discrepancies between ice core records and large dispersion of the $N_2O$ data in the 1850-1970 period, more firn and ice datasets could be used to evaluate the trend used in CMIP6 historical runs (e.g. Machida et al., 1995; Battle et al., 1996; Ishijima et al., 2007)

Sections 3.4 and 3.5 lack focus on the most important species in terms of radiative

forcing and bibliographic references.

lines 536-545: bibliographic references should be provided for the nineteenth century mixing ratio estimates.

lines 717-721 and Figure 9: the reason why the early part of the CMIP6 $CO_2$ trend is smoother than the CMIP5 trend whereas the early part of the CMIP6 $N_2O$ trend is less smooth than the CMIP5 trend is unclear to me. Could this choice be commented?

Section 5.2: Figures 10, 11, 13 and 14 are not directly comparable to Figure 2 and could be placed in a Supplement, whereas Figure 12 could include a representation of the CMIP6 scenarios in similar format as the CMIP5 mixing ratio outputs.

Section 5.5: the comparison with other literature studies lacks priorities in terms of radiatively most important species and a check of the independence of the data used for evaluation with respect to those used to generate the assimilated fields. For example, the $CO_2$ and $CH_4$ high Northern latitude trends in Buizert et al., 2012 were provided by V. Petrenko (see file SCENARIO_NEEM2008_....xls) and are mostly based on NOAA ESRL and Law Dome data (see Section 2.4.2 in file Supplement Buizert ....pdf). On the other hand, the comparison with early $CO_2$ atmospheric data (Keeling et al., 1976) is not commented. I'm surprized that the Cape Grim air archive data are not commented for $N_2O$ (Park et al., 2012) and other species (e.g. Newland et al., 2013). It could be mentioned that the Martinerie et al. (2009) trends for halocarbons are based on industrial emission histories and are used in Buizert et al. (2012) for the time period preceding atmospheric measurements.

Lines 860-870: I could not see the Buizert et al. (2012) trends on the Figures. Are the commented differences within uncertainties provided in Buizert et al. (2012)?

lines 871-877: would the pioneer study by Butler et al. (1999) be more consistent with other trends before 1950 if the South Pole firn air age spread was taken into account?

Section 6: major additional uncertainties for the early part of CMIP6 historical simula-

[Figure]

tions such as the lack of constraints on nineteenth century $CO_2$ meridional gradients could be mentioned.

Tables 2, 4, 5, 6, 7, 8, 9, 11 (technical documentation) could be placed in a Supplement

Technical corrections:

lines 60, 708, 709 etc. and references: Meinshausen et al., 2011, 2011a and 2011b seem to be the same article

lines 158-159: the first figure quoted in the manuscript should be Figure 1 rather than Figure 22

lines 189-193 and 281-282: It would be clearer to describe the scale change of the firn and ice data together with the scale description of the atmospheric data.

line 230: define EOF notation at first use

line 595: is it really needed to quote Eyring et al., GMDD, 2015 instead of Eyring et al., GMD, 2016?

lines 799-800: check the writing. This sentence seems contradictory with lines 167-171, 801-802 and 824-829.

lines 904-905: Trudinger et al. (2004) is not more recent than WMO (2014) and Velders et al. (2014)

lines 1408-1410: incorrect list of authors

Figure 1: horizontal scale issue for panel c. A complete reference should be provided for each dataset. I saw only $CO_2$ data (no $CH_4$ and $N_2O$ data) in Rubino et al. (2013), and Table 1 mentions different references for Law Dome $CH_4$ and $N_2O$ data.

Figure 6: panels g, i, k, m, o, q would be much easier to read if the horizontal scale started in 1850 or 1900

Figure 12: I can't see the 12 five lines mentioned in the caption, and the shaded areas

are not described in the caption

Figure 15: wrong reference for the $CH_4$ "NEEM" scenario (see Supplement of Buizert et al. 2012, Section 2.4.2 in file Supplement Buizert ....pdf, and file SCE-NARIO_NEEM2008_....xls, $CO_2$ and $CH_4$ scenarios were made by Vas Petrenko). The NOAA global mean and WDCGG global mean results should be made easier to distinguish.

References not provided in the manuscript:

Battle et al., Nature, 383(6597), 231-235, 1996

Hall et al., Atmos. Meas. Tech., 7, 469-490, 2014

Ishijima et al., J. Geophys. Res., 112, D03305, 2007, doi:10.1029/2006JD007208

Machida et al., Geophys. Res. Lett., 22(21), 2921-2924, 1995

Newland et al., Atmos. Chem. Phys., 13, 5551–5565, 2013.

Park et al., Nature Geoscience, 2012, DOI: 10.1038/NGEO1421.

Rhoderick et al., Elementa Sci Anth 3: 000075, 2015, doi:10.12952/journal.elementa.000075

---

## Short Comment (SC3) · 29 Sep 2016

This comment raises the fairly narrow question as to whether the level of detail provided by this reconstruction of greenhouse gas concentrations is appropriate for CMIP.

Without diminishing the tremendous amount of work represented by this reconstruction nor its possible value in other contexts, discussions with modeling centers over the protocol for the Radiative Forcing MIP (https://dx.doi.org/10.5194/gmd-9-3447-2016) suggests that much of the detail in these specifications will not implemented as part of the CMIP6 protocol. In particular:

1) We fear that few modeling centers will implement the latitudinal variations or vertical profiles of well-mixed greenhouse gases as described in section 4. We discussed this only with GFDL and the Met Office, so we may well be wrong, but these are two top-notch centers with strong local interest and expertise in radiation issues, and both will

use time-dependent scalar values.

2) The three options for describing atmospheric composition (lines 369-377) offer a useful range of compromises but option 1, using a subset of gases, might be improved by greatly restricting the number of halocarbon species provided. It is not clear how many, if any, line-by-line models include the long list of species. It is certain that no climate model radiation codes include more than a few. Based on rough calculations using radiative efficiencies from IPCC AR4, including only CFC-11, CFC-12, HCFC-22, CFC-113, HFC-134a, and CCl4 reproduces the total instantaneous radiative forcing in 2014 to within 0.045 W/m2.

If the protocol includes levels of details that modeling centers are unlikely to observe it may be more appropriate to reduce to level of detail to something practical.

On a distinct issue, we are surprised that the protocol does not include estimates of CO. Rough estimates suggest a clear-sky instantaneous radiative forcing at 2014 of roughly 0.05 W/m2, or the same contribution from one gas as from 35 of the specified halocarbons in total.

With best regards - Robert Pincus, University of Colorado (lead coordinator, RFMIP) Eli J. Mlawer, Atmospheric Environment Research

---

## Author Comment (AC1) · 29 Nov 2016

Given the naming convention for this Special Issue, we propose to change the title to: "Historical greenhouse gas concentrations for climate modelling (CMIP6)"

---

## Author Comment (AC2) · 29 Nov 2016

Reply to comment on "Historical greenhouse gas concentrations" by Anonymous Referee #2

Comment 1: The manuscrfipt presents a data synthesis and assimilation study which aims at producing historical greenhouse gas concentration fields to be used in CMIP6 historical simulations. While I am overall confident that the results are of sufficient quality, I found the manuscript difficult to read and confusing in several aspects. It is not enough focused on the most relevant time period (1850-2014) and species (in terms of radiative forcing). The task of precisely documenting and illustrating the data for all considered periods and 43 species may have been too ambitious, and results in incomplete documentation of the data even for the most important period and species: incomplete and hard to read figure captions, match of the data and scenarios hard to appreciate on figures in 1850-1950, missing references etc. (see below), however it

should not be too difficult to improve the results presentation and discussion for the most important species and CMIP6 simulations relevant period.

Reply 1: First of all, we would like to thank the reviewer for the extensive and without doubt time-consuming review comments that helped to, we believe, enhance the quality of the manuscript substantially.

We share the reviewer's observation that the overall project has turned out to be rather ambitious in terms of covering lots of gases, especially because the time series and concentration fields had been built up from individual station data. Unlike for CMIP5, when existing time series were simply "glued together", the ambition to provide monthly and latitudinal optional data coverage for CMIP6 and extrapolation meant that a simple merging of existing data was not possible. This comprehensive effort however allowed to produce more consistent time series.

For example, the CMIP5 global-mean time series was taken from the Law Dome record data, which did not represent a best guess for the global-mean concentration (which was in particular an issue for methane). The absence of global-mean concentration time series in the literature back to 1850 meant anyway that latitudinally gradients had to be taken account of when aggregating station data – which meant that the step to providing the latitudinal consistent fields was comparatively small, given they had to be produced anyway in the background to produce a best estimate of global-mean surface concentrations back in time.

We do however acknowledge the that focus on 1850-2014 for the three big gases could be strengthened. We hence adapted our old overview figure and added three panels specifically for that 1850-2014 period for $CO_2$, $CH_4$ and $N_2O$ – in conjunction with some comparison and input data for our study. See the new Figure 6.

The new Figure 6 is: [SEE NEW FIGURE 6 ATTACHED AT BOTTOM OF THIS COMMENT]

Caption Fig. 6 - Atmospheric CO2, CH4 and N2O mixing ratios over different time-scales, from 800 thousand years ago until today (panel a), over the last 2000 years (panel b) and over 1850 to 2014 (panel c, d, e). The shown data is for CO2: Mauna Loa data by Keeling et al. (Keeling et al., 1976); the Law Dome ice record (Etheridge et al., 1998b; MacFarling Meure et al., 2006; Rubino et al., 2013); NOAA ESRL station data (NOAA, 2013; NOAA ESRL GMD, 2014a, b, c); the EPICA composite data (Ahn and Brook, 2014; Bereiter et al., 2015; Bereiter et al., 2012; Lüthi et al., 2008; MacFarling Meure et al., 2006; Marcott et al., 2014; Monnin et al., 2004; Petit et al., 1999; Schneider et al., 2013; Siegenthaler et al., 2005) and the WAIS data (Bauska et al., 2015). For CH4, the shown data is the Law Dome data (Etheridge et al., 1998a; MacFarling Meure et al., 2006), the instrumental data from the NOAA and AGAGE networks (see Table 3), NEEM ice core measurements (Rhodes et al., 2013) the EPICA composite (Barbante et al., 2006a; Barbante et al., 2006b) the long record by Loulergue et al. (2008) as well as the GISP2D, WDC05A and WDC06A records by Mitchell et al. (2013). In case of N2O, the shown data is the Law Dome record (MacFarling Meure et al., 2006), the Talos Dome record (Schilt et al., 2010b), the GISPII record (Sowers et al., 2003) and the EPICA record (Fluckiger et al., 2002; Schilt et al., 2010a; Spahni et al., 2005) in addition to the H15 ice core record from Antarctica (Machida et al., 1995), the South Pole firn record (Battle et al., 1996), the Law Dome firn record "Park" (Park et al., 2012) and a modelling synthesis by Ishijima (2007). For data sources behind "this study's" composite product, see Table 2, Table 3 and Table 4.

Furthermore, we produced another new figure (Figure 7), which specifically looks at the 1950 to 1990 period of CO2 and provides an additional discussion of the CO2 time series over that time frame in comparison with Scripps station data.

We furthermore highlighted in the abstract the CMIP6 purpose of the data and point towards this specific section high up in the introduction – to ease the modeler's digestion of the manuscript. Specifically, we added the pointer to section 4 in the Introduction, stating: "The description of the datasets geared towards CMIP6 modelling groups is

provided in section 4, including a description of available data formats and CMIP6 minimum recommendations. "

[SEE NEW FIGURE 7 ATTACHED AT THE BOTTOM OF THIS COMMENT]

Caption Fig. 7 – Comparison of 1950 to 1990 $CO_2$ concentrations with early Scripps station data (Keeling et al., 2001) for each 15°-degree latitudinal band. Also, the Law Dome ice record data is shown (panel k) with our 3rd degree polynomial smoothing. This study's monthly $CO_2$ zonal means were derived from station data from 1984 onwards. Before that, this study used Mauna Loa MLO annual average and smoothed Law Dome data (see Table 1 and section 2 "Methods"). The shown comparison with monthly Scripps station data before 1984 is a qualitative validation of the applied methodology to regress latitudinal gradient and seasonality changes to times before 1984. See text.

Comment 2: The scientific aspects of the study would be better highlighted if figures and tables that are little or not commented in the text were placed in a supplement more focussed on the technical documentation of the data. I noted several potential circular arguments that should be clarified.

Reply 2: We would welcome more detail on the "potential circular arguments" so that we can address them. We assume that they related to below comment "14" in regard to the scaling of $CO_2$ seasonality changes with a composite indicator that includes global-mean temperatures. On the supplement: We organized the manuscript such that all the so-called factsheets for all gases (except for $CO_2$, $CH_4$ and $N_2O$) are presented in Appendix A and two additional CMIP5 ESM analysis figures in Appendix B. Following Reviewer #1 comments, we placed all other figures and tables in chronological order with the text.

Comment 3: In a worldwide IPCC context study, I was sad to read that the data used are nearly exclusively American network data for the atmospheric measurements (whereas for example WDCGG conveniently provides a large dataset in consistent format) and Australian data firn and ice core data (whereas considering all existing firn/ice datasets for the CMIP6 historical simulations period and most important species should not require a tremendous bibliographic effort). In a CMIP7 perspective, I think that more efforts could be made to relate the building of model inputs to an IPCC worldwide data synthesis, and include uncertainty estimates.

Reply 3: We fully agree with the author that the WDCGG is a tremendously valuable initiative and indeed that it should be considered as the starting point for a CMIP7 product. While we do not think that the results would be much different (due to largely overlapping datasets) between our predominant use of AGAGE and NOAA network data and the WDCGG, we do acknowledge this to be a limitation of our study. We now added to the limitations section:

"For the recent instrumental period, our study is predominantly based the NOAA and the international AGAGE network data. Consistent quality control and consistent scales are advantages of that approach. Ideally however, our study should have started out from a yet more inclusive representation, e.g. including the multiple additional station datasets gathered and archived by the World Data Centre for Greenhouse Gases (WDCGG) that are neither part of AGAGE or NOAA networks. The WDCGG station raw data is available at: http://ds.data.jma.go.jp/gmd/wdcgg/cgi-bin/wdcgg/catalogue.cgi . While the methodology of our study could be maintained or built upon, we hence recommend for any future updates, that those additional datasets are considered – with the appropriate quality control and scale conversion efforts."

However, we would like to point out that we provide and acknowledge a large set of comparison data products and studies. For example, the new Figure 1 also features the West Antarctic Ice record (Bauska et al.) for $CO_2$ and the factsheets for individual gases include several comparison products (mostly displayed in panels f, g and h of Figures 10, 12, 13 and Figures 24 to 63 in Appendix A (Figure numbers refer to the revised manuscript).

Furthermore, AGAGE is an international cooperative network that relies on substantial contributions (measurements and more) from international partners (Australians, Brits, Swiss, Japanese, Chinese, Koreans, among others. NOAA also relies heavily on international partners to conduct their global network. The characterization as 'American network data' is hence not quite correct, we find. Regarding the missing uncertainty estimates. The reviewer is also correct to point out this limitation. We already had mentioned that in our limitations section and fully acknowledge that. Given that the primary purpose and starting point for our study has been the provision of a dataset for the CMIP6 historical model runs, uncertainty estimates were not fundamental. In fact, given that a multi-model intercomparison should be run with a single standard input to be able to compare differences in the model responses, an uncertainty estimate would not have been of particular use for CMIP6. Nevertheless, given that the dataset is likely being used outside CMIP6, we again acknowledge this limitation. Given the multiple distinct sources of uncertainty, some of them correlated in space and time, a proper uncertainty analysis would have multiplied the effort that went into this study. We hence consider this as beyond the scope of our study and encourage future synthesis efforts to include a statistically correct uncertainty analyses.

Detailed comments:

Comment 4: CMIP6 could be mentioned in the abstract.

Reply 4: Done.

Comment 5: The lists and number of species at lines 33-36 and 138-142 are not consistent.

Reply 5: Thanks for spotting that and our apologies. We made both sections consistent now. The abstract's listing is: "We provide consolidated datasets in various spatiotemporal resolutions for carbon dioxide (CO2), methane (CH4) and nitrous oxide (N2O), as well as 40 other GHGs, namely 17 ozone depleting substances, 11 hydrofluorocarbons (HFCs), 9 perfluorocarbons (PFCs), sulfur hexafluoride (SF6), nitrogen trifluoride

(NF3) and sulfuryl fluoride (SO2F2)."

And the more detailed listing in the methods section is: "We consider a total of 43 GHGs: CO2, CH4, N2O, a group of 17 ozone depleting substances made up of five CFCs (CFC-12, CFC-11, CFC-113, CFC-114, CFC-115), three HCFCs (HCFC-22, HCFC-141b, HCFC-142b), three halons (Halon-1211, Halon-1301, Halon-2402), methyl chloroform (CH3CCl3), carbon tetrachloride (CCl4), methyl chloride (CH3Cl), methylene chloride (CH2Cl2), chloroform (CHCl3), and methyl bromide (CH3Br), and 23 other fluorinated compounds made up of 11 HFCs (HFC-134a, HFC-23, HFC-32, HFC-125, HFC-143a, HFC-152a, HFC-227ea, HFC-236fa, HFC-245fa, HFC-365mfc, HFC-43-10mee), nine PFCs (CF4, C2F6, C3F8, C4F10, C5F12, C6F14, C7F16, C8F18, and c-C4F8), NF3, SF6, and SO2F2."

Comment 6: The introduction or Section 4 could mention how other important greenhouse gases (e.g. O3), greenhouse gas producers (e.g. CO, organics), aerosol source species (e.g. organics and sulfur compounds) and/or aerosols should be handled in historical simulations.

Reply 6: Thanks. Rather than providing a description ourselves, we however opted to point the modelers to the experiment specific protocols. The addition to section 4 reads: "Depending on the specific CMIP6 experiment, different protocols and recommendations can apply. Modelers should hence also check the experiment specific descriptions (see special issue available at http://www.geosci-model-dev.net/special_issue590.html), including protocols regarding the important other forcing input datasets like aerosols, their emissions and optical properties, landuse patterns, but also short-lived GHGs like tropospheric and stratospheric ozone for models without interactive ozone chemistry.."

Comment 7: lines 52-56: this sentence is misleading, other contributions could be emphasized such as Buizert et al. (2012), WDCGG etc.

Reply 7: We revised the respective section, so that it reads now: "To date, reconstructions of millennial global-mean time series based on ice and firn data have been performed, e.g. for CO2 over the last millennia (Ahn et al., 2012; MacFarling Meure et al., 2006; Rubino et al., 2013). For the more recent past, several studies investigated firn and ice data to constrain halocarbons (Buizert et al., 2012; Martinerie et al., 2009; Mühle et al., 2010a; Sturrock et al., 2002; Trudinger et al., 2016), some of them with hemispheric resolution. In terms of latitudinally-resolved monthly data, there have only been a few synthesis products, namely for CO2, CH4 and N2O over the instrumental record over the past 20 to 40 years (NOAA, 2013; NOAA ESRL GMD, 2014a, b, c). For this recent past, the World Data Centre for Greenhouse Gases (WDCGG) (ds.data.jma.go.jp/gmd/wdcgg/) also provides a synthesis with global and hemispheric means for CO2, CH4 and N2O (Tsutsumi et al., 2009)."

Comment 8: lines 76-78 and Tables 3 and 10: it would be useful to provide a radiative forcing ranking of the 43 species considered.

Reply 8: Table 3 provides an atmospheric abundance ranking (for all species) and Table 10 provides a radiative forcing ranking (for the first 15 species). We hence feel that inserting a radiative forcing ranking also in Table 3 would be a redundancy. We hope the reviewer is ok with this clarification of why we limited one ranking to Table 3 and one to Table 10.

Comment 9: lines 83-85: only a few references are provided here, as well as in Sections 3.4 and 3.5 and the Supplement.

Reply 9: Thank you. We acknowledge that our list of references was not complete. We now complemented the list of provided references in the introduction and also added a Table 12, which details all the literature studies we show and use. The new text now reads:

"Furthermore, many detailed literature studies (Arnold et al., 2013; Arnold et al., 2014; Aydin et al., 2010; Butler et al., 1999; Ivy et al., 2012; Martinerie et al., 2009; Montzka et al., 2014; Mühle et al., 2010b; Oram et al., 2012; Sturrock et al., 2002; Trudinger et

al., 2004; Trudinger et al., 2016; Velders et al., 2014; Vollmer et al., 2016; Worton et al., 2006) for radiatively less important species are compared with our data product in the factsheet figures for the specific gases (Table 12 and Appendix A with Fig. 20 to Fig. 59) or synthesised where direct observational records from the above networks were not available. Furthermore, while we added a couple of references to the Results section 3.4 and 3.5, we referenced the new Table 12 again in section 5.6 "Comparison with other literature studies" as we believe this is the most appropriate place.

[SEE NEW TABLE 12 ATTACHED AT BOTTOM OF THIS REPLY]

Comment 10: lines 101-112: the role of the ocean could be mentioned in the discussion of past latitudinal CO2 gradients.

Reply 10: Given the extent of the manuscript, we would prefer not to enter into a discussion on the ocean in pre-industrial gradients. We did however clarify that our reference to the carbon cycle relates to both the ocean and land domain. The new sentences now reads: "One complication to retrieve the latitudinal pre-industrial CO2 concentration profile is that CO2 fertilization and temperature effects on the carbon cycle, both over ocean and land, change both the magnitude and spatial patterns of natural CO2 fluxes..."

Comment 11: line 121: more recent references could be provided for CH4.

Reply 11: Thanks. We added the recent review by Kirschke et al. (2013) in Nature Geoscience.

Comment 12: lines 176-180: references are provided for a subset of AGAGE data (not CH2Cl2 discussed at lines 167-168) but not NOAA data. More generally, it is not clear to me if the datasets for all species are published and/or publicly available in AGAGE or NOAA databases.

Reply 12: In regard to CH2Cl2 data reference from the AGAGE network: We reproduced the key references as stated here:

http://cdiac.ornl.gov/ftp/ale_gage_Agage/AGAGE/. Our apologies that we did not provide a balanced referencing of the NOAA data (that would be Spivakovsky, 2000). We now deleted the respective AGAGE section and moved all references to the new Table 12, reproduced above. This Table now provides a comprehensive overview of the used data sources, both from the NOAA and AGAGE networks. We also provide all the ftp links in Table 12. Yes, the data is publicly available.

Comment 13: lines 181-196 (calibration scales): for the major halocarbons in terms of radiative forcing, calibration scale intercomparison studies (e.g. Hall et al., 2014; Rhoderick et al., 2015) could be used at least to evaluate uncertainties. Scale names for the seven species mentioned at lines 186-188 suggest that measurements were not made by AGAGE or NOAA. Is it the case? If yes, could the data source / reference be provided?

Reply 13: We thanks for the suggestion and inserted two new sentences reading: "Gas measurements on different measurement scales, and even when using the same scales by different laboratories, are subject to uncertainties (Hall et al., 2014). For halocarbons, the difference in calibration scales has been estimated as small, but not negligible, i.e. within 2.5%, often within 1% (Rhoderick et al., 2015)." The earlier scales were used in some of AGAGE network data (which includes a global range of partners), before the scales were converted to the most recent standard).

Comment 14: lines 252-262: I'm not at ease with the principle of scaling CO2 variations with temperature variations while producing inputs for models aiming at evaluating the impact of CO2 on temperature. On Figure 2 a.3 the seasonality change is provided only after about 1950. I think that the earlier CO2 seasonality change should be illustrated and discussed.

Reply 14: We now added a new figure to discuss both various alternatives as well as pre-1950 times in regard to their assumed CO2 seasonality changes. As we've indicated in our text, we had tested several regression options, including using only

CO2 concentrations at regressor of the seasonality changes. The added text now reads:

"Specifically, we tested global-mean CO2 concentrations, global-mean annual average surface air temperatures and lagged averages of surface air temperatures as regressors (see Fig. 5). The R-squared values of the regressions over the 1984-2014 period are relatively similar across all regressors, around 0.8. The marked difference is that the regression with only CO2 concentrations would result in a stronger reduction of seasonality around 1940-1960 and before 1900. By 1850, the reduction of summertime CO2 concentrations in the zonal band around 52.5°N would be around 8.6 ppm compared to 2014 (multiply the differences of the seasonality scaling difference between 1850 and 2014, about 21, with the 0.41 ppm maximum of the EOF pattern, shown in Fig. 9 a.2). In contrast, the other regression options would limit the maximal seasonality change to about 5.7ppm, closer to the maximal seasonality change detected within the period 1984-2014, of 4.5ppm (cf. Fig. 5e). Given the uncertainty in regard to pre-1960 seasonality, we opted for the more conservative extrapolation method that implies a less significant change outside the observational period and chose the regressor with the least variability, namely our composite regressor combining temperature and CO2 concentrations.

Despite the differences in the regressors, it should be noted that early CO2 observations are too sparse to come to a definite conclusion in regard to which regressor is best suited – given the induced differences around 1960s and 1970s are fairly small compared to the noise in the observations (see panel f and g of Fig. 5). Furthermore, . . ."

Regarding the first point, i.e. the potentially circular argument of scaling CO2 concentrations with temperatures. We appreciate the concern. However, we use global-mean temperatures in our composite indicator with which we change the seasonality changes. Given that any scaling of our zero-mean seasonality patterns does not change global-mean CO2 concentrations, the actual influence in CMIP6 model runs

of higher or lower seasonality patterns should be a rather small second-order effect related to the differential global-mean and annual average warming resulting from the high-northern latitude winter and summer-time $CO_2$ concentrations that are not offset by correspondingly lower to $CO_2$ concentration changes elsewhere. The error that results from a uniform application of global-mean annual average $CO_2$ concentrations in the concentration-driven runs will likely be orders of magnitude higher. We are therefore not clear, why the scaling of the seasonality changes with an observed record of global-mean annual average temperatures would materially affect the CMIP6 modelling results.

The new figure to address Comment 14 is: [SEE NEW FIGURE 5 AT END OF THIS COMMENT]

Caption Fig. 5 – Comparison of various scaling options for the change of seasonality of $CO_2$ concentrations over time. The first EOF of the residual fields of observations minus the mean 1984-2014 $CO_2$ seasonality (Fig. 9 a.2) is scaled with an EOF score. Before 1984, this EOF score is regressed against a composite of global-mean $CO_2$ concentrations and global-mean surface air temperatures (see text and panel b). Alternative regressors include global-mean $CO_2$ concentrations (panel a), lagged averages of monthly global-mean surface air temperatures (panel c) and raw global-mean annual average surface air temperatures (HadCRUT4v) (Morice et al., 2012) (panel d). The regressed EOF score back in time is shown in panel e. A comparison to the first $CO_2$ measurements of higher northern latitudes at so-called Station P (STP) and Point Barrow in Alaska (PTB), where the seasonality change is most pronounced, is provided in panels f and g, respectively (see text for discussion).

Comment 15: lines 278-281, 292-294 and Figures 1, 2, 4, 5 and 9. The ad hoc smoothing of Law Dome data and very high sampling resolution of non Law Dome recent ice core data (e.g. Bauska et al., 2015, Mitchell et al., 2013, Rhodes et al., 2013) makes the choice of using Law Dome only data less obvious than some years ago. The choices of time scales in the Figures make it very hard to appreciate the match of the

scenarios with available firn and ice data especially for the beginning of the CMIP6 historical runs (1850-1950). One specific issue is that in view of the N2O data dispersion, I'm not convinced that the dip in the N2O scenario around 1850 is really reliable.

Reply 15: We hope that the updated Figure 6 addresses the concern and provides better comparability of the different firn and ice records. In regard to CO2, we choose our settings for the median-preserving smoothing such that the shape of the WAIS record by Bauska et al. 2015 is approximately matched, although corrected by the offset. Thus, while the smoothing originates from the Law Dome record, we took the evidence of the other ice records into account and opted not to follow the more pronounced variations in the Law Dome record (although we are happy to produce such a higher-frequency CO2 history for interested modelling groups). In regard to N2O: Indeed the divergence between different ice record histories is large and unresolved. It is outside the scope of this study to arrive at a conclusive best-estimate synthesis of all available ice core records. We hence would like to see our N2O histories to be seen as one plausible history.

Comment 16: lines 305-308: in this very technical description, I did not understand the main message. Why does the Gosh et al. (2015) data need to be updated? Why excluding North GRIP? Does that induce significant changes?

Reply 16: We clarified the text to read now: "We used the NEEM CH4 firn measurements from Buizert et al (2012) (2008 campaign), with effective ages from Ghosh et al. (2015) based on the iterative dating method of Trudinger et al (2002b), corrected for the effect of gravity (as applied in other firn data) and put onto the NOAA 2006 primary calibration scale."

Comment 17: lines 312-315: the methodology is unclear to me here. Mixing ratio data are always local. Global or hemispheric means should already be the result of an assimilation procedure. Is there some circularity in constraining an assimilation procedure with assimilated data?

[Figure]

Reply 17: We would not call it circularity. We simply base the CMIP6 datasets on available literature studies in some cases. For some gases that lack a good representation in the main measurement networks (at least for some periods), like the PFCs listed in this sentence, we opt for replicating the same NH and SH averages as the cited literature studies. Given that we do not claim of having derived those latitudinal gradients from raw station data, we are not sure we understand the issue.

Comment 18: lines 314-321: the list of key studies should be focused on key species in terms of radiative forcing. It would be useful to provide the references of all data used in the supplementary tables.

Reply 18: Thank you and apologies for our oversight of not having provided a full reference list before. We hope that the new Table 12 in conjunction with the updated Factsheet figures addresses that concern. We also revised the text to provide missing references. The ranking in terms of the 15 most radiatively important species is provided in Table 5.

Comment 19: lines 323-325: what are the data used to constrain the major halocarbon trends (e.g. CFC-11, CFC-12, HCFC-22) before about 1978?

Reply 19: We clarified that by adding the sentence: "The three radiatively most important fluorinated species CFC-12, CFC-11 and HCFC-22 (Table 5) follow the global mean concentrations provided by Velders et al. (2014), in conjunction with separately derived latitudinal gradients and seasonality."

Comment 20: lines 373-377: I don't understand the motivation for grouping the species as ozone depleting versus non ozone depleting. Splitting the species between those destroyed in the troposphere or not seems more obvious to me as they have very different vertical structures. Could the ozone depleting choice be commented?

Reply 20: That choice arises from an informal survey among modellers, with some CMIP6 modelling frameworks being set up in that way (and ignoring vertical gradients

anyway). We agree with the reviewer that many other choices would make more sense. As in CMIP5, modelling groups can and likely will also apply their own aggregations to the provided original gas-by-gas data. We encourage modelling groups to do so.

Comment 21: lines 380-390: are the inputs used for CMIP5 simulations (CO2 fluxes?) and the CMIP6 input scenarios discussed here fully independent?

Reply 21: Yes and no. Yes in regard to the spatial patterns and the seasonality. No, in regard to the global-mean average values. For CMIP5, only global-mean annual average $CO_2$ surface concentrations were provided as recommendations. Thus, in terms of input recommendations, everything but the global-mean annual-averages are independent (as the latter are partially based on the same data in both CMIP5 and CMIP6, such as Law Dome records). To what extent CMIP5 models used NOAA network data to constrain their internal and spatially varying $CO_2$ fields is outside of the scope of this study as we did not undertake a systematic survey in this regard.

Comment 22: lines 399-409 and Figure 1: the consistency of the different datasets for the CMIP6 simulation period (after 1850) should be made more visible on the figure and should be commented in the manuscript.

Reply 22: The new Figure 6 panel c that focusses on the 1850 to 2014 period hopefully addresses this concern. We've also added a new sentence in that regard: "The differences between the WAIS and the Law Dome record persist in 1850 to 1890 with subsequent data points being more aligned with each other (Fig. 6c)."

Comment 23: lines 448-449: Section 3.1 starts with discussing discrepancies of several ppb between ice core datasets and large uncertainties on meridional gradients. Providing estimates of the uncertainties on the global mean CO2 at the dates mentioned would be useful.

Reply 23: To assist the reader, we added the following sentence: "Our methodology does not include a formal uncertainty analysis. As a minimum uncertainty for the

1850's pre-industrial values, we refer to the 1.2ppm variability stated by Etheridge et al. (1996), also used in Rubino et al. (2013) and Trudinger et al. (2002a) as minimum uncertainty for that period." Comment 24: lines 497-498 and Figure 1f: in view of the large discrepancies between ice core records and large dispersion of the N2O data in the 1850-1970 period, more firn and ice datasets could be used to evaluate the trend used in CMIP6 historical runs (e.g. Machida et al., 1995; Battle et al., 1996; Ishijima et al., 2007) Sections 3.4 and 3.5 lack focus on the most important species in terms of radiative forcing and bibliographic references.

Reply 24: We took the reviewer's suggestion on board, plotted the Machida et al. H15 ice core record, the South Pole firn data by Battle et al. (1996) and also the modelling study results presented in Ishijima et al. (2007). Please see our revised Figure 6 e above. The amended discussion of our data in comparison with these alternative ice and firn records reads now: "A temporary local maximum indicated by individual Law Dome data in the 15th century is not resolved by our smoothing, and a similar spike in the 17th century is only just reflected (Fig. 6f). Several data points indicate a small decrease after a 1750 maximum with a minimum in 1850 of around 273.02 ppb. This 1750ish maximum and subsequent minimum around 1800-1850 is also apparent in the H15 ice core record by Machida (1995) (we scale-corrected the Machida data downwards by 1 ppb as in Battle et al. (1996)) (Fig. 6b). After 1850, N2O mixing ratios increased markedly, reaching 1900, 1950, 2000 and 2014 values of 279.5, 289.7, 315.8 and 327.0 ppb, respectively (Table 6). Comparing the different firn and ice records, the 1920 – 1940 period seems particularly uncertain with some high measurements close to and beyond 290ppb from both Law Dome and H15, while some of the Law Dome data is still at levels around 285 ppb or even 280 ppb in the case of H15 (Fig. 6e). The South Pole firn data (Battle et al., 1996) suggest lower N2O concentrations in the 1920s and around 1960 – compared to both the smoothed Law Dome data (thin dashed line in Fig. 6e) and consequently our even higher global-mean estimate. Although the Ishijima estimate (Ishijima et al., 2007) (their Figure 6a) around 1952 is almost identical to our global-mean, their modelling study suggests slightly lower values around 1960

before being closely matching again from 1970 onwards. The Law Dome firn record (Park et al., 2012) suggests slightly higher N2O concentrations for the high southern latitudes compared to our global-mean (Fig. 6e). "

Comment 25: lines 536-545: bibliographic references should be provided for the nineteenth century mixing ratio estimates.

Reply 25: We apologize for not having been clear on the origin of those pre-industrial concentration estimates. We now extended the paragraph to read: "Four of the considered chlorinated and ozone depleting substances are assumed to have natural emissions and hence non-zero pre-industrial mixing ratios. We estimate those pre-industrial natural background concentration by a simple budget equation under the assumption of a constant lifetime (IPCC, 2013) of 1 year for CH3Cl and 0.8 years for CH3Br – minimizing the error term when taking into account anthropogenic emission and atmospheric concentration estimates over 1950 to 1990 by Velders et al. (2014). Specifically, methyl chloride (CH3Cl) is assumed to have pre-industrial global-mean mixing ratios of 457 ppt, and methyl bromide (CH3Br) of 5.3 ppt. Chloroform (CHCl3) is assumed to have a pre-industrial mixing ratio of about 6 ppt, approximately in line with findings by Worton et al. (2006) and the estimation by Aucott et al. (1999) that in 1990 CHCl3 was at about 8 ppt, with 80% of emissions assumed to be of natural origin. Lastly, in the absence of other information (a good understanding of the natural vs anthropogenic source fraction or historical industrial production records) the available firn measurements (e.g., Trudinger et al., 2004) supplying information about methylene chloride (CH2Cl2) mole fractions in the early 20th century are used to suggest a 6.9 ppt pre-industrial mean mixing ratio with a strong latitudinal gradient that results in northern (southern) hemisphere average mixing ratios of 12.8 (1.0) ppt. The transition of mixing ratios of some species between the observational station data and pre-industrial levels are also uncertain. For CH2Cl2, our derivation is in line with the smooth trajectory of Trudinger et al. (2004), indicating an almost monotonic transition between 1997 values and pre-industrial mixing ratios (Fig. 26f). Our assimilation approach (which is based on the

Walker et al. data (2000)) causes our carbon tetrachloride (CCl4) reconstruction to have a near-zero pre-industrial concentration of 0.025 ppt (0.025% of its peak value of 100ppt).We note that Walker et al. (2000) suggest zero pre-industrial concentrations before 1910, although the lowest empirical evidence from firn records suggest <5ppm ppt (Butler et al., 1999) or 3-4ppt as measured by S. Montzka for 1863 firn air and reported in Liang et al. (2016)"

Comment 26: lines 717-721 and Figure 9: the reason why the early part of the CMIP6 CO2 trend is smoother than the CMIP5 trend whereas the early part of the CMIP6 N2O trend is less smooth than the CMIP5 trend is unclear to me. Could this choice be commented?

Reply 26: This was not a deliberate "choice" but is rather an outcome of the chosen methodology. The CMIP5 extension of the RCP histories was an assemblage of existing timeseries (see Meinshausen et al. 2011). The CMIP6 derivation is much different by using raw measurement data points to derive a global field of concentrations and then back out the global mean. Thus, there was no deliberate choice involved regarding the smoothness of the timeseries in regard to CMIP5.

Comment 27: lines Section 5.2: Figures 10, 11, 13 and 14 are not directly comparable to Figure 2 and could be placed in a Supplement, whereas Figure 12 could include a representation of the CMIP6 scenarios in similar format as the CMIP5 mixing ratio outputs.

Reply 27: We followed the suggestion to move the Figures 10, 11, 13 and 14 into the Supplementary Appendix B. In regard to Figure 12, we opted for also moving this to the Appendix and not overloading the figure by additional timeseries. We however show the ensemble mean of the bottom panel in former Figure 12 in the CO2 overview (new Figure 9, panel b).

Comment 28: lines Section 5.5: the comparison with other literature studies lacks priorities in terms of radiatively most important species and a check of the independence

of the data used for evaluation with respect to those used to generate the assimilated fields. For example, the CO2 and CH4 high Northern latitude trends in Buizert et al., 2012 were provided by V. Petrenko (see file SCENARIO_NEEM2008_....xls) and are mostly based on NOAA ESRL and Law Dome data (see Section 2.4.2 in file Supplement Buizert ....pdf). On the other hand, the comparison with early CO2 atmospheric data (Keeling et al., 1976) is not commented. I'm surprized that the Cape Grim air archive data are not commented for N2O (Park et al., 2012) and other species (e.g. Newland et al., 2013). It could be mentioned that the Martinerie et al. (2009) trends for halocarbons are based on industrial emission histories and are used in Buizert et al. (2012) for the time period preceding atmospheric measurements.

Reply 28: Thank you for those helpful comments and clarifications. • Petrenko. We now honored the origin of the CO2 and CH4 timeseries by labelling them "Petrenko-2010" and included them in the source Table copied above. • Furthermore, we now inserted a new section as comparison to the mostly independent Keeling Scripps CO2 station data before 1984 (exception is Mauna Loa of which we use the annual averages) – see Figure X copied above. • We inserted the Park et al. Law Dome firn data in Fig 6 panel e. • We added the Newland Halon data on the UEA scale to the Halon factsheets and referred to it in the text, such as: o "Halon-2402 is also an illustration of how big differences in some measurement scales can potentially be. The Cape Grim data analysed by Newland with a volumetric UEA scale indicates 10-15% lower mixing ratios (Fig. 33f) (Newland et al., 2013)."

Comment 29: lines 860-870: I could not see the Buizert et al. (2012) trends on the Figures. Are the commented differences within uncertainties provided in Buizert et al. (2012)?

Reply 29: We now increased the strength of the triangular data symbol, which is now labelled Petrenko-2010 in the CO2 overview figure. With a vector-graphic reproduction of the figures in the final manuscript, we hope that the data is better visible. Regarding the differences: Yes, the Petrenko-2010 data is in line with the seasonal cycle for high

northern latitudes. It might be a different story for the variations seen in the CH4 record from Petrenko/Buizert, which seems to show a higher year to year variability than our data suggests. For example, the jump from December 1955 CH4 mixing ratios of 1251ppb to January 1956 mixing ratios of 1307 is not reproduced by our data. (see new Figure 11f, former Figure 4f).

We now clarified the text so that the mentioned differences between the high-northern seasonal cycle and the northern-hemispheric average are not misunderstood as inconsistent. The new text reads: "For CO2, the Petrenko data set has, as expected for the high northern latitudes, a very strong seasonal cycle, consistent with our less pronounced northern hemispheric-average cycle, as the data represents higher northern latitudes (Fig. 9f, g, and h). The long-term mixing ratio trend over time in the Petrenko CO2 record seems similar to the global CMIP5 data set which in turn was based on previous Law Dome data, indicating a slight local maximum in 1890 and lower 1940s plateau (cf. Fig. 9g and Fig. 15)."

Comment 30: lines 871-877: would the pioneer study by Butler et al. (1999) be more consistent with other trends before 1950 if the South Pole firn air age spread was taken into account?

Reply 30: If a wider age distribution were assumed in the analysis of Butler then the derived CCl4 history would indeed be consistent with a later onset, and steeper increase towards higher values. So, yes, a wider air age spread assumption in the Butler methodology would in this case close the gap towards the Velders et al. 2014 dataset. We added a sentence in that regard, reading: "The difference between the Butler and Velders datasets can probably be explained by the wider firn air age distribution in the study by Butler.."

Comment 31: lines Section 6: major additional uncertainties for the early part of CMIP6 historical simulations such as the lack of constraints on nineteenth century CO2 meridional gradients could be mentioned. Tables 2, 4, 5, 6, 7, 8, 9, 11 (technical documentation) could be placed in a Supplement

Reply 31: We structured the limitations section now with clear subheadings and added that particular point by saying "More generally, further research into observational and modelling-derived constraints regarding pre-1950 latitudinal gradients of CO2 could allow future studies to go beyond our simplified assumption of a zero pre-industrial gradient in the light of the uncertainty." In section 6.6 We would prefer the keep the tables with the data descriptions in the main manuscript so that the full list of references is retained – given the importance of many people's data contributions. However, we are happy to reconsider based on the editor's guidance.

Technical corrections:

Comment 32: lines 60, 708, 709 etc. and references: Meinshausen et al., 2011, 2011a and 2011b seem to be the same article

Reply 32: Thanks. Corrected.

Comment 33: lines 158-159: the first figure quoted in the manuscript should be Figure 1 rather than Figure 22

Reply 33: Thanks. Corrected.

Comment 34: lines 189-193 and 281-282: It would be clearer to describe the scale change of the firn and ice data together with the scale description of the atmospheric data.

Reply 34: Suggestion implemented.

Comment 35: line 230: define EOF notation at first use

Reply 35: Done.

Comment 36: line 595: is it really needed to quote Eyring et al., GMDD, 2015 instead of Eyring et al., GMD, 2016?

Reply 36: No. Thanks. We updated all Eyring references to the final GMD manuscript.

Comment 37: lines 799-800: check the writing. This sentence seems contradictory with lines 167- 171, 801-802 and 824-829.

Reply 37: Thank you. We corrected the sentence and clarified the difference between the NOAA MBL and other NOAA products.

Comment 38: lines 904-905: Trudinger et al. (2004) is not more recent than WMO (2014) and Velders et al. (2014)

Reply 38: Thank you. The "more recent" statement referred to the Butler et al. 1999 study that WMO and Velders are based on. Sentences corrected to read now: "Here, we follow again the WMO (2014) and (not independent) Velders et al. (2014) reconstruction that are based on Butler et al. (1999) firn reconstructions. However, we note that the more recent Trudinger et al. (2004) CH3Cl reconstruction indicates both a significantly lower mixing ratio for southern latitudes in the 1970s and a smoother increase compared to the more sudden rise of mixing ratios around 1940 as implied in this study (Fig. 29g)."

Comment 39: lines 1408-1410: incorrect list of authors

Reply 39: Our apologies. Corrected.

Comment 40: Figure 1: horizontal scale issue for panel c. A complete reference should be provided for each dataset. I saw only CO2 data (no CH4 and N2O data) in Rubino et al. (2013), and Table 1 mentions different references for Law Dome CH4 and N2O data.

Reply 40: Thanks. We changed Figure 1 (see above) and corrected the references.

Comment 41: Figure 6: panels g, i, k, m, o, q would be much easier to read if the horizontal scale started in 1850 or 1900

Reply 41: Thanks. We will adapt the figure correspondingly.

Comment 42: Figure 12: I can't see the 12 five lines mentioned in the caption, and the shaded areas are not described in the caption

Reply 42: Thanks. We now describe the shaded areas (they are the min-max ranges over those 12 lines) and hope that the vector graphic figure will come out more clearly.

Comment 43: Figure 15: wrong reference for the CH4 "NEEM" scenario (see Supplement of Buizert et al. 2012, Section 2.4.2 in file Supplement Buizert ....pdf, and file SCENARIO_NEEM2008_....xls, CO2 and CH4 scenarios were made by Vas Petrenko). The NOAA global mean and WDCGG global mean results should be made easier to distinguish.

Reply 43: Thanks. Apologies. We corrected the data label to Petrenko.

Comment 44: References not provided in the manuscript: • Battle et al., Nature, 383(6597), 231-235, 1996 • Hall et al., Atmos. Meas. Tech., 7, 469-490, 2014 • Ishijima et al., J. Geophys. Res., 112, D03305, 2007, doi:10.1029/2006JD007208 • Machida et al., Geophys. Res. Lett., 22(21), 2921-2924, 1995 • Newland et al., Atmos. Chem. Phys., 13, 5551–5565, 2013. • Park et al., Nature Geoscience, 2012, DOI: 10.1038/NGEO1421. • Rhoderick et al., Elementa Sci Anth 3: 000075, 2015, doi:10.12952/journal.elementa.000075

Reply 44: Thanks. We do include these references now in the reference list.

References.

Ahn, J., Brook, E. J., Mitchell, L., Rosen, J., McConnell, J. R., Taylor, K., Etheridge, D., and Rubino, M.: Atmospheric CO2 over the last 1000 years: A high‐resolution record from the West Antarctic Ice Sheet (WAIS) Divide ice core, Global Biogeochemical Cycles, 26, 2012.

Arnold, T., Harth, C. M., Mühle, J., Manning, A. J., Salameh, P. K., Kim, J., Ivy, D. J., Steele, L. P., Petrenko, V. V., and Severinghaus, J. P.: Nitrogen trifluoride global emissions estimated from updated atmospheric measurements, Proceedings of the

National Academy of Sciences, 110, 2029-2034, 2013.

Arnold, T., Ivy, D. J., Harth, C. M., Vollmer, M. K., Mühle, J., Salameh, P. K., Paul Steele, L., Krummel, P. B., Wang, R. H., and Young, D.: HFC‐43‐10mee atmospheric abundances and global emission estimates, Geophysical Research Letters, 41, 2228-2235, 2014.

Aydin, M., Montzka, S., Battle, M., Williams, M., De Bruyn, W. J., Butler, J., Verhulst, K., Tatum, C., Gun, B., and Plotkin, D.: Post-coring entrapment of modern air in some shallow ice cores collected near the firn-ice transition: evidence from CFC-12 measurements in Antarctic firn air and ice cores, Atmospheric Chemistry and Physics, 10, 5135-5144, 2010.

Buizert, C., Martinerie, P., Petrenko, V. V., Severinghaus, J. P., Trudinger, C. M., Witrant, E., Rosen, J. L., Orsi, A. J., Rubino, M., Etheridge, D. M., Steele, L. P., Hogan, C., Laube, J. C., Sturges, W. T., Levchenko, V. A., Smith, A. M., Levin, I., Conway, T. J., Dlugokencky, E. J., Lang, P. M., Kawamura, K., Jenk, T. M., White, J. W. C., Sowers, T., Schwander, J., and Blunier, T.: Gas transport in firn: multiple-tracer characterisation and model intercomparison for NEEM, Northern Greenland, Atmos. Chem. Phys., 12, 4259-4277, 2012.

Butler, J. H., Battle, M., Bender, M. L., Montzka, S. A., Clarke, A. D., Saltzman, E. S., Sucher, C. M., Severinghaus, J. P., and Elkins, J. W.: A record of atmospheric halocarbons during the twentieth century from polar firn air, Nature, 399, 749-755, 1999.

Cooperative Global Atmospheric Data Integration Project: Multi-laboratory compilation of synchronized and gap-filled atmospheric carbon dioxide records for the period 1979-2012 (obspack_co2_1_GLOBALVIEW-CO2_2013_v1.0.4_2013-12-23). NOAA Global Monitoring Division: Boulder, C., U.S.A (Ed.), 2013.

Ivy, D. J., M. Rigby, M. Baasandorj, J. B. Burkholder and R. G. Prinn (2012). "Global

emission estimates and radiative impact of C4F10, C5F12, C6F14, C7F16 and C8F18." Atmos. Chem. Phys. 12(16): 7635-7645.

MacFarling Meure, C., Etheridge, D., Trudinger, C., Steele, P., Langenfelds, R., Van Ommen, T., Smith, A., and Elkins, J.: Law Dome CO2, CH4 and N2O ice core records extended to 2000 years BP, Geophysical Research Letters, 33, 2006.

Martinerie, P., Nourtier-Mazauric, E., Barnola, J. M., Sturges, W. T., Worton, D. R., Atlas, E., Gohar, L. K., Shine, K. P., and Brasseur, G. P.: Long-lived halocarbon trends and budgets from atmospheric chemistry modelling constrained with measurements in polar firn, Atmos. Chem. Phys., 9, 3911-3934, 2009.

Montzka, S., McFarland, M., Andersen, S., Miller, B., Fahey, D., Hall, B., Hu, L., Siso, C., and Elkins, J.: Recent trends in global emissions of hydrochlorofluorocarbons and hydrofluorocarbons: Reflecting on the 2007 adjustments to the Montreal Protocol, The Journal of Physical Chemistry A, 119, 4439-4449, 2014. Mühle, J., Ganesan, A., Miller, B., Salameh, P., Harth, C., Greally, B., Rigby, M., Porter, L., Steele, L., and Trudinger, C.: 
[revised manuscript text omitted]

**Fig. 6.** New Table 12 - Page 3

---

## Author Comment (AC3) · 29 Nov 2016

Comment 1: The paper by Meinshausen and colleagues presents historical climate model scenarios of GHGs. There are numerous extensions over their CMIP5 efforts that together make considerable progress on a number of fronts. Particularly impressive is is knitting together of observations and models. This will also help the two communities understand the others needs. The paper represents a huge amount of work by the author team and it provides huge community good. I was very impressed.

Reply 1: Thanks.

Comment 2: The paper is long, has complex figures and contains a lot of technical detail. I would argue that this is appropriate and necessary for the GMD approach to allow clarity of methods and their reproducibility. The paper goes beyond simply documenting the method and showing the data. It details comparison to other data,

and has a very interesting discussion and limitations sections that has insights into ESM uncertainty and possible effects on climate.

Reply 2: Thanks. We appreciate the understanding for the lengthy paper format.

My suggestions of corrections are of only a technical nature, outlined below

Comment 3: Ln 26-27. I would argue that lots of things change climate not just GHGs and aerosol. I would maybe phrase as GHGs largely responsible for the warming and associated climate change?

Reply 3: Suggestion taken on board.

New text reads: "Those elevated GHG concentrations warm the planet and - together with net cooling effects by aerosols - are largely responsible for the observed warming over the past 150 years."

Comment 4: line 28 and 40. The future climate change comments seemed strange in abstract as to me the future is another but related problem - the paper really helps sort the past, but your call

Reply 4: Thanks. We deleted "future" climate change in line 28/29. At the end of the abstract, we keep the outlook for why the modelling of past climate change can be important for the future (i.e. reduce uncertainty).

Comment 5: Abstract. Time period is not mentioned - only 1850 date. Also not at all clear you are talking about historical scenarios - or changes through time. 0-2014 ins mentioned but analysts concentrates on 1850-2014? Will historical runs end at the end of 2014? I thought it was 2015, but I may well be wrong!

Reply 5: We clarified the abstract. And no, the data only runs until the end of 2014 – a convention across multiple historical CMIP6 datasets. We also clarified that the astronomical year 0 (the year before 1 AD) equals the Gregorian or Julian year 1BC.

New text: "The focus rests on the period 1850 to 2014 for historical CMIP6 runs, while

the data is provided from the beginning of year 0 (year 1 BC) towards the end of 2014"

Comment 6: Maybe I'm stupid but it did not seem clear where the data could be accessed?

Reply 6: There is a dedicated "Data Availability" section at the very end of the GMD manuscript before the references. We now included the main links also in the abstract.

New text: "The data is available at https://pcmdi.llnl.gov/search/input4mips/ and www.climatecollege.unimelb.edu.au/cmip6"

Comment 7: The paper would benefit from a careful proof read. I am afraid that it is beyond my community spirit to do this! But examples are 1. Sections are not referred to consistently, sometimes by names, sometimes by numbers sometimes both cf ln 70,128,210 (e.g.) 2. There are typos in places e.g.1026-1027 (to prove i read to end!) 3. The odd statement is repeated 4. Equations are not presented as uniformly as they might be e.g. 230-236 e.g. nxm 5. Do you want asterisks or for multiply or something else? line 210 Figures 20 and 21 might also be useful here -these don't seem to be referred to in text?

Reply 7: Our apologies. We now performed another proof read by "fresh eyes". Thanks to Zebedee Nicholls in that respect. Specifically:

1. We now refer to section always by the section number, and in some cases, where the title assists the reader also by the title, i.e. when referring to the "Limitations" section. 2. 3. Thanks. We had a native speaker proofreading the manuscript again. 4. 5. We made the deleted duplications, where we found them. 6. 7. We corrected nxm to 'n $\times$ m', i.e. using the standard multiplier sign, and streamlined other equations to some degree, e.g. by deleting superfluous multiplier asterisks '*' or replacing it by the 'x' sign where useful for readability. 8. 9. Thanks for spotting that Figure 20 and 21 were not referenced in the text. That was due to a non-dynamic Figure link, as the figure references in the following paragraph meant to point to exact those figures. Now

corrected. 10.

Comment 8: Figures don't seem to appear in the correct order - I'm not sure what your logic is here?

Reply 8: Thanks. The figures have now been sorted. Figures 1 to 22 in the main manuscript should be ordered in the order of appearance of their reference. Appendix A contains the factsheets of other gases. And Appendix B contains to supplementary figures that are related to Figure 16.

Comment 9: line 167 - that are these AGAGE? files - maybe giving a web address early on where files could be found would help? Or adding more explanation?

Reply 9: Suggestion implemented. The website for the data is now provided, i.e. http://agage.eas.gatech.edu/data_archive/agage/.

Comment 10: 189-193 - it is not clear which scalings are being referred to for what gases?

Reply 10: We rephrased lines 189-193 to hopefully make it clearer what happens to $CO_2$ (nothing), $CH_4$ (Tohuko is converted to NOAA04, if necessary) and $N_2O$ (nothing), so that those lines read:

"In the case of $CO_2$, we source all our $CO_2$ station data from the NOAA network, which means no scale conversion is necessary. In the case of $CH_4$, we account for different calibration scales by converting AGAGE $CH_4$ data (Tohuko University scale) to the NOAA scale (NOAA04) (multiplication by 1.0003). In the case of $N_2O$, both the AGAGE (SIO1998) and NOAA network calibration scales (NOAA-2006) are compatible without the need for a conversion factor (WMO, 2012)."

Comment 11: I found Table 1 really useful in helping me understand your methods - could this be referred to earlier?

Reply 11: That was our oversight. We now inserted the reference to Table 1 early on

in section 2.1, by stating "….Figure 1 and tabulated also for the three main GHGs in Table 1."

Comment 12: The figures are generally good considering their complexity - but details are hard to see even when zooming in online, such as the small "5" on fig 22 referred to in the text.

Reply 12: We acknowledge the complexity and dense information content of the figures. We hope that the final manuscript, with a vector rendering of all the figures (rather than the medium raster resolution that is present in the discussion paper) will make small details much more readable, especially when using zoom.

Comment 13: I also found it hard to see the CMIP5 lines on the CMIP6/CMIP5 comparison figures.

Reply 13: Thanks. We changed the thin CMIP5 lines towards thicker red lines.

Comment 14: On the science I had a few questions 1. It might be useful to quote 1750 PI concentrations and 2011 concentrations to compare to IPCC. A comparison might also be fun with IPCC historical forcings?

Reply 14: We do provide 1750 and 2011 concentrations in Table 6 for the main gases. Furthermore, we compare our 2011 concentrations to the various' networks' estimates as stated in IPCC AR5 WG1 in our Table 7.

We hesitate to compare our linearized illustrative forcings to IPCC – at least for $CO_2$, $CH_4$ and $N_2O$ (Figure 6 upper panels), as our forcings are only using linear radiative efficiencies, not even simple saturation formula. However, we will introduce comparison IPCC AR5 numbers for year 2011 in the middle and lower panels of Figure 6 for comparison.

Comment 15: 2. I guess your forcing estimates were all made with global radiative efficiency formula - are you going to run your fields through a radiation call to estimate actual forcings. Give me a shout if you would like someone to do this!

Reply 15: Thank you very much for your kind offer. We will take you up on that for a future project.

---

## Author Comment (AC4) · 29 Nov 2016

Thanks to R. Pincus for these additional comments.

Comment 1: This comment raises the fairly narrow question as to whether the level of detail provided by this reconstruction of greenhouse gas concentrations is appropriate for CMIP.

Reply 1: We agree with R. Pincus that the level of detail is more than what the CMIP6 end-user will need. For this, a simple description of the datasets, the formats, the scope and some key limitations would have been sufficient. We hence provided a section 4 that is dedicated to the CMIP6 needs. However, without the details of the data derivation, the developed method would be intransparent. We hence hope that this structure of the manuscript with a dedicated section 4 is a suitable compromise to bridge the two communities, i.e. the measurement community as well as the modelling

community.

Comment 2: Without diminishing the tremendous amount of work represented by this reconstruction nor its possible value in other contexts, discussions with modeling centers over the protocol for the Radiative Forcing MIP (https://dx.doi.org/10.5194/gmd-9-3447-2016) suggests that much of the detail in these specifications will not implemented as part of the CMIP6 protocol. In particular:

Reply 2: We are aware that some models choose to only implement globally uniform concentrations and that is ok. We feel there is some imbalance in terms of how much detail is implemented in some other aspects of the CMIP6 input data, which would have similar or lesser effect on seasonal and latitudinally dependent forcings, though. If CO were implemented (see comment below), that would be an example. Time will tell, whether the broader community regards the additional detail of seasonality and latitudinal dependence as something important.

Comment 3: 1) We fear that few modeling centers will implement the latitudinal variations or vertical profiles of well-mixed greenhouse gases as described in section 4. We discussed this only with GFDL and the Met Office, so we may well be wrong, but these are two topnotch centers with strong local interest and expertise in radiation issues, and both will use time-dependent scalar values.

Reply 3: We were made aware at a late stage of our project that the default minimum recommendation will be to only use time-dependent scalar values and we fully respect that some modelling centres will only follow the minimum recommendation. However, given the importance of regional inhomogenous forcings, for example to correctly undertake historical constraining excercises and detection and attribution (e.g. Shindell, 2014, doi:10.1038/nclimate2136), it seems worthwhile to provide the choice to the modelling community. Only using time-dependent scalar values for all latitudes will insert easily avoidable biases into the models (although they are not dramatic of course). We are aware that there are some models that can choose to nudge their internally generated mixing ratio fields with a similar seasonality and latitutudinal resolution towards the scalar global average values and that is a perfectly legitimate approach, too (although it might hinder slightly the historical comparability given the differences in those internally generated fields during CMIP5, as we highlight in some of the CMIP5 figures in Appendix B).

Comment 4: 2) The three options for describing atmospheric composition (lines 369-377) offer a useful range of compromises but option 1, using a subset of gases, might be improved by greatly restricting the number of halocarbon species provided. It is not clear how many, if any, line-by-line models include the long list of species. It is certain that no climate model radiation codes include more than a few. Based on rough calculations using radiative efficiencies from IPCC AR4, including only CFC-11, CFC-12, HCFC-22, CFC-113, HFC-134a, and CCl4 reproduces the total instantaneous radiative forcing in 2014 to within 0.045 W/m2.

Reply 4: We fully agree that the subset of species can be further reduced and we provide the impact of that choice by stating the cumulative percentage radiative forcing change covered by the top 15 species. (see our Table 5, reproduced at the bottom of this reply). It is of course up to the modelling centre to only choose the top, say, 8, species and then cover 99.1% instead of 99.7% of the GHG-induced radiative forcing. The choices by modelling centres will vary. It is important though that those choices get clearly documented by the modelling centres.

Comment 5: If the protocol includes levels of details that modeling centers are unlikely to observe it may be more appropriate to reduce to level of detail to something practical.

Reply 5: We agree. And this is the reason why we provide Option 2 and 3, in which case modelling centres only have to include 5 (equivalent) gases to cover together 100% of the forcing change. Of course, the more top notch modelling centres can (and have done in the past) come up with their own aggregations.

Comment 6: On a distinct issue, we are surprised that the protocol does not include

estimates of CO. Rough estimates suggest a clear-sky instantaneous radiative forcing at 2014 of roughly 0.05 W/m2, or the same contribution from one gas as from 35 of the specified halocarbons in total.

Reply 6: CO did not fall into our scope of "long-lived" GHG concentrations, so we would like to refer to the CMIP panel for who will cover CO concentration (fields). We agree that it would be important to include. And by the same token, it can be important to include the seasonality of CO2 and latitudinal gradient of CH4 and other GHGs also, which has locally an influence of multiple times that of CO forcing... ïĄŁ

Thanks for taking the time for these additional comments.

―――――――――――――――――

[Figure]

Table 5- Options for reducing the number of GHGs to be taken into account to approximate full radiative forcing of all GHGs. In Option 1, a climate model explicitly resolves actual GHG mixing ratios. With 8 and 15 species, 99.1% and 99.7% of the total radiative effect can be captured. In Option 2, only CFC-12 is modelled next to $CO_2$, $CH_4$, and $N_2O$; all other gases are summarized in a CFC-11-equivalence mixing ratio. In Option 3, all ODS are summarized in a CFC-12-equivalence mixing ratio, and all other fluorinated substances are summarized in HFC-134a-equivalence mixing ratios. The first column indicates the importance of gases in terms of the radiative effect change between 1750 and 2014. Note that below shares are approximations, as linear radiative forcing efficiencies are assumed here for all gases, also for $CO_2$, $N_2O$ and $CH_4$.

| Rank | The GHG contribution to climate change since 1750. Shares of change of total warming effect since 1750: Approx. Radiative forcing contribution between 1750 and 2014 relative to that of all GHGs | | Option 1 — Using subset of actual mixing ratios, no equivalent gases. Shares of total warming effect: Approx. Radiative effect compared to effect of all GHGs (absolute in 2014, not relative to 1850) | | Option 2 — Summarizing all gases of lower importance than CFC-12 into CFC-11eq. | | Option 3 — Summarizing all ODS into CFC-12-eq and all other fluorinated gases into HFC134a-eq | |
|---|---|---|---|---|---|---|---|---|
| 1 | CO₂ | 64.0% | CO₂ | 72.9% | CO₂ | 72.9% | CO₂ | 72.9% |
| 2 | CH₄ | 79.5% | N₂O | 86.1% | N₂O | 86.1% | N₂O | 86.1% |
| 3 | CFC12 | 86.0% | CH₄ | 95.0% | CH₄ | 95.0% | CH₄ | 95.0% |
| 4 | N₂O | 92.2% | CFC12 | 97.2% | CFC12 | 97.2% | CFC12-eq | 99.5% |
| 5 | CFC11 | 94.5% | CFC11 | 98.0% | CFC11-eq | 100.0% | HFC134a-eq | 100% |
| 6 | HCFC22 | 96.4% | HCFC22 | 98.6% | | | | |
| 7 | CFC113 | 97.2% | CFC113 | 98.9% | | | | |
| 8 | CCl₄ | 97.8% | CCl₄ | 99.1% | | | | |
| 9 | HFC134a | 98.3% | HFC134a | 99.3% | | | | |
| 10 | CFC114 | 98.5% | CF₄ | 99.4% | | | | |
| 11 | HFC23 | 98.7% | CH₃Cl | 99.5% | | | | |
| 12 | SF₆ | 98.8% | CFC114 | 99.5% | | | | |
| 13 | CF₄ | 99.0% | HFC23 | 99.6% | | | | |
| 14 | HCFC142b | 99.2% | SF₆ | 99.7% | | | | |
| 15 | HCFC141b | 99.3% | HCFC142b | 99.7% | | | | |
| … | 28 more GHGs | 100% | 28 more GHGs | 100% | | | | |

**Fig. 1.** Table 5

---

## Author Comment (AC5) · 29 Nov 2016

Reply to Comment by Olaf Morgenstern:

Thank you for the comment. We would however feel uncomfortable to follow the suggestion of using ppmv, ppbv and pptv units for the following reason (I thank in particular Ray Weiss from the author team for discussions and wording contributions here):

Volume ratios of real gases are dependent on temperature and pressure. Most gas volumes are given at STP (standard temperature and pressure), namely 0 degrees C and 1 atmosphere. The concept behind the "v", in ppmv for example, is that if you could separate the gas in question from the rest of the dry air you could measure each volume (nominally at STP), and then report the ratio of these volumes. But many of the gases we measure in atmospheric chemistry, such as CFC-11, for example, are actually liquids at STP so the concept is meaningless in these cases. Even for $CO_2$,

which is much more ideal than CFC-11, the difference between the ideal gas volume and the real gas volume is about 1 part in 150 at STP, yet we report results that strive to be accurate to about 1 part in 4000 (i.e. 0.1 ppm). The concept of the "v" in ppmv, ppbv, etc. may make sense in the ideal world of an introductory high school chemistry course, but breaks down completely under even a rudimentary level of scrutiny. Mole fractions are the only sensible way to report these results.

We hence suggest to add to the final manuscript a clarification like this:

"All mixing ratios given here are as dry air mole fractions, denoted as parts per million (ppm), parts per billion (ppb) and parts per trillion (ppt). Note that dry air mole fractions are independent of temperature and pressure, while volume mixing ratios for real non-ideal gases are not, and at standard conditions can differ significantly from their corresponding mole ratios."

---

## Author Response (AR2)

23 March 2017

To:

Dr Olaf Morgenstern, Editor GMD, olaf.morgenstern@niwa.co.nz

Re: Manuscript GMD-2016-169

Dear Olaf, dear Copernicus editorial team,

Thank you for your acceptance "with corrections" of our CMIP6 historical GHG concentrations paper.

I provide here a brief reply to the editing suggestions.

Editor suggestion: I suggest to make the following corrections: p7l12 and elsewhere: I'm fine with using "dry air mole fraction" and "mixing ratio" interchangeably, using your phrase for clarification. They reduce to the same thing in the limit of ideal gases. However, the term "concentration" is patently a different quantity. "Concentration" is not conserved under transport and is not suitable to force a climate model with. It would come in units of mol/m^3, whereas mole fractions are dimensionless. So I suggest to remove all occurrences of "concentration" from the text.

Reply: After the previous discussions about the units (ppmv versus mole fractions), we had extensive discussions in the author team on the correct terms. Given that the CMIP6 panel asked us to provide the GHG CONCENTRATIONS, and given that this is the generally used term, we feel obliged to use "concentrations" both in the title as well as in the manuscript. We do however clearly define that we use this term interchangeably with "mole fractions". The respective paragraph reads now:

"All concentrations given here are dry air mole fractions and we use 'mole fractions' and 'concentrations' interchangeably and synonymously with 'molar mixing ratios'. For simplicity, we denote the dry air mole fractions 'µmol mol-1', 'nmol mol-1' and

**Australian-German Climate and Energy College**
Lab14, Level 1, 700 Swanston Street, University of Melbourne, Parkville, Victoria 3010, Australia
**T:** +61(0)3 8344 4124   **E:** manager@climate-energy-college.org   **W:** www.climateenergycollege.unimelb.edu.au

'pmol mol-1' as parts per million (ppm), parts per billion (ppb) and parts per trillion (ppt), respectively. Note that dry air mole fractions are independent of temperature and pressure, while volume mixing ratios (e.g. ppmv) for mixtures of non-ideal real gases are not, and at standard temperature and pressure conditions can differ significantly from their corresponding mole ratios."

Furthermore, we agree with the Editor that "mole per volume" is not a conserved quantity. However, we would argue that concentrations are not ubiquitously understood as a "mole per volume" unit. In the general usage of the words, e.g. $CO_2$ concentrations are not considered to be lowering throughout the atmospheric column because of a height / thinning of atmosphere reason. In the specific modelling community, concentrations are clearly associated with mole fractions, as it is denoted in the CMIP6 modelling protocols; the RCPs "Reference Concentration Pathways" are used to denote the concentrations as mole fractions or mole mixing ratios (e.g. ppm), and not as a non-preserved quantity.

We hence suggest to keep a wording throughout the article for simplicity reasons, that is consistent with the CMIP6 panels' request to provide GHG CONCENTRATIONS and hope that the provided clarifications within the ms are sufficient.

Editor's comment - p6l6: The list of 43 gases considered in the paper include quite a few with negligible or zero GWP; they are there because they are ozone-depleting substances. Calling them all "GHGs" is therefore inaccurate. I suggest to replace "GHGs" with "GHGs/ODSs" or to find another way of consistently addressing this concern.

Reply: All the considered substances were chosen because they do have a (sometimes rather small) radiative forcing effect. Thus, none of the substances was considered in this CMIP6 exercise merely because it constitutes an ODS (although that property might be of primary importance for the stratospheric ozone community). We are hence not sure which substances are referred to as not having a GWP. We list all the 43 substances radiative forcing efficacies and lifetimes in the top right corners of the 43 factsheets. The convolution of atmospheric residence and radiative forcing efficacies builds the foundation for deriving GWPs, and hence is not strictly zero for any of the considered gases as neither lifetimes nor radiative forcing efficiencies are zero. We do agree (and discuss in the paper) that some species have negligible radiative forcing effects – which hence gives rise to

**Australian-German Climate and Energy College**
Lab14, Level 1, 700 Swanston Street, University of Melbourne, Parkville, Victoria 3010, Australia
**T:** +61(0)3 8344 4124   **E:** manager@climate-energy-college.org   **W:** www.climateenergycollege.unimelb.edu.au

calculating equivalence species. Again, none of the substances were considered simply because they are ODS. We hence would argue to keep the current naming as purely referring to GHGs.

p9l16: An EOF is an "empirical orthogonal function" not field, as far as I know.

Thanks. Corrected.

p9l20: Capitalize "Gram".

Thanks. Corrected.

p20l22: Some ODSs are indeed also GHGs, but not all, see above. I suggest to insert "Some" at the start of the sentence.

See above. All ODS's have a climate effect indirectly via ozone destruction (as the Editor points out, also via dynamically induced changes). Thus, the introduction of the word 'some' would render the sentence incorrect in our view.

p20l27: The cooling effect of ozone depletion is substantially complicated by its dynamical impact (effect on circulation) which is the real reason people have identified ozone depletion as the leading cause of recent climate change in the Southern Hemisphere.

Thanks. We adapted the sentence that now reads:

"The impact of ODSs on climate is somewhat complicated by their destruction of stratospheric ozone, which induces dynamical effects on circulation patterns, and has a net cooling effect on the global climate."

There is one additional issue. In the course of compiling the latest version of the manuscript, we noticed a small error. We hence suggest to introduce the following "known issues" section in the limitations section:

[Figure]

[Figure]

"6.8  Known issues

There is one known issue in the historical dataseries before the year 2002 for $CF_4$, $C_2F_6$ and $C_3F_8$. We use the Trudinger et al. {, 2016 #4733} datasets and our algorithm categorised them as mid-year values, but the data were estimates for start-of-year values. Thus, while Trudinger et al. {, 2016 #4733} is well aligned with the Mühle et al. {, 2010 #4551} over that time period (given that the same in-situ and archive data was used), our historical timeseries suggest half a year's growth rate, i.e. up to maximally 0.63 ppt, 0.065 and 0.015 ppt, too low mole fractions for $CF_4$, $C_2F_6$, $C_3F_8$, respectively for the pre-2002 timeframe. In terms of radiative forcing, this difference amounts to approximately 0.00022 Wm-2, 0.000016 Wm-2 and 0.0000043 Wm-2 in the years with the maximal growth rates (1980, 1999 and 2002, respectively). Given that some CMIP6 models had started using the historical data by the time of discovering this error (which will have no significant effect on CMIP6 outputs), we opted for not revising this study's CMIP6 datasets."

Thank you very much for your time and effort and guidance. The final manuscript and figures are uploaded separately.

Sincerely,

A/Prof. Malte Meinshausen
Director of the Australian-German Climate & Energy College

**Australian-German Climate and Energy College**
Lab14, Level 1, 700 Swanston Street, University of Melbourne, Parkville, Victoria 3010, Australia
**T:** +61(0)3 8344 4124   **E:** manager@climate-energy-college.org   **W:** www.climateenergycollege.unimelb.edu.au